# European summer weather linked to North Atlantic freshwater anomalies in preceding years

Marilena Oltmanns[1], N. Penny Holliday[1], James Screen[2], Ben I. Moat[1], Simon A. Josey[1], D. Gwyn Evans[1], and Sheldon Bacon[1]

[1]National Oceanography Centre, Southampton, UK
[2]University of Exeter, Exeter, UK

**Correspondence:** Marilena Oltmanns (marilena.oltmanns@noc.ac.uk)

**Abstract.** Amplified Arctic ice loss in recent decades has been linked to increased occurrence of extreme mid-latitude weather. The underlying mechanisms remain elusive, however. One potential link occurs through the ocean as the loss of sea ice and glacial ice leads to increased freshwater fluxes into the North Atlantic. Thus, in this study, we examine the link between North Atlantic freshwater anomalies and European summer weather. Combining a comprehensive set of observational products, we
show that stronger freshwater anomalies are associated with a sharper sea surface temperature front between the subpolar and the subtropical North Atlantic in winter, an increased atmospheric instability above the sea surface temperature front, and a large-scale atmospheric circulation that induces a northward shift in the North Atlantic Current, shifting and strengthening the sea surface temperature front. In the following summer, the lower tropospheric winds are deflected northward along the enhanced sea surface temperature front and the European coastline, forming part of a large-scale atmospheric circulation anomaly
that is associated with warmer and drier weather over Europe. The identified statistical links are significant on timescales from years to decades and indicate an enhanced predictability of European summer weather at least a winter in advance, with the exact regions and amplitudes of warm and dry weather anomalies over Europe being sensitive to the location, strength, and extent of North Atlantic freshwater anomalies in the preceding winter.

## 1 Introduction

Arctic near-surface temperature is currently warming twice as fast as the global average (Cohen et al., 2019), which manifests itself in an average sea ice volume loss of $3.0 \pm 0.2 \cdot 1000$ km$^3$ decade$^{-1}$, based on the period 1979 to 2018 (Kumar et al., 2020). Similarly large losses are observed for land ice, particularly from the Greenland ice sheet, amounting to $3.0 \pm 0.3 \cdot 1000$ km$^3$ decade$^{-1}$, based on the period 2003 to 2012 (Khan et al., 2015). Earlier studies noticed statistical links between an amplified sea ice loss at high latitudes and an increased occurrence of weather extremes at mid-latitudes (Francis and Vavrus,
2012; Tang et al., 2014; Screen and Simmonds, 2013; Cohen et al., 2014). However, the robustness of these links has been questioned and the underlying mechanisms are poorly understood (Barnes, 2013; Overland et al., 2015; Blackport and Screen, 2020).

One potential connection occurs through the ocean. Specifically, the loss of sea ice and glacial ice in the Arctic and sub-Arctic regions constitutes a source of freshwater for the North Atlantic (Bamber et al., 2018; Carmack et al., 2016). Large North Atlantic freshwater anomalies, moreover, were found to give rise to cold surface anomalies and the development of storms in the subpolar region in winter (Oltmanns et al., 2020). In turn, cold anomalies in the subpolar region in winter were found to precede heat waves over Europe in the subsequent summer (Duchez et al., 2016; Mecking et al., 2019). The heat waves were attributed to a stationary jet stream over the North Atlantic (Duchez et al., 2016) and were successfully reproduced in model simulations initialised with the cold anomaly (Mecking et al., 2019). Thus, by triggering cold anomalies in winter, increased surface freshening could initiate a deterministic chain of events that first leads to cold anomalies and storms in winter and then heat waves in the subsequent summer.

While earlier studies support individual connections between the North Atlantic sea surface temperature (SST) and the jet stream (Woollings et al., 2010), or between shifts in the jet stream and European heat waves (Dong et al., 2013; Gervais et al., 2020), the role of freshwater in initiating this causal chain is unclear. Yet, given that the Arctic and sub-Arctic regions are expected to continue to warm and release freshwater from melting sea ice and glacial ice into the North Atlantic, it is critical to understand how the resulting feedbacks could affect weather in Europe.

The gap in our knowledge around the potential influences of North Atlantic freshwater anomalies on European summer weather arises from the difficulty to simulate salinity. Freshwater enters the subpolar region through narrow boundary currents and mesoscale eddies requiring ocean models with a fine grid spacing of ∼1/12° (Marzocchi et al., 2015; Böning et al., 2016; Müller et al., 2019). Most current coupled global climate models have a coarser grid spacing, giving rise to salinity biases (Mecking et al., 2017; Menary et al., 2015; Wu et al., 2018). From an observational perspective, estimating freshwater variations is also difficult. In-situ observations of sea surface salinity mostly stem from Argo floats which cannot fully capture the large spatial variability at high temporal resolution. Moreover, satellite observations of sea surface salinity are associated with large uncertainties and only available since 2009 (Bao et al., 2019; Xie et al., 2019).

Given the limitations associated with currently available model and observational products of sea surface salinity, we use a new approach to estimate freshwater variations, taking advantage of a dynamical constraint of the sea surface salinity on the SST. In the subpolar region in autumn and winter, the air is colder than the ocean surface. Thus, the surface water is cooled by the atmosphere, becomes denser, and sinks. Enhanced surface freshening reduces the surface density and requires additional cooling before the surface water is dense enough to sink. This constraint of freshwater on the surface cooling can be used to infer its variability using a mass balance analysis (Oltmanns et al., 2020).

In the following, we describe the involved data products (Section 2). We then explain the approach to estimate freshwater variability from a surface mass balance (Section 3). In Section 4, we examine the ocean-atmosphere evolution associated with freshwater anomalies, and assess their links with European summer weather based on statistical analyses (Section 4). We conclude by discussing the dynamical role of freshwater anomalies in the identified ocean-atmosphere evolution and the implications for predictability (Section 5).

## 2 Data

First, we describe the observational products involved in this study and describe any processing steps. Since the analyses are based on statistical methods, a high data quality is important. Thus, we focussed on the period since 1979, motivated by the increased data quality associated with the onset of satellite observations in 1979.

### 2.1 Datasets

The analysis of ocean variability includes a merged SST product consisting of Hadley Centre HadISST1 data (Rayner et al., 2003; Hurrell et al., 2008) and optimal-interpolated, remote sensing-based SST data from NOAA (Reynolds et al., 2002). The merged Hadley – NOAA data product has a monthly temporal resolution, a $1° \times 1°$ spatial resolution and is available from https://gdex.ucar.edu/dataset/158_asphilli.html.

To assess changes in surface currents, we further used absolute dynamic topography data since 1993, derived from altimetry (Le Traon et al., 1998). Absolute dynamic topography represents the sea level anomaly with respect to the geoid and thus, the stream function of the geostrophic surface flow. The monthly, gridded, absolute dynamic topography dataset has a spatial resolution of $0.25° \times 0.25°$ and is distributed by the Copernicus Marine Environment Monitoring Service (https://marine.copernicus.eu/). Geostrophic surface velocities were calculated from the absolute dynamic topography using $u_g = -\frac{g}{fR_E} \frac{\partial \eta}{\partial \theta}$ and $v_g = \frac{g}{fR_E cos(\theta)} \frac{\partial \eta}{\partial \phi}$, where $u_g$ and $v_g$ are the zonal and meridional velocities, $\eta$ is the absolute dynamic topography, $g$ is the gravitational acceleration, $f$ is the Coriolis parameter, $R_E$ is the Earth's radius, and $\theta$ and $\phi$ are the latitude and longitude respectively.

Moreover, to compare freshwater anomalies, estimated from the surface mass balance analysis, with in-situ observations from the subpolar North Atlantic, we included a hydrographic, mixed layer database. The dataset provides mixed layer depths, mixed layer salinities, and mixed layer temperatures, derived from Argo float profiles (Holte et al., 2017). It is freely available at http://mixedlayer.ucsd.edu.

The ocean data is complemented by monthly output from the ERA5 atmospheric reanalysis model from the European Centre for Medium-Range Weather Forecasts since 1979 (Hersbach et al., 2018). In addition to the standard variables, we estimated the maximum Eady growth rate using monthly mean output from ERA5 to qualitatively assess the baroclinic instability in the lower troposphere over increased meridional SST gradients. Following earlier studies (Lindzen and Farrell, 1980; Dierer et al., 2005), the maximum Eady growth rate in the 1000 hPa to 750 hPa layer was calculated as $\sigma_E \approx 0.31 \frac{f}{N} \mid \frac{u_{750} - u_{1000}}{z_{750} - z_{1000}} \mid$, where $f$ is again the Coriolis frequency, $u$ is the zonal wind, $z$ the height, $N$ the Brunt-Väisälä frequency and the subscripts refer to the associated pressure levels.

A key parameter, used to derive freshwater indices, is the mean North Atlantic Oscillation (NAO), obtained from the National Oceanic and Atmospheric Administration (NOAA) Climate Prediction Center. The NAO index was calculated using Rotated Principal Component Analysis, applied to the monthly standardised 500 hPa geopotential height anomalies between $20°N$ and $90°N$ (Barnston and Livezey, 1987) and identified as the dominant mode of variability in the northern hemisphere. A detailed derivation can be found at https://www.cpc.ncep.noaa.gov/products/precip/CWlink/pna/nao.shtml.

Lastly, we included data from the Greenland climate model MAR to assess potential causes of freshwater anomalies. We used version 3.12, run at a resolution of 20 km forced by the ERA5 reanalysis (Fettweis et al., 2017) and distributed by the Laboratory of Climatology at the University of Liège. For the purpose of this study, we considered the runoff over the full ice sheet from 1950 through to the end of 2022 at monthly resolution. The dataset is available at ftp://ftp.climato.be/fettweis/MARv3.12.

## 2.2 Preprocessing

Over the investigated period, the climate has been characterised by increasing greenhouse gas concentrations, leading to enhanced surface warming (Lashof and Ahuja, 1990). Over the last two decades, moreover, the freshening has also been increasing (Tesdal et al., 2018), particularly because of increased runoff and melting from Greenland (Bamber et al., 2012, 2018). Thus, the surface warming resulting from increased greenhouse gases could superimpose on potential surface cooling or warming signals resulting from changes in the ocean or atmospheric circulations associated with increased North Atlantic surface freshening. This superposition could distort the interpretation of the statistical analyses when assessing the specific influences of changes in the ocean and atmospheric circulations associated with increased North Atlantic freshening.

Considering that the freshening trend is an important part of the signal we are investigating, removing trends at each location (or grid point) would remove an important part of a signal that we are interested in. Thus, to reduce the influence of increasing greenhouse gas concentrations on European air temperatures, we subtract regionally averaged trends from the air temperature. The method of subtracting regionally averaged trends is motivated by the observation that greenhouse gases are distributed comparatively uniformly in the atmosphere (Reuter et al., 2020) whereas the observed surface warming exhibits large regional differences (Simmons, 2022). These regional differences in surface warming result from changes in the ocean and atmospheric circulations, which are redistributing the excess heat. Since, in this study, we are specifically interested in these dynamic processes associated with changes in the ocean and atmospheric circulations, we are subtracting a spatially uniform warming trend associated with increasing greenhouse gases.

We tested different regions and found that the results are not sensitive to the exact area that is used for the averaging, as long as it is sufficiently large. Here, we averaged over the main area of investigation from 25 °N to 65 °N and from 60 °W to 60 °E, resulting in an average trend of $\sim$0.04 °C year$^{-1}$ in the 2-m air temperature from ERA5. Extending the region in any direction does not appreciably change this trend, nor the subsequent results, consistent with the assumption that the direct warming trend that is solely due to increasing greenhouse gases is distributed relatively uniformly.

While the summer air temperature is strongly affected by a spatially uniform warming trend, the other variables exhibit no or only minor trends after they have been averaged over a large area. Thus, after removing the trend in the air temperature prior to the analyses, we obtain a signal that is dynamically consistent across all investigated variables. If, on the other hand, we do not remove the trend in the air temperature, we still obtain the same patterns throughout the results but there would be a large-scale, uniform warming signal superimposed over the full domain.

We did not apply any other averaging, smoothing, filtering, or further preprocessing steps to the datasets.

## 3 Estimation of freshwater anomalies

The objective of this study is to investigate feedbacks initiated by freshwater anomalies. However, high-quality global salinity measurements have only been routinely available since 2002, and mostly in the open ocean from Argo floats. Moreover, satellite observations of the sea surface salinity are of relatively low accuracy and only available since 2009 (Bao et al., 2019; Xie et al., 2019). Considering the limitations associated with currently available salinity products, we use a surface mass balance analysis to estimate the variability of freshwater.

### 3.1 Mass balance

The mass budget for the surface mixed layer in the subpolar region in winter can be expressed as:

$$\frac{\partial}{\partial t}\left(\int_{-h(t)}^{\eta} \rho\, dz\right) = -\frac{B}{g} - \nabla \cdot \left(\int_{-h(t)}^{\eta} \rho \boldsymbol{u}\, dz\right), \tag{1}$$

where $\rho$ is the mixed layer density, $h$ is the mixed layer depth, $\eta$ is the surface elevation above the geoid (which is equivalent to the absolute dynamic topography), $g$ is the gravitational acceleration, $B$ is the buoyancy flux through the surface, and $\boldsymbol{u}$ corresponds to the velocity vector (Gill, 2016; Griffies and Greatbatch, 2012). The term $-\nabla \cdot \left(\int_{-h(t)}^{\eta} \rho \boldsymbol{u}\, dz\right)$ represents the convergence of mass, which we separate into an active component $A$ and a passive component $E$. The passive component is defined as the entrainment of mass into the mixed layer that results from mixed layer deepening as the mixed layer density increases. The active component results from externally forced, horizontal and vertical mass fluxes, such as wind-driven Ekman transports and upwelling. The passive component can only change the mixed layer depth, but not its density, while the active component does change the mixed layer density.

Next, we assume that the density is homogeneous in the mixed layer and that $\eta$ in winter is much smaller than the mixed layer depth $h$. After integrating Eq. (1) from summer to winter and neglecting the contribution of the surface elevation $\eta$ relative to the mixed layer depth on the lefthand side of Eq. (1), we thus obtain:

$$\rho h \approx h_0 \rho_0 + \left(-\frac{B}{g} + A + E\right) \cdot \Delta t, \tag{2}$$

where $h_0$ and $\rho_0$ represent a mixed layer depth and density at the end of the summer (for instance in September), $h$ and $\rho$ refer to the depth and density in winter (January to March), and $\Delta t$ is the corresponding integration interval from summer to winter.

While the climatological mean mixed layer density increases during the winter, the mixed layer deepens. Thus, before the winter, the mixed layer is several tens of metres deep while during the winter, it reaches several hundred metres. Since the density anomaly in the initial shallow mixed layer becomes distributed over a much larger depth range, the first term on the righthand side is negligible compared to the other terms. Any density anomalies beneath the initial, shallow mixed layer are still included in $E$. Eq. (2) thus simplifies to:

$$\rho h \approx \left(-\frac{B}{g} + A + E\right) \cdot \Delta t. \tag{3}$$

We further separate each term into a mean and an anomaly $n$, with $n$ referring to the n'th winter of an arbitrary subset of $N$ winters and the mean representing the mean over these winters. Since we have defined $E$ as a passive component that can only change the mixed layer depth, not its density, we can write it as $E_n \cdot \Delta t = h_n \cdot (\rho_n + \rho_{mean})$. Moreover, assuming that the mean state is in balance, we subtract the mean values from Eq. (3), resulting in:

$$\rho_n h_{mean} + \rho_{mean} h_n + \rho_n h_n \approx \left(-\frac{B_n}{g} + A_n\right) \cdot \Delta t + h_n \cdot (\rho_n + \rho_{mean}), \tag{4}$$

where the terms involving $h_n$ cancel each other.

  Lastly, we express the density as a function of temperature and salinity by considering variations in the density around a reference state, which we choose to be the mean over the $N$ selected winters. Since local density variations due to pressure are several orders of magnitude smaller than those, due to changes in salinity and temperature (Talley, 2011), we only consider temperature and salinity variations: $\rho_n \approx \rho_{mean}(-\alpha \cdot T_n + \beta \cdot S_n)$, where $T$ is the temperature, $S$ is the salinity, $\alpha$ and $\beta$ are

the thermal and haline expansion coefficients. Plugging this expression into Eq. (4), we obtain:

$$\rho_{mean}(-\alpha \cdot T_n + \beta \cdot S_n) \cdot h_{mean} \approx \left(-\frac{B_n}{g} + A_n\right) \cdot \Delta t. \tag{5}$$

  The objective of the following analysis is to find conditions in which the density anomalies associated with temperature anomalies are much larger than the effect of potential, active drivers of density anomalies on the righthand side of Eq. (5): $\rho_{mean} \cdot h_{mean} \cdot |\alpha \cdot T_n| \gg |\left(-\frac{B_n}{g} + A_n\right)| \cdot \Delta t$. Under these conditions, the temperature and salinity anomalies must compen-

sate each other in their influence on density, allowing us to estimate the salinity anomalies from the associated temperature anomalies: $\beta S_n \approx \alpha T_n$.

  The idea that such conditions exist is motivated by the observation that salinity changes are not only a response to surface fluxes and entrainment but can, in turn, constrain the drivers of density anomalies. Large freshwater anomalies in winter can impede convection and entrainment and thus limit the oceanic heat release to the atmosphere ($B$). At the same time, a stronger

surface cooling is required to mix freshwater down, influencing the mixed layer temperature $T$. Considering the competing influences of salinity and temperature on stratification, the conditions in which freshwater may impact the temperature, can only occur in autumn and winter, when surface water is cooled by the atmosphere, becomes denser and sinks. In summer, the temperature and salinity do not compete in their influence on stratification and thus, do not constrain each other.

  To exploit this constraint of salinity on temperature and identify these potential conditions, we assume that the surface

mixed layer in winter is relatively well mixed, so we can approximate the mixed layer temperature $T$ with the SST. We then search for potential freshwater indices that exhibit a strong linear relationship with subpolar SST anomalies, regress Eq. (5) onto these indices, and compare the magnitude and spatial characteristics of the resulting terms. If enhanced surface freshening substantially affected the SST, we expect the terms $A_n$ and $B_n$ to drop out of Eq. (5) after the regression. In essence, the indices serve as filters that help us to identify conditions in which freshwater anomalies have been sufficiently large to influence the

heat exchange between the ocean and the atmosphere, either within the subpolar region or before entering it. Later (in Section 4.5), we will further assess if these conditions in which the air-sea heat exchange and, in turn, the SST have been affected by freshwater anomalies hold generally over the North Atlantic or only for selected indices.

## 3.2 Derivation of freshwater indices

The challenge in detecting the conditions in which freshwater anomalies may have affected the SST, consists in the complexity of SST and freshwater variability in the subpolar region. In theory, changes in surface freshwater can be influenced by river runoff, sea ice and glacial melting, evaporation and precipitation, mixing, and ocean currents. Considering that multiple factors can contribute to freshwater variations over a range of timescales and spatial scales, it may not be possible to reduce the complexity of freshwater variability in space and time into a single, one-dimensional index.

To overcome this challenge, we construct indices over subsets of years that allow us to closely constrain the variability of the SST over the selected subset. Thus, this approach is different to traditional methods in which the dynamical mechanisms are known a priori, and statistical methods are used to assess the significance of these mechanisms. Here, we first select indices with a strong and significant statistical relationship with the SST, and then look for potential freshening mechanisms that can explain the relationship, assuming that these mechanisms exist but may be masked by other drivers.

As a first, educated guess to identify suitable freshwater indices, we start with the NAO index in summer (Fig. 1a), motivated by its dynamical links to freshwater. On the one hand, a more negative NAO phase in summer has been associated with enhanced runoff and melting over Greenland (Hanna et al., 2013, 2021), which is a source of freshwater to the North Atlantic (Bamber et al., 2018; Dukhovskoy et al., 2019). On the other hand, a more positive NAO phase has been associated with an intensified subpolar gyre circulation, leading to enhanced freshwater imports into the subpolar region (Häkkinen et al., 2011a, 2013; Holliday et al., 2020). Yet, even if the freshening occurs in summer (when melting and runoff is strongest), the effect of the freshwater on the SST would only become visible in autumn and winter (when the freshwater impedes the sinking of surface water). Thus, we focus on the SST in winter to infer the occurrence of freshwater anomalies.

Consistent with the existence of multiple possible drivers of freshwater and SST anomalies in winter, we obtain a qualitatively different relationship between the summer NAO index in July and August ($NAO_S$) and the temperature difference between the subpolar and subtropical gyres in the subsequent winter below and above a threshold of $\sim -0.5$ in $NAO_S$ (Fig. 1a to d). Below the threshold of $\sim -0.5$, there is a progressively larger SST difference between the northern subtropical region and the southern subpolar region for decreasing $NAO_S$ phases in the preceding summer (Fig. 1b). Above this threshold, there is a progressively larger SST difference for increasing $NAO_S$ phases in the preceding summer. Also, the associated cold, subpolar SST anomaly is weaker and displaced further to the northwest (Fig. 1c). The threshold of $\sim -0.5$ was initially identified using box regions for the subpolar and subtropical regions (for instance with latitudinal boundaries between 45 °N and 60 °N for the subpolar region and between 30 °N and 45 °N for the subtropical region), but the identified relationships are not sensitive to the exact region.

Next, we strengthen the identified relationships between the two NAO subsets (above and below $-0.5$) and the subsequent SST anomalies through subsampling. Specifically, if $x_i$ corresponds to the $NAO_S$ subset years, and $y_i$ corresponds to the SST anomaly in the subsequent winter, we strive to derive a linear relationship $y = ax + b$, where $a$ and $b$ are constants and in which $|a|$ is high. The higher the magnitude of $a$ is, the higher is the magnitude of $\alpha T$ on the lefthand side of Eq. (5) after regressing Eq. (5) onto the index. Thus, we aim to select NAO years for which the magnitude of the slope $a = \frac{y_i - y}{x_i - x_0}$ is large, where

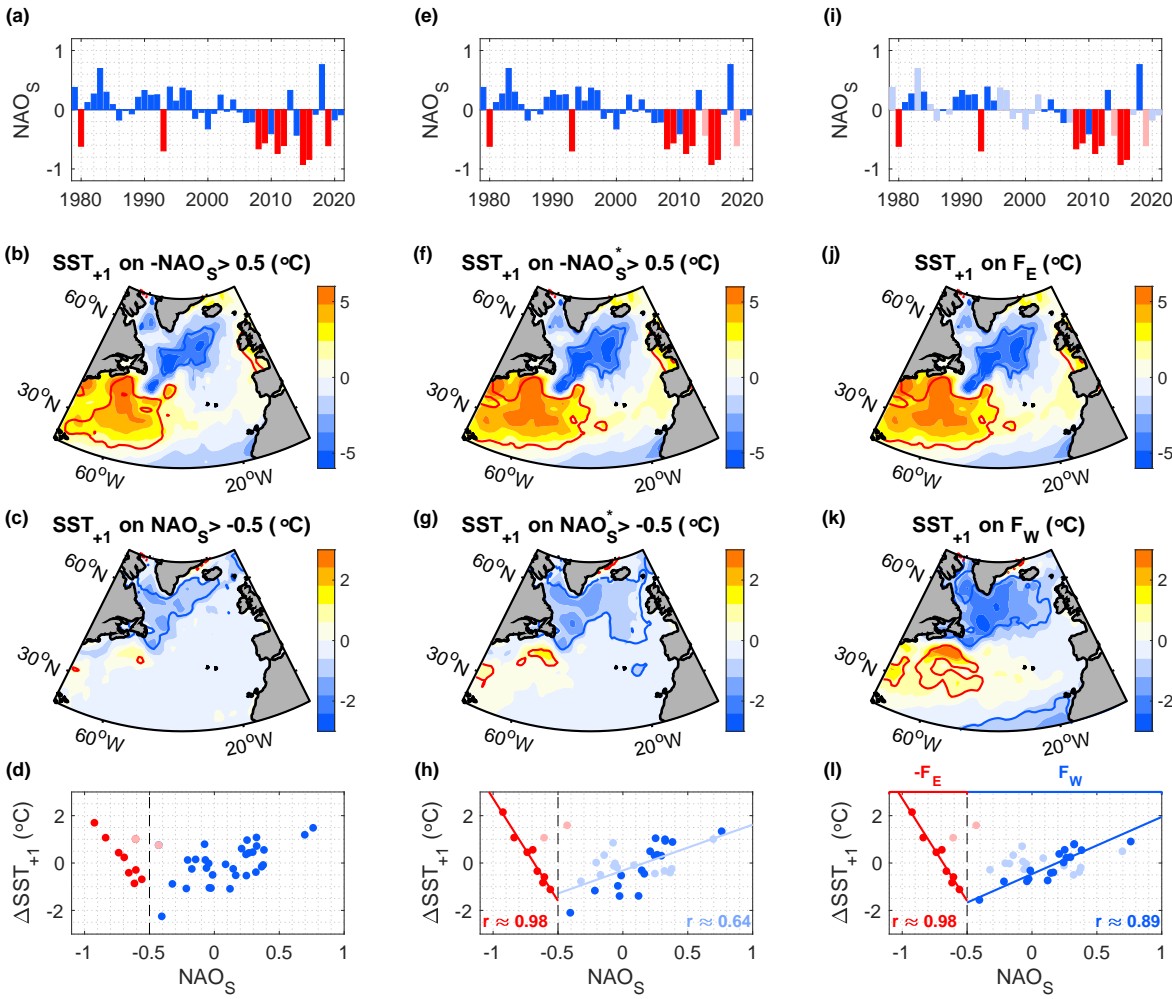

**Figure 1.** (a,e,i) Variability of the NAO index in July and August (NAO$_S$). The strong red coloured bars represent the NAO years used for the regressions in the second row (panels b, f, and j), and the strong blue coloured bars represent the NAO years used for the regressions in the third row (panels c, g, and k). Light coloured bars indicate the years that were removed in the course of the subsampling. (b,c) Regression of the SST in winter (January through to March) onto (b) $-1 \times$NAO$_S$ in all years in which NAO$_S < -0.5$, (c) NAO$_S$ in all years in which NAO$_S > -0.5$. The SST anomalies correspond to the winter following the NAO$_S$ summers (indicated by the '+1' in the titles). (f,g) As in b and c but for NAO$_S^*$ corresponding to NAO$_S$ without the two light red coloured outliers. (j,k) As in b and c but for the final two freshwater indices F$_E$ and F$_W$ (shown by the strong coloured years in i and l). Contours encompass regions that are significant at the 95% confidence level. Please note the different colour scales. (d,h,l) Relationship between NAO$_S$ and the subsequent $\Delta$SST in winter, where $\Delta$SST represents the SST difference between the red, subtropical and the blue, subpolar 95% confidence regions in the respective panels above (panels b, f, and j for the red years and panels c, g, and k for the blue years), relative to the temporal means over each subset. Light coloured dots correspond to years that are rejected in the course of the subsampling. The final indices F$_E$ and F$_W$ are shown as strong coloured dots in l.

$x_0 = x|_{y=\bar{y}_i}$ and $\bar{y}_i$ represents the (temporal) mean over the subset $y_i$. At the same time, we strive to obtain a high correlation between the subset and the subsequent SST anomalies. Thus, we aim to select NAO years where $(x_i - x_0)^2$ is large, since this increases the variance of the SST anomalies that can be explained by the index.

The values $x_i$ included in each subset directly correspond to the respective $NAO_S$ values (Fig. 1c) without scaling them, while the values $y_i$ correspond to the observed SST difference between the subpolar and the subtropical gyres in any given year ($\Delta SST$), rather than only the SST anomaly at a single location. Using the SST difference has the advantage that we filter out any spatially uniform, radiative warming signals due to increasing greenhouse gas concentration. In addition, we only average the SST over regions in which the SST anomalies are significant (Fig. 1b and c), allowing us to directly inspect the robustness of the correlations and ensure that they are not due to outliers or clusters. Thus, we identify two outliers, corresponding to the NAO summers in 2014 and 2019 (faint red years in Figure 1d), which we exclude from the subsequent subsampling to obtain a faster convergence of the results. Individual inspection of both years showed that they were associated with pronounced cold, subpolar SST and freshwater anomalies. However, their relation to $NAO_S$ differed from those in the other years, precluding them from being included in the subsets.

Following the above objectives to maximise the slope and variance of the subsampled index, we select the $N$ years where the term $(y_i - y) \cdot (x_i - x_0)$ is highest. Here, the subscript $i$ refers to all years in each subset, excluding the two outliers, and $y$ is the associated, linear regression of $y_i$ on $x_i$ (Fig. 1e to h). Graphically, the subsampling is equivalent to increasing the slope of the regression line (the light blue line in Figure 1h) while keeping a high variance. Thus, the method aims at increasing the statistical relationship between two variables and thus identifying dynamical links, based on the assumption that noise, and other mechanisms, can mask these links. Once a strong statistical connection has been established, the physical basis will be assessed by investigating the potential, underlying dynamical links.

There is a trade-off between the number of years $N$ included in each subset and the resulting correlations between the NAO subset and subsequent SST pattern. Considering that the number of years is already low for the cold anomalies preceded by negative NAO summers, where $NAO_S$ is smaller than $-0.5$ ($N = 8$), we do not apply any further subsampling (Fig. 1i and j).

For the other subset (corresponding to the $NAO_S$ years higher than $-0.5$), we select $N = 17$ years as a reasonable compromise for obtaining a high correlation while keeping a relatively large sample size (Fig. 1i and k). Thus, we achieve an increase in the correlation between the subsampled $NAO_S$ index and the resulting SST difference between the subpolar and subtropical region from $\sim$0.64 to $\sim$0.89, resulting in a low p-value of $\sim$1.8 $\cdot$ $10^{-6}$ (Fig. 1h and l).

In Section 4.5 and Appendix B, we show that the results are not sensitive to the subsampling or the number of years included. However, having a close relationship between the index and the SST results in reduced uncertainties when estimating the associated freshwater anomalies. In addition, the high correlations help us to identify and assess potential dynamical links more clearly: Freshwater indices that are only poorly correlated with freshwater are only of limited use when assessing links between freshwater and other ocean or atmospheric parameters. Since the indices will be used as a tool for representing freshwater anomalies, high correlations between the indices, the SST and potential freshwater anomalies are a prerequisite, not a conclusion, and we make no assumptions on the suitability of both subsets outside the selected years.

Through the subsampling, we have derived two subsets with close, linear relationships with the SST difference between the subpolar and subtropical gyre (Fig. 1i to l). To distinguish the two subsets from each other, we name them according to the location of the associated cold SST anomalies. Since the maximum cold anomalies associated with $NAO_S < -0.5$ are strongest over the southeastern subpolar region (Fig. 1j), we refer to the selected 8 years as $F_E$ index — shown by the clear red coloured bars in Figure 1i. For the other subset, the maximum cold anomalies extend over the full subpolar gyre, including the western part (Fig. 1k). Thus, we refer to the selected 17 years as $F_W$ index — shown by the clear blue coloured bars in Figure 1i. The corresponding years included in each index are additionally listed in Table A1. In the following analyses, we will examine the dynamical links of both indices to freshwater anomalies and the associated air-sea feedbacks.

## 4 Results

Having selected two NAO subsets, we will first assess their suitability for representing freshwater anomalies. Thus, we evaluate the associated mass balance to estimate freshwater anomalies and examine their potential causes. We will then use the indices to investigate links between the estimated freshwater anomalies and the large-scale ocean and atmospheric conditions in winter and summer, and test if the identified links hold generally by using an un-subsampled index. Lastly, we will assess the role of North Atlantic freshwater anomalies as a predictor for Europe's warmest summers by constructing composites of the 10 warmest relative to the 10 coldest summers between 1979 and 2022 and comparing the preceding freshwater anomalies.

### 4.1 Estimation of freshwater anomalies

Taking advantage of the strong relationships between the selected $NAO_S$ subsets and subsequent SST anomalies, we regress each term in Eq. (5) on the corresponding indices $F_E$ and $F_W$. We then evaluate the surface mass balance over the subpolar cold SST anomaly regions within the regions enclosed by the 95% confidence lines. In the following, we present the key analysis steps and results while a detailed evaluation and comparison with in-situ observations is provided in Appendix A.

To estimate the temperature term $\alpha T_n$, we again assume that the mixed layers are relatively homogenous and approximate the mixed layer temperature with the SST, averaged over the winter (January through to March). Even if the SST is slightly warmer or colder than the mixed layer temperature, the relationship between the mixed layer temperature and the mixed layer salinity will still remain the same as that between the SST and the sea surface salinity, due to having a constant density profile in the mixed layer. To estimate the mean mixed layer depth $h_{mean}$, moreover, we averaged the mean mixed layer depth, obtained from Argo floats (Holte et al., 2017), over the same regions and months as the SST, resulting in a mean mixed layer depth of ~250 m for the $F_E$ subset and ~280 m for the $F_W$ subset.

We further compute $\alpha$ and $\beta$ using the Gibbs Seawater Routines (McDougall et al., 2009), in accordance with the highest standards of current knowledge. Noting that the effects of salinity and pressure on $\alpha$ and $\beta$ are small and only affect the second decimal place or less, we use nominal values of 35 g kg$^{-1}$ and 10 db for the subpolar region in winter to compute $\alpha$ and $\beta$. The dependence of $\alpha$ and $\beta$ on temperature is larger, however. For instance, $\alpha$ can vary from $5 \cdot 10^{-4}$ to $18 \cdot 10^{-4}$ °C$^{-1}$ across the subpolar ocean surface. Thus, for an enhanced accuracy, we allow $\alpha$ and $\beta$ to vary with the SST.

Next, we estimate the terms on the righthand side of Eq. (5). On the timescales and spatial scales considered, oceanic flows are predominantly in geostrophic balance, redistributing heat and freshwater. However, geostrophic flows cannot contribute to a net mass input. Over the open ocean, away from topographic boundaries, on interannual timescales, the winds and air-sea fluxes represent the largest energy sources that can result in vertical mixing or horizontal mass convergence (Ferrari and Wunsch, 2009; Wunsch and Ferrari, 2004). Other sources of energy include pressure loading by the atmosphere, geothermal heating, biological activity, and the tides but we estimate them to be negligible over the investigated timescales and spatial scales. Thus, the terms on the righthand side of Eq. (5) are confined to the surface buoyancy fluxes, horizontal Ekman transports and wind-driven vertical fluxes, all of which are estimated using the atmospheric reanalysis ERA5. Considering the nonlinearity within the individual terms of Eq. (5), we first evaluate each term before regressing it onto the indices.

After estimating the density anomalies associated with the cold anomalies, and the buoyancy fluxes, the horizontal Ekman transports and the vertical Ekman velocity, and regressing them onto the freshwater indices (Appendix A), we find: Regardless of the exact region and mean mixed layer depth, and regardless of which month is selected as starting month of the integration, the density increase implied by the cold anomalies associated with $F_E$ and $F_W$ is always more than an order of magnitude larger than the density changes associated with $A_n$ or $B_n$. Moreover, neither of these fluxes is significantly correlated with the subsets, and their spatial patterns are inconsistent with the obtained SST patterns, regardless of whether we include the full subpolar region where the SST anomaly is negative or only the region enclosed by the 95% confidence lines, or whether we start the integration in October or only consider the winter months January to March.

With the buoyancy fluxes, vertical Ekman transports and horizontal advection being negligible, we conclude that the density increase associated with the cold anomalies must be balanced by a density decrease associated with freshwater anomalies: $\alpha SST_E \approx \beta SSS_E$, and $\alpha SST_W \approx \beta SSS_W$, where $SSS$ is the sea surface salinity and the subscripts refer to the anomalies obtained from the regressions onto the respective index. This result implies a close connection between freshwater and SST anomalies included in each subset. A demonstration of the connection between SSS and SST anomalies with hydrographic observations shows that, even in winters with most intense air-sea fluxes, freshwater anomalies can still be inferred from the SST with reasonably small uncertainties (Appendix A).

Using the obtained density compensation between SSS and SST anomalies, we estimate SSS anomalies from the two NAO subsets. Thus, we find that the maximum freshwater anomalies (or minimum SSS anomalies) associated with $F_E$ occur over the central subpolar region (corresponding to the southeastern subpolar gyre) and are spatially more confined than the maximum freshwater anomalies associated with $F_W$ (Fig. 2a and b). Moreover, the significant area of $F_W$ freshwater anomalies extends further eastward, westward, and northward compared to $F_E$ freshwater anomalies and the anomalies have a smaller amplitude, consistent with the associated cold, subpolar SST anomalies (Fig. 1e and f).

Since the buoyancy fluxes represent the largest term on the righthand side of Eq. (5), they determine the uncertainty of the obtained salinity estimates, amounting to 4% for the $F_E$ subset and 6% for the $F_W$ subset (Appendix A). These uncertainties apply to the cold anomaly regions, enclosed by the 95% lines. Uncertainties at each individual grid point can differ. Moreover, if the freshwater forcing is very large, the surface mass balance may underestimate the freshening because freshwater anomalies can (in theory) increase up to a salinity threshold of zero, while SST anomalies cannot drop below the air temperature. Still,

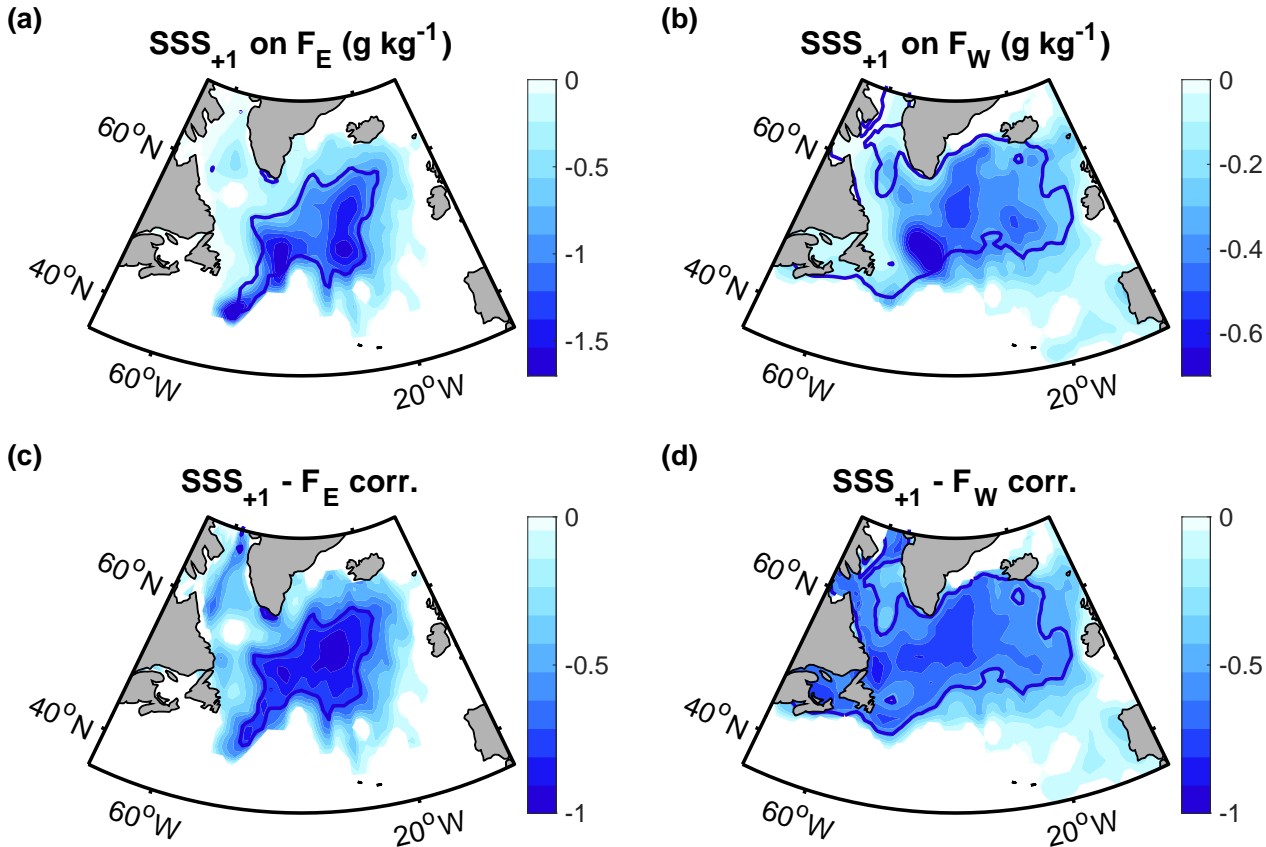

**Figure 2.** (a,b) Regression of the sea surface salinity in winter (January through to March) on the two freshwater indices from the preceding summer (Fig. 1c). The contours delineate the regions that are significant at the 95% confidence level. (c,d) Correlations between the sea surface salinity in winter and the freshwater indices from the preceding summer, with the thick contours delineating the regions that are significant at the 95% confidence level, assessed by means of two-sided t-tests. The underlying sea surface salinity variability has been estimated from the surface mass balance by assuming density compensation with the SST anomalies.

we find that even during the strong observed freshwater anomalies in 2015 and 2016, the surface mass balance provided a good approximation (Appendix A), suggesting that a potential underestimation is only small.

In addition to the low overall uncertainties of the SSS estimates, another implicit advantage of the selected NAO subsets $F_E$ and $F_W$ is that they are, by construction, highly correlated with the obtained freshwater estimates in the subsequent winter, with the magnitude of the correlations between the SSS anomalies and the freshwater indices exceeding 0.9 (Fig. 2c and d). The SSS correlations with the $F_E$ subset reach their highest magnitude over the southeastern subpolar gyre while the highest magnitudes of the SSS correlations with $F_W$ occur over the central and northern subpolar region, covering an overall larger

area, like the corresponding regressions. Considering the low uncertainties of the obtained freshwater estimates, and their high correlations with the two NAO subsets, we conclude that both subsets represent suitable freshwater indices.

## 4.2 Causes of freshwater anomalies

Freshwater anomalies may result from enhanced sea ice or glacial melt, river runoff, surface fluxes (precipitation minus evaporation), and circulation changes. After investigating the surface fluxes from ERA5, glacial runoff from the Greenland climate model MAR, and the regional gyre circulation from altimetry, we find a significant anti-correlation between the summer NAO and runoff (Fig. 3a), pointing to runoff as potential freshwater source for the $F_E$ freshwater anomalies since they are preceded by a strongly negative summer NAO. While other sources of meltwater, such as sea ice, may also contribute to enhanced freshening, the correlation between runoff and the summer atmospheric circulation is consistent with other studies evaluating individual links between the summer atmospheric forcing and glacial runoff (Hanna et al., 2013, 2021), and the resulting freshwater input into the North Atlantic (Bamber et al., 2018; Dukhovskoy et al., 2019).

With the majority of seasonal runoff arriving in the subpolar gyre during autumn (Fratantoni and McCartney, 2010; Schmidt and Send, 2007), the change in the surface salinity from summer (August) to winter (January to March) has previously been estimated by evaluating Eq. (1) for a shallow surface layer (Oltmanns et al., 2020). Thus, the summer NAO, multiplied by '$-1$' was identified as a suitable index for the seasonal freshwater that reaches the subpolar region between August and winter (Fig. 3b). The timing of the increased seasonal freshwater input associated with $-1 \times NAO_S$ supports the role of seasonal runoff and melting for the $F_E$ freshwater anomalies. However, we point out that the relationship between the NAO index and runoff cannot explain differences in the freshwater anomalies within the $F_E$ subset. Instead, it explains the existence of the $F_E$ subset in the first place since runoff and increased surface melting are the only drivers of freshwater anomalies that are anti-correlated with the summer NAO.

While the full, un-subsampled summer NAO is a suitable indicator of runoff and the seasonal surface freshening from summer to winter, it is not necessarily correlated with absolute SSS anomalies in winter. Once a seasonal mixed layer is eroded, the SST and surface salinity are expected to be influenced by other factors, consistent with the nonlinear relationship between the summer NAO and subsequent winter SST anomalies (Fig. 1d).

Among the dominant drivers of deeper freshwater anomalies is the subpolar gyre circulation. Specifically, a stronger subpolar gyre circulation, particularly in the northwestern subpolar region, has been found to lead to enhanced inflow of fresh and cold polar water from the coastal Labrador Current into the subpolar gyre (Häkkinen and Rhines, 2009; Häkkinen et al., 2011a, 2013; Koul et al., 2020). Since the subpolar gyre circulation is, in turn, largely forced by the wind stress (Häkkinen et al., 2011b; Spall and Pickart, 2003), earlier studies have identified a significant link between a stronger wind stress curl over the subpolar North Atlantic and a reduced salinity in the subpolar gyre (Häkkinen and Rhines, 2009; Häkkinen et al., 2011a, 2013; Hátún et al., 2005; Holliday et al., 2020).

To assess the role of the wind stress curl and subpolar gyre circulation for the cold and fresh anomalies associated with higher summer NAO states, we inspect the associated absolute dynamic topography in winter. The full, un-subsampled summer NAO only displays a weak and mostly non-significant relationship with the geostrophic surface circulation in the southwest subpolar region (Fig. 3c). When using the sub-sampled summer NAO corresponding to the $F_W$ subset, however, the absolute dynamic topography north of 50 °N in winter is significantly reduced, implying a more cyclonic and hence, stronger subpolar

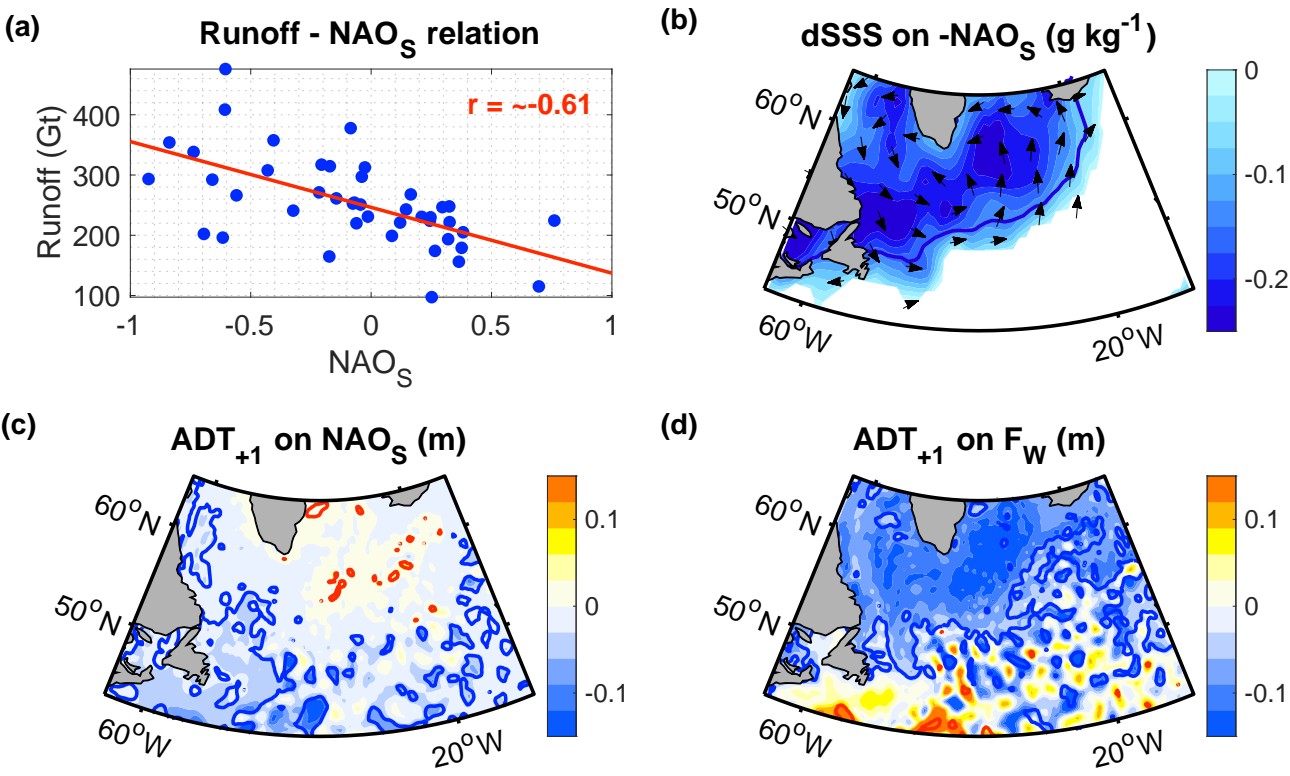

**Figure 3.** (a) Relationship between the NAO in July and August (NAO$_S$) and the runoff integrated over the Greenland ice sheet and over July and August. (b) Regression of the newly arriving, seasonal surface freshening between summer (August) and winter (January to March) onto $-$NAO$_S$ from the preceding summer. dSSS corresponds to the SSS change from summer and winter, estimated using a mass balance analysis (Oltmanns et al., 2020). The arrows represent the mean geostrophic surface flow, obtained from the absolute dynamic topography, averaged from August to March (the period of the freshening). Multiplying the summer NAO by '$-1$' serves the purpose of using an index that is positively correlated with the surface freshening. (c,d) Regression of the absolute dynamic topography in winter (January to March) onto (c) NAO$_S$ and (d) F$_W$ from the preceding summer. Contours encompass regions that are significant at the 95% confidence level.

gyre circulation in the northwest subpolar region (Fig. 3d). The strengthened relationship between the subsampled summer NAO and the subpolar gyre circulation thus supports the subsampling by providing a physical explanation for the freshwater anomalies associated with $F_W$ (Fig. 2b). For the rejected years (in the second step of the subsampling), the dependence of hydrographic anomalies on the subpolar gyre circulation still holds (Häkkinen et al., 2011a, 2013), but the NAO index is not a suitable indicator of this circulation.

While a detailed quantification of freshwater budget is beyond the scope of this study, the proposed physical causes of the obtained freshwater estimates are supported by their spatial characteristics and intensities. $F_W$ freshwater anomalies are largest over the western subpolar region, where the subpolar gyre circulation is strongest and where the surface heat fluxes are largest, and can erode seasonal freshwater anomalies more easily. $F_E$ freshwater anomalies are largest over the southeastern part of the subpolar region where surface fluxes and the subpolar gyre circulation are weaker, and where the mixed layer depths are shallower. We also examined the associated surface fluxes (precipitation minus evaporation) but found that they were too small to account for freshwater anomalies. In autumn and winter, moreover, the surface freshwater fluxes were evaluated as part of the buoyancy fluxes in the surface mass balance and found to be negligible. The implication that seasonal runoff and melt may cause absolute freshwater anomalies in winter is new and suggests that the strong fresh and cold anomalies in the subpolar North Atlantic in the mid 2010s were forced by a different mechanism to those in earlier decades.

### 4.3 Atmosphere-ocean circulation in winter

Next, we examine the large-scale atmosphere circulation associated with both types of freshwater anomalies. We focus on the anomalies that are represented by the $F_E$ subset (Fig. 2a) due to their sharper SST signals. However, freshwater anomalies associated with the $F_W$ subset show qualitatively similar atmospheric responses, both in winter (not shown) and in summer (Section 4.4). Since the meridional SST gradient is increased in winters after stronger freshwater anomalies, there is a sharper SST front between the subtropical and the subpolar gyre, particularly over the western North Atlantic (Fig. 1j). Directly above this sharper SST front, we observe an amplified baroclinic instability in the atmosphere, indicated by an enhanced Eady growth rate (Fig. 4a).

The amplified baroclinic instability manifests itself in a distinct atmospheric circulation anomaly. When an air parcel travels northward across the SST front, it rises because it is warmer than the surrounding air masses. By rising, the air column stretches, acquiring positive vorticity. The opposite occurs when an air parcel travels southward across the front. Consistent with the resulting enhanced baroclinic wave activity, the observations show a cyclonic anomaly north of the SST front and an anticyclonic anomaly to the south (Fig. 4b), representative of a positive NAO phase. Accordingly, we find that after all but the two weakest $F_E$ years, the NAO changed from its strongly negative state in summer into a positive state in the subsequent winter. Moreover, over the full period 1979 to 2022, without conditioning on $F_E$ years, the correlation between the NAO in summer (July and August) and the NAO in the subsequent winter (January to March) is r $\approx 0.12$, which is not significant (p $\approx$ 0.46). With conditioning on $F_E$ years, the correlation is r$\approx -0.74$, which is significant (p $\approx 0.03$).

The obtained atmospheric circulation anomaly drives a convergent Ekman transport between the subtropical and subpolar gyre (Fig. 4b), leading to an increase in sea level. This Ekman transport is an instantaneous response to the wind forcing

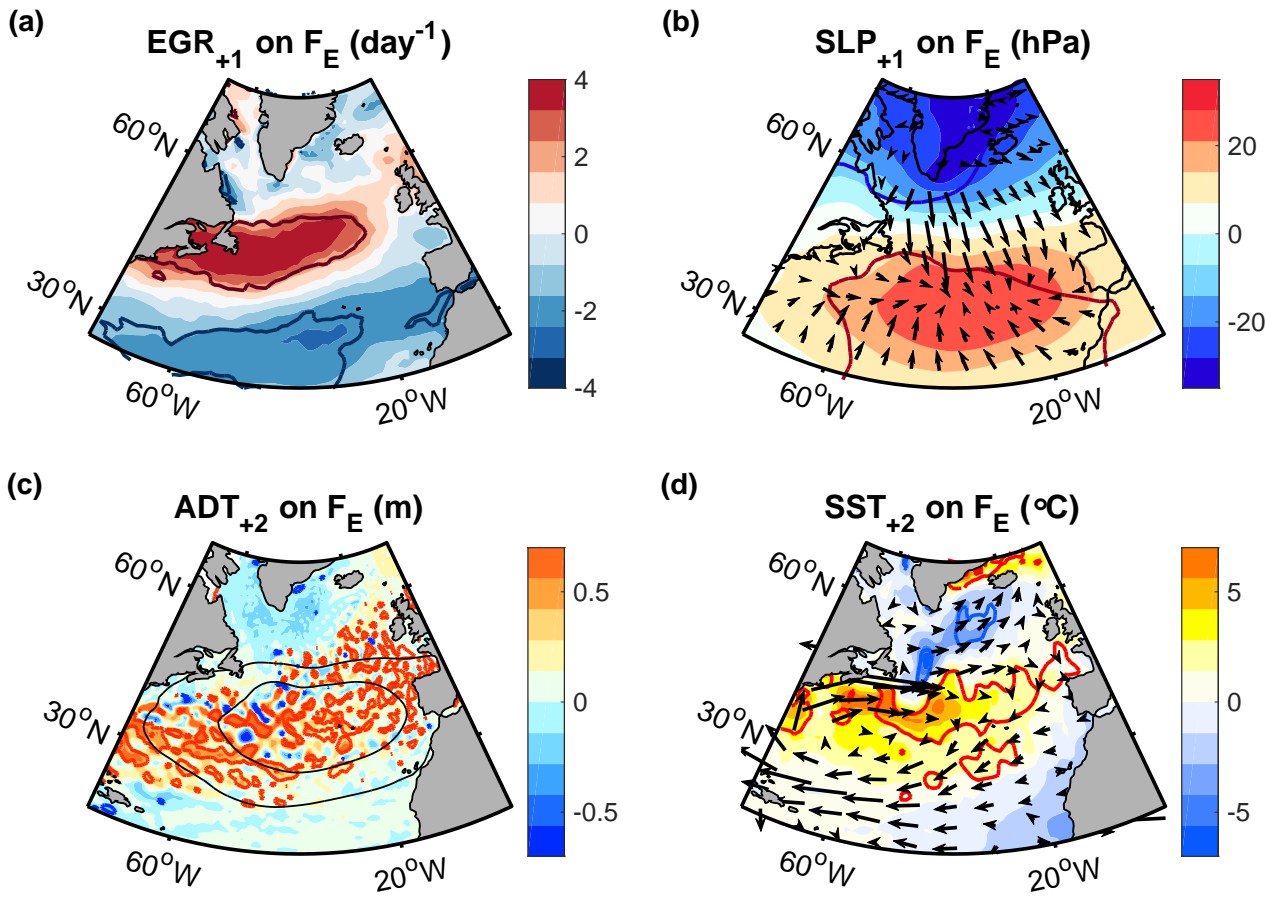

**Figure 4.** Regressions of (a) the maximum Eady growth rate in the lower troposphere, (b) the sea level pressure, (c) the absolute dynamic topography (ADT) and (d) the SST in winter (January through to March) on $F_E$. (a) and (b) are in the first winter after the anomalies whereas (c) and (d) are in the second winter after $F_E$ (indicated by the '+1' and '+2' in the title). The arrows in (b) show the direction of the associated Ekman transports. The arrows in (d) represent the smoothed geostrophic flow implied by the ADT anomaly. The thin black contours in (c) show the region of Ekman flow convergence from (b). Thick contours in all panels encompass regions that are significant at the 95% level.

(on the investigated timescales) but the resulting increase in sea level and horizontal pressure gradients has longer lasting
        effects. Thus, we find that the increased sea level and associated ocean instabilities manifest themselves in a broad band of
        anti-cyclonic eddies that extends into the second winter after the freshwater anomalies (Fig. 4c). The eddies are not visible
        in the SST due to the coarser $1° \times 1°$ grid spacing of the SST product, compared to the $0.25° \times 0.25°$ grid spacing of the
        absolute dynamic topography product. Considering that the mean flow along the eddies is eastward, representing the North
Atlantic Current, the integrated effect of the anti-cyclonic eddies is a reduced eastward speed at the southern edge of the band
        and an increased eastward speed at the northern edge (Fig. 4d). This circulation pattern has been referred to as inter-gyre gyre
        circulation (Marshall et al., 2001) and is equivalent to a northward shift of the North Atlantic Current (Kostov et al., 2021;
        Zhao and Johns, 2014).

        The northward shift of the North Atlantic Current implies a warm SST anomaly to the south of the subpolar cold anomaly,
not because the water inside the current is anomalously warm but because the current occurs at an anomalously northward
        location. Thus, the warm SST anomaly to the south of the subpolar cold anomaly is reinforcing the SST gradient, driven by
        the large-scale winds. It is seen in the first, and the second winter after freshwater anomalies (Fig. 1e and 4d). However, in
        the first winter, the northward shift is partially obscured by the southward expansion of the cold SST anomaly over the eastern
        North Atlantic, potentially driving enhanced mixing and erosion of the SST front. At the same time, the spatial distribution of
the surface heat fluxes does not match the SST field (Fig. A1d), indicating that the contribution of the surface heat fluxes to
        the warm SST anomaly is limited. While this mechanism has been demonstrated using the $F_E$ subset, the signals for the $F_W$
        subset are qualitatively the same but confined to only the first winter.

        We summarise that freshwater anomalies are associated with cold anomalies in the subpolar region in winter (Fig. 1e and f).
        The cold anomalies form part of an enhanced meridional SST gradient, implying a sharper SST front between the subpolar gyre
and the subtropical gyre. The sharper SST front is associated with an amplified baroclinic instability in the atmosphere (Fig.
        4a) that is characterised by a more cyclonic circulation anomaly over the subpolar gyre and a more anticyclonic anomaly to
        the south (Fig. 4b). This atmospheric circulation anomaly sets up surface pressure gradients through Ekman transports, which
        drive a geostrophic flow that contributes to the warm anomaly south of the subpolar cold anomaly (Fig. 4c and d).

        The overall effect of the ocean-atmosphere coupling is a sharper SST gradient between the subtropical warm anomaly
and the subpolar cold anomaly, which is characteristic of the large-scale SST tripole pattern and associated feedbacks (Czaja
        and Frankignoul, 2002; Marshall et al., 2001). By being highly correlated with the SST anomalies, the freshwater indices
        serve as valuable tools for visualising the associated ocean and atmospheric circulations, reinforcing each other (Figs. 1 and
        4). However, we do not causally attribute the SST pattern to freshwater anomalies, and we do not infer that the freshwater
        anomalies act as a trigger for the characteristic tripole SST pattern.

## 4.4   Links to European summer weather

        The preceding analysis revealed a close statistical link between freshwater anomalies and associated winter conditions. Next,
        we investigate the SST and atmospheric conditions in subsequent summers. To facilitate the integration of the results into a
        larger context, we are comparing the regression anomalies obtained from the two subsets with the climatological mean SST

and atmospheric conditions in summer, which are similar to the mean conditions across each of the two subsets. We start by investigating the SST field after freshwater anomalies (Fig. 5a to c). In the first summer after stronger freshwater anomalies (again represented by $F_E$), we find that the SST is characterised by an enhanced subpolar cold SST anomaly covering part of the North Atlantic Current in the central North Atlantic (Fig. 5b). In the second summer, the northward shift of the North Atlantic Current is the most pronounced signal, visible as a band of increased SST that extends northeastward across the North Atlantic from Nova Scotia towards the British coast (Fig. 5c).

The SST signal in both summers after the freshwater anomalies implies an increased SST difference between the warm subtropical gyre and the cold subpolar gyre. The exact location of the SST front between the subtropical gyre and the subpolar gyre can differ between the years and is therefore poorly constrained, resulting in reduced significances at individual grid points. However, the increased SST gradient — which is of greater dynamical relevance than absolute SST anomalies — is highly significant. For instance, the SST difference between the region in which the SST anomaly exceeds 2 °C and the region in which the SST anomaly falls below −2 °C, includes a substantial area of the extra-tropical North Atlantic (Fig. 5b and c) and is significantly correlated with the $F_E$ index with a correlation coefficient above 0.7 in both summers (r ≈ 0.76 and 0.84 in the first and second summer respectively), with p-values well below 0.05.

As in the preceding winters, we find that the atmospheric circulation is aligned with the underlying SST in both the first and second summer after the freshwater anomalies, with the winds at 700 hPa circulating cyclonicly around the subpolar cold SST anomalies (Fig. 5b and c). Accordingly, we observe a northward deflection of the lower tropospheric winds downstream of the cold SST anomaly along the coast (Fig. 5d to f). In the first summer, the northward deflection occurs west off northern Africa, Spain, Portugal, France and the British coastline (Fig. 5e). In the second summer, the northward deflection of the winds occurs further north to the northwest of the Scandinavian coastline (Fig. 5f), consistent with the more northerly SST front over the North Atlantic (Fig. 5b and c).

The northward deflection of the lower tropospheric winds forms part of a large-scale atmospheric circulation anomaly, with an increased baroclinic instability across the European coastline, a more cyclonic circulation anomaly over the ocean and a more anticyclonic circulation anomaly over the continent (not shown but indicated by the arrows in Figure 5b and c, and the southward wind deflection to the east of the northward deflection in Figure 5e and f). Thus, the large-scale atmospheric circulation is similar to the conditions described in winter (Fig. 4b). In both seasons, there is a cyclonic atmospheric circulation anomaly above the cold SST anomaly. In summer, however, the anticyclonic circulation occurs over the continent instead of over the subtropical North Atlantic. We hypothesise that the difference is due to the faster surface heating of the land in spring and summer, which increases the surface temperature difference between the colder subpolar ocean surface and the warmer continent, and hence favours an increased atmospheric instability across the coastline.

In line with the shifted large-scale atmospheric circulation anomalies, we observe relatively warmer and drier air over northern Africa and southwest Europe in the first summer after stronger freshwater anomalies, and relatively warmer and drier air over northwest Europe in the second summer (Fig. 5g-l). In the first summer, the maximum warm anomalies extend from Morocco and Algeria northward to France and southern Germany, while the maximum dry anomalies occur further to the east covering large parts of southwest Europe, including Italy and Greece. In the second summer after the freshwater

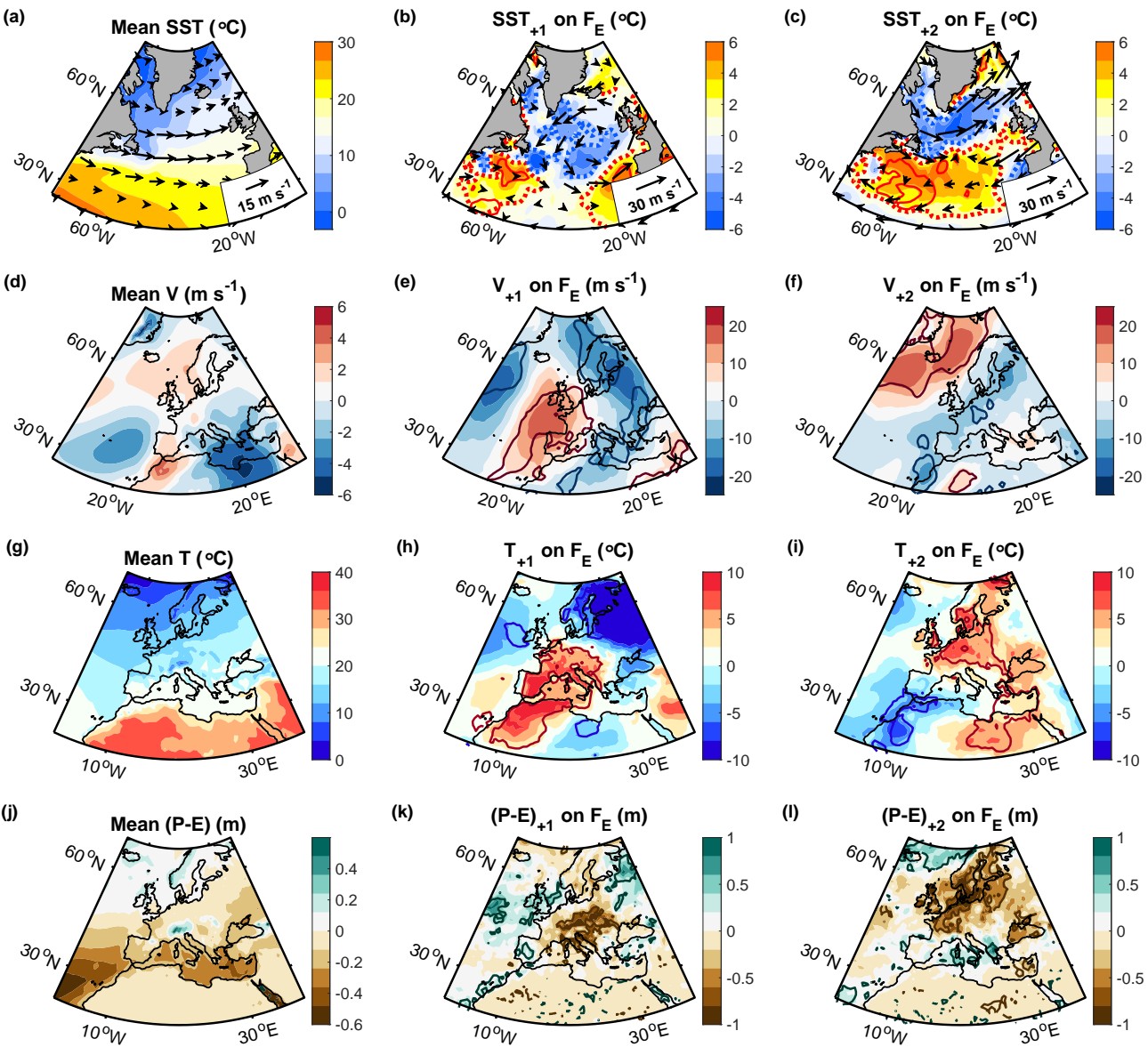

**Figure 5.** Climatological mean (a) SST, (d) meridional winds at 700 hPa, (g) 2-m air temperature and (j) precipitation minus evaporation in summer (May through to August). Regressions of (b,c) the SST (colour shading) and 700 hPa winds (arrows), (e,f) the meridional winds at 700 hPa, (h,i) the 2-m air temperature and (k,l) the accumulated precipitation minus evaporation on $F_E$ in (b,e,g,k) the first and (c,f,h,l) the second summer (May through August) after the freshwater anomalies (indicated by the '+1' and '+2' in the titles). We removed large-scale trends from the air temperature to reduce the direct warming effect of greenhouse gases (Section 2), and we excluded the anomaly in 2016 since it was associated with a larger cold anomaly that extended further southeast than the other anomalies (Fig. C1). Thick contours encompass regions that are significant at the 95% confidence level and the red and blue dotted lines in panels a and b delineate the regions in which the the SST anomalies exceed 2 °C and fall below −2 °C.

anomalies, the maximum warm anomalies occur over central to northern Europe, including Germany, France, the UK, Poland and southern Sweden, while the maximum dry anomalies again extend further eastward, including Finland and the Baltic countries. Considering that precipitation anomalies preferentially occur along trailing cold fronts and are shifted southward relative to cyclone centres (Booth et al., 2018; Kodama et al., 2019), the observed displacement of the dry anomalies relative to the warm anomalies is expected from their locations within individual weather systems and consistent with other studies (Yu et al., 2023).

Placing the identified atmospheric anomalies into a larger context described in the literature, we find that it is representative of blocking anticyclones (Brunner et al., 2018; Kautz et al., 2022). In summer, blocking anticyclones over Europe are typically associated with increased surface pressure and higher surface air temperatures (Brunner et al., 2018; Kautz et al., 2022). Considering that the maximum temperature anomalies in summers after enhanced $F_E$ freshwater anomalies occur east of the northward wind deflection, in the centre of the anticyclonic circulation anomaly, the location of the increased air temperature anomalies is consistent with earlier studies which have attributed the warm anomalies to enhanced shortwave radiation (Kautz et al., 2022; Pfahl, 2014; Sousa et al., 2018). Moreover, the occurrence of the dry anomalies to the east of the warm surface air temperature anomalies likely results from a reduced passage of cyclonic weather systems, which are blocked by the large-scale anticyclonic circulation anomalies (Sousa et al., 2017).

A downside of the statistical approach arises from the sensitivity of European summer weather to the exact location of the SST front between the subtropical and subpolar gyres. Small deviations in the spatial characteristics of the SST pattern and lower tropospheric circulation between two years can lead to shifts in the location of the maximum warm and dry anomalies over Europe, partially cancelling each other. Thus, we found that the spatial patterns in Summer 2016 did not match those of the other years included in the $F_E$ subset (Fig. C1). The cold SST anomaly extended further south of the North Atlantic Current, resulting in enhanced mixing and a patchy meridional SST gradient just west of the European coast with two cold anomalies of reduced amplitudes. Consistent with the underlying SST field, we identified a split zonal wind between ∼0 °E to ∼10 °E, with one branch extending northward along the European coastline, and another one crossing the southern Mediterranean Sea. Accordingly, one warm surface air temperature anomaly covered northern Africa and another warm anomaly occurred along the northwest European coastline (Fig. C1). So, even though the spatial SST pattern in Summer 2016 did not match those in the other summers, we still identify the same close relationship between the SST, tropospheric winds, and European weather anomaly.

Similar to the $F_E$ freshwater anomalies, freshwater anomalies associated with the $F_W$ subset are also followed by a cold SST anomaly in the subsequent summer. However, compared to $F_E$ freshwater anomalies, the cold SST anomalies associated with the $F_W$ index are smaller and confined to the central and western North Atlantic off the coast of Newfoundland, with the regressions peaking in July and August (Fig. 6a). Consequently, we observe a sharp northward deflection of the winds just eastward of the cold anomaly, and further west compared to the $F_E$ subset (Fig. 6b). Likewise, the warm air temperature anomalies over Europe also occur further west and are centred over France, Great Britain, Belgium and northern Spain, extending westward of the coast, while the dry anomalies extend eastward to the Baltic Sea region and northern Poland (Fig. 6c and d).

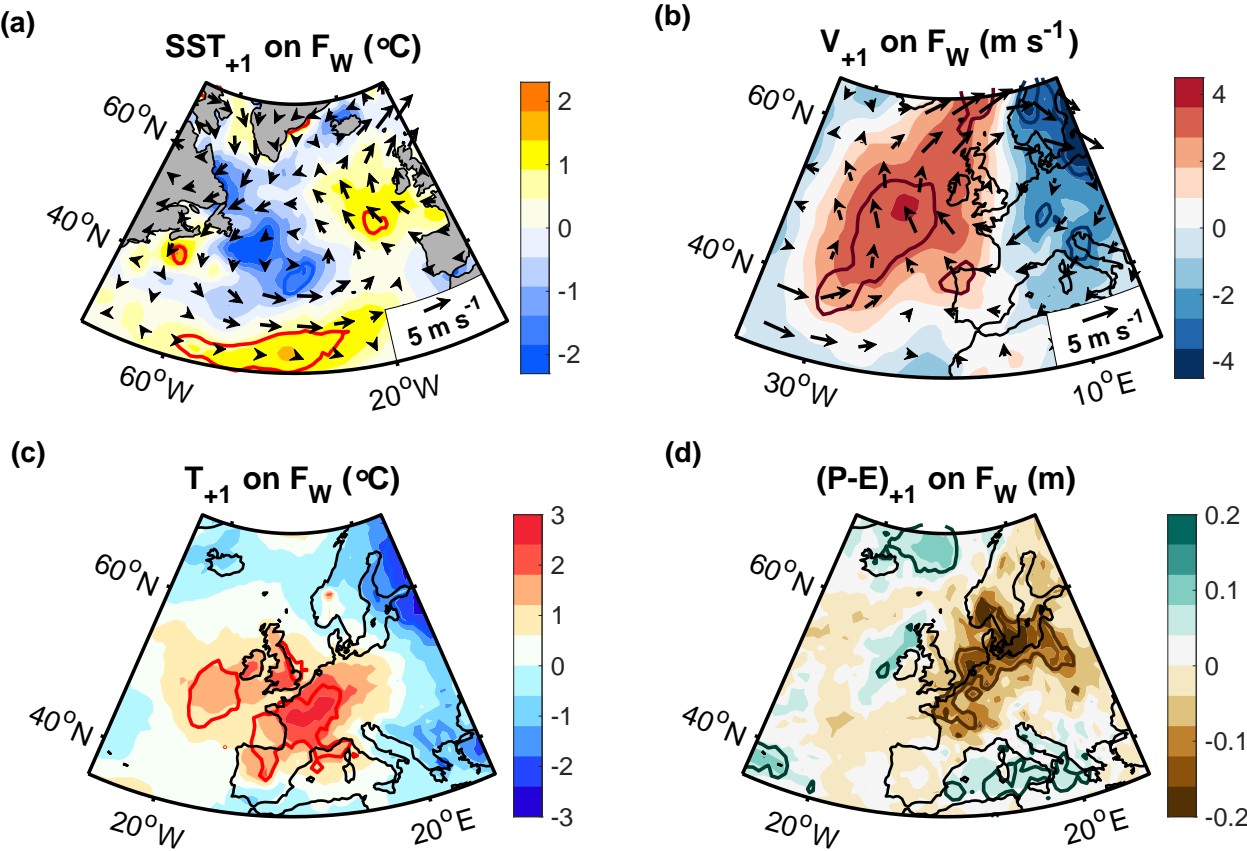

**Figure 6.** Regressions of (a) the SST (colour shading) and the 700-hPa winds (arrows), (b) the meridional winds at 700 hPa, (c) the 2-m air temperature and (d) the precipitation minus evaporation in summer (July and August) on $F_W$ from the preceding summer, again after subtracting large-scale trends from the air temperature. The thick contours encompass regions that are significant at the 95% confidence level.

Overall, we find that the regressions of the SST and atmospheric circulation on $F_W$ are weaker compared to those on $F_E$, consistent with weaker freshwater anomalies (Fig. 2) and smaller regression slopes (Fig. 1l), implying weaker sensitivities to the freshwater index and associated atmospheric circulation in the preceding summer. Yet, despite differences in the location and magnitude of the anomalies, the overall patterns are qualitatively similar after $F_E$ and $F_W$ freshwater anomalies: Both types of freshwater anomalies are characterised by a cold SST anomaly and northward deflection of the lower tropospheric winds over the North Atlantic in the subsequent summer. In both cases, the obtained, large-scale atmospheric circulation anomaly is associated with warmer and drier weather over parts of Europe. Moreover, considering that — from all the years included in each subset (17 and 8 respectively) — only the Summer 2016 exhibited a spatially diverging SST pattern, the results suggest that (1) the statistical method is overall successful in selecting years with similar spatial structures, and (2) the spatial coherency for which we selected in winter is, in most cases, maintained through to the subsequent summer.

## 4.5 Significance and robustness

The significance of the relationships between the freshwater indices and the ocean and atmospheric conditions in the subsequent winter and summer was assessed by Student t-tests. Importantly, the subsampling was based on the SST and freshwater anomalies only. Thus, it does not affect the relationship between the subsampled index and any variable that is statistically independent of freshwater (or the SST). If a random variable has no actual connection to freshwater anomalies, the probability for randomly obtaining a significant statistical link by chance remains the same.

Based on the Student t-tests, we obtained statistically significant links above the 95% confidence level, indicating that the probability for randomly obtaining the identified connections between North Atlantic freshwater anomalies in winter and the subsequent European summer weather is less than 5%. To ensure that the results are robust, we repeated the regressions by changing the number of years included in the subsampling, and by excluding anomalies in consecutive years (Appendix B). In all cases, we find that the identified links are robust, which is consistent with the scatter diagram (Fig. 1l), showing that there are no outliers or clusters of values responsible for the high correlations.

A downside of the $F_E$ and $F_W$ indices arises from the limited set of years, raising the question if the relationship between the SST, SSS and subsequent atmospheric anomalies holds generally or only over the selected subsets. To address this question, we use an un-subsampled SST-based index covering all years. As before, we avoid potential influences of a spatially uniform warming trend by using the spatial SST differences between the subpolar and the subtropical gyre ('ΔSST'), rather than absolute cold anomalies. Specifically, we use the observed SST difference between regions enclosed by the 95% lines in Figure 1j in any given year. We selected these regions since they cover such a large area of the subpolar and northern subtropical region and clearly define both regions. However, the results are not sensitive to this choice. The resulting time series is shown in Figure 7a.

Evaluating the surface mass balance associated with the new ΔSST index, we again find that none of the terms on the righthand side of Eq. (5) can account for the mass increase, implied by the associated cold, subpolar SST anomaly (Appendix A). Thus, we conclude that the cold anomaly can only be explained by the simultaneous existence of a freshwater anomaly, allowing us to infer the variability and spatial distribution of surface freshwater with an overall uncertainty of ∼6% which results from assuming density compensation. The correlation of the estimated freshwater variability with the ΔSST index extends over the full subpolar region, with maximum amplitudes of up to ∼0.8 occurring in the eastern subpolar gyre (Fig. 7b). This correlation is slightly smaller than those obtained for the other two freshwater indices but the index now covers all 44 years.

Considering the significant link between the ΔSST index and surface freshwater in the supolar region, we use it as a new freshwater index and examine the ocean and atmospheric conditions in the subsequent summer. Inspection of the SST shows that a stronger ΔSST index in winter is associated with pronounced cold SST anomaly over the central subpolar region in the subsequent summer (Fig. 7c). The atmospheric circulation is aligned with the underlying SST field, with the winds at 700 hPa circulating cyclonically around the cold, subpolar SST anomaly (Fig. 7c). To the east of the cold SST anomaly, the winds are deflected northward along the European coastline (Fig. 7d). Again the northward deflected winds form part of a large-scale

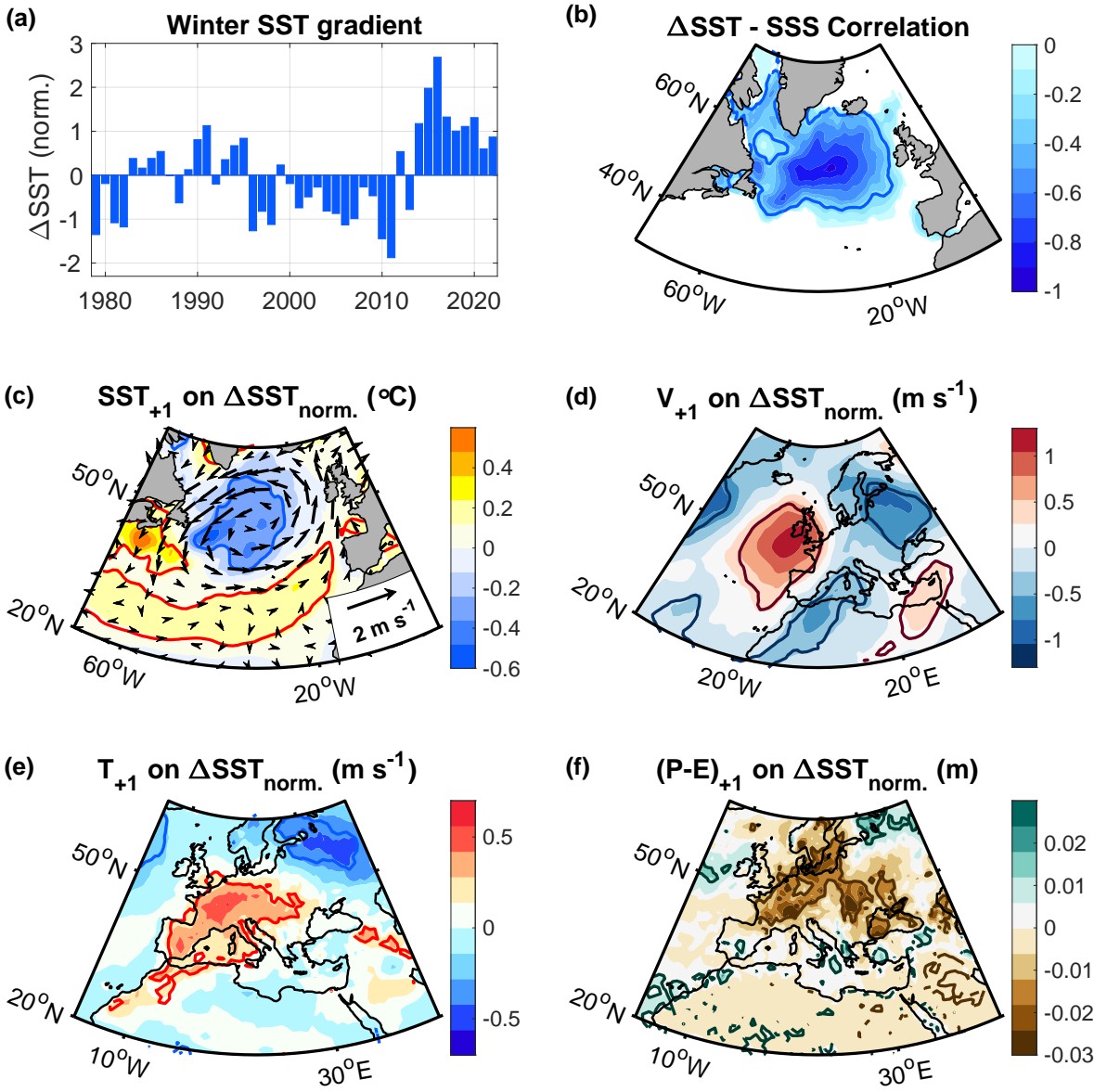

**Figure 7.** (a) ΔSST index corresponding to the SST difference between the subtropical warm anomaly and the subpolar cold anomaly, enclosed within the 95% confidence lines in Figure 1j. The time series of the SST difference has been normalised by its standard deviation. (b) Correlation between the ΔSST index, shown in panel a, and the sea surface salinity anomaly in the same winter (January through March), estimated from the surface mass balance. (c-f) Regressions of (c) the SST (colour shading) and 700 hPa winds (arrows), (d) the meridional winds at 700 hPa, (e) the 2-m air temperature and (f) the precipitation minus evaporation in summer (July and August) onto the ΔSST index from the preceding winter (panel a), again after subtracting the large-scale trends from the air temperature. The thick contours encompass regions that are significant at the 95% confidence level.

atmospheric circulation anomaly that is associated with warmer and drier weather over Europe. The associated warm anomalies extend over Spain, Italy France, the Netherland and parts of Germany eastward to Austria, Hungary and Slovakia, while the dry anomalies occur further northeastward, covering France, the Netherlands, Denmark, and parts of northern Germany, Poland and Ukraine (Fig. 7e and f).

Unlike the summer NAO, the new, SST-based index has higher autocorrelations (Fig. 8a), which we attribute to enhanced low-frequency variability of the North Atlantic climate in winter. We still assume that interannual variability substantially contributes to the correlations, due to the high, interannual variability of European summer weather, reflected in low autocorrelations (Fig. 8b). Nonetheless, to assess the contribution of low-frequency variability to the obtained links, we lowpass filter European summer weather with a hanning filter, using a window size of 3 summers to approximate the higher autocorrelations of the $\Delta$SST index (Appendix B). After accounting for the reduced number of independent samples in the significance tests with $N^* = \frac{N\Delta t}{2T_e} - 2$, where $N$ here refers so the number of data points, $\Delta t$ is the time interval between them, and $T_e$ is the e-folding timescale of the autocorrelations (Leith, 1973), we still obtain statistically significant relationships but the amplitudes of the regressions are reduced (Fig. B8), indicating that high-frequency, interannual variability substantially contributed to the relationship obtained from the unfiltered time series (Fig. 7).

To further assess the timescales on which the identified relationship holds and is significant, we carry out a multi-taper coherence analysis. Specifically, we calculate the coherence between the $\Delta$SST index and the temperature and precipitation minus evaporation anomalies in the regions in which we identified a significant link from the regressions (Fig. 7e and f). Inspection of the coherence estimate shows that both, temperature and precipitation minus evaporation over Europe, are significantly linked to freshwater variations in the subpolar region on timescales from a few years to decades (Fig. 8c and d). The coherence between the $\Delta$SST index and the precipitation minus evaporation anomaly is particularly high and well above the 95% significance line (Fig. 8d). The associated phase shifts are relatively constant at 0° for the air temperature (indicating a positive correlation) and 180° for precipitation minus evaporation (implying anti-correlation). We used 8 tapers, which is a standard value. However, the results are not sensitive to this choice.

We conclude that the link between cold, fresh ocean anomalies in the subpolar North Atlantic region in winter and warm, dry atmospheric anomalies over Europe in the subsequent summer is robust, significant at both higher and lower frequencies, and independent of the spatial and temporal characteristics of the freshwater index that is used.

### 4.6 Predictability of European summer weather

The preceding analyses revealed significant links between North Atlantic freshwater anomalies and European summer weather in subsequent years. This raises the question to what extent this link can be used to predict European summer weather in advance. Thus, we next assess the predictability based on the explained variance in the observations, estimated by means of the squared correlation coefficient with the freshwater indices.

The variance of the near surface temperature and precipitation minus evaporation anomalies, explained by the $F_E$ subset, reaches and even exceeds 80% over large parts of Europe (Fig. 9a-d). For the $F_W$ subset, the explained variance drops to ~50% (Fig. 9e and f), and for the $\Delta$SST index, the explained variance drops further to ~20% (Fig. 9g and h), as expected from the

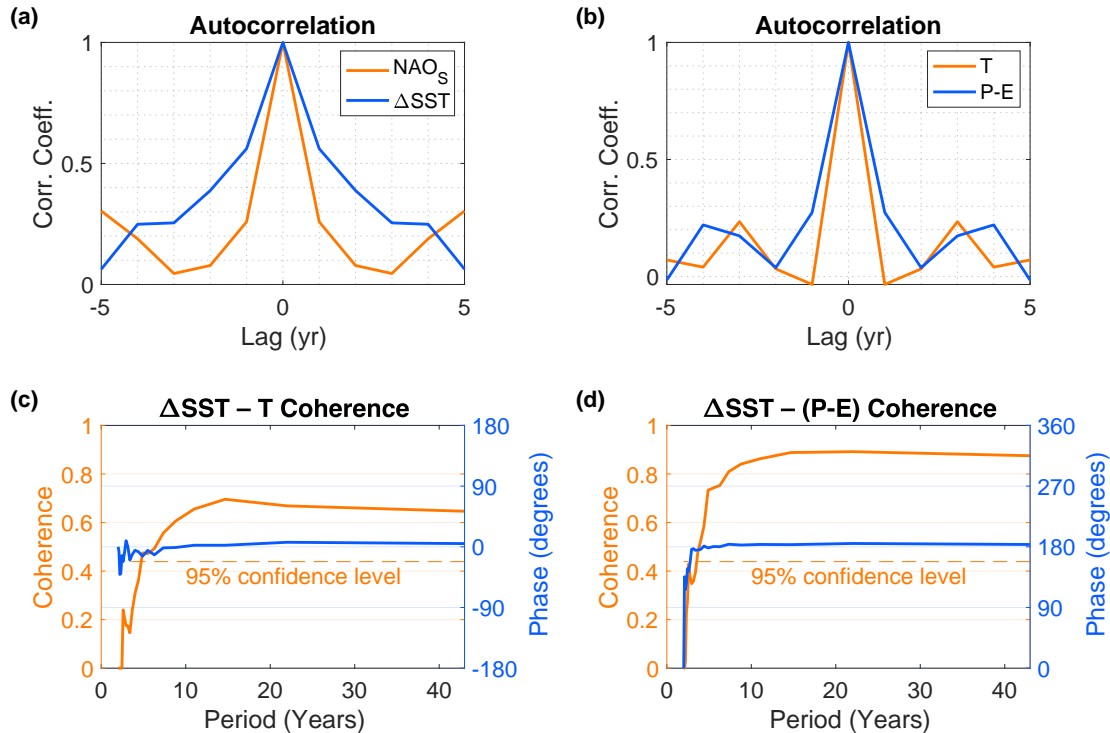

**Figure 8.** (a,b) Autocorrelations of (a) the NAO index in summer (July and August) and the $\Delta$SST index in winter (January to March), (b) the 2-m air temperature and precipitation minus evaporation anomalies in summer (July and August), averaged over the regions enclosed by the 95% confidence lines in Figure 7e and f. (c,d) Multi-taper coherence and phase shift estimates for the $\Delta$SST index in winter (January to March) and (c) the 2-m air temperature and (d) precipitation minus evaporation in the subsequent summer (July and August), again within the regions enclosed by the 95% confidence lines in Figure 7e and f. We used 8 tapers. The 95% confidence estimates are based on Amos and Koopmans (1963) after correcting for the bias inherent in coherence estimates (Priestley, 1982).

trade-off between the number of years included in the index and the associated correlations with freshwater anomalies in the subpolar North Atlantic region in winter and European weather anomalies in the subsequent summer.

Overall, we find: The higher the correlation is between the initial freshwater anomaly and its index, the higher is also the variance of European summer weather that the index subsequently explains. The $F_E$ index, in particular, has an extremely high
correlation with the initial freshwater anomaly of over $\sim$0.9 (Fig. 2c) and explains over 80% of the variance of European summer weather. Considering that all indices represent fresh and cold SST anomalies in the subpolar region, and notwithstanding the small sample sizes or the reduced correlations, these results indicate that accurate estimates of the sea surface salinity in the subpolar region can serve as valuable constraints for the subsequent European summer weather. Specifically, the amount of the variance in European summer weather, explained by the freshwater anomaly, depends on the location, amplitude and type
of the freshwater anomaly in the subpolar region in the preceding winter.

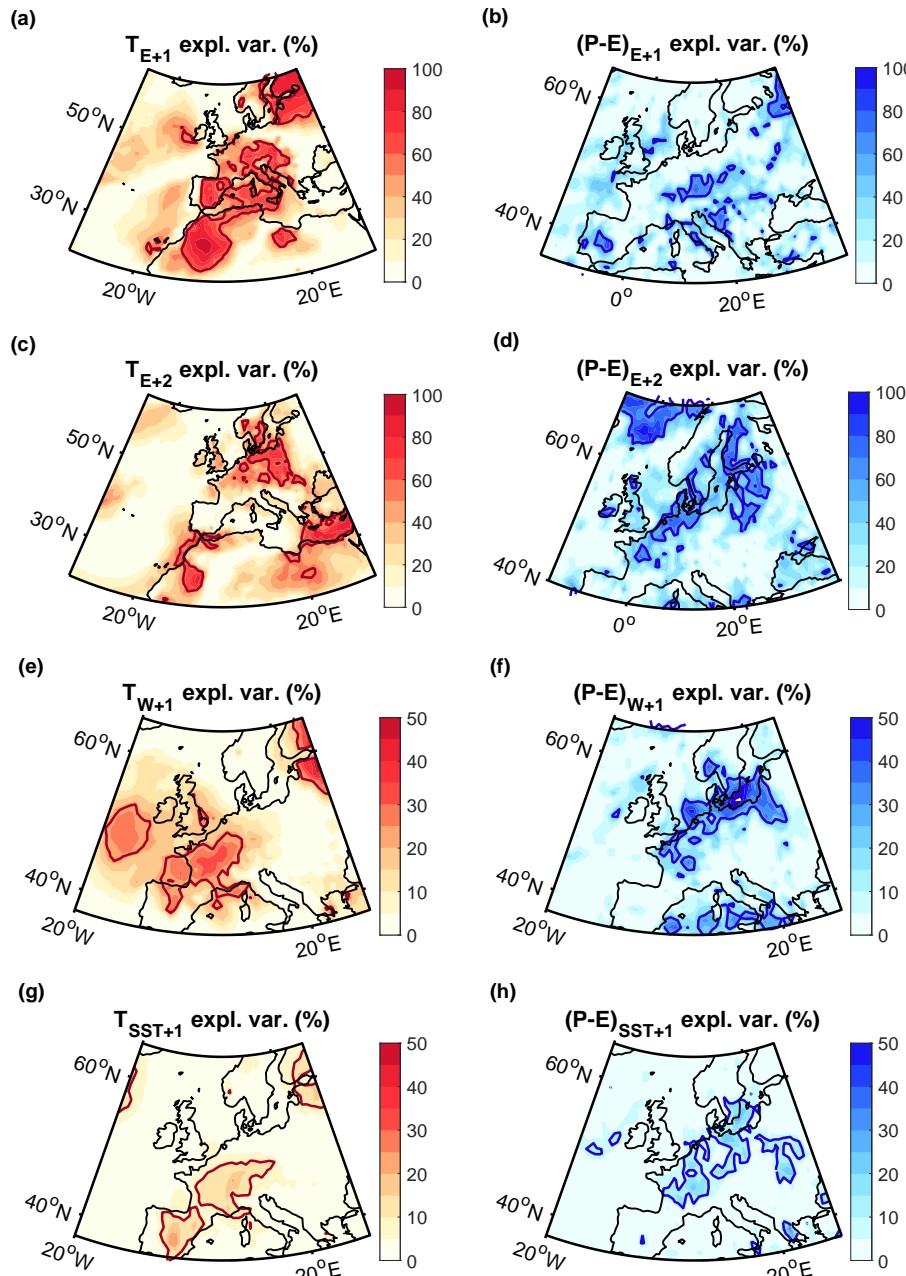

**Figure 9.** Variances explained by (a-d) $F_E$, (e,f) $F_W$ and (g,h) the $\Delta$SST index of (a,c,e,g) the 2-m air temperature and (b,d,f,h) precipitation minus evaporation after freshwater anomalies. '+1' and '+2' in the titles refer to the first and second summer after the freshwater anomaly. We again excluded the 2016 freshwater anomaly from $F_E$ since its responses were covered by the 2017 anomaly. Thick contours delineate the regions in which the correlation is significant at the 95% confidence level, assessed by means of two-sided Student t-tests. The explained variances were obtained from the squared correlation coefficients. Please note the different colour scales.

## 4.7 Warm summers in Europe

The preceding analyses showed that two types of freshwater anomalies with opposite atmospheric drivers (characterised by a high and a low NAO states in the preceding summer) are associated with cold SST anomalies over the North Atlantic in winter. The resulting increased SST fronts are maintained through the subsequent summer, when (aligned with the underlying SST fronts) the lower tropospheric winds are deflected northward east of the cold anomaly and along the European coastline. The winds form part of large-scale atmospheric circulation anomalies that lead to warmer, drier weather over Europe, with the location of the warm and dry anomalies being sensitive to the exact location, strength and type of freshwater anomaly in the preceding winter. In this last section, we investigate if the warmest European summers can in turn be linked back to a freshwater anomaly in the preceding year. Thus, we assess the extent to which enhanced freshwater anomalies are not only a sufficient but also a necessary condition for warmer European summers.

Based on composites, we find that the 10 warmest relative to the 10 coldest summers in western Europe were associated with a dry anomaly to the east of the warm air temperature anomaly, a northward deflection of the wind at 700 hPa west of Portugal, France and Britain, and a pronounced cold SST anomaly in the subpolar North Atlantic (Fig. 10a-e). Using a surface mass balance (Appendix A), we again identify a significant freshwater anomaly in the preceding winter, with the freshwater anomaly covering a large part of the subpolar North Atlantic (Fig. 10f). Selecting different regions for the temperature variability over Europe (Fig. 10a and b) shifts the location of the atmospheric circulation pattern and the location of the maximum North Atlantic SST gradient but does not qualitatively alter the results.

The similarity of the ocean and atmospheric conditions with those described in the preceding sections supports the relevance of freshwater anomalies in winter for the subsequent ocean-atmosphere evolution into the summer. In addition, the composites suggest that enhanced freshwater anomalies in the subpolar North Atlantic in winter could serve as early warning signs of Europe's warmest and coldest summers approximately half a year in advance.

## 5 Conclusions

In this study, we examined the link between North Atlantic freshwater anomalies and European weather in subsequent summers. Given the limitations of currently available salinity observations, we estimated the variability of freshwater based on a surface mass balance analysis. We further investigated the statistical links between the obtained freshwater anomalies in winter and the subsequent European summer weather by applying regression and correlation analyses, composite analyses and multi-tapered coherence analyses.

Combined, the analyses reveal a significant relationship between freshwater anomalies in winter and European weather in subsequent summers. Specifically, we found that an enhanced freshwater anomalies are associated with colder, subpolar SST anomalies and an increased SST difference between the warm subtropical and the cold subpolar gyre in winter. The increased, meridional SST gradient is linked to an amplified atmospheric instability and a large-scale atmospheric circulation anomaly with a more cyclonic circulation over the subpolar region and an anticyclonic anomaly to the south. This atmospheric circulation anomaly induces a northward shift in the North Atlantic Current which contributes to a warm anomaly to the south

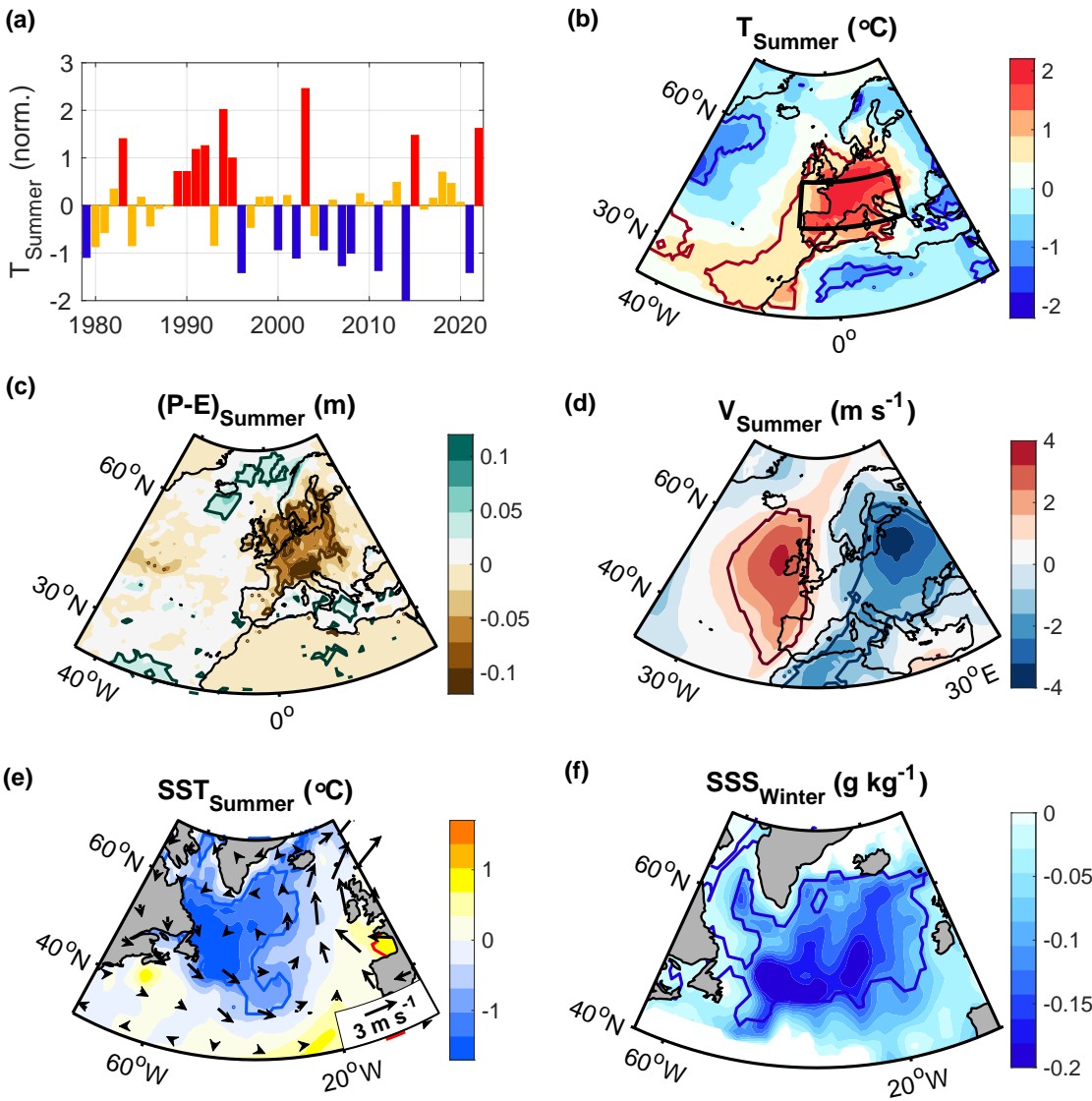

**Figure 10.** (a) Variability of the de-trended 2-m air temperature anomaly over land within the box shown in (b) during summer (July and August). (b,c,d,e) Composites of (b) the 2-m air temperature, (c) precipitation minus evaporation, (d) the meridional winds at 700 hPa, and (e) the SST (colour shading) and the 700 hPa winds (arrows) for the 10 warmest minus the 10 coldest summers, shown in (a). (f) Same as in (b-e) but for the sea surface salinity anomaly in the preceding winter, obtained from a surface mass balance (Appendix A). Contours delineate the regions that are significant at the 95% confidence level, assessed by means of two-sample t-tests.

of the subpolar cold anomaly, amplifying the meridional SST gradient. In subsequent summers, the lower tropospheric winds are deflected northward over the North Atlantic in the wake of the cold SST anomaly, aligned with the underlying SST fronts. This northward deflection of the winds forms part of a large-scale atmospheric circulation anomaly, consisting of a more cyclonic circulation over the subpolar North Atlantic region and a more anticyclonic circulation over parts of Europe, giving rise to warmer and drier weather over Europe.

The atmospheric circulation in summer is characteristic of blocking anticyclones over Europe described in earlier studies (Brunner et al., 2018; Kautz et al., 2022). Thus, the warm surface anomalies can be explained by increased shortwave radiation in the centre of the anticyclones (Kautz et al., 2022; Pfahl, 2014; Sousa et al., 2018) while the dry anomalies to the east can be explained by the blocking of cyclonic weather systems (Sousa et al., 2017). Further studies are required to quantify the relative contributions of ocean and atmospheric drivers to the large-scale atmospheric circulation anomaly in summer, their uncertainties, and the role of freshwater as potential trigger of the identified chain of events. However, the obtained evolution of freshwater anomalies follows the chain of events expected from theory. In addition, the statistical links identified in this study suggest that estimation of the extent, amplitude and type of freshwater anomaly in any given winter can constrain the subsequent European summer weather, based on the evolution of past freshwater anomalies and the associated explained variances.

Current numerical weather prediction systems show very limited to no forecast skill for European summer weather (Arribas et al., 2011; Dunstone et al., 2018). Thus, the existence of a link between North Atlantic freshwater anomalies and European summer weather indicates new potential to enhance the predictability of European summer weather a year in advance. Further studies that improve the representation of North Atlantic freshwater variations in models, and that quantify the predictability arising from them, are therefore desirable. In addition, targeted observational networks that monitor the variability of freshwater anomalies may help improve current forecast systems.

Linking European summer weather with North Atlantic freshwater anomalies, as opposed to linking it to SST anomalies only, has the advantage that the occurrence of freshwater anomalies is easier to predict into the future than SST anomalies, due to having more narrowly defined drivers. Thus, in this study, we attributed the freshwater anomalies to only two main drivers on interannual timescales. One type of freshwater anomaly was linked to change in the subpolar gyre circulation. The other type of freshwater anomaly was linked to enhanced runoff and melting. Runoff and melting, specifically, largely occur in summer, giving rise to a longer predictive timescale, half a year in advance of the cold and fresh anomalies in winter, and one year in advance of the subsequent European summer weather.

Moreover, over the coming decades, the melting of land and sea ice are expected to increase (Notz and Stroeve, 2018; Briner et al., 2020), resulting in an enhanced freshwater discharge into the North Atlantic. With stronger freshwater anomalies, our results indicate an increase in the risk of warm, dry European summers and of heat waves and droughts accordingly. Unfortunately, global climate models have difficulties in capturing the hydrographic structure and freshwater distribution in the subpolar North Atlantic (Menary et al., 2015; Heuzé, 2017; Liu et al., 2017; Sgubin et al., 2017; Mecking et al., 2017; Wu et al., 2018). Considering the identified links between freshwater anomalies and subsequent ocean-atmosphere evolution, our results suggest that models may miss a key source of climate variability and potential long-range predictability.

*Code and data availability.* This study is only based on publicly available data and standard analysis techniques. The SST and NAO data are available from NOAA (https://psl.noaa.gov/data/gridded/data.noaa.oisst.v2.html and https://www.cpc.ncep.noaa.gov/products/precip/CWlink/pna/nao.shtml). The Hadley SST data is available from https://www.metoffice.gov.uk/hadobs/hadisst/ and a complete merged NOAA and Hadley SST product can be obtained from https://gdex.ucar.edu/dataset/158_asphilli.html. Absolute dynamic topography data is distributed by the Copernicus Marine Environment Monitoring Service (https://marine.copernicus.eu/). ERA5 data can be obtained from the European Centre for Medium-Range Weather Forecasts (https://www.ecmwf.int/en/forecasts/datasets/reanalysis-datasets/era5) and the ECHAM5 and CAM5 model output can be downloaded from the Facility of Climate Assessments repository (https://psl.noaa.gov/repository/facts). Matlab codes can be obtained from the corresponding author.

## Appendix A:  Mass balance analyses

The following sections include the evaluation of the mass balances obtained from the freshwater indices $F_E$, $F_W$ and $\Delta$SST (A1), the surface mass balance obtained from the SST composite (A2), and a demonstration of the mass balance with hydrographic observations (A3).

### A1   Surface mass balance for freshwater indices

Taking advantage of the strong relationships between the $\text{NAO}_S$ subsets ($F_E$ and $F_W$) and the subsequent SST anomalies, we regress each term of Eq. (5) onto the indices and evaluate the surface mass balance over the subpolar cold anomaly regions within the 95% confidence lines (Figs. A1a and A2a).

Considering that mean mixed layer deepens from summer to winter, reaching its maximum in late winter, the integrated anomalies in the surface heat and buoyancy fluxes during autumn are predominantly driven by existing anomalies in the density profile. For instance, an anomalously warm and light layer of water will lead to increased ocean heat and buoyancy losses once it has been entrained (Timlin et al., 2002). Thus, given that the anomalies in $B_n$ and $M_n$ are expected to largely compensate for each other when integrated over autumn (the period of rapid mixed layer deepening), we focussed on the winter period (January through to March), when the amplitude and variability of the surface fluxes is largest. However, if we integrate the terms on the righthand side of Eq. (5) over autumn and winter, instead of only winter, the magnitude of the integrated anomalies does not appreciably change and their signs remain the same.

First, we estimate the convergence of mass ($A$). On the timescales and spatial scales considered, the strongest horizontal velocities result from the geostrophic surface flow (including eddies and the subpolar gyre circulation). These surface flows do not contribute to a net mass increase as they occur along lines of constant density and pressure. The largest ageostrophic surface flow in the open ocean results from the wind forcing, which we evaluate using the wind stresses from the atmospheric reanalysis ERA5. Integrated over the winter period (January through to March), we find that neither the horizontal Ekman transports nor the vertical Ekman pumping can account for the density increase associated with the cold anomaly. They are not significantly correlated with the freshwater indices, their amplitudes are too small, and their directions are inconsistent with the cold anomaly (Figs. A1a, b and A2a, b).

Next, we estimate the surface buoyancy flux anomalies with:

$$B = \frac{g\alpha}{c_p}Q + g\beta S\left(P - E\right), \tag{A1}$$

where $c_p$ is the heat capacity, $Q$ is the heat flux (positive downward) and $P - E$ is the freshwater flux in kg m$^{-2}$ s$^{-1}$ (Gill, 2016). The thermal and haline expansion coefficients $\alpha$ and $\beta$ were estimated using Gibbs Seawater Routines (McDougall et al., 2009). Specifically, we used a nominal pressure of 10 db, a salinity of 34.5 g kg$^{-1}$ and the observed skin temperature from ERA5. Likewise, the direct salinity was in Eq. (A1) was also estimated with 34.5 g kg$^{-1}$, corresponding to a typical salinity in the subpolar region. However, the results are not sensitive to the exact values.

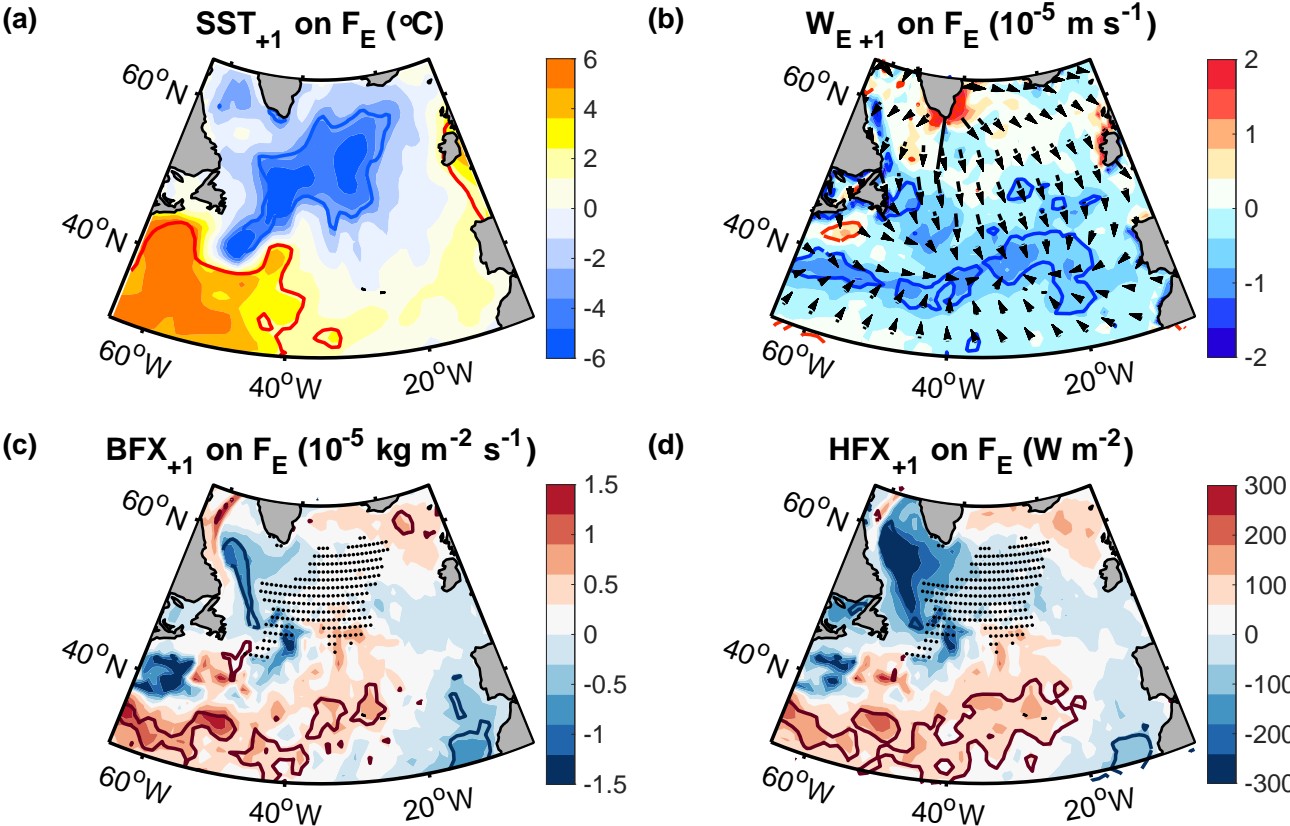

**Figure A1.** Regression of (a) the SST, (b) the vertical Ekman velocity (positive upward), (c) the buoyancy flux anomaly (positive downward) and (d) the surface heat fluxes (also positive downward) in winter (January through March) on $F_E$ from the preceding summer. The arrows in (b) indicate the direction of the horizontal Ekman transports and the dots in (c) and (d) show the region used for the mass balance calculations, corresponding to the cold anomaly region. Contours encompass regions that are significant at the 95% confidence level.

After evaluating the buoyancy fluxes with 6-hourly ERA5 output and regressing them on the freshwater indices, we find that they do not match the distribution of the SST (Figs. A1c and A2c). The surface heat fluxes, which have the largest contribution

to the buoyancy fluxes, are also not significantly correlated with the indices (Figs. A1d and A2d). When averaged over the cold anomaly regions, enclosed by the 95% confidence lines, and integrated over the winter, the buoyancy flux anomaly associated with the $F_E$ subset reflects an anomalous mass decrease of $\sim$7 kg m$^{-2}$ whereas the cold anomaly implies a mass increase of $\sim$204 kg m$^{-2}$. Likewise, the buoyancy flux anomaly associated with the $F_W$ subset reflects a mass decrease of $\sim$4 kg m$^{-2}$, whereas the cold anomaly implies a mass increase of $\sim$69 kg m$^{-2}$. For both subsets, we used a mean density of $\rho_{mean} \approx 1000$

700 kg m$^{-3}$ to estimate the mass anomaly associated with the cold anomaly.

Since none of the potential, active drivers of density anomalies on the righthand side of Eq. (5) can account for the density increase associated with the cold anomalies, we conclude that the density increase associated with the cold anomalies must be

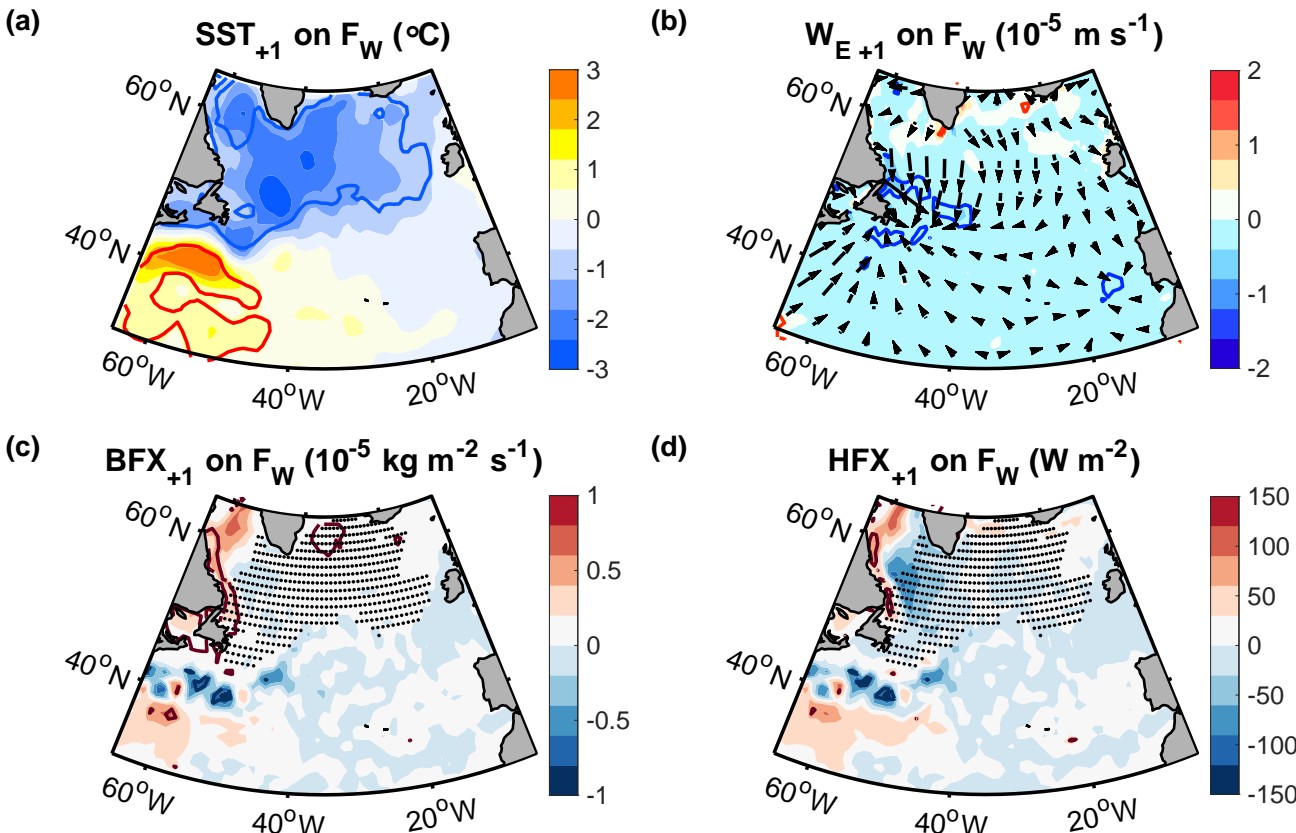

**Figure A2.** Regression of (a) the SST, (b) the vertical Ekman velocity (positive upward), (c) the buoyancy flux anomaly (positive downward) and (d) the surface heat fluxes (also positive downward) in winter (January through March) on $F_W$ from the preceding summer. The arrows in (b) indicate the direction of the horizontal Ekman transports and the dots in (c) and (d) show the region used for the mass balance calculations. Contours encompass regions that are significant at the 95% confidence level.

balanced by a density decrease associated with freshwater anomalies. The buoyancy fluxes represent the largest term on the righthand side of Eq. (5), and thus determine the uncertainty of the obtained salinity estimates, amounting to ~4% for the $F_E$

subset and ~6% for the $F_W$ subset.

To verify the robustness of the results, we tested different integration periods and regions for the mass balance calculations. For instance, we also integrated the transports and surface fluxes from September to March instead of January to March, and we extended the investigated region over the full cold anomaly region, over which the SST anomaly is negative. In each case, the results did not change appreciably.

In addition, we repeated the analyses for the un-subsampled $\Delta$SST index (Fig. A3). In this case, we obtain a mean mixed layer depth of ~250 m, a negative mass anomaly of $\sim -1$ kg m$^{-2}$ resulting from the surface buoyancy fluxes and a positive mass anomaly of $\sim +18$ kg m$^{-2}$ associated with the cold SST anomaly. Thus, estimating the sea surface salinity anomaly by

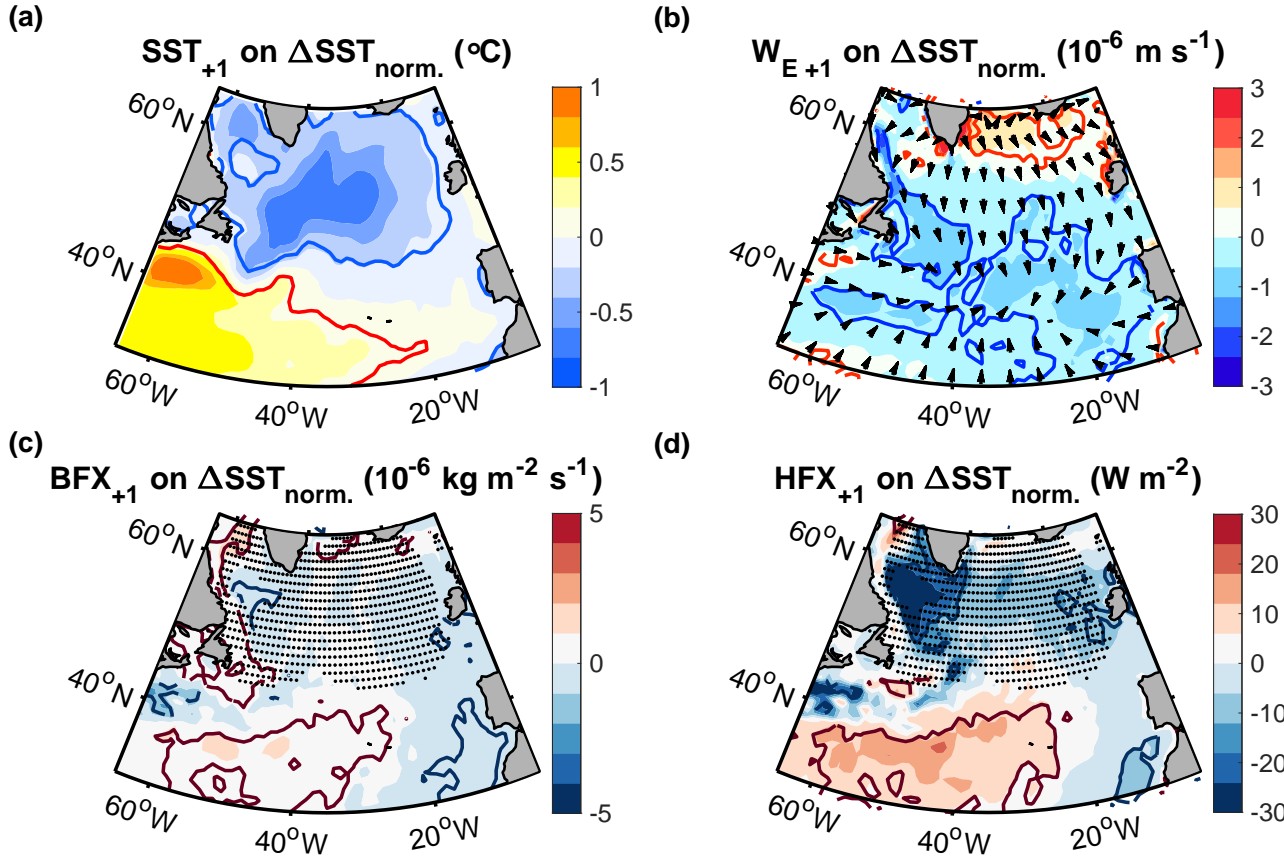

**Figure A3.** Regression of (a) the SST, (b) the vertical Ekman velocity (positive upward), (c) the buoyancy flux anomaly (positive downward) and (d) the surface heat fluxes (also positive downward) in winter (January through March) on the normalised $\Delta SST$ index (Fig. 7a). The arrows in (b) indicate the direction of the horizontal Ekman transports and the dots in (c) and (d) show the region used for the mass balance calculations. Contours encompass regions that are significant at the 95% confidence level.

assuming density compensation, we obtain a sea surface salinity of $\sim -0.10$ g kg$^{-1}$, averaged over the cold anomaly region enclosed by the 95% confidence lines with an overall uncertainty of 6% that results from neglecting the terms on the righthand side of Eq. (5).

## A2 Surface mass balance of the SST composite and trend

We further carried out a mass balance analysis for the composites of the cold anomaly in the winters preceding the 10 warmest relative to the 10 coldest summers over Europe (Fig. 10). Thus, we again evaluated the terms in Eq. (5) over the cold anomaly region and the winter, where now, the subscript $n$ refers to anomalies associated with the composites.

After evaluating each term in the mass balance equation, we obtain similar patterns compared to those associated with the two NAO subsets (Fig. A4). Again, we find that none of the density drivers on the righthand side of Eq. (5) show a significant signal over the cold anomaly region, and their amplitudes cannot account for the density increase implied by the cold anomaly. The mean mixed layer depth in the cold anomaly region is now $\sim$290 m, the surface buoyancy flux, which is the largest term on the righthand side of Eq. (5), amounts to $\sim +1.2$ g kg$^{-1}$ while the density anomaly implied by the cold anomaly is $\sim -44$

725    g kg$^{-1}$. Thus, the uncertainty of the estimated freshwater anomaly (Fig. 10e) amounts to $\sim$3%.

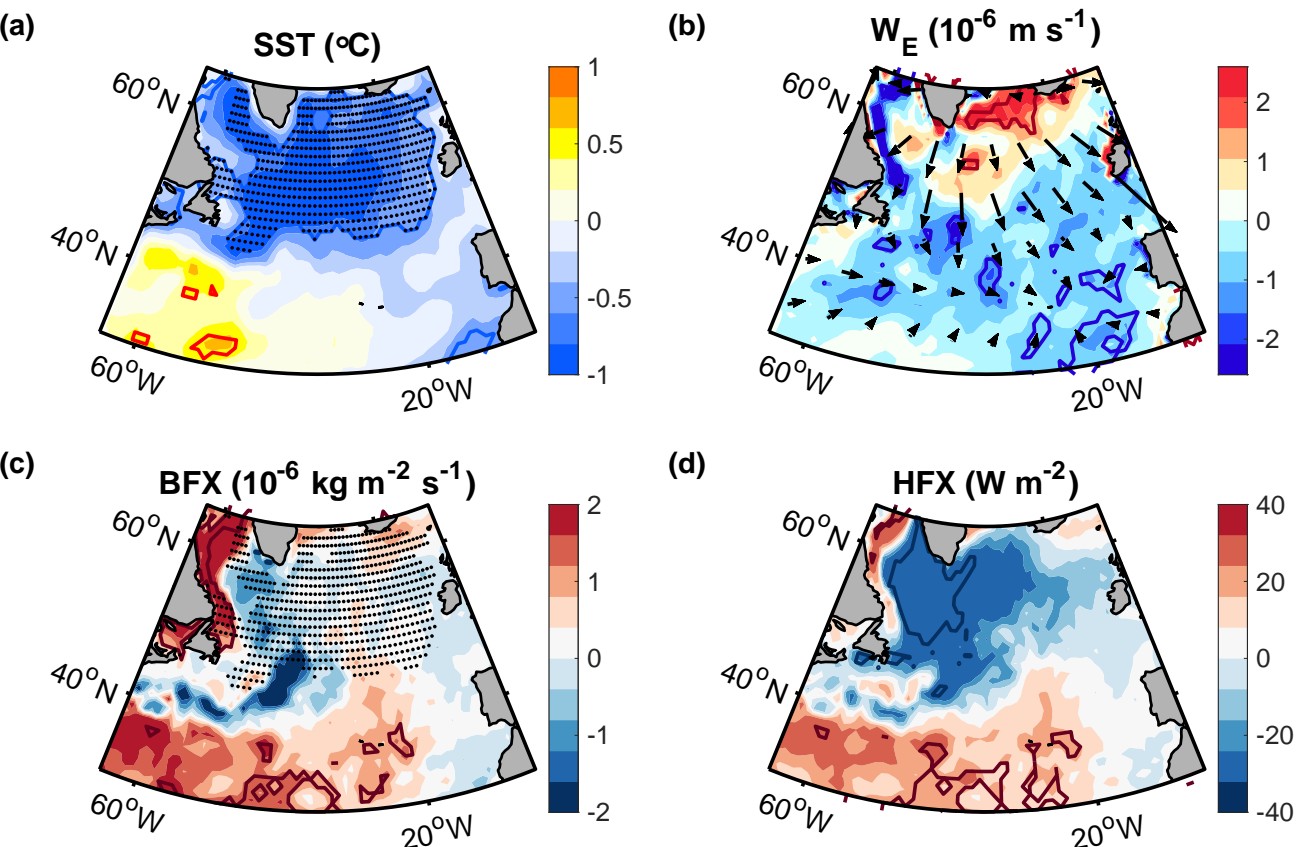

**Figure A4.** (a,b,c) Anomaly of (a) the SST, (b) the vertical Ekman velocity (positive upward), (c) the buoyancy flux anomaly (positive downward), and (d) the surface heat flux (also positive downward) in the 10 winters (January through March) before the warmest summers minus the 10 winters before the 10 coldest summers (Fig. 10). The arrows in (b) indicate the direction of the horizontal Ekman transports and the dots in (a) and (c) mark the region of the mass balance calculations.

## A3 Comparison with in-situ observations

To demonstrate the density compensation between temperature and salinity anomalies, we use mixed layer profiles from Argo floats in the subpolar region (Holte et al., 2017). We focus on the extreme winters 2015 and 2016, which were characterised by particularly large surface fluxes and deep convection (Yashayaev and Loder, 2017; Piron et al., 2017).

In both winters, the temperature and salinity anomalies are well-correlated with each other ($r \approx 0.72$, $p \approx 5 \cdot 10^{-242}$, based on 1532 profiles). Moreover, the observed salinity anomalies are well-aligned with the estimated salinity anomalies, obtained by assuming density compensation (Fig. A5). The root mean square error associated with the mass balance estimate amounts to $\sim 0.09$ g kg$^{-1}$, which is smaller than that of currently available salinity products (Bao et al., 2019; Xie et al., 2019).

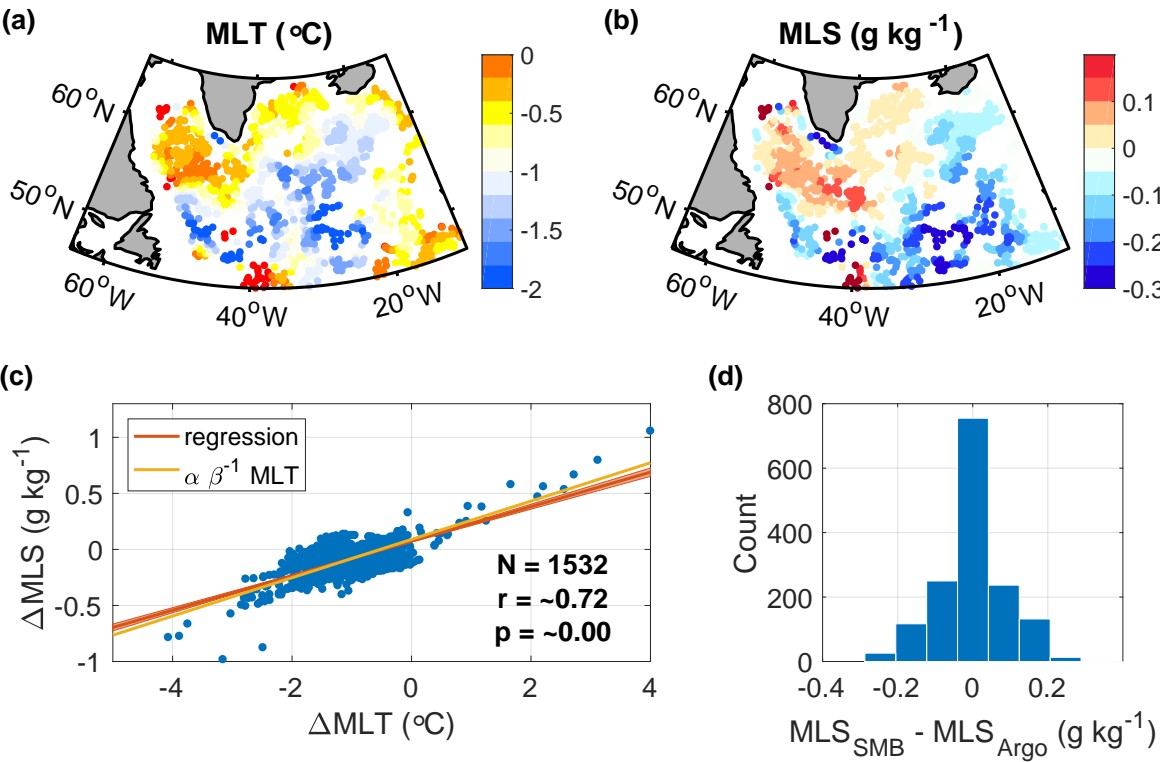

**Figure A5.** (a,b) Mixed layer temperature (MLT) and salinity (MLS) anomalies in the winters of 2015 and 2016 (January to April), derived from Argo profiles (Holte et al., 2017). The anomalies are relative to the climatological mean, estimated by averaging all other wintertime profiles within $2°$ longitude and $1°$ latitude. (c) Linear regression of the observed MLS anomalies on the MLT anomalies (red line), and the MLS estimate obtained by assuming density compensation (yellow line). (d) Differences between the estimated and observed MLS anomalies. The associated root mean square error is $\sim 0.09$ g kg$^{-1}$.

## Appendix B:  Robustness and significance tests

In this section, we conduct sensitivity tests to assess the robustness of the indices $F_E$, $F_W$, and $\Delta$SST. Considering the different weaknesses of each index, the analyses for each index are different. $F_E$ only includes a low number of years. Thus, we will test the sensitivity of the results to including and excluding individual years. For $F_W$, the sample size is larger, such that adding or removing individual years has almost no effect on the results. Thus, we will test the sensitivity of the results to the number of years included in increments of 5 years. Finally, the $\Delta$SST index is characterised by high autocorrelations. Thus, we will test if the results remain significant if we lowpass filter European summer weather and assume a lower number of degrees of freedom.

We start with the $F_E$ subset. If we include the outlier in 2019 (which was removed as part of the subsampling in Section 3), we still obtain similar ocean and atmospheric conditions in the subsequent summers, compared to those with the outlier excluded (Figure B1 and Figure 5). Specifically, we still identify an increased SST difference between the subpolar cold SST anomaly and the subtropical warm SST anomaly, with the location of the SST front being shifted northward in the second summer relative to the first summer (Fig. B1a and b). Moreover, we still identify northward deflections of the winds and warm and dry anomalies over Europe (Fig. B1c to h), with the locations closely resembling the regressions with the $NAO_S$ year in 2019 excluded (Fig. 5c to h). Likewise, if we only include the second anomaly in all consecutive anomalies, the results also remain similar and significant (Fig. B2).

Next we examine the sensitivity of the results to the number of years included in the $F_W$ subset. Since we find that the results do not change appreciably when we include or exclude single years, we show how the results are changing when we add or remove years in increments of 5 years. Thus, we show the regressions for $N = 7$ (Fig. B3), $N = 12$ (Fig. B4), $N = 17$ (Fig. 6), $N = 22$ (Fig. B5), $N = 27$ (Fig. B6) and $N = 32$ (Fig. B7), with $N$ corresponding to the number of years included. The choice of years follows the same method as before (Section 3), with the objective of maximising the regression slope and the variance (resulting in high correlations). Thus, we rank all years according to the term $(y_i - y) \cdot (x_i - x_0)$ (Section 3.2) and then select the $N$ highest terms.

The regression for the case, where no further subsampling is applied after excluding the $F_E$ subset and the two outliers (corresponding to $N = 33$, not shown), do not change appreciably compared to the regression where $N = 32$. Likewise, if we exclude the only remaining consecutive year from the case where $N = 7$ (not shown), the results do not change appreciably compared to the $N = 7$ case. Overall, we find: The lower the number of years, the higher is the amplitude of the correlations and regressions, which compensate for the reduced number of degrees of freedom in the significance estimates. In addition to the higher amplitudes, the location of the maximum warm and dry anomalies can shift in accordance of the location of the associated summer SST anomalies. However, in all cases we identify an increased SST difference between the subpolar cold anomaly and the warm North Atlantic Current, a northward deflection of the lower tropospheric winds west of the European coastline, and warm and dry atmospheric anomalies over parts of western Europe or the eastern North Atlantic.

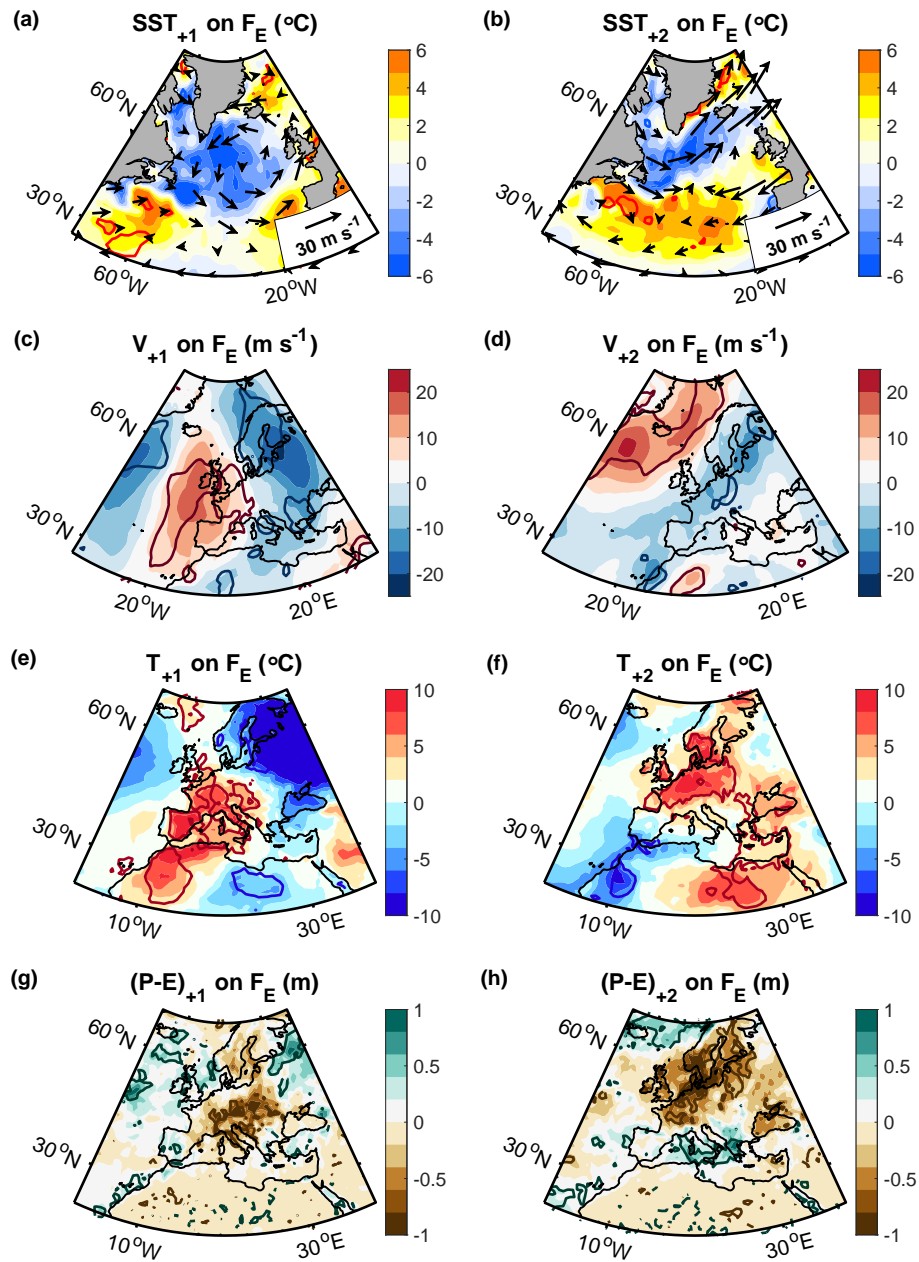

**Figure B1.** As in Figure 5 of the main manuscript, but with the NAO$_S$ index in 2019 included in the F$_E$ subset.

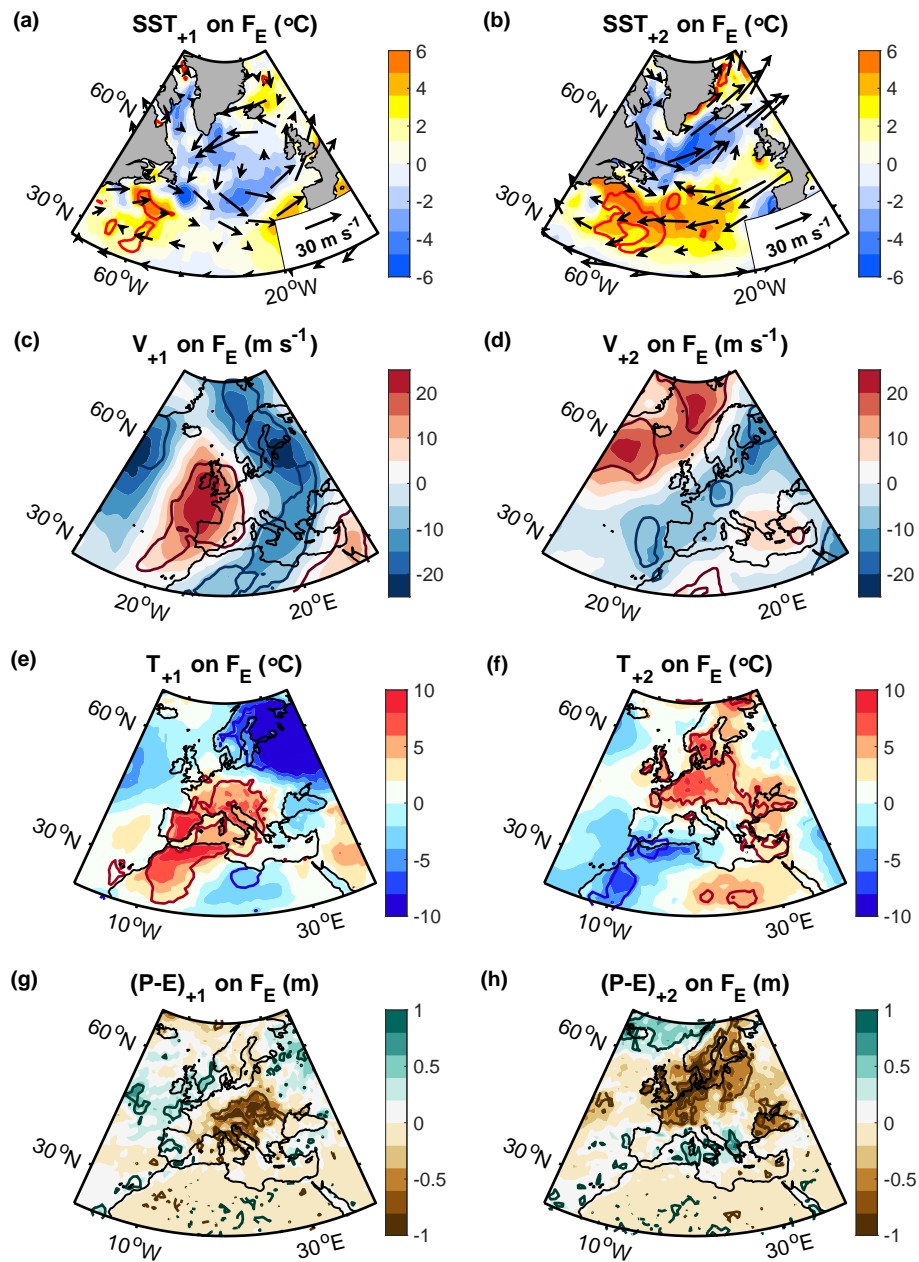

**Figure B2.** As in Figure 5 of the main manuscript, but excluding all consecutive years. Specifically, we only included the second year in all consecutive years. Thus, the regression is based on the NAO$_S$ years 1980, 1993, 2009, 2012 and 2016.

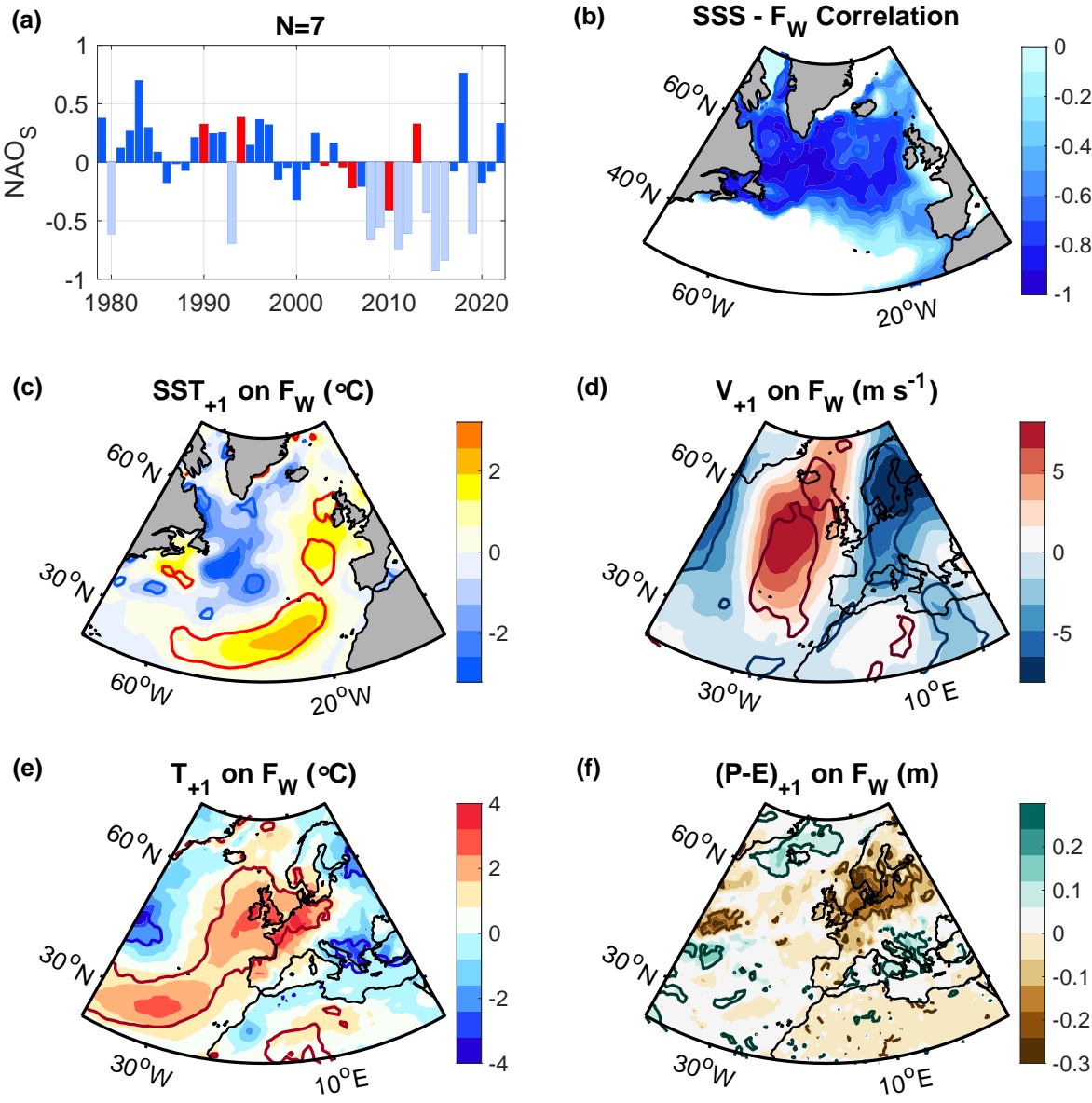

**Figure B3.** (a) $NAO_S$ index with the years included in the $F_W$ subset shown in red. Years excluded prior to the subsampling (including the 8 $F_E$ years and the two outliers) are shown as faint blue bars. (b) Correlation between the $F_W$ subset (the red $NAO_S$ years in panel a) and the associated SSS anomaly in the subsequent winter (January to March), estimated from the surface mass balance by assuming density compensation. (c-f) Regressions as in Figure 6 of the manuscript but including only 7 years in the $F_W$ subset (shown in panel a). The thick contours encompass regions that are significant at the 95% confidence level.

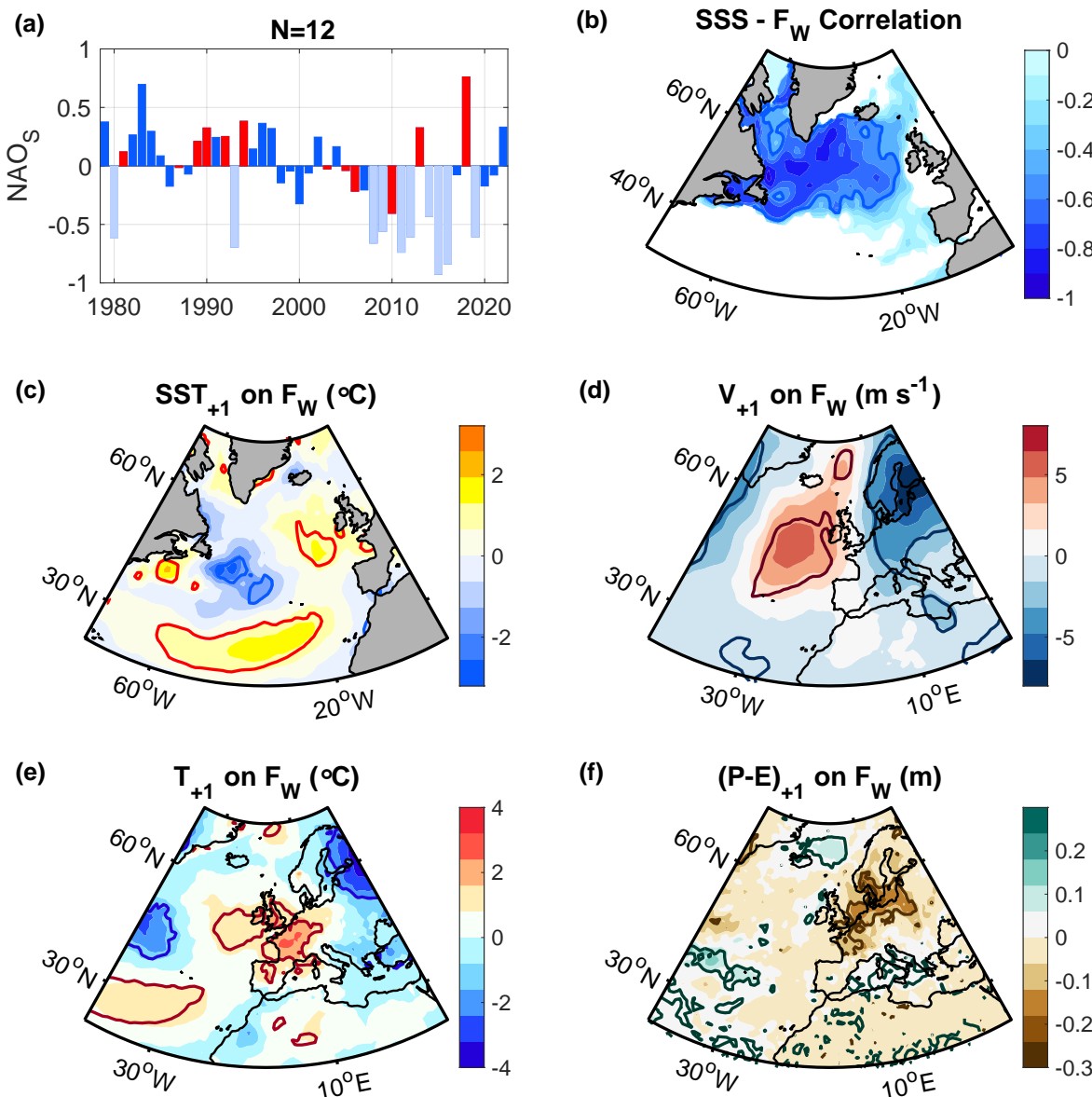

**Figure B4.** As in Figure B3 but for $N = 12$ years.

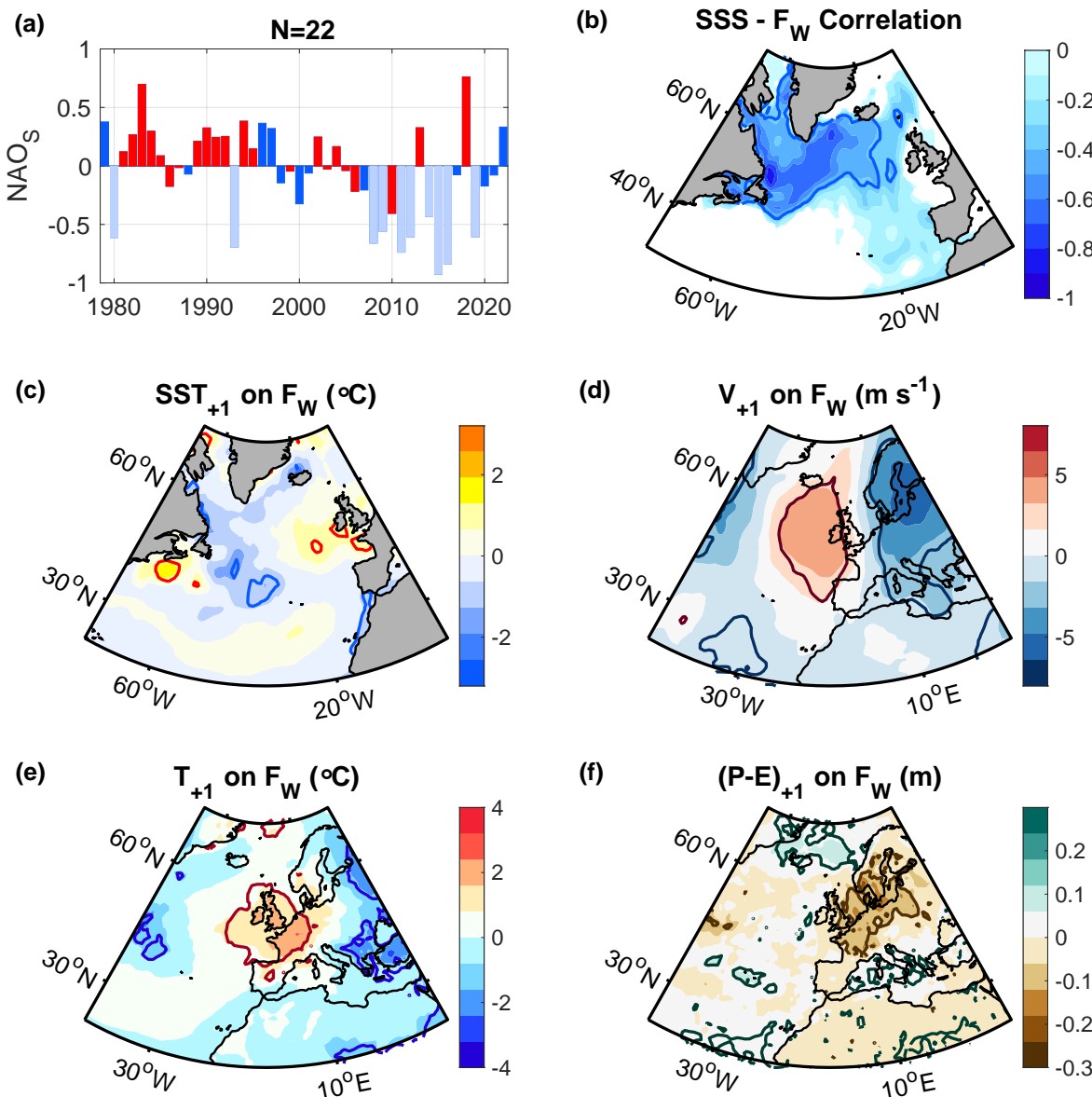

**Figure B5.** As in Figure B3 but for $N = 22$ years.

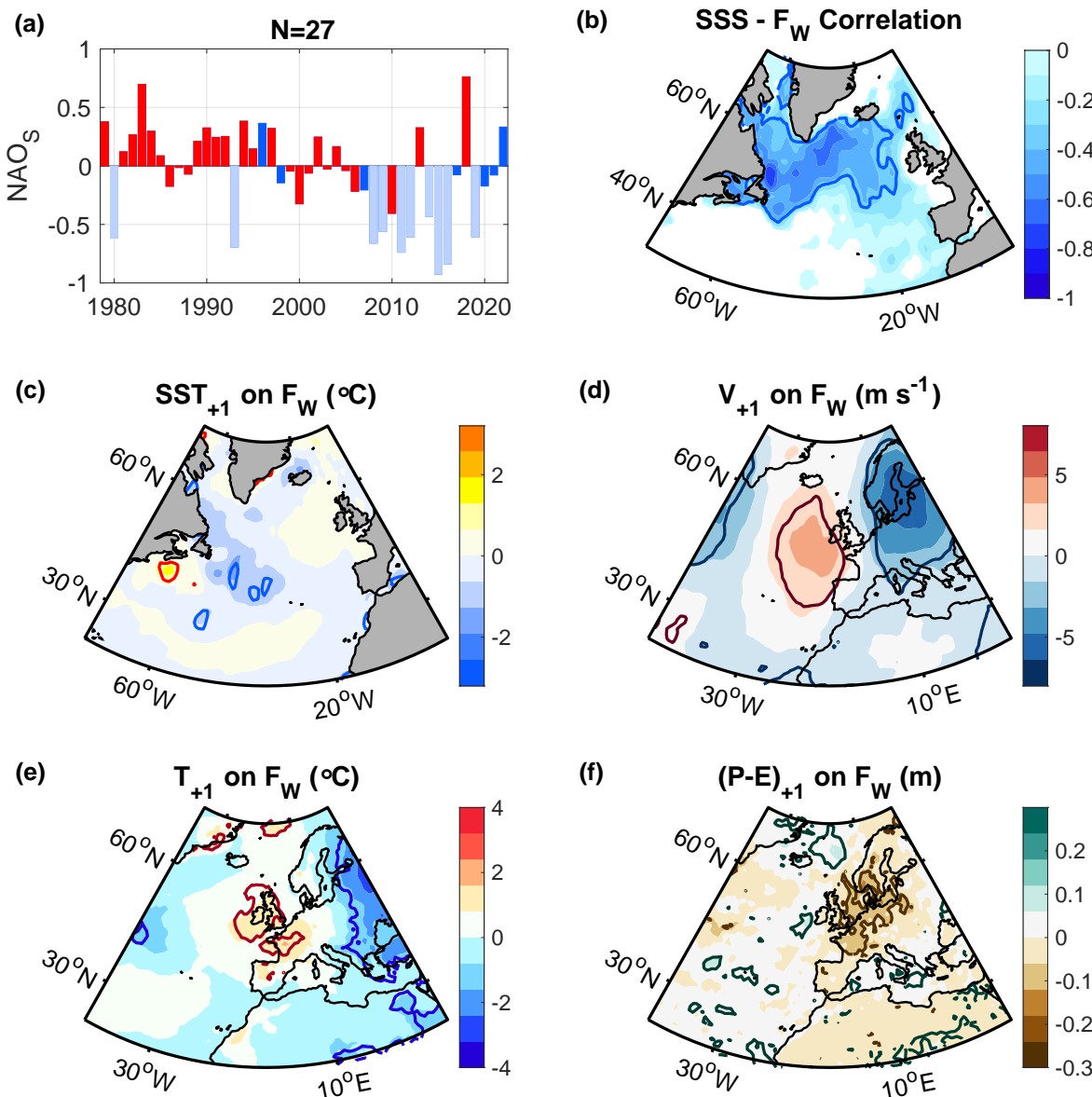

**Figure B6.** As in Figure B3 but for $N = 27$ years.

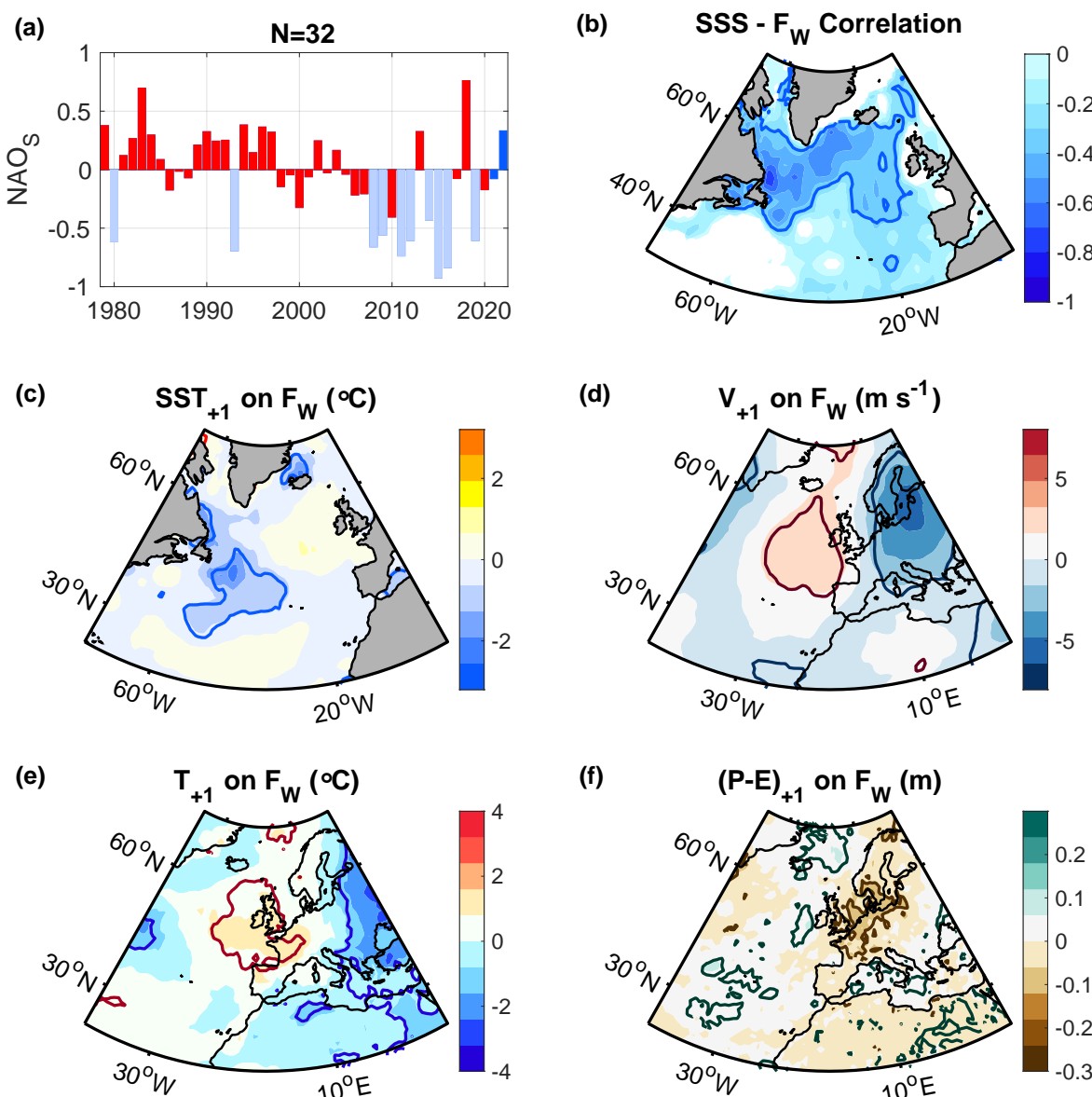

**Figure B7.** As in Figure B3 but for $N = 32$ years.

Lastly, we tested if the results obtained with the $\Delta$SST index (Fig. 7) remain significant if we lowpass filter the atmospheric anomalies in summer and assume a smaller number of degrees of freedom. In the lowpass filtering, we only consider the summer months (July and August). The filter does not include any other months. Thus, we lowpass filter the SST, winds at 700 hPa, the 2-m air temperature and precipitation minus evaporation variability with a 3-summer hanning filter. After the filtering,

the resulting autocorrelations of European summer weather are still smaller than the one for the $\Delta$SST index. Nonetheless, we estimate the number of degrees of freedom based on the $\Delta$SST index, resulting in $N^* = \frac{N\Delta t}{2T_e} - 2$ degrees of freedom, where $N$ here is the number of data points, $\Delta t$ is the time interval between them, and $T_e$ is the e-folding timescale of the autocorrelations (Leith, 1973), which is 2 years for the $\Delta$SST index (Fig. 8a). While the regressions remain significant, their amplitude weakens (Fig. B8), indicating that the interannual variability, which has been filtered out, must have contributed to

the increased relationship in the regressions obtained from the unfiltered time series (Fig. 7).

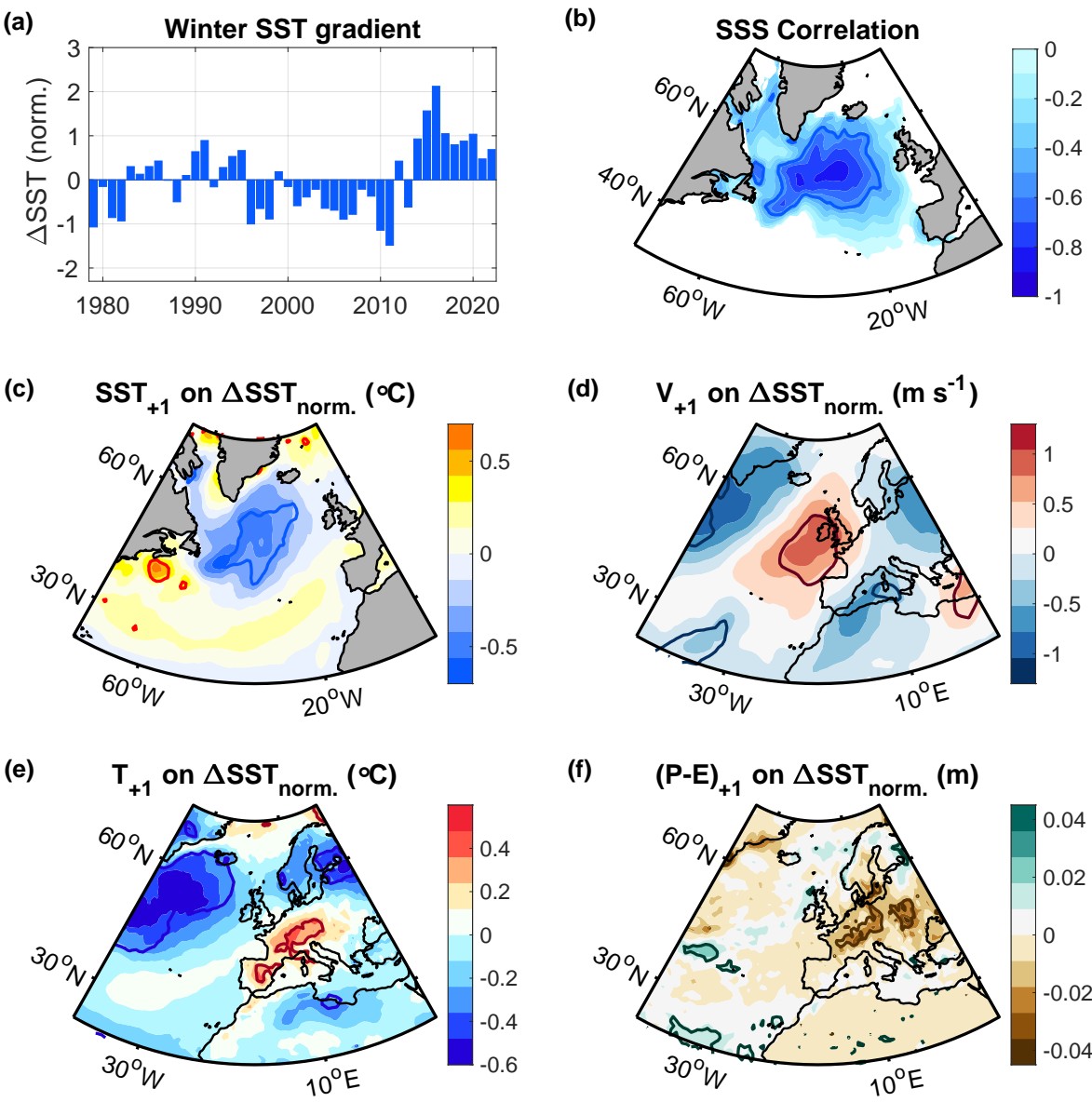

**Figure B8.** As in Figure 7 of the manuscript, after lowpass filtering the SST in summer (July and August), the 700 hPa winds in summer, and the 2-m air temperature and precipitation minus evaporation anomalies in summer, with a 3-summer hanning filter. The total number of degrees of freedom in the significance tests in all panels was estimated with $\frac{N\Delta t}{2T_e} - 2 = 9$, where $N$ is the number of years (which is 44), $\Delta t$ is one year, and $T_e$ is the lag where the correlation drops to the e-folding value ($\sim$0.37), corresponding to $\sim$2 years for the $\Delta$SST index.

**Appendix C: Supplementary Figures**

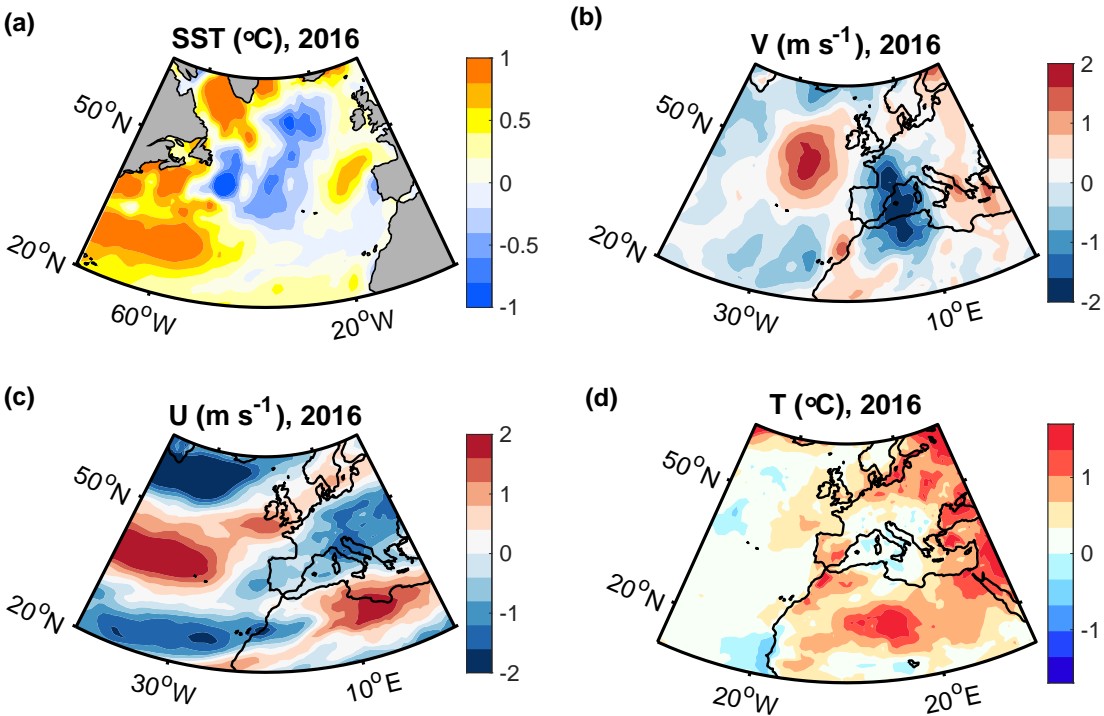

**Figure C1.** Anomalies of (a) SST, (b) the meridional winds and (c) the zonal winds at 700 hPa, and (d) the 2-m air temperature in summer (May through to August) in 2016, relative to the climatological mean.

**Table A1.** List of years included in the three freshwater indices $F_E$, $F_W$ and $\Delta$SST. The years listed for $F_E$ and $F_W$ correspond to the years of the summer NAO index in July and August, while the period listed for $\Delta$SST corresponds to the years of the SST anomalies in winter (January to March).

| $F_E$ | $F_W$ | $\Delta$SST |
|-------|-------|-------------|
| 1980 | 1981 | 1979 – 2022 |
| 1993 | 1982 | |
| 2008 | 1984 | |
| 2009 | 1987 | |
| 2011 | 1989 | |
| 2012 | 1990 | |
| 2015 | 1991 | |
| 2016 | 1992 | |
| | 1994 | |
| | 1995 | |
| | 2003 | |
| | 2004 | |
| | 2005 | |
| | 2006 | |
| | 2010 | |
| | 2013 | |
| | 2018 | |

*Author contributions.* M.O. conceived the study, carried out the analyses and was lead writer of the text. P.H. facilitated the implementation of the study; J.S. provided guidance in the model analysis; S.B. helped to revise the paper.

*Competing interests.* The authors declare that they have no conflict of interest.

*Acknowledgements.* We thank NOAA/OAR/ESRL and the Hadley Centre for providing the SST data, the Copernicus Marine Service for distributing the altimetry products, the European Centre for Medium-Range Weather Forecasts for developing the reanalysis ERA5 product, and the NOAA Physical Sciences Laboratory for facilitating access to the climate model outputs. We also thank Xavier Fettweis for providing output from the Greenland climate model MAR. The Argo data were collected and made freely available by the international Argo project and the national programs that contribute to it (http://doi.org/10.17882/42182). This study was funded through the grants ACSIS (NE/N018044/1), CLASS (NE/R015953/1) and CANARI (NE/W004984/1) from the UK National Environmental Research Council.

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
