# Peer review of "European summer weather linked to North Atlantic freshwater anomalies in preceding years"

_Weather and Climate Dynamics, 2023_

## Referee Comment (RC1)

The authors use statistical analysis of observations and reanalysis data to support their hypothesis that warmer and drier summer weather in Europe can be linked to freshwater anomalies in the North Atlantic subpolar gyre region during the preceding year. The proposed mechanism for this link is a northward shift of the North Atlantic current leading to a similar deflection of the jet stream and therefore altering the advection pathway of maritime air masses. The foundation of the analysis are freshwater indices derived from a mass balance equation that are used to identify freshwater anomalies in relation to simultaneous sea surface temperature (SST) anomalies linked to the North Atlantic Oscillation (NAO).

I understand that this is a re-submission of an earlier version of the manuscript, but I was not involved in the previous review process. Therefore, I cannot assess how the manuscript has been improved, but rather provide a fresh pair of eyes.

**I. General Comments and Suggestions:**

1. One of central results of this study is the description of "a coherent, deterministic mechanism that links North Atlantic freshwater events to European summer weather" (l. 315-316). However, the actual role of the identified freshwater anomalies in the subpolar gyre remains unclear. Given the lack of salinity observations, SST anomalies in relationship to the NAO are used as proxy for freshwater anomalies. In turn, a substantial part of the described link between the freshwater anomalies and European summer weather is based on the enhanced meridional SST gradient between the subpolar and subtropical gyre, and its influence on the storm track over the North Atlantic. This raises the question to what extent the freshwater anomalies actually influence the proposed mechanism and the downstream response?
2. At times, it is difficult to follow the analysis which might be in part due to the overall structure of the manuscript and lack of some details in the text (often they are only mentioned in figure captions). Additionally, some of the terminology is unclear or inconsistent throughout the manuscript which is possibly an artifact from the refactoring of the previous version. Hopefully, the comments below will help to streamline the text and make it more accessible for the reader.
3. Some of the figures are hard to read as individual panels are small or details are obscured. For most maps, the colorbars and axis labels take up valuable "real estate". I suggest to decrease their size and use the free white space to increase the maps wherever possible. Furthermore, I recommend to decrease the thickness of the coastlines since they can be quite distracting, especially on maps with vectors. It might also be worthwhile to mention differences in color scales in the caption wherever it can help guide the reader (e.g., Figure 2).

**II. Main Comments:**

1. Section 3/Appendix A: It took me a long time, including going back and forth between the main text and appendix to follow the approach. Given the importance of the freshwater indices as

foundation for the subsequent analysis, I suggest to combine Sections 3.1, 3.2, with Appendices A1, and A2 to describe the derivation in the main text including the clarification of the following points:

1. l. 126: Please state this equation.
2. l. 129: I think that M is not the same "downward mixing", but rather entrainment of water masses below the pycnocline into the surface ocean mixed layer as a result of a deepening mixed layer. In the context of this sentence, you refer to increased stratification due to large freshwater anomalies that inhibits a deepening of the mixed layer due to convection.
3. l. 135-137: This sentence becomes only understandable after reading the appendix.
4. Section 3.2/Appendix A2: The whole derivation of the freshwater indices is based on the NAO. I can't help but notice the striking similarity between the spatial pattern of the cold anomaly in the subpolar gyre for $F_E$ and regression pattern of SST anomalies on the North Atlantic SST Index of the Atlantic Multidecadal Oscillation (AMO) in that region (see Fig. 11 in Deser et al., 2010). The AMO has been in its warm phase since the mid-1990s and thus during the time of most of the $F_E$ years. This raises the question to what extent longer-term climate variability influences the relationship between the NAO and freshwater anomalies and if this can be utilized in the design of the freshwater indices?
5. l. 152-153: How do you estimate the correlation between the index and freshwater anomalies and how well the index represents the initial freshwater anomalies if they "are not known a priori" (cf. l. 148-149)? I think you are referring to SST anomalies which serve as proxies for the freshwater anomalies.
6. l. 156: Please define $F_E$ and $F_W$ explicitly. Without going through the appendix, the reader might ask themselves why there are two indices? What do the subscripts refer to? How are the two related?
7. l. 160: Please refer to Appendix A3 to show how these uncertainty estimates were obtained.
8. l. 161-162 & Figure 3: I think Figure 3 deserves more prominence in the text as these are the actual freshwater anomalies your hypothesis is based on. I suggest to move this sentence into its own paragraph and add more details, e.g., by being explicit that the shown salinity anomalies are estimate based on the surface mass balance (I think?), how you obtained the relationships, and what the white areas represent. Please define SSS.
9. Figure 2: Panels (a) and (b) should be the same as Panels (a) and (e) in Figure A1, but the structure of the largest values looks different. Is this just because of the differences in the color scale?
10. Equation (A2): Please define $\rho_0$.
11. l. 350: Strictly speaking, given Equation (A2) is the result of an integral over time, $h_n$ is the mixed layer at the end of winter.
12. l. 359-361: How realistic are these conditions and at what timescales do you expect this assumption to hold?
13. l. 369 & l. 371: "lower NAO index" and "higher NAO index" – do these refer to the magnitude and/or phase of the NAO?
14. l. 375-377: It would be helpful for the reader if you add a sentence how the relationship was obtained. This is partially described in the caption of Figure A1, which makes it more difficult to follow the arguments in the text.

15. l. 377: How did you determine the threshold?

16. l. 377-378: I think it is important to also mention the significant positive SST anomaly in the subtropical gyre/western North Atlantic (Figure A1a) which has a substantial contribution to ΔSST for NAO < -0.5.

17. l. 380-400: The description of the optimization process is unclear. I understand the rationale of increasing the signal-to-noise ratio, however, the selection of included years appears very subjective. How did you choose the number of years to include in the index? How did you select the discarded years? What about the two outlier years?

18. l. 387: Please define the SST gradient in the text.

19. l. 388-389: It is unclear what you mean by "spatial gradients are more robust to local variations in the surface fluxes". What if the spatial gradient is the result of local heat flux variations as one might expect from the response to the NAO (e.g., Cayan, 1992; Marshall et al., 2001; Deser et al., 2010).

20. Figure A1: It would be more intuitive and consistent with the text if you wrote $NAO_S < -0.5$ in the title of Panel (a) and in the caption. Do you include the significant cold tongue off western Africa in the calculation of ΔSST?

2. Section 4.1:

   1. The circulation anomaly you describe (Figure 4b) is reminiscent of the positive NAO phase for which the atmospheric variability patterns and corresponding ocean response are known (e.g., Cayan, 1992; Marshall et al., 2001), and are in line with your findings. It would make this section stronger if you make an explicit link of your results to the winter NAO.

   2. Changes in the wind field associated with the NAO not only change the Ekman transport as you discuss (l. 198), but also lead to changes in latent and sensible heat fluxes. Can you elaborate to what extend these changes in air-sea heat fluxes are important for creating and maintaining the meridional SST gradient?

   3. l. 200: It is unclear why you bring in the second winter. A short motivation will help to keep the reader on board.

   4. l. 210-213: From the presented figures, I cannot see a northward shift of the North Atlantic Current during the first winter (I think it shows up nicely int Figure 4d for the second winter). Is it possible that different timescales between heat fluxes and Ekman transport can explain the differences between the first and second winter? A SST gradient which is set up in the first winter and shifted northward during the second year seems also more in line with the summer SST pattern that you describe in Section 4.2 (l. 227-229).

3. Section 4.2: This section seems rather short given that it addresses one of the main results of the study. It would be helpful for the reader if you add more details and clarify the following points:

   1. l. 231: "more northerly location" compared to what?

   2. l. 237, 238, 241: "cold anomaly" in the ocean or atmosphere?

   3. l. 240: "over Europe" is rather vague (e.g., the warm and dry anomalies (Figures 6c and d) occur in different regions). See also next comment.

   4. l. 241: Is it actually true that "the overall patterns are similar after $F_E$ and $F_W$ freshwater anomalies"? The significant air temperature anomalies one year after $F_E$ extend across the Iberian Peninsula all the way to northern Africa while they are more centered around over

France and Great Britain after $F_W$. Similar the the dry anomaly occurs over the Alps and eastern Europe during the first summer after $F_E$, but more over Baltic region after $F_W$ which is more similar to second summer response after $F_E$.

5. It seems like that patterns after $F_W$ are one order of magnitude smaller compared to the patterns after $F_E$. Is this an artifact of the smaller correlation in the construction of the freshwater indices or is it due to the stronger meridional SST gradient that exists in the $F_E$ subset with significant positive SST anomalies in the subtropical gyre region?

6. Is there a reason why you show the zonal wind at 700 hPa for the $F_E$ subset and the meridional component for the $F_W$ subset?

4. Sections 4.4 and 4.5.: I have to admit that I got lost here. In general, I am wondering whether the analysis of the model simulations adds any additional information that warrants its inclusion in the manuscript.

   1. l. 271: It is not clear which pattern ($F_E$ or $F_W$) you project the on.
   2. l. 273: Most of the analysis in Sections 4.1 and 4.2 is focused on the first summer after both $F_E$ and $F_W$ years with only a brief discussion of the second summer after $F_E$. It is unclear why you construct a new index for the analysis of the model simulations based on the SST pattern in the second summer.
   3. l. 289: cold anomalies in the ocean or atmosphere?
   4. l. 294-295: Given your derivation of the freshwater indices using the surface mass balance, any cold anomaly coinicides with with a freshwater anomaly, by construction. Your analysis of the observations points out the importance of the the meridional SST gradient and its influence on the position of the jetstream. This raises the question whether the freshwater anomalies are just side effect of the mechanism that sets up the SST gradient. It is unclear to me how the model simulations help to answer this question.

**III. Additional Comments and Suggestions:**

1. l. 37, l. 83, l. 84: It would be more appropriate to use "grid spacing" instead of resolution (e.g., Grasso 2000).
2. l. 44: It's not just cold air, but also stronger winds that increase heat fluxes.
3. l. 46: Please summarize the conditions here or refer to the derivation of the freshwater indices.
4. Section 2.1: Please add details about grid spacing, temporal resolution, and any processing (e.g., calculation of anomalies, spatial interpolation, etc.). This would help make the study more reproducible. It might also be worthwhile to specify in this section which months you refer to by "summer" and "winter" throughout the text, especially since they are different from the standard definitions June-August (JJA) and December-February (DJF), respectively.
5. l. 63: How did you combine the two datasets given their different temporal and spatial resolutions?
6. l. 94-97: This sentence is unclear. I do not understand why warm anomalies due to shift in the jet stream "must" be balanced by a cold anomaly elsewhere.
7. Here are a few wordings that are either inconsistent or remnants of the previous version of the manuscript:

1. l. 185: "in winters after stronger freshwater anomalies" – based on the construction of your freshwater indices, the the anomalies should occur during winter.
    2. l. 211, 271, 316, 498, 499: what are "freshwater events"?
    3. l. 273, l. 419: What are "melt-driven" or "melt-induced" events? How are they connected to $F_E$ and $F_W$?
    4. l. 421: What are "circulation-induced freshwater events"?
8. Figure 4: I suggest to mask out the Ekman transport vectors over land. This would make it more intuitive that they refer to an ocean variable
9. l. 253: Please add a reference for the statement that "most current coupled global climate models have large freshwater biases".
10. l. 408: Do you integrate the wind stresse or resulting Ekman transport over the winter period?
11. l. 429: In l. 412-413, you define the heat flux (Q) as positive downward. A positive surface buoyancy flux (B) anomaly means Q needs to be positve (unless it its overcompensated by the freshwater flux), i.e., the ocean gains heat.
12. l. 439-440: What is the uncertainty in the freshwater fluxes due to the constant mixed layer depth used in your analysis?
13. Figure A4: It is unclear whether these are composites just for the winters before the warmest summers or also the difference with the coldest summers.
14. l. 498: This goes back to my first general comment (I.1.): If the SST pattern drives the observed atmospheric response, what is the role of the freshwater anomalies?

**IV. Typos/Wording:**

l. 53, 301: "ocean atmosphere" to "ocean-atmosphere"
l. 273: "over the central North Atlantic" to "in the central North Atlantic"

**References:**
Cayan, Daniel R. "Latent and Sensible Heat Flux Anomalies over the Northern Oceans: The Connection to Monthly Atmospheric Circulation." *Journal of Climate* 5, no. 4 (April 1, 1992): 354–69. https://doi.org/10.1175/1520-0442(1992)005<0354:LASHFA>2.0.CO;2.
Deser, Clara, Michael A. Alexander, Shang-Ping Xie, and Adam S. Phillips. "Sea Surface Temperature Variability: Patterns and Mechanisms." *Annual Review of Marine Science* 2, no. 1 (2010): 115–43. https://doi.org/10.1146/annurev-marine-120408-151453.
Grasso, Lewis D. "The Differentiation between Grid Spacing and Resolution and Their Application to Numerical Modeling." *Bulletin of the American Meteorological Society* 81, no. 3 (March 1, 2000): 579–86. https://doi.org/10.1175/1520-0477(2000)081<0579:CAA>2.3.CO;2.
Marshall, John, Helen Johnson, and Jason Goodman. "A Study of the Interaction of the North Atlantic Oscillation with Ocean Circulation." *Journal of Climate* 14, no. 7 (April 1, 2001): 1399–1421. https://doi.org/10.1175/1520-0442(2001)014<1399:ASOTIO>2.0.CO;2.

---

## Author Comment (AC1)

**Responses to Reviewer 1**

The authors use statistical analysis of observations and reanalysis data to support their hypothesis that warmer and drier summer weather in Europe can be linked to freshwater anomalies in the North Atlantic subpolar gyre region during the preceding year. The proposed mechanism for this link is a northward shift of the North Atlantic current leading to a similar deflection of the jet stream and therefore altering the advection pathway of maritime air masses. The foundation of the analysis are freshwater indices derived from a mass balance equation that are used to identify freshwater anomalies in relation to simultaneous sea surface temperature (SST) anomalies linked to the North Atlantic Oscillation (NAO).

I understand that this is a re-submission of an earlier version of the manuscript, but I was not involved in the previous review process. Therefore, I cannot assess how the manuscript has been improved, but rather provide a fresh pair of eyes.

We strongly thank the reviewer for providing a fresh pair of eyes and reviewing our manuscript. The review was extremely thorough and detailed. Moreover, the comments and suggestions were constructive and have helped us to improve the manuscript.

**I. General Comments and Suggestions:**

1. One of central results of this study is the description of "a coherent, deterministic mechanism that links North Atlantic freshwater events to European summer weather" (l. 315-316). However, the actual role of the identified freshwater anomalies in the subpolar gyre remains unclear. Given the lack of salinity observations, SST anomalies in relationship to the NAO are used as proxy for freshwater anomalies. In turn, a substantial part of the described link between the freshwater anomalies and European summer weather is based on the enhanced meridional SST gradient between the subpolar and subtropical gyre, and its influence on the storm track over the North Atlantic. This raises the question to what extent the freshwater anomalies actually influence the proposed mechanism and the downstream response?

Thank you for asking about the role of the freshwater anomalies. In the previous version, we only showed that freshwater is statistically linked with subsequent European summer weather and proposed a dynamical explanation. In the revised manuscript, we are more specific about the role of freshwater and use a more cautious narrative. Therefore, we have implemented four main changes:

(1) We have included a description of the drivers of freshwater anomalies. Thus, we provide a physical explanation for the start of the chain of feedbacks. Since the identified drivers (runoff for $F_E$ events and circulation changes for $F_W$ events) are not otherwise linked to European summer weather on the investigated timescales, and occur exactly one year in advance, they challenge the idea that an unknown third mechanism drives both freshwater anomalies and warmer European summers without the two being physically connected.

(2) We have included an analysis of the involved timescales of variability and considered alternative drivers of European summer weather acting on these timescales. Thus, we now show that surface freshening has a trend (Fig. 1a), superimposed on substantial interannual variability that is reflected in the variability of runoff, obtained from the Greenland climate model MAR (Version 3.12, forced with ERA5). Consistent with the identified drivers of freshwater anomalies, the variability of the summer cold anomaly is significantly correlated with the runoff from the preceding summer (July and August): $r \approx 0.59$, $p \approx 9 \times 10^{-8}$ over the last 70 years, which remains significant after detrending ($r \approx 0.45$, $p \approx 1 \times 10^{-4}$). After considering alternative drivers acting on these timescales, we conclude:

"Combined, the close relationships between the freshwater trend, the SST trends in summer and winter, the superimposed, high interannual variability of the cold anomaly in summer and of runoff in the year before, point to runoff as a potential trigger of the cold anomaly in summer (Fig. 1c). No other currently known mechanism in the tropics, stratosphere or outside the North Atlantic region, has such high interannual variability, is simultaneously characterised by a strong and significant trend over the last 70 years, leads to fresh and cold anomalies in winter, and occurs exactly one year before the characteristic summer SST pattern."

(3) To assess whether runoff is a trigger of the SST pattern, rather than only a predictor, we have included an analysis of the ocean-atmosphere feedbacks that contribute to the evolution of the SST pattern in summer. After investigating the surface heat and momentum fluxes and their influences on the ocean and atmosphere with ERA5, remote sensing data, in-situ hydrographic observations from the cold anomaly region, and with the models, we find that the momentum fluxes and the resulting wind-driven transports contribute to the intensification of the SST signal, while surface heat fluxes are, in turn, driven by the SST anomalies, contributing to the baroclinic instability in the atmosphere. Thus, we conclude:

"The large-scale SST pattern in winter and its evolution from winter to summer can be explained by air-sea coupling over the full North Atlantic. On the one hand, wind-induced transports and in-situ hydrographic observations from the cold anomaly region demonstrate the relevance of atmospheric forcing in intensifying the SST signals over the North Atlantic. On the other hand, model simulations, forced with prescribed, observed SST reveal the importance of the SST for the large-scale atmospheric circulation, including European summer weather. Given the importance of the involved ocean-atmosphere feedbacks, freshwater cannot be understood as the sole driver of European summer weather. It can, however, initiate the chain of ocean-atmosphere feedbacks that, in turn, affects European summer weather."

(4) We are more cautious about the wording. Throughout the analyses, we refer to freshwater as a predictor. In the conclusion, we discuss evidence that points to freshwater as a trigger rather than only a predictor, but we are explicit about the involved uncertainties, and we have removed phrases that may previously have caused confusion.

[Figure]

*Figure 1: (a,b,c) Linear trend of (a) the SSS, obtained from a surface mass balance, and (b,c) the SST in (a,b) winter (January to March) and (c) summer (July and August) over the last 70 years. (d) Regression of the SST in summer onto the time variability of the detrended SST trend pattern in summer (shown by the red bars in e), obtained by (1) projecting the spatial variability of the SST in summer onto the trend pattern, between 35 °N and 65 °N and between 10 °W and 70 °W (shown by the box in c), and by (2) detrending the resulting time series. Thus, the dTP time series represents the high-frequency component of the variability of the SST trend pattern. Contour lines in panels a-d delineate the regions that are significant at the 95% confidence level. (e) Variability of the SST trend pattern in summer (blue bars) and the de-trended time series (red bars). (f) Autocorrelations of the time variability of the trend pattern (blue bars in e), the detrended time series (red bars in e) and the full (un-detrended) summer NAO in July and August.*

2. At times, it is difficult to follow the analysis which might be in part due to the overall structure of the manuscript and lack of some details in the text (often they are only mentioned in figure captions). Additionally, some of the terminology is unclear or inconsistent throughout the manuscript which is possibly an artifact from the refactoring of the previous version. Hopefully, the comments below will help to streamline the text and make it more accessible for the reader.

Thank you for providing very specific comments below on the exact places in the manuscript that have been unclear. We have now clarified each of these instances and took care to include all details in the text. In addition, we have removed all inconsistencies in terminology that you have spotted.

3. Some of the figures are hard to read as individual panels are small or details are obscured. For most maps, the colorbars and axis labels take up valuable "real estate". I suggest to decrease their size and use the free white space to increase the maps wherever possible. Furthermore, I recommend to decrease the thickness of the coastlines since they can be quite distracting, especially on maps with vectors. It might also be worthwhile to mention differences in color scales in the caption wherever it can help guide the reader (e.g., Figure 2).

Thank you for these suggestions. We have now reduced the size of the axis labels and font sizes around the colour bars, decreased the thickness of the coastline and increased the map sizes for specific figures. Thus, we ensured that all relevant signals are included in the maps. We also mention differences in colour scales wherever they occur.

**II. Main Comments:**

1. Section 3/Appendix A: It took me a long time, including going back and forth between the main text and appendix to follow the approach. Given the importance of the freshwater indices as foundation for the subsequent analysis, I suggest to combine Sections 3.1, 3.2, with Appendices A1, and A2 to describe the derivation in the main text including the clarification of the following points:

Thank you for indicating that the derivation was difficult to follow. We have now combined Appendices A1 and A2 with Sections 3.1 and 3.2.

1. l. 126: Please state this equation.

The equation is now fully stated.

2. l. 129: I think that M is not the same "downward mixing", but rather entrainment of water masses below the pycnocline into the surface ocean mixed layer as a result of a deepening mixed layer. In the context of this sentence, you refer to increased stratification due to large freshwater anomalies that inhibits a deepening of the mixed layer due to convection.

We are now more precise and specify "vertical mixing and entrainment".

3. l. 135-137: This sentence becomes only understandable after reading the appendix.

We have removed this sentence.

4. Section 3.2/Appendix A2: The whole derivation of the freshwater indices is based on the NAO. I can't help but notice the striking similarity between the spatial pattern of the cold anomaly in the subpolar gyre for $F_E$ and regression pattern of SST anomalies on the North Atlantic SST Index of the Atlantic Multidecadal Oscillation (AMO) in that region (see Fig. 11 in Deser et al., 2010). The AMO has been in its warm phase since the mid-1990s and thus during the time of most of the $F_E$ years. This raises the question to what extent longer-term climate variability influences the relationship between the NAO and freshwater anomalies and if this can be utilized in the design of the freshwater indices?

Thank you for this suggestion. We did indeed attempt to divide the full period in high and low AMO phases. However, this did not turn out to be useful. We found that the summer NAO already filters out the interannual variability of freshwater and that the targeted approach of subsampling was more effective in optimising the indices.

In a related study, we find that the trend of the cold anomaly over the last 70 years has recently overtaken the AMO signal and now has a larger amplitude than the AMO. This has far-reaching implications for North Atlantic climate variability, including a shift of power towards interannual timescales. While a detailed analysis of the AMO signal is beyond the scope of this manuscript, we now mention its relationship to freshwater anomalies in the revised manuscript.

Overall, we find that freshwater can explain the variability of the cold anomaly pattern on a range of timescales, including that of the AMO, due to the different causes of freshwater. Thus, it may also not be desirable to completely filter out this signal.

5. l. 152-153: How do you estimate the correlation between the index and freshwater anomalies and how well the index represents the initial freshwater anomalies if they "are not known a priori" (cf. l. 148-149)? I think you are referring to SST anomalies which serve as proxies for the freshwater anomalies.

Yes, you are right. We present the correlation together with the uncertainty since both estimates belong together. We first estimate the freshwater anomalies from the SST anomalies and obtain an uncertainty of 4% and 6% respectively. We then calculate the correlation between the estimated freshwater anomalies and the NAO index. This is now clarified in the text.

6. l. 156: Please define $F_E$ and $F_W$ explicitly. Without going through the appendix, the reader might ask themselves why there are two indices? What do the subscripts refer to? How are the two related?

Since we have removed the appendices, all relevant information about $F_E$ and $F_W$ is now included in the main text.

7. l. 160: Please refer to Appendix A3 to show how these uncertainty estimates were obtained.

We have now shifted part of Appendix A3 and A4 into the results section. Thus, the results section now includes a detailed explanation of the freshwater anomalies, their uncertainty estimates, and how they were obtained.

8. l. 161-162 & Figure 3: I think Figure 3 deserves more prominence in the text as these are the actual freshwater anomalies your hypothesis is based on. I suggest to move this sentence into its own paragraph and add more details, e.g., by being explicit that the shown salinity anomalies are estimate based on the surface mass balance (I think?), how you obtained the relationships, and what the white areas represent. Please define SSS.

Thank you for pointing out the importance of Figure 3. We have now shifted it into its own section in the results and clearly explain it. We also define SSS as the sea surface salinity.

9. Figure 2: Panels (a) and (b) should be the same as Panels (a) and (e) in Figure A1, but the structure of the largest values looks different. Is this just because of the differences in the color scale?

Yes, we had adjusted the colour scale in Figure A1e to make it the same as that of Figure A1b. However, we agree that the choice of the colour scale concealed the structure of the largest values. After combining the Appendix with the approach section, we have removed this figure.

10. Equation (A2): Please define $\rho_0$.

We now define it as the density before the start of the winter.

11. l. 350: Strictly speaking, given Equation (A2) is the result of an integral over time, $h_n$ is the mixed layer at the end of winter.

Thank you for spotting this. We have now corrected it.

12. l. 359-361: How realistic are these conditions and at what timescales do you expect this assumption to hold?

At this location in the text, it is only a motivation. It is an objective of the approach to achieve these conditions, but we do not make any assumptions. We have now rephrased the sentence for clarification.

Moreover, after evaluating the mass balances, we find that (on interannual timescales) these conditions always hold within a reasonable uncertainty range. Even for the cases without any subsampling, the results hold with an uncertainty of up to 10%. Since this is an important result, we have now added:

"The result implies a remarkably close connection between freshwater and SST anomalies. A demonstration of this result with hydrographic observations is included in the Appendix, where we find that, even in the case of the most extreme air-sea fluxes, it is possible to infer freshwater anomalies from the SST with a reasonable uncertainty that is below that of currently available satellite products."

13. l. 369 & l. 371: "lower NAO index" and "higher NAO index" – do these refer to the magnitude and/or phase of the NAO?

They refer to the phase. To clarify this, we have replaced all instances of "NAO index" by "NAO state" or "phase".

14. l. 375-377: It would be helpful for the reader if you add a sentence how the relationship was obtained. This is partially described in the caption of Figure A1, which makes it more difficult to follow the arguments in the text.

Thank you for pointing this out. We have now rephrased this sentence to clarify that the relationship was not "obtained" in that we did not take any steps or created it ourselves. Instead, we observe this relationship from visual inspection of the scatter plot.

15. l. 377: How did you determine the threshold?

We determined it from visual inspection of the relationship between the summer NAO meridional SST gradient. We have now rewritten this paragraph to clarify it.

16. l. 377-378: I think it is important to also mention the significant positive SST anomaly in the subtropical gyre/western North Atlantic (Figure A1a) which has a substantial contribution to $\Delta$SST for NAO < -0.5.

Thank you for pointing this out. It is indeed important, and we now mention it at the location you indicated.

We also explain the development of the warm anomaly through air-sea coupling in more detail in Section 4.3. Overall, we are now more explicit about the role of large-scale atmospheric feedbacks contributing to the SST signal beyond the subpolar region.

17. l. 380-400: The description of the optimization process is unclear. I understand the rationale of increasing the signal-to-noise ratio, however, the selection of included years appears very subjective. How did you choose the number of years to include in the index? How did you select the discarded years? What about the two outlier years?

Thank you for indicating that the optimisation process was insufficiently described. As you correctly point out, the objective of the process is to increase the signal-to-noise ratio.

To better describe the process, we now further explain: "There is a trade-off between the number of years included and the resulting correlation. Here, we selected N=17 years as a reasonable compromise for obtaining a high correlation of 0.90 while keeping a relatively large sample size, reflected in low p-values (p < 2.6 x10$^{-6}$). Selecting N=16 or N=18 increases the p-value again. However, the results are not sensitive to this choice."

To add further support for the subsampling, we have included a new section in the results that links the two sub-sampled indices to physical causes of freshwater anomalies, supporting the optimisation process with a dynamical explanation.

The two outlier years correspond to years where the NAO index was not a useful indicator for the fresh and cold anomalies. However, upon investigating both years more closely we found that the relationship between the salinity and temperature anomalies still holds.

18. l. 387: Please define the SST gradient in the text.

We now define the SST gradient in the text and explain that it refers to the difference between the subtropical warm and subpolar cold anomaly regions, enclosed within the 95% confidence lines.

19. l. 388-389: It is unclear what you mean by "spatial gradients are more robust to local variations in the surface fluxes". What if the spatial gradient is the result of local heat flux variations as one might expect from the response to the NAO (e.g., Cayan, 1992; Marshall et al., 2001; Deser et al., 2010).

Thank you for pointing out that the sentence was unclear. By using spatial gradients, we filter out the uniform warming effect of increasing greenhouse gases. Another advantage of using spatial gradients is that the area, used for the derivation of the indices is based on overall larger areas and therefore less sensitive to regional fluctuations in surface currents or fluxes. This is now clarified.

We also point out that the surface fluxes do not contribute to the SST patterns. The surface fluxes were evaluated as part of the mass balance. Over the cold anomaly, the surface flux anomaly is positive implying that the ocean loses less heat. Thus, the ocean anomalously cools the atmosphere rather than the other way round.

20. Figure A1: It would be more intuitive and consistent with the text if you wrote NAOs < -0.5 in the title of Panel (a) and in the caption. Do you include the significant cold tongue off western Africa in the calculation of ΔSST?

The change in the title you suggest would not simply be a change in nomenclature. Instead, it would change the sign of the regressor. Thus, everything in the figure would be opposite. The sign of all the obtained SST anomalies would be reversed. However, the point of the figure is to show that we obtain similar anomalies as in panel b, where the sign of the NAO is opposite.

The cold tongue can be understood as a feedback since the associated large-scale atmospheric circulation anomaly induces upwelling off western Africa. However, we do not include the cold tongue off western Africa in the calculation of the SST gradient. We have now clarified this in the text. Also, throughout the manuscript we are more explicit about the large-scale atmospheric feedbacks that contribute to the SST pattern.

2. Section 4.1:
   1. The circulation anomaly you describe (Figure 4b) is reminiscent of the positive NAO phase for which the atmospheric variability patterns and corresponding ocean response are known (e.g., Cayan, 1992; Marshall et al., 2001), and are in line with your findings. It would make this section stronger if you make an explicit link of your results to the winter NAO.

Thank you for pointing this out. Following your suggestion, we now make an explicit link to the winter NAO.

   2. Changes in the wind field associated with the NAO not only change the Ekman transport as you discuss (l. 198), but also lead to changes in latent and sensible heat fluxes. Can you elaborate to what extend these changes in air-sea heat fluxes are important for creating and maintaining the meridional SST gradient?

We now state that we do not find any significant impact of the surface heat fluxes on maintaining the SST gradient, neither in winter nor in summer. On the contrary, we find that the SST anomaly drives the surface flux anomalies. Since the cold anomaly is associated with a positive heat flux anomaly, it implies reduced ocean heat losses. Thus, it contributes to the baroclinic instability in the lower troposphere. This is now clarified in the description of the air-sea feedbacks, both in winter and in summer.

3. l. 200: It is unclear why you bring in the second winter. A short motivation will help to keep the reader on board.

Thank you for suggesting this. We now motivate the investigation of the second winter with the climatic importance of North Atlantic Current shifts.

4. l. 210-213: From the presented figures, I cannot see a northward shift of the North Atlantic Current during the first winter (I think it shows up nicely int Figure 4d for the second winter). Is it possible that different timescales between heat fluxes and Ekman transport can explain the differences between the first and second winter? A SST gradient which is set up in the first winter and shifted northward during the second year seems also more in line with the summer SST pattern that you describe in Section 4.2 (l. 227-229).

Thank you for pointing out that this was unclear.

The Ekman transports are an instantaneous response. They already lead to a warm anomaly in the first winter after the freshwater index. However, the northward current shift is only visible in the warm anomaly to the south of the cold anomaly (the Gulf Stream deflection point). It does not extend across to the east coast.

We now clarified in the text: "The water inside the North Atlantic Current is not anomalously warm but it occurs at an anomalously northward location. Thus, the northward shift of the North Atlantic Current is reflected in the warm anomaly to the southwest of the subpolar cold anomaly. The warm anomaly is already visible in the first winter after the summer NAO but it does not extend to the east coast."

We also investigated the surface heat fluxes (shown in the appendix) but they were not able to explain the SST patterns. This is now clarified.

3. Section 4.2: This section seems rather short given that it addresses one of the main results of the study. It would be helpful for the reader if you add more details and clarify the following points:

Thank you for your suggestions. Following your suggestions below, we have now expanded the section.

1. l. 231: "more northerly location" compared to what?

"More northerly compared to the previous summer". This is now clarified.

2. l. 237, 238, 241: "cold anomaly" in the ocean or atmosphere?

We referred to the negative SST anomaly. This is now clarified.

3. l. 240: "over Europe" is rather vague (e.g., the warm and dry anomalies (Figures 6c and d) occur in different regions). See also next comment.

Following your comment below, we are now more specific.

4. l. 241: Is it actually true that "the overall patterns are similar after $F_E$ and $F_W$ freshwater anomalies"? The significant air temperature anomalies one year after $F_E$ extend across the Iberian Peninsula all the way to northern Africa while they are more centered around over France and Great Britain after $F_W$. Similar the the dry anomaly occurs over the Alps and eastern Europe during the first summer after $F_E$, but more over Baltic region after $F_W$ which is more similar to second summer response after $F_E$.

Thank you for suggesting this. We are now more specific and say that the mechanism is the same, but it occurs over a different region, consistent with the underlying SST anomalies. We now exactly specify the regions in the text.

5. It seems like that patterns after $F_W$ are one order of magnitude smaller compared to the patterns after $F_E$. Is this an artifact of the smaller correlation in the construction of the freshwater indices or is it due to the stronger meridional SST gradient that exists in the $F_E$ subset with significant positive SST anomalies in the subtropical gyre region?

Yes, you are right that the magnitudes of the obtained signals in European summer weather are carried over from the larger magnitudes in the freshwater and SST signals.

Please note these are not absolute anomalies but regressions. The large magnitudes result from steep regression slopes. These steep regression slopes occur because the underlying changes in the $F_E$ index are much smaller compared to the changes in the $F_W$ index. An implication of this is that, once the seasonal surface freshening (or the $F_E$ index) exceeds a critical threshold, a relatively small further increase is linked to relatively large feedbacks. This is now better explained in the text, both in the section where we introduce the indices and where we describe the subsequent European summer weather.

6. Is there a reason why you show the zonal wind at 700 hPa for the $F_E$ subset and the meridional component for the $F_W$ subset?

The winds closely follow the SST fronts, indicated by the arrows. To best show the link between the SST and the wind field, we selected the wind component that best matches the shape of the underlying SST pattern. This is now clarified.

4. Sections 4.4 and 4.5.: I have to admit that I got lost here. In general, I am wondering whether the analysis of the model simulations adds any additional information that warrants its inclusion in the manuscript.

Following your suggestion, we have removed the previous model analysis. We have now replaced it by another analysis in which the link between the selected SST anomalies for the model analysis and the freshwater anomalies is clearer.

Since we have removed the analysis, the comments below no longer apply.

1. l. 271: It is not clear which pattern ($F_E$ or $F_W$) you project the on.
2. l. 273: Most of the analysis in Sections 4.1 and 4.2 is focused on the first summer after both $F_E$ and $F_W$ years with only a brief discussion of the second summer after $F_E$. It is unclear why you construct a new index for the analysis of the model simulations based on the SST pattern in the second summer.

3. l. 289: cold anomalies in the ocean or atmosphere?
4. l. 294-295: Given your derivation of the freshwater indices using the surface mass balance, any cold anomaly coinicides with a freshwater anomaly, by construction. Your analysis of the observations points out the importance of the the meridional SST gradient and its influence on the position of the jetstream. This raises the question whether the freshwater anomalies are just side effect of the mechanism that sets up the SST gradient. It is unclear to me how the model simulations help to answer this question.

In the new analyses, we examine the link between the freshwater anomaly in winter and the SST anomaly in summer more thoroughly. While the freshwater anomalies can explain the initial trigger of the chain of events, the final SST anomalies in summer can only be understood as the result of large-scale air-sea coupling processes. Thus, we are more cautious in the wording in the revised manuscript.

**III. Additional Comments and Suggestions:**

1. l. 37, l. 83, l. 84: It would be more appropriate to use "grid spacing" instead of resolution (e.g., Grasso 2000).

Thank you for pointing this out. We have corrected this now.

2. l. 44: It's not just cold air, but also stronger winds that increase heat fluxes.

This is true but in the preceding sentence we explain that the air is always colder than the ocean in winter. This naturally implies a mean climatological ocean heat loss. We do not exclude that stronger winds also increase the surface fluxes. They are just not relevant in the context of this paragraph. We have now rewritten the sentence more clearly.

3. l. 46: Please summarize the conditions here or refer to the derivation of the freshwater indices.

We now state the conditions.

4. Section 2.1: Please add details about grid spacing, temporal resolution, and any processing (e.g., calculation of anomalies, spatial interpolation, etc.). This would help make the study more reproducible. It might also be worthwhile to specify in this section which months you refer to by "summer" and "winter" throughout the text, especially since they are different from the standard definitions June-August (JJA) and December-February (DJF), respectively.

Thank you for these suggestions. This is now included in the data section.

5. l. 63: How did you combine the two datasets given their different temporal and spatial resolutions?

We did not combine them ourselves but downloaded the merged dataset from NCAR. This is now clarified by including the link to the datafile in the data section.

6. l. 94-97: This sentence is unclear. I do not understand why warm anomalies due to shift in the jet stream "must" be balanced by a cold anomaly elsewhere.

In contrast to greenhouse gas warming, freshwater-linked temperature anomalies do not result in a net imbalance in the Earth's surface energy budget. The warming over Europe is balanced by a cooling over the ocean since the underlying baroclinic wave activity consists of an anticyclonic anomaly on one side of the jet stream, and a cyclonic anomaly on the other side. Thank you for pointing out that this was unclear. We have clarified this in the revised manuscript.

7. Here are a few wordings that are either inconsistent or remnants of the previous version of the manuscript:

Many thanks for spotting the below inconsistencies. We have now clarified all instances.

  1. l. 185: "in winters after stronger freshwater anomalies" – based on the construction of your freshwater indices, the anomalies should occur during winter.

  We have removed this sentence.

  2. l. 211, 271, 316, 498, 499: what are "freshwater events"?

  We have replaced "freshwater events" by "freshwater anomalies" everywhere.

  3. l. 273, l. 419: What are "melt-driven" or "melt-induced" events? How are they connected to $F_E$ and $F_W$?

  Thank you for spotting this. We have removed the terms "melt-driven" and "melt-induced" at both locations. Also, we now added a section that explains the term.

  4. l. 421: What are "circulation-induced freshwater events"?

  We have removed the term.

8. Figure 4: I suggest to mask out the Ekman transport vectors over land. This would make it more intuitive that they refer to an ocean variable

Indeed, thank you. We have now masked them out.

9. l. 253: Please add a reference for the statement that "most current coupled global climate models have large freshwater biases".

We have now added references.

10. l. 408: Do you integrate the wind stress or resulting Ekman transport over the winter period?

Yes, we do. We have now clarified this in the text.

11. l. 429: In l. 412-413, you define the heat flux (Q) as positive downward. A positive surface buoyancy flux (B) anomaly means Q needs to be positive (unless it its overcompensated by the freshwater flux), i.e., the ocean gains heat.

Yes, this is true for the anomalies. However, in the mean winter, the subpolar ocean loses heat. Thus, we considered it more appropriate to state "the ocean loses less heat" instead of "the ocean gains heat".

12. l. 439-440: What is the uncertainty in the freshwater fluxes due to the constant mixed layer depth used in your analysis?

We assume that you are referring to "constant" as in "constant over different years", not "constant over the winter" since we do not assume the mixed layer depth to be constant over the winter.

The influence of a variable mixed layer depth over different winters on the results can be understood by considering two cases:

(1) If the mixed layer depth is positively correlated with the NAO indices (that means the mixed layer would be deeper for larger indices), the terms on the righthand side of the mass balance equation would be even less relevant and the actual uncertainties of the freshwater anomalies would be even smaller than the ones provided.

(2) If the mixed layer depth is negatively correlated with the NAO indices (that means the mixed layer would be shallower), we do not need to evaluate the mass balance. In that case, the combination of shallower mixed layers and negative temperature anomalies implies that salinity anomalies must dominate stratification.

In this scenario, the freshwater anomalies even overcompensate the density increase by the temperature anomalies. However, shallower mixed layers also imply that less ocean heat is available to drive the atmosphere, reflected in the positive surface flux anomalies. Since the identified surface flux anomalies are very small and not significant, we conclude that this overcompensation is negligible up to the uncertainty range provided in the text.

In the text, we have now added: "Since for both $F_W$ and $F_E$, the surface buoyancy flux anomalies are positive, the ocean loses less heat ("M drives B"), and the mixed layer is slightly shallower and lighter for increased indices, when averaged over the cold anomaly regions. As shown above, however, the density changes implied by the surface fluxes associated with both $F_W$ and $F_E$ are over one order of magnitude smaller than the density changes implied by the cold anomalies. Thus, the change in the mixed layer depth, and any overcompensation of the density anomaly – by a surplus of surface freshening, a slowdown of the buoyancy-driven overturning circulation, pre-existing density anomalies, or any other buoyancy-driven mechanism – is negligible on the timescales considered."

13. Figure A4: It is unclear whether these are composites just for the winters before the warmest summers or also the difference with the coldest summers.

For consistency with the heat wave composites, the composites refer to the difference between the ten warmest and the ten coldest summers. This is now clarified.

14. l. 498: This goes back to my first general comment (I.1.): If the SST pattern drives the observed atmospheric response, what is the role of the freshwater anomalies?

We are now more specific in the role of the freshwater anomalies (please see first comment for more details).

**IV. Typos/Wording:**

l. 53, 301: "ocean atmosphere" to "ocean-atmosphere"

This is now corrected.

l. 273: "over the central North Atlantic" to "in the central North Atlantic"

This is now corrected.

**References:**

Cayan, Daniel R. "Latent and Sensible Heat Flux Anomalies over the Northern Oceans: The Connection to Monthly Atmospheric Circulation." *Journal of Climate* 5, no. 4 (April 1, 1992): 354–69. https://doi.org/10.1175/1520-0442(1992)005<0354:LASHFA>2.0.CO;2.

Deser, Clara, Michael A. Alexander, Shang-Ping Xie, and Adam S. Phillips. "Sea Surface Temperature Variability: Patterns and Mechanisms." *Annual Review of Marine Science* 2, no. 1 (2010): 115–43. https://doi.org/10.1146/annurev-marine-120408-151453.

Grasso, Lewis D. "The Differentiation between Grid Spacing and Resolution and Their Application to Numerical Modeling." *Bulletin of the American Meteorological Society* 81, no. 3 (March 1, 2000): 579–86. https://doi.org/10.1175/1520-0477(2000)081<0579:CAA>2.3.CO;2.

Marshall, John, Helen Johnson, and Jason Goodman. "A Study of the Interaction of the North Atlantic Oscillation with Ocean Circulation." *Journal of Climate* 14, no. 7 (April 1, 2001): 1399–1421. https://doi.org/10.1175/1520-0442(2001)014<1399:ASOTIO>2.0.CO;2.

We strongly thank the reviewer for reading our manuscript so carefully. The level of detail in this review was extraordinary. Thank you for providing so many detailed, helpful and constructive comments and suggestions!

---

## Author Comment (AC2)

**Responses to Reviewer 2**

This paper documents an apparent impact of subpolar freshwater anomalies onto the North Atlantic sea surface temperatures (SSTs) in winter and the subsequent summer. These summer SSTs are then argued to drive changes in the atmospheric circulation that drive significant changes in summer temperatures though analysis of reanalysis data and model simulations. Overall, the authors argue that freshwater changes in the North Atlantic can drive a large amount of the variance in European summer temperatures and so argue that such a mechanism could provide significant extra skill in seasonal predictions.

Overall this is an interesting study and presents an exciting set of results. However, I have a number of issues with the manuscript in its current form, which I list below in my major comments. Therefore, I do not recommend publication at this time.

We strongly thank the reviewer for reviewing our manuscript and providing many detailed and constructive comments and suggestions! Your comments and suggestions have helped us to clarify and improve the manuscript.

**Major comments**

1. My main concern with this manuscript is that the most important results are buried in the appendix - that is the definition of the freshwater budget method, and the results evaluating the impact of freshwater on the North Atlantic SSTs. I note that this is similar to the structure of Oltmanns et al, 2020, in GRL. This usage of the appendix meant that the paper was quite challenging to read as I had to keep flipping back and forth in the paper to try to understand the logic. Furthermore, I would argue that the most important result in the paper is the evaluation of impact of freshwater onto the resulting SST (e.g., section A3). Therefore, I would strongly recommend the authors to move more of the important background information into the main paper and systematically step the reader through the ideas.

Thank you for indicating that the usage of the appendix made it difficult to follow the mass balance approach, and that the link between the freshwater and SST is also an important result. Following your suggestion, we have now integrated Appendix A3 and A4 into the results section. In addition, we have combined Appendix A1 and A2 with the approach section. Thus, the full derivation is now included in the approach section and the results of the mass balance analysis are now included in the results sections.

Still, we think that the link between the freshwater and SST anomalies in winter is only part of the results. The link between freshwater anomalies and the subsequent European summer weather is also an important finding. Thus, we have included additional analyses focusing on the link between the fresh and cold anomalies in winter and the SST anomalies in the subsequent summer, including the role of air-sea feedbacks throughout their evolution (further details are included in your specific comments below).

2. Although I found the results exciting, ultimately the results on the potential impact of freshwater anomalies on Europe were much more uncertain than I feel the authors described. The uncertainty spawns, I think, from the following important reasons.

- There is little physical understanding of how the summer NAO is leading large freshwater driven SST changes across the subpolar North Atlantic. For example, there is much discussion about how the sign of the relationship between summer NAO and the winter SST anomaly changes at ~0.5, but no reason for this is given. How can we be certain that the freshwater analysis is picking up real physical changes related to changes in freshwater?

Thank you for suggesting adding a physical explanation for the link between the summer NAO and the subsequent freshwater anomalies. In the revised manuscript, we now review the causes of freshwater anomalies associated with both high and low NAO summers.

Consistent with earlier studies, we find that a lower NAO phase is associated enhanced seasonal runoff and melting while a higher NAO phase is associated with an enhanced wind stress curl over the subpolar region, resulting in a stronger subpolar gyre circulation and enhanced advection of fresh, polar water into the subpolar region (Fig. 1). The threshold of ~ -0.5 in the summer NAO corresponds to a critical surface freshening above which the shallower, seasonal freshwater anomalies are mixed down, such that deeper, circulation-driven anomaly signals dominate the hydrography of the subpolar gyre. We also investigated the surface freshwater fluxes (precipitation minus evaporation) but found them to be unimportant for the identified freshwater anomalies.

[Figure]

*Figure 1: (a) Relationship between the NAO in July and August ($NAO_S$) and the total runoff, integrated over the Greenland ice sheet, in July and August. (b) Regression of the seasonal surface freshening from summer (August) to winter (January to March) onto - $NAO_S$ from the preceding summer. Multiplying the summer NAO by '-1' serves the purpose of obtaining an index that is positively correlated with the surface freshening. The seasonal surface freshening has been obtained from a surface mass balance (Oltmanns et al., 2020) and the arrows indicate the mean subpolar gyre circulation, derived from altimetry. (c,d) Regression of the absolute dynamic topography (ADT) in winter (January to March) onto the (c) $NAO_S$ and (d) $F_W$ from the preceding summer. A negative ADT anomaly implies a more cyclonic circulation and hence, increased advection of polar water into the region (Häkkinen et al. 2013). The contours in panels b-d delineate the regions that are significant at the 95% confidence level.*

The obtained links are consistent with earlier studies on the role of runoff (Bamber et al., 2018; Dukhovskoy et al., 2019) and the subpolar gyre circulation (Häkkinen and Rhines 2009; Häkkinen et al., 2011, 2013; Holliday et al., 2020) for freshwater anomalies in the subpolar North Atlantic.

- o Related to the above, the paper focuses on interannual changes - however, there is plenty of observed evidence that there is significant decadal time-scale variability in the subpolar North Atlantic, and so questions arise on what the freshwater analysis, which appears to assume independence between years, is picking up. For example, figure 2c shows that it is the decadal time-scales that dominate the summer NAO time-series, consistent with a southward shift of the North Atlantic Jet (e.g., Dong et al, 2013). Given the low-frequency changes, how can the authors be sure that they are seeing the interannual impact of freshwater changes as opposed to a different, or related, mechanism? For example, is the summer NAO and winter SST responding to another mechanism which is driving both (e.g., large-scale ocean circulation, external forcings etc)? As mentioned below, I think overlaying a timeseries of summer NAO with subpolar SSTs would be useful in seeing the potential importance of these longer timescales.

Thank you for pointing out that the involved timescales of freshwater variability were unclear. In the revised manuscript, we have included an analysis of the longer-term freshwater trend over the last 70 years (Fig. 2). Thus, we find that the summer NAO largely reflects the high frequency, interannual variability of the cold anomaly pattern in the subsequent summer. We also now show the autocorrelation of the summer NAO (Fig. 2f), which is negligible at one-year lag and consistent with the trend of ~-0.002 $yr^{-1}$, which is only small and not significant.

In the past, decadal switches of the Arctic Ocean Oscillation have led to periodic exports of fresh and cold anomalies from the Arctic into the subpolar North Atlantic (Proshutinsky et al., 2015), which can explain the decadal variability of the SST pattern in winter. However, by using the negative summer NAO as regressor, we filter out low-frequency variability and focus on interannual timescales, consistent with the timescales of European summer weather, which also has a high interannual variability. Since the autocorrelations of the NAO indices, used as regressors in this study, are negligible, we do not expect decadal variability to substantially affect the results.

[Figure]

*Figure 2: (a,b,c) Linear trend of (a) the SSS, obtained from a surface mass balance, and (b,c) the SST in (a,b) winter (January to March) and (c) summer (July and August) over the last 70 years. (d) Regression of the SST in summer onto the time variability of the detrended SST trend pattern in summer (shown by the red bars in e), obtained by (1) projecting the spatial variability of the SST in summer onto the trend pattern, between 35 °N and 65 °N and between 10 °W and 70 °W (shown by the box in c), and by (2) detrending the resulting time series. Thus, the dTP time series represents the high-frequency variability of the SST trend pattern. Contour lines in panels a-d delineate regions that are significant at the 95% confidence level. (e) Variability of the SST trend pattern in summer (blue bars) and the de-trended time series (red bars). (f) Autocorrelations of the time variability of the trend pattern (blue bars in e), the detrended time series (red bars in e) and the full (un-detrended) summer NAO in July and August.*

o  The model experiments are interesting, but they do not, of course, explore the impact of freshwater changes on European weather. Indeed, the models are feeling the influence of both SSTs and, presumably, external forcings, and the assumption is that the influence of freshwater can be isolated by focusing on the summer NAO analysis. However, and related to point b, this relies that there are no other mechanisms in play that could explain both the summer NAO and the winter SSTs.

Thank you for pointing out that the role of freshwater anomalies for European summer weather was unclear. To address your comment, we have implemented four main changes:

(1) We have included a description of the drivers of freshwater anomalies. Thus, we provide a physical explanation for the start of the chain of feedbacks. Since the identified drivers (runoff for $F_E$ events and circulation changes for $F_W$ events) are not otherwise linked to European summer weather on the investigated timescales, and occur exactly one year in advance, they challenge the idea that an unknown third mechanism drives both freshwater anomalies and warmer European summers without the two being physically connected.

(2) We have included an analysis of the involved timescales of variability and considered alternative drivers of European summer weather acting on these timescales. Thus, we now show that surface freshening has a trend, superimposed on substantial interannual variability that is reflected in the variability of runoff, obtained from the Greenland climate model MAR (Version 3.12, forced with ERA5). Consistent with the identified drivers of freshwater anomalies, the variability of the summer cold anomaly is significantly correlated with the runoff from the preceding summer (July and August): $r \approx 0.59$, $p \approx 9$ x $10^{-8}$ over the last 72 years, which remains significant after detrending ($r \approx 0.45$, $p \approx 1$ x $10^{-4}$). After considering alternative drivers acting on these timescales, we conclude:

"Combined, the close relationships between the freshwater trend, the SST trends in summer and winter, and the superimposed, high interannual variability of the cold anomaly in summer and of runoff in the year before, point to runoff as a potential trigger of the cold anomaly in summer (Fig. 2c). No other currently known mechanism in the tropics, stratosphere or outside the North Atlantic region, has such a high interannual variability, is simultaneously characterised by a strong and significant trend over the last 70 years, leads to fresh and cold anomalies in winter, and occurs exactly one year before the characteristic summer SST pattern."

(3) To assess whether runoff is a trigger of the SST pattern, rather than only a predictor, we have included an analysis of the ocean-atmosphere feedbacks that contribute to the evolution of the SST pattern in summer. After investigating the surface heat and momentum fluxes and their influences on the ocean and atmosphere with ERA5, remote sensing data, in-situ hydrographic observations from the cold anomaly region, and with the models, we find that the momentum fluxes (wind-driven Ekman transports) contribute to the intensification of the SST signal through the summer, while surface heat fluxes are, in turn, driven by the SST anomalies, contributing to the baroclinic instability in the atmosphere. Thus, we conclude:

"The large-scale SST pattern in winter and its evolution from winter to summer can be explained by air-sea coupling over the full North Atlantic. On the one hand, wind-induced transports and in-situ hydrographic observations from the cold anomaly region demonstrate the relevance of atmospheric forcing in intensifying the SST signals over the North Atlantic. On the other hand, model simulations, forced with prescribed, observed SST reveal the importance of the SST for the large-scale atmospheric circulation, including European summer weather. Given the importance of the involved ocean-atmosphere feedbacks, freshwater cannot be understood as the sole driver of European summer weather. It can, however, initiate a chain of ocean-atmosphere feedbacks that affects European summer weather."

(4) We are more cautious about the wording. Throughout the analyses, we refer to freshwater as a predictor, not as a driver. In the conclusion, we discuss evidence that points to freshwater as a trigger rather than only a predictor, but we are explicit about the involved uncertainties, and we have removed phrases that may previously have caused confusion.

Indeed, I can't help noticing that the SST patterns being explored in the models (figure 8a and figure 9a and b) look like the cold AMV pattern (e.g., Zhang et al, 2019). Therefore, how can the authors be confident that the atmospheric circulation patterns are being driven by just the subpolar North Atlantic, rather than the tropical North Atlantic?

Thank you for indicating that large-scale ocean-atmosphere feedbacks over the full North Atlantic were insufficiently described. In the revised manuscript we describe them more clearly, both in winter and in summer.

The SST in the subtropical and subpolar regions is highly coupled due to large-scale atmospheric feedbacks. For instance, the large-scale atmospheric circulation anomaly, associated with the subpolar cold anomaly, leads to convergent, wind-induced surface currents in the inter-gyre region, and thus a warm anomaly that emphasises the SST gradient. At the same time, the anti-cyclonic, atmospheric circulation anomaly to the south of the SST front leads to upwelling off the coast of Africa, contributing to the characteristic tri-pole pattern of the AMV.

We have now removed the previous model analyses and replaced it with a new analysis where the underlying $SST_{FW}$ pattern in summer is more clearly linked to the preceding freshwater anomalies. Consistent with the observation that the SST anomalies are largest in the subpolar region and decrease towards the south, it can be explained by the large-scale atmospheric feedbacks associated with freshwater anomalies. This includes a range of timescales, including that of the AMV.

However, we are very cautious in the wording about the role of freshwater in triggering these feedbacks. We refer to freshwater anomalies as predictor. While we discuss evidence in the conclusions that suggests freshwater may act as a trigger of the ocean-atmosphere feedbacks, rather than only a predictor, we clearly state the involved uncertainties.

**Minor comments**

Line 1 - The paper discusses here and in the introduction the possibility of long-term forced changes in the arctic having an impact on the mid-latitude weather, but then exclusively focuses on interannual variability? This left me slightly confused about what I was supposed to be taking from the analysis, so maybe it worth rethinking the motivation?

Thank you for pointing this out. We have now added an analysis of the long-term variability of freshwater and the implications. Thus, we find that there is a significant freshening trend over the last 70 years, superimposed on substantial interannual variability.

Line 63 - why not just use HadISST rather than merge the two datasets?

We did not manually merge the two datasets ourselves. The merged dataset is freely available and benefits from the optimal interpolation technique used in the NOAA data. We have now added the link in the data description to avoid misunderstandings.

Section 2.2 - how do these simulations differ from CMIP style AMIP runs (e.g., Eyring et al, 2016)?

The simulations were designed to facilitate comparison between different models, like those in Eyring et al. (2016). Thus, they have a pre-determined forcing to facilitate the comparison, which is now clarified. In this case, the simulations were performed with the prescribed, observed SST and sea ice cover and time varying greenhouse gases and ozone.

Section 2.3 - I am confused by the discussion of a trend, in part due to the framing of the paper on understanding the impact of long-term forced changes in the Arctic on the wider climate, but I think the point is that trends over land may alter the relationship with ocean SST anomalies? Please clarify.

The point of this analysis was to remove the effect of Greenhouse gases, not to remove trends, which is now clarified. Thus, we do not distinguish between trends over land and ocean. We have also adjusted the subsequent paper to the motivation in the introduction. Thus, we have added an analysis of the trends in the freshening and SST over the last 70 years.

Line 133 - You state that "we derive indices that exhibit a strong relationship to subpolar temperatures but not to the drivers of density anomalies", but then argue that the summer NAO drives changes in freshwater (which will drive the density anomalies) - can you clarify your point, please?

Thank you for pointing out that this was unclear. We referred to the drivers of density (or in turn temperature and freshwater) anomalies. We consider freshwater as state variable that in turn has drivers (like precipitation or runoff). We have now clarified this in the manuscript. In addition, we have added a section describing the physical links between the summer NAO and the freshwater anomalies in the subsequent winter.

Line 140 - you mention the potential impact for the summer NAO on freshwater, but won't it also affect the heat fluxes too? How do you take account of these in your analysis?

Yes, we do consider the surface fluxes in the mass balance analysis. The surface heat fluxes are included in the buoyancy fluxes, and we also show them separately. Surface heat fluxes can be a driver of SST anomalies or a response. In this case, we find that that the surface fluxes cannot explain the SST anomalies. Instead, the surface flux anomalies are positive over the cold anomaly regions, implying that the ocean loses less heat to the atmosphere, contributing to the increased baroclinic instability in the lower troposphere. We have stated this more clearly now in the revised manuscript.

Line 392 - "...we select all the years that lead to an increase in the slope of the regression line." - I'm uncomfortable about this - it sounds a bit like cherry picking to get a stronger signal (which in turn would affect your conclusions about how important freshwater changes are). Please could the authors justify this choice physically?

The objective of this method is to improve the signal-to-noise ratio between the index and freshwater anomalies. The strong signal between the index and the freshwater anomalies is a pre-requisite for the index to represent freshwater anomalies. We do not use it as the conclusion. We now explain the method more clearly and have integrated the appendices into the main text, making it easier to read.

Following your first suggestion, we have also added a description of the causes of freshwater anomalies associated with high and low NAO summers. Thus, we find that the subsampled indices are also more strongly linked to the physical causes of freshwater anomalies, supporting the subsampling with a dynamical explanation. Thank you for suggesting this!

Figure A1 - I can't help but to see that there is a long-term change in the summer NAO with most negative years occurring after 2000. This is consistent with the long-term trend in summer jet. The question this raises is how important is this longer term variability in the sNAO for your results? One thing that is missing is a time series of subpolar SST, which also shows significant decadal timescale variability. Please add this to figure A1 at least. However, following up on this, previous studies have linked these low frequency variability in the jet to the changes in the subpolar temperatures (e.g., Dong et al, 2016) - therefore, how certain are you that the changes in summer NAO are not reflecting decadal time-scale changes in the subpolar gyre?

We now show the timeseries of the SST pattern linked to freshwater events, including the cold anomaly pattern since it is this pattern that is relevant for European summer weather. While there has been significant decadal variability in the causes of freshwater anomalies in the past, the negative NAO picks out the high-frequency variations, associated with the interannual variability of runoff (see also our response to your second comment).

Line 405 - It's not clear to me why the geostrophic currents are not important - there is nothing in the 95% confidence lines that are related to constant lines of density. Please can you clarify your reasoning here?

Geostrophic currents are, by definition, along lines of constant density. Thus, they cannot lead to an increase or a decrease of density. Importantly, they can, and do, drive freshwater and temperature anomalies. However, the density anomalies associated with freshwater and temperature changes need to compensate each other in geostrophically balanced blows. We have clarified this in the manuscript.

Section A3 - it is not clear why you are regressing the JFM heat fluxes onto your summer NAO? In order to rule out the impact of surface heat fluxes on the JFM SSTs it is the integrated heat fluxes between the summer and JFM that are important?

Thank you for pointing out that this was unclear. We are examining the SST signal in winter, which is typically not influenced by the surface fluxes of the preceding summer because the variability of the surface fluxes in winter is much larger than in summer. To be sure, we have repeated the surface mass balance for the period from summer to winter and found that the results did not change appreciably. This is now stated more clearly in the revised manuscript.

417 - the mean mixed layer is the annual mean mixed layer?

Since we are investigating the SST anomalies in winter, we used the mean mixed layer in winter. As the Reviewer 1 pointed out, however, we should have used the mixed layer depth at the end of the winter. Thus, we have corrected this now.

Line 440 - Please elaborate on the origin of the uncertainty numbers.

Thank you for pointing out that this was not sufficiently clear. The uncertainty originates from neglecting the terms on the righthand side of the mass balance equation. It is obtained by comparing the size of these terms to the size of the terms on the lefthand side. We now devote a full section in the main text to the derivation of freshwater anomalies and their uncertainties.

Figure 3 - what is the data used to plot the SSS anomalies, or are they implied from your mass balance analysis?

Yes, the estimates were obtained from the mass balance analysis. This is now clarified in the revised manuscript. Thank you for pointing out that this was unclear.

Section 4.2 - I think when you mean first summer, you mean the same summer as the summer NAO event? However, it was not clear as you're talking about looking in subsequent summers which I took to be after the winter - please clarify. This is particularly important for figure 6, which is unclear what summers you are looking at.

No, we refer to the European weather in the summer one year after the NAO index. It is not the same summer. This is why the summer NAO can be a such a valuable predictor of European summer weather, one year in advance. We have now clarified this.

Figure 7 - Are you computing the variance explained using all years, or are you only focusing on the years that you have large changes? Either way, how important is the long-term trend in atmospheric circulation shown in figure 2c for your analysis?

The variance is computed over all the years for which the indices are defined. In addition, we have added an analysis of the long-term trends over 70 years. Thus, we find that the NAO mostly picks up the interannual variability. Considering that the autocorrelation of the indices is nearly zero at one-year lag, we do not expect the long-term trend to substantially affect the explained variances. This is now clarified.

Figure 9 - Is the only difference between these figures and that shown in figure 8 (d and f) that they are split across model ensembles? Not clear to me why this is relevant - could you just include the ensemble spread in figure 8 ?

The main point of the figure was to show the spread across the ensemble members and how they compare with the observations. However, we have removed this figure now.

We strongly thank the reviewer for carefully reading and reviewing our manuscript and providing so many detailed and constructive comments and suggestions. Your comments have helped us to clarify and improve the manuscript!

References:

Bamber, J. L., Tedstone, A. J., King, M. D., Howat, I. M., Enderlin, E. M., Van Den Broeke, M. R., & Noel, B. (2018). Land ice freshwater budget of the Arctic and North Atlantic Oceans: 1. Data, methods, and results. *Journal of Geophysical Research: Oceans*, *123*(3), 1827-1837.

Dukhovskoy, D. S., Yashayaev, I., Proshutinsky, A., Bamber, J. L., Bashmachnikov, I. L., Chassignet, E. P., ... & Tedstone, A. J. (2019). Role of Greenland freshwater anomaly in the recent freshening of the subpolar North Atlantic. *Journal of Geophysical Research: Oceans*, *124*(5), 3333-3360.

Hakkinen, S., & Rhines, P. B. (2009). Shifting surface currents in the northern North Atlantic Ocean. *Journal of Geophysical Research: Oceans*, *114*(C4).

Häkkinen, S., Rhines, P. B., & Worthen, D. L. (2011). Warm and saline events embedded in the meridional circulation of the northern North Atlantic. *Journal of Geophysical Research: Oceans*, *116*(C3).

Häkkinen, S., Rhines, P. B., & Worthen, D. L. (2013). Northern North Atlantic sea surface height and ocean heat content variability. *Journal of Geophysical Research: Oceans*, *118*(7), 3670-3678.

Holliday, N. P., Bersch, M., Berx, B., Chafik, L., Cunningham, S., Florindo-López, C., ... & Yashayaev, I. (2020). Ocean circulation causes the largest freshening event for 120 years in eastern subpolar North Atlantic. *Nature communications*, *11*(1), 1-15.

Proshutinsky, A., Dukhovskoy, D., Timmermans, M. L., Krishfield, R., & Bamber, J. L. (2015). Arctic circulation regimes. *Philosophical Transactions of the Royal Society A: Mathematical, Physical and Engineering Sciences*, *373*(2052), 20140160.

---

## Author Response (AR1)

Dear Prof. Pfahl,
Dear Reviewers,

We are pleased to submit a revised version of the manuscript "European summer weather linked to North Atlantic freshwater anomalies in preceding years". Based on the detailed feedback provided by both reviewers, we have clarified and improved the manuscript.

Both reviews were helpful, thorough, and constructive throughout. The main comments were related to the placement of important information in the appendices, an uncertainty about the physical causes of freshwater anomalies, and a lack of clarity regarding the actual role of freshwater anomalies for European summer weather. To address these comments, we have implemented the following main changes:

- We have shifted the previous Appendices A1 and A2 into the approach section, and Appendices A3 and A4 into the results section. The new appendix now only contains the calculations and the comparison with hydrographic observations. All critical information is provided in the main text.

- We have added an analysis of the causes of freshwater anomalies, which supports earlier studies. While a detailed examination of the freshwater budget and origin of the freshwater anomalies are beyond the scope of this study, the analysis provides a physical explanation for having two different types of freshwater anomalies linked to opposite atmospheric circulation patterns in the preceding summer.

- We are explicit about the involved timescales. The focus of this study is on interannual timescales, reflected in low autocorrelations of the freshwater indices and the summer NAO. Nevertheless, freshwater variations are also linked to lower-frequency variations of the North Atlantic SST. Thus, we mention that periodic freshwater releases from the Arctic may have contributed to Atlantic Multidecadal Variability. Also, we show that the freshening has a trend over the last 70 years, and we discuss the implications.

- We have added a clearer analysis of the involved air-sea coupling processes in winter and summer. Given the importance of atmospheric feedbacks in reinforcing the SST anomaly, we do not attribute the anomalies in European summer weather to freshwater variations alone. Considering the spatial and temporal characteristics of the characteristic SST pattern, and the lag of one year, we propose in the discussion that enhanced freshening can act as a trigger of subsequent, large-scale air-sea feedbacks, which in turn affect European summer weather. However, we are cautious about the wording and clearly state the involved uncertainties.

We further provided more details and explanations, removed any ambiguities, and unclear or inconsistent terminologies. Detailed changes are explained in the response letters, where the responses to the reviewers' comments are shown in blue, and the resulting changes in the manuscript are shown in red. Figure and line numbers refer to the revised manuscript without tracked changes:

- Reviewer 1: page 2
- Reviewer 2: page 15

We think the manuscript has greatly benefitted from the review, and we are confident that is now clarified, easier to read, and a worthwhile contribution to Weather and Climate Dynamics. We thank both reviewers and the editor for their efforts in reviewing and handling this manuscript.

Sincerely,

Marilena Oltmanns, on behalf of all authors

**Responses to Reviewer 1**

The authors use statistical analysis of observations and reanalysis data to support their hypothesis that warmer and drier summer weather in Europe can be linked to freshwater anomalies in the North Atlantic subpolar gyre region during the preceding year. The proposed mechanism for this link is a northward shift of the North Atlantic current leading to a similar deflection of the jet stream and therefore altering the advection pathway of maritime air masses. The foundation of the analysis are freshwater indices derived from a mass balance equation that are used to identify freshwater anomalies in relation to simultaneous sea surface temperature (SST) anomalies linked to the North Atlantic Oscillation (NAO).

I understand that this is a re-submission of an earlier version of the manuscript, but I was not involved in the previous review process. Therefore, I cannot assess how the manuscript has been improved, but rather provide a fresh pair of eyes.

We sincerely thank the reviewer for providing a fresh pair of eyes and reviewing our manuscript. The review was extremely thorough and detailed. Moreover, the comments and suggestions were constructive and have helped us to improve the manuscript.

**I. General Comments and Suggestions:**

1. One of central results of this study is the description of "a coherent, deterministic mechanism that links North Atlantic freshwater events to European summer weather" (l. 315-316). However, the actual role of the identified freshwater anomalies in the subpolar gyre remains unclear. Given the lack of salinity observations, SST anomalies in relationship to the NAO are used as proxy for freshwater anomalies. In turn, a substantial part of the described link between the freshwater anomalies and European summer weather is based on the enhanced meridional SST gradient between the subpolar and subtropical gyre, and its influence on the storm track over the North Atlantic. This raises the question to what extent the freshwater anomalies actually influence the proposed mechanism and the downstream response?

Thank you for asking about the role of the freshwater anomalies. In the previous version, we only showed that freshwater is statistically linked with subsequent European summer weather and proposed a dynamical explanation. In the revised manuscript, we are more specific about the role of freshwater and use a more cautious narrative. Therefore, we have implemented four main changes:

(1) We have included a description of the drivers of freshwater anomalies (Section 4.2). Thus, we provide a physical explanation for the start of the chain of feedbacks. Since the identified drivers (runoff for $F_E$ events and circulation changes for $F_W$ events) are not otherwise linked to European summer weather on the investigated timescales, and occur exactly one year in advance, they challenge the idea that an unknown third mechanism drives both freshwater anomalies and warmer European summers without the two being physically connected.

(2) We are explicit about the involved timescales of variability. While the focus is on interannual timescales (line 296), we now show that surface freshening has a trend (Fig. 9a of the revised manuscript) that is superimposed on substantial interannual variability. The interannual variability is correlated with the variability of the summer NAO and runoff from the preceding year, obtained from the Greenland climate model MAR. (Section 4.6). After considering alternative drivers acting on these timescales, we find (line 405):

"Apart from seasonal runoff- and melt-driven freshening, there are currently no conceivable, physical mechanisms in the tropics, stratosphere or outside the North Atlantic which, at the same time, have a significant trend over the last 70 years, exhibit a similarly high interannual variability, can explain the occurrence of freshwater anomalies in the subpolar region in the subsequent winter, and are significantly correlated with the characteristic summer SST pattern a year before it occurs. This suggests that seasonal freshening may not only be a predictor of the SST pattern but also a potential trigger."

(3) To assess whether runoff is a trigger of the SST pattern, rather than only a predictor, we have included an analysis of the ocean-atmosphere feedbacks that contribute to the evolution of the SST pattern in summer (Section 4.7). After investigating the surface heat and momentum fluxes and their influences on the ocean and atmosphere with ERA5, remote sensing data, in-situ hydrographic observations from the cold anomaly region, and with the models, we find that wind-driven transports may contribute to the intensification of the SST signal. Thus, we conclude (line 453):

"The atmospheric forcing contributes to the development of the SST field but is in turn, forced by it. On the one hand, the consistency of the spatial SST pattern with the wind-driven transports, the intensification of the SST signal in mid-summer, and the vertical extent of the hydrographic anomaly point to the relevance of the atmospheric forcing in driving the SST signal over the North Atlantic. On the other hand, model simulations, forced with the prescribed, observed SST, reveal the importance of the SST field for the large-scale atmospheric circulation, including for European summer weather. The underlying SST pattern covers the entire North Atlantic with the strongest signal occurring in the subpolar region (Fig. 11b). While further studies are necessary to confirm the dynamical contribution of freshwater anomalies to the large-scale SST pattern, its spatial and temporal characteristics, and the time lag of one year, indicate that enhanced surface freshening from the preceding year may have initiated the chain of air-sea feedbacks."

(4) We are more cautious about the wording. Throughout the analyses, we refer to freshwater as a predictor (e.g. line 350). In the conclusion, we discuss evidence that points to freshwater as a trigger rather than only a predictor, but we are explicit about the involved uncertainties, and we have removed phrases that may previously have caused confusion.

2. At times, it is difficult to follow the analysis which might be in part due to the overall structure of the manuscript and lack of some details in the text (often they are only mentioned in figure captions). Additionally, some of the terminology is unclear or inconsistent throughout the manuscript which is possibly an artifact from the refactoring of the previous version. Hopefully, the comments below will help to streamline the text and make it more accessible for the reader.

Thank you for providing very specific comments below on the exact places in the manuscript that have been unclear. We have now clarified each of these instances and took care to include all details in the text. In addition, we have removed all inconsistencies in terminology that you have spotted.

3. Some of the figures are hard to read as individual panels are small or details are obscured. For most maps, the colorbars and axis labels take up valuable "real estate". I suggest to decrease their size and use the free white space to increase the maps wherever possible. Furthermore, I recommend to decrease the thickness of the coastlines since they can be quite distracting, especially on maps with vectors. It might also be worthwhile to mention differences in color scales in the caption wherever it can help guide the reader (e.g., Figure 2).

Thank you for these suggestions. We have now slightly reduced the size of the axis labels and font sizes around the colour bars. In addition, we have increased the map sizes for specific figures further to the south to ensure that all relevant signals are included in the maps. We further decreased the thickness of the coastline in all figures, so the vectors are clearly visible. We also mention differences in colour scales wherever they occur.

**II. Main Comments:**

1. Section 3/Appendix A: It took me a long time, including going back and forth between the main text and appendix to follow the approach. Given the importance of the freshwater indices as foundation for the subsequent analysis, I suggest to combine Sections 3.1, 3.2, with Appendices A1, and A2 to describe the derivation in the main text including the clarification of the following points:

Thank you for indicating that the derivation was difficult to follow. We have now combined Appendices A1 and A2 with Sections 3.1 and 3.2.

    1.   l. 126: Please state this equation.

    The equation is now fully stated.

    2.   l. 129: I think that M is not the same "downward mixing", but rather entrainment of water masses below the pycnocline into the surface ocean mixed layer as a result of a deepening mixed layer. In the context of this sentence, you refer to increased stratification due to large freshwater anomalies that inhibits a deepening of the mixed layer due to convection.

    We are now more precise and specify "entrainment".

    3.   l. 135-137: This sentence becomes only understandable after reading the appendix.

    We have removed this sentence.

    4.   Section 3.2/Appendix A2: The whole derivation of the freshwater indices is based on the NAO. I can't help but notice the striking similarity between the spatial pattern of the cold anomaly in the subpolar gyre for $F_E$ and regression pattern of SST anomalies on the North Atlantic SST Index of the Atlantic Multidecadal Oscillation (AMO) in that region (see Fig. 11 in Deser et al., 2010). The AMO has been in its warm phase since the mid-1990s and thus during the time of most of the $F_E$ years. This raises the question to what extent longer-term climate variability influences the relationship between the NAO and freshwater anomalies and if this can be utilized in the design of the freshwater indices?

Thank you for this suggestion. We did indeed attempt to divide the full period in high and low AMO phases. However, this did not turn out to be useful. We found that the summer NAO already filters out the interannual variability of freshwater and that the targeted approach of subsampling was more effective in optimising the indices.

In a related study, we find that the trend of the cold anomaly over the last 70 years has recently overtaken the AMO signal and now has a larger amplitude than the AMO. This has far-reaching implications for North Atlantic climate variability, including a shift of power towards interannual timescales.

While a detailed analysis of the AMO signal is beyond the scope of this manuscript, we now mention its potential relationship to freshwater anomalies in the revised manuscript through periodic Arctic freshwater releases (line 293). Overall, we find that freshwater can explain the variability of the cold anomaly pattern on a range of timescales, including that of the AMO, and including the cold anomaly trend (Section 4.6), due to the different causes of freshwater anomalies. Thus, it may not be desirable to completely filter out these signals.

    5.   l. 152-153: How do you estimate the correlation between the index and freshwater anomalies and how well the index represents the initial freshwater anomalies if they "are not known a

priori" (cf. l. 148-149)? I think you are referring to SST anomalies which serve as proxies for the freshwater anomalies.

Yes, you are right. We present the correlation together with the uncertainty since both estimates belong together. We first estimate the freshwater anomalies from the SST anomalies and obtain an uncertainty of 4% and 7% respectively. We then calculate the correlation between the estimated freshwater anomalies and the NAO index.

This is now clarified in Section 4.1 (line 241) and the caption of Figure 3. We have also removed the sentence you are referring to, since it was confusing.

6. l. 156: Please define $F_E$ and $F_W$ explicitly. Without going through the appendix, the reader might ask themselves why there are two indices? What do the subscripts refer to? How are the two related?

Since we have removed the appendices, all information about $F_E$ and $F_W$ is now included in the main text (Sections 3.2, 4.1 and 4.2). Thank you for suggesting this.

7. l. 160: Please refer to Appendix A3 to show how these uncertainty estimates were obtained.

We have now shifted Appendix A3 and A4 into the results section (Sections 4.1 and 4.2). Thus, Section 4.1 now includes a detailed explanation of the freshwater anomalies, their uncertainty estimates, and how they were obtained.

8. l. 161-162 & Figure 3: I think Figure 3 deserves more prominence in the text as these are the actual freshwater anomalies your hypothesis is based on. I suggest to move this sentence into its own paragraph and add more details, e.g., by being explicit that the shown salinity anomalies are estimate based on the surface mass balance (I think?), how you obtained the relationships, and what the white areas represent. Please define SSS.

Thank you for pointing out the importance of Figure 3. We have now shifted it into its own section in the results and clearly explain it (Section 4.1). We also define SSS as the sea surface salinity (line 234).

9. Figure 2: Panels (a) and (b) should be the same as Panels (a) and (e) in Figure A1, but the structure of the largest values looks different. Is this just because of the differences in the color scale?

Yes, we had adjusted the colour scale in Figure A1e to make it the same as that of Figure A1b. However, we agree that the choice of the colour scale concealed the structure of the largest values. After combining the appendix with the approach section, we have removed this figure.

10. Equation (A2): Please define $\rho_0$.

We now define it as the density before the winter, for instance September or October, with dt referring to the resulting time interval of integration. We also explain in Section 4.1 that the results are not sensitive to the exact starting point if the mixed layer is still relatively shallow (as in September or October).

11. l. 350: Strictly speaking, given Equation (A2) is the result of an integral over time, $h_n$ is the mixed layer at the end of winter.

Importantly, the time of the mixed layer depth ($h_n$) must be the same as that of the temperature term ($T_n$). While it is possible to use March only – and the results do not change appreciably – we consider the full winter period for increased robustness. In some months, the maximum mixed layer depth may already be reached in February.

You are correct that Eq. (2) [which is now Eq. (4)] is an integral over the time. Here, we integrate from an arbitrary time before the winter (for instance September or October, when the mixed layers are still shallow) to the winter period (January to March), with dt referring to the corresponding time of integration. We now clearly state these periods (line 133), motivate the choices and explain that the results are not sensitive to them (line 226, line 580, line 585).

12. l. 359-361: How realistic are these conditions and at what timescales do you expect this assumption to hold?

At this location in the text, it is only a motivation. It is an objective of the approach to achieve these conditions. We do not make any assumptions. We have now rephrased the sentence for clarification.

After evaluating the mass balances, we find that (on interannual timescales) these conditions always hold within a reasonable uncertainty range. Even without any further subsampling of the years in which $NAO_s > -0.5$, the results would still hold with an uncertainty of ~10%. Since this is an important result, we have now added (line 248):

"The result implies a close connection between freshwater and SST anomalies. A demonstration of this connection with hydrographic observations shows that, even in winters of most intense air-sea fluxes, it is possible to infer freshwater anomalies from the SST with a reasonably small uncertainty that is below that of currently available satellite products (Appendix A)."

We now also show that the results hold for longer timescales, specifically the trend over the last 70 years (Section 4.6). However, we find that the resulting uncertainties are larger for longer timescales.

13. l. 369 & l. 371: "lower NAO index" and "higher NAO index" – do these refer to the magnitude and/or phase of the NAO?

They refer to the phase. To clarify this, we have replaced all instances of "NAO index" by "NAO state" or "phase".

14. l. 375-377: It would be helpful for the reader if you add a sentence how the relationship was obtained. This is partially described in the caption of Figure A1, which makes it more difficult to follow the arguments in the text.

Thank you for pointing this out. We have now rephrased this sentence to clarify that the relationship was not obtained in that we did not take any steps or created it ourselves. Instead, we observe this relationship from visual inspection of the scatter plot.

15. l. 377: How did you determine the threshold?

We determined it from visual inspection of the relationship between the summer NAO meridional SST gradient. We have now rewritten this paragraph to clarify this (line 184).

16. l. 377-378: I think it is important to also mention the significant positive SST anomaly in the subtropical gyre/western North Atlantic (Figure A1a) which has a substantial contribution to ∆SST for NAO < -0.5.

Thank you for pointing this out. It is indeed important, and we now mention it at the location you indicated. We also explain the development of the warm anomaly through air-sea coupling in more detail in Section 4.3. Overall, we are now more explicit about the role of large-scale atmospheric feedbacks contributing to the SST signal beyond the subpolar region.

17. l. 380-400: The description of the optimization process is unclear. I understand the rationale of increasing the signal-to-noise ratio, however, the selection of included years appears very subjective. How did you choose the number of years to include in the index? How did you select the discarded years? What about the two outlier years?

Thank you for indicating that the optimisation process was insufficiently described. As you correctly point out, the objective of the process is to increase the signal-to-noise ratio.

To better describe the process, we now explain (line 196): "There is a trade-off between the number of years included and the resulting correlation. Here, we selected N=17 years as a reasonable compromise for obtaining a high correlation of 0.90 while keeping a relatively large sample size, reflected in low p-values ($p < 2.6 \times 10^{-6}$). However, the results are not sensitive to this choice."

To add further support for the subsampling, we have included a new section in the results that links the two sub-sampled indices to physical causes of freshwater anomalies, supporting the optimisation process with a physical explanation (Section 4.2).

The two outlier years correspond to years where the NAO index was not a useful indicator for the fresh and cold anomalies. However, upon investigating both years more closely and carrying out a mass balance for both years, we found that the relationship between the salinity and temperature anomalies still held. The discrepancy between the NAO index and these two freshwater anomalies can be explained by a superposition of both, enhanced runoff and changes in the circulation. Thus, the NAO index underestimated the freshwater anomalies. Since we think that these analyses did not add new information to the manuscript, we have not included them in the manuscript.

18. l. 387: Please define the SST gradient in the text.

We now define the SST gradient in the text and explain that it refers to the difference between the subtropical warm and subpolar cold anomaly regions, enclosed within the 95% confidence lines.

We further explain (line 190): "Here, we specifically used the 95% confidence regions as a means to directly inspect the robustness of the correlations and ensure that they are not due to outliers or clusters. Another advantage of using spatial differences is that we filter out any potential, spatially uniform, radiative warming signals. However, the identified relationships are not sensitive to the exact regions."

19. l. 388-389: It is unclear what you mean by "spatial gradients are more robust to local variations in the surface fluxes". What if the spatial gradient is the result of local heat flux variations as one might expect from the response to the NAO (e.g., Cayan, 1992; Marshall et al., 2001; Deser et al., 2010).

Thank you for pointing out that the sentence was unclear. By using spatial gradients, we filter out the uniform warming effect of increasing greenhouse gases. Another advantage of using spatial gradients is that the area, used for the derivation of the indices is based on overall larger areas and therefore less sensitive to regional fluctuations in surface currents or fluxes.

We now state this in the manuscript, and we also mention that the results are not sensitive to the exact regions (line 191).

The surface fluxes were evaluated as part of the mass balance. They do not substantially contribute to the SST patterns (Figs. A1d and A2d). We now explicitly state this in the manuscript (line 336).

20. Figure A1: It would be more intuitive and consistent with the text if you wrote NAOS < -0.5 in the title of Panel (a) and in the caption. Do you include the significant cold tongue off western Africa in the calculation of ΔSST?

Thank you for pointing out that the title of the figure was confusing. We have now removed the figure.

The cold tongue can be understood as feedback since the associated large-scale atmospheric circulation anomaly induces upwelling off western Africa. However, we did not include the cold tongue off western Africa in the calculation of the SST gradient.

We have now clarified that we only use the subpolar cold anomaly in the text (line 175, line 188) and in the figure caption (Figure 2). Moreover, throughout the manuscript we are more explicit about the atmospheric feedbacks contributing to the large-scale SST pattern in winter (Section 4.3) and we have added a new description of these feedbacks in summer (Section 4.7). Thus, we also mention the enhanced upwelling off the coast of Africa (line 348).

2. Section 4.1:
    1. The circulation anomaly you describe (Figure 4b) is reminiscent of the positive NAO phase for which the atmospheric variability patterns and corresponding ocean response are known (e.g., Cayan, 1992; Marshall et al., 2001), and are in line with your findings. It would make this section stronger if you make an explicit link of your results to the winter NAO.

Thank you for suggesting this. Following your suggestion, we now make an explicit link to the winter NAO (line 318). We also mention that this signal implies a switch of the NAO sign from more negative in summer to more positive in winter after strong $F_E$ freshwater anomalies.

    2. Changes in the wind field associated with the NAO not only change the Ekman transport as you discuss (l. 198), but also lead to changes in latent and sensible heat fluxes. Can you elaborate to what extend these changes in air-sea heat fluxes are important for creating and maintaining the meridional SST gradient?

We now state that we do not find any significant impact of the surface heat fluxes on maintaining the SST gradient, neither in winter (line 336, Fig. A1) nor in summer (line 426, Fig. C1). In the mass balance analysis, moreover, the surface fluxes were found to be too weak to contribute to the cold anomaly. However, wind-driven Ekman transport can, and do contribute to the large-scale SST patterns by setting up large-scale surface pressure gradients, consistent with earlier studies (e.g., Marshall et al. 2001; Zhao and Johns, 2014). This is now clarified in the description of the air-sea feedbacks, both winter (Section 4.3) and in summer (Section 4.7).

    3. l. 200: It is unclear why you bring in the second winter. A short motivation will help to keep the reader on board.

Thank you very much for suggesting this. We now motivate the investigation of the second winter with the climatic repercussions of North Atlantic Current shifts (line 323).

    4. l. 210-213: From the presented figures, I cannot see a northward shift of the North Atlantic Current during the first winter (I think it shows up nicely int Figure 4d for the second winter). Is it possible that different timescales between heat fluxes and Ekman transport can explain the differences between the first and second winter? A SST gradient which is set up in the first

winter and shifted northward during the second year seems also more in line with the summer SST pattern that you describe in Section 4.2 (l. 227-229).

Thank you very much for pointing out that this was unclear.

The Ekman transports are an instantaneous response. They already lead to a warm anomaly in the first winter after the freshwater index. However, the northward current shift is only visible in the warm anomaly to the south of the cold anomaly (the Gulf Stream deflection point). It does not extend across to the east coast.

We now clarified (line 332): "The northward shift of the North Atlantic Current implies a large-scale warm anomaly to the south of the subpolar cold anomaly, not because the water inside the current is anomalously warm but because the current occurs at an anomalously northward location. It is seen in the first, and the second winter after freshwater anomalies (Fig. 2a and 5d). However, in the first winter, the northward shift is obscured by the wind-driven, southward expansion of the cold anomaly over the eastern North Atlantic, potentially driving enhanced mixing and erosion of the SST front."

3.  Section 4.2: This section seems rather short given that it addresses one of the main results of the study. It would be helpful for the reader if you add more details and clarify the following points:

Thank you for all your suggestions! Following your suggestions below, we have now expanded the section and clarified these points.

1.  l. 231: "more northerly location" compared to what?

"More northerly compared to the previous summer". This is now clarified.

2.  l. 237, 238, 241: "cold anomaly" in the ocean or atmosphere?

We referred to the negative SST anomaly. This is now clarified.

3.  l. 240: "over Europe" is rather vague (e.g., the warm and dry anomalies (Figures 6c and d) occur in different regions). See also next comment.

Following your comment below, we are now more specific (line 367).

4.  l. 241: Is it actually true that "the overall patterns are similar after $F_E$ and $F_W$ freshwater anomalies"? The significant air temperature anomalies one year after $F_E$ extend across the Iberian Peninsula all the way to northern Africa while they are more centered around over France and Great Britain after $F_W$. Similar the the dry anomaly occurs over the Alps and eastern Europe during the first summer after $F_E$, but more over Baltic region after $F_W$ which is more similar to second summer response after $F_E$.

Thank you for suggesting this. We are now more specific and say that the mechanism is the same, but it occurs over a different region, consistent with the underlying SST anomalies. We now exactly specify the regions in the text (line 367).

5.  It seems like that patterns after $F_W$ are one order of magnitude smaller compared to the patterns after $F_E$. Is this an artifact of the smaller correlation in the construction of the freshwater indices or is it due to the stronger meridional SST gradient that exists in the $F_E$ subset with significant positive SST anomalies in the subtropical gyre region?

Yes, you are right that the magnitudes of the obtained signals in European summer weather are carried over from the larger magnitudes in the freshwater and SST signals. The large magnitudes result from steep regression slopes. These steep regression slopes occur because the underlying changes in the $F_E$ index are much smaller compared to the changes in the $F_W$ index. We now clarified (line 370):

"The regressions of the SST and atmospheric circulation on $F_W$ are weaker compared to those on $F_E$ consistent with weaker freshwater anomalies (Fig. 3) and smaller regression slopes (Fig. 2d). Physically, the higher regressions on the $F_E$ subsets imply a higher sensitivity of the ocean and atmospheric conditions to $F_E$ freshwater anomalies. Once the seasonal freshening exceeded a critical threshold (corresponding to a threshold of ~0.5 in the $NAO_S$ index), a relatively small further increase was linked to substantially warmer and drier summers."

This high sensitivity can have severe implications for European summer weather, as the associated, critical threshold in the seasonal freshening may be exceeded more frequently. Thus, we now also refer to this high sensitivity in the implications for predictability (line 486) and in the discussion (line 526).

6. Is there a reason why you show the zonal wind at 700 hPa for the $F_E$ subset and the meridional component for the $F_W$ subset?

To best show the link between the SST and the wind field, we selected the wind component that best matches the shape of the underlying SST pattern.

For consistency across the analyses, and since we specifically emphasise the northward deflection, we now only show the meridional component in all analyses (Figs. 6, 7, 8 and 10).

4. Sections 4.4 and 4.5.: I have to admit that I got lost here. In general, I am wondering whether the analysis of the model simulations adds any additional information that warrants its inclusion in the manuscript.

Following your suggestion, we have removed the previous model analysis. We have now replaced it by another analysis in which the link between the selected SST anomalies for the model analysis and the freshwater anomalies is clearer.

Since we have removed the analysis, the comments below no longer apply.

1. l. 271: It is not clear which pattern ($F_E$ or $F_W$) you project the on.
2. l. 273: Most of the analysis in Sections 4.1 and 4.2 is focused on the first summer after both $F_E$ and $F_W$ years with only a brief discussion of the second summer after $F_E$. It is unclear why you construct a new index for the analysis of the model simulations based on the SST pattern in the second summer.
3. l. 289: cold anomalies in the ocean or atmosphere?
4. l. 294-295: Given your derivation of the freshwater indices using the surface mass balance, any cold anomaly coincides with a freshwater anomaly, by construction. Your analysis of the observations points out the importance of the the meridional SST gradient and its influence on the position of the jetstream. This raises the question whether the freshwater anomalies are just side effect of the mechanism that sets up the SST gradient. It is unclear to me how the model simulations help to answer this question.

In the new analyses, we examine the link between the freshwater anomaly in winter and the SST anomaly in summer more thoroughly. While the freshwater anomalies can explain the initial trigger of the chain of events, the final SST anomalies in summer can only be understood as the result of

large-scale air-sea coupling processes. Thus, we are more cautious in the wording in the revised manuscript.

We further clarified the meaning of the previous Section 4.5, which includes the composite analysis of warm European summers. Thus, we now explain (line 389):

"The analysis of Europe's warmest and coldest summers supports the statistical link between freshwater anomalies and European summer weather. It demonstrates that the link is robust to the analysis technique and independent of the indices. While the regressions on the freshwater indices showed that freshwater anomalies can constrain the variability of the subsequent European summer weather, the composites additionally show that, on interannual timescales, Europe's largest temperature anomalies were preceded by freshwater anomalies. This indicates that enhanced freshwater anomalies are not only a sufficient but also a necessary condition for warmer European summers."

**III. Additional Comments and Suggestions:**

1. l. 37, l. 83, l. 84: It would be more appropriate to use "grid spacing" instead of resolution (e.g., Grasso 2000).

Thank you very much for pointing this out. We have corrected this.

2. l. 44: It's not just cold air, but also stronger winds that increase heat fluxes.

This is true but in the preceding sentence we explain that the air is always colder than the ocean in winter. This naturally implies a mean climatological ocean heat loss. We do not exclude that stronger winds also increase the surface fluxes. They are just not relevant in the context of this paragraph.

We have now removed the sentence since it raised confusion.

3. l. 46: Please summarize the conditions here or refer to the derivation of the freshwater indices.

We now state the conditions.

4. Section 2.1: Please add details about grid spacing, temporal resolution, and any processing (e.g., calculation of anomalies, spatial interpolation, etc.). This would help make the study more reproducible. It might also be worthwhile to specify in this section which months you refer to by "summer" and "winter" throughout the text, especially since they are different from the standard definitions June-August (JJA) and December-February (DJF), respectively.

Thank you for these suggestions. Grid spacing, resolution, and processing steps are now all included in the data section.

We still state the definitions of the seasons where they occur, and we motivate these definitions. For instance, we use the NAO in July and August because runoff is largest in these months. In the analysis of European summer weather, we consider the full period from May through to August (Fig. 6) because we include the full periods over which we observe strong signal. While the exact periods may thus differ, we make sure that we always clearly state the periods.

5. l. 63: How did you combine the two datasets given their different temporal and spatial resolutions?

We did not combine them ourselves, but we downloaded the merged dataset from NCAR. This is now clarified by including the link to the datafile in the data section.

6. l. 94-97: This sentence is unclear. I do not understand why warm anomalies due to shift in the jet stream "must" be balanced by a cold anomaly elsewhere.

In contrast to greenhouse gas warming, freshwater-linked temperature anomalies do not result in a net imbalance in the Earth's surface energy budget. The warming over Europe is balanced by a cooling over the ocean since the underlying baroclinic wave activity consists of an anticyclonic anomaly on one side of the jet stream, and a cyclonic anomaly on the other side.

Thank you for pointing out that this was unclear. We have clarified this in the revised manuscript (line 109).

7. Here are a few wordings that are either inconsistent or remnants of the previous version of the manuscript:

Many thanks for spotting the below inconsistencies! We have now clarified all instances.

1. l. 185: "in winters after stronger freshwater anomalies" – based on the construction of your freshwater indices, the anomalies should occur during winter.

We have removed this sentence.

2. l. 211, 271, 316, 498, 499: what are "freshwater events"?

We have replaced "freshwater events" by "freshwater anomalies" everywhere.

3. l. 273, l. 419: What are "melt-driven" or "melt-induced" events? How are they connected to $F_E$ and $F_W$?

Thank you for spotting this. We have removed the terms "melt-driven" and "melt-induced" at both locations. Also, we now added a section that explains the term.

4. l. 421: What are "circulation-induced freshwater events"?

We have removed the term.

8. Figure 4: I suggest to mask out the Ekman transport vectors over land. This would make it more intuitive that they refer to an ocean variable

Indeed, thank you very much. We have now masked them out.

9. l. 253: Please add a reference for the statement that "most current coupled global climate models have large freshwater biases".

We have now removed the statement at this location since we are already making it in the introduction (line 37) and the discussion (line 528). At both locations, we provide the associated references.

10. l. 408: Do you integrate the wind stress or resulting Ekman transport over the winter period?

We integrated the Ekman transports over the winter period, after computing it from the wind stresses. This is now clarified.

11. l. 429: In l. 412-413, you define the heat flux (Q) as positive downward. A positive surface buoyancy flux (B) anomaly means Q needs to be positive (unless it its overcompensated by the freshwater flux), i.e., the ocean gains heat.

Yes, this is true for the anomalies. However, in the mean state in winter, the subpolar ocean loses heat. Thus, we considered it more appropriate to state "the ocean loses less heat" instead of "the ocean gains heat".

We have now removed the sentence because the anomaly in the surface heat fluxes is very small, and the results are not decisive. The sign of the integrated surface heat fluxes is more sensitive to the exact region compared to the buoyancy fluxes. However, the magnitude of both types of fluxes remains negligible, independent of the exact region.

The observation that the signs of B and Q do not exactly match is not due to the P-E term but due to the dependence of B on alpha, which depends on the skin temperature.

12. l. 439-440: What is the uncertainty in the freshwater fluxes due to the constant mixed layer depth used in your analysis?

We have carefully revised the mass balance and now explicitly account for a variable mixed layer depth (Section 3.1 and 4.1). This does not affect the results.

13. Figure A4: It is unclear whether these are composites just for the winters before the warmest summers or also the difference with the coldest summers.

For consistency with the heat wave composites, the composites refer to the difference between the ten warmest and the ten coldest summers. The surface mass balance was also carried out for the differences between the ten warmest and coldest, which is now clarified.

14. l. 498: This goes back to my first general comment (I.1.): If the SST pattern drives the observed atmospheric response, what is the role of the freshwater anomalies?

Thank you for asking about the role of freshwater anomalies for European summer weather.

We have now included further analyses and are more explicit about the role of the freshwater anomalies. Given the importance of ocean-atmosphere feedbacks in the evolution and intensification of the associated SST signals, we do not attribute the anomalies in European summer weather to freshwater variations alone. Instead, considering the spatial and temporal characteristics of the characteristic SST signals, we propose in the discussion that enhanced freshening may act as a trigger of subsequent, large-scale air-sea feedbacks, which in turn affect European summer weather. Please see first our response to your first comment for more details.

**IV. Typos/Wording:**

l. 53, 301: "ocean atmosphere" to "ocean-atmosphere"

Thank you for spotting this. We have now corrected it.

l. 273: "over the central North Atlantic" to "in the central North Atlantic"

This sentence has been removed.

**References:**

Cayan, Daniel R. "Latent and Sensible Heat Flux Anomalies over the Northern Oceans: The Connection to Monthly Atmospheric Circulation." *Journal of Climate* 5, no. 4 (April 1, 1992): 354–69. https://doi.org/10.1175/1520-0442(1992)005<0354:LASHFA>2.0.CO;2.

Deser, Clara, Michael A. Alexander, Shang-Ping Xie, and Adam S. Phillips. "Sea Surface Temperature Variability: Patterns and Mechanisms." *Annual Review of Marine Science* 2, no. 1 (2010): 115–43. https://doi.org/10.1146/annurev-marine-120408-151453.

Grasso, Lewis D. "The Differentiation between Grid Spacing and Resolution and Their Application to Numerical Modeling." *Bulletin of the American Meteorological Society* 81, no. 3 (March 1, 2000): 579–86. https://doi.org/10.1175/1520-0477(2000)081<0579:CAA>2.3.CO;2.

Marshall, John, Helen Johnson, and Jason Goodman. "A Study of the Interaction of the North Atlantic Oscillation with Ocean Circulation." *Journal of Climate* 14, no. 7 (April 1, 2001): 1399–1421. https://doi.org/10.1175/1520-0442(2001)014<1399:ASOTIO>2.0.CO;2.

We strongly thank the reviewer for reading our manuscript so carefully. The level of detail was extraordinary, and the review included many helpful, and constructive comments and suggestions.

Thank you!

**Responses to Reviewer 2**

This paper documents an apparent impact of subpolar freshwater anomalies onto the North Atlantic sea surface temperatures (SSTs) in winter and the subsequent summer. These summer SSTs are then argued to drive changes in the atmospheric circulation that drive significant changes in summer temperatures though analysis of reanalysis data and model simulations. Overall, the authors argue that freshwater changes in the North Atlantic can drive a large amount of the variance in European summer temperatures and so argue that such a mechanism could provide significant extra skill in seasonal predictions.

Overall this is an interesting study and presents an exciting set of results. However, I have a number of issues with the manuscript in its current form, which I list below in my major comments. Therefore, I do not recommend publication at this time.

We strongly thank the reviewer for reviewing our manuscript and providing so many detailed and constructive comments and suggestions! Your comments and suggestions have helped us to clarify and improve the manuscript.

**Major comments**

1. My main concern with this manuscript is that the most important results are buried in the appendix - that is the definition of the freshwater budget method, and the results evaluating the impact of freshwater on the North Atlantic SSTs. I note that this is similar to the structure of Oltmanns et al, 2020, in GRL. This usage of the appendix meant that the paper was quite challenging to read as I had to keep flipping back and forth in the paper to try to understand the logic. Furthermore, I would argue that the most important result in the paper is the evaluation of impact of freshwater onto the resulting SST (e.g., section A3). Therefore, I would strongly recommend the authors to move more of the important background information into the main paper and systematically step the reader through the ideas.

Thank you for indicating that the usage of the appendix made it difficult to follow the mass balance approach, and that the link between the freshwater and SST is also an important result.

Following your suggestion, we have now integrated Appendix A3 and A4 into the results section. In addition, we have combined Appendix A1 and A2 with the approach section. Thus, the full derivation and results of the mass balance analysis are now included in the main manuscript.

We still think that the link between the freshwater and SST anomalies in winter is only part of the results. The link between freshwater anomalies and the subsequent European summer weather is also an important finding. Thus, we have included additional analyses focusing on the link between freshwater anomalies in winter and the SST and European weather in subsequent summers, including the role of air-sea feedbacks. Further details are included in response to your specific comments below.

2. Although I found the results exciting, ultimately the results on the potential impact of freshwater anomalies on Europe were much more uncertain than I feel the authors described. The uncertainty spawns, I think, from the following important reasons.

   o There is little physical understanding of how the summer NAO is leading large freshwater driven SST changes across the subpolar North Atlantic. For example, there

is much discussion about how the sign of the relationship between summer NAO and the winter SST anomaly changes at ~0.5, but no reason for this is given. How can we be certain that the freshwater analysis is picking up real physical changes related to changes in freshwater?

Thank you very much for suggesting adding a physical explanation for the link between the summer NAO and the subsequent freshwater anomalies.

In the revised manuscript, we have now added an analysis of the causes of freshwater anomalies associated with both high and low NAO summers (Section 4.2).

Consistent with earlier studies, we find that a lower NAO phase is associated enhanced seasonal runoff and melting while a higher NAO phase is associated with an enhanced wind stress curl over the subpolar region, resulting in a stronger subpolar gyre circulation and enhanced advection of fresh, polar water into the subpolar region (Fig. 2 of the revised manuscript). While a detailed examination of the freshwater budget is beyond the scope of this study, our findings are consistent with earlier studies on the role of runoff (Bamber et al., 2018; Dukhovskoy et al., 2019) and the subpolar gyre circulation (Häkkinen and Rhines 2009; Häkkinen et al., 2011, 2013; Holliday et al., 2020) for freshwater anomalies in the subpolar North Atlantic.

The threshold of ∼ -0.5 in the summer NAO corresponds to a critical surface freshening above which the shallower, seasonal freshwater anomalies are mixed down, such that deeper, circulation-driven signals dominate the hydrography of the subpolar gyre. We also investigated the surface freshwater fluxes (precipitation minus evaporation) but found that there were unimportant for the identified freshwater anomalies.

- o Related to the above, the paper focuses on interannual changes - however, there is plenty of observed evidence that there is significant decadal time-scale variability in the subpolar North Atlantic, and so questions arise on what the freshwater analysis, which appears to assume independence between years, is picking up. For example, figure 2c shows that it is the decadal time-scales that dominate the summer NAO time-series, consistent with a southward shift of the North Atlantic Jet (e.g., Dong et al, 2013). Given the low-frequency changes, how can the authors be sure that they are seeing the interannual impact of freshwater changes as opposed to a different, or related, mechanism? For example, is the summer NAO and winter SST responding to another mechanism which is driving both (e.g., large-scale ocean circulation, external forcings etc)? As mentioned below, I think overlaying a timeseries of summer NAO with subpolar SSTs would be useful in seeing the potential importance of these longer timescales.

Thank you for pointing out that the involved timescales of freshwater variability were unclear.

In the revised manuscript, we are now more specific about the involved timescales. Thus, we state that the focus of this study is on interannual timescales, consistent with the high interannual variability of European summer weather (line 297). We also mention potential links to low-frequency variability in the causes of freshwater anomalies (line 296). Thus, in the past, decadal switches of the Arctic Ocean Oscillation have led to periodic exports of fresh and cold anomalies from the Arctic into the subpolar North Atlantic (Proshutinsky et al., 2015), which likely contributed to the decadal variability of the SST pattern in winter.

In addition, we now show the long-term freshwater trend over the last 70 years (Fig. 9, Section 4.6). Thus, we conclude that cold anomaly patterns in winter and summer are linked to freshwater anomalies over a range of timescales. While the described, dynamical feedback mechanisms act on short, sub-seasonal to interannual timescales (baroclinic wave activity in

the atmosphere, Ekman transports, geostrophic adjustments), the broad range of timescales in SST and freshwater variability can be explained by the physical causes of freshwater anomalies, acting on seasonal to decadal timescales.

The autocorrelation of the summer NAO is only small at one-year lag (Fig. 9f), and the trend in the summer NAO over the last 70 years is $\sim$–0.002 yr$^{-1}$. This suggests that the summer NAO mostly reflects high frequency, interannual variability of the cold anomaly pattern in the subsequent summer. Since the autocorrelations of the NAO indices are only small, we do not expect decadal variability to substantially affect the results.

- o The model experiments are interesting, but they do not, of course, explore the impact of freshwater changes on European weather. Indeed, the models are feeling the influence of both SSTs and, presumably, external forcings, and the assumption is that the influence of freshwater can be isolated by focusing on the summer NAO analysis. However, and related to point b, this relies that there are no other mechanisms in play that could explain both the summer NAO and the winter SSTs.

Thank you for pointing out that the role of freshwater anomalies for European summer weather was unclear. To address your comment, we have implemented four main changes:

(1) We have included a description of the drivers of freshwater anomalies (Section 4.2). Thus, we provide a physical explanation for the start of the chain of feedbacks. Since the identified drivers (runoff for $F_E$ events and circulation changes for $F_W$ events) are not otherwise linked to European summer weather on the investigated timescales, and occur exactly one year in advance, they challenge the idea that an unknown third mechanism drives both freshwater anomalies and warmer European summers without the two being physically connected.

(2) We are explicit about the involved timescales of variability. While the focus is on interannual timescales (line 296), we now show that surface freshening has a trend (Fig. 9a of the revised manuscript), superimposed on substantial interannual variability that is reflected in the variability of the summer NAO and runoff from the preceding year, obtained from the Greenland climate model MAR. (Section 4.6). After considering alternative drivers acting on these timescales, we find (line 405):

"Apart from seasonal runoff- and melt-driven freshening, there are currently no conceivable, physical mechanisms in the tropics, stratosphere or outside the North Atlantic which, at the same time, have a significant trend over the last 70 years, exhibit a similarly high interannual variability, can explain the occurrence of freshwater anomalies in the subpolar region in the subsequent winter, and are significantly correlated with the characteristic summer SST pattern a year before it occurs. This suggests that seasonal freshening may not only be a predictor of the SST pattern but also a potential trigger."

(3) To assess whether runoff is a trigger of the SST pattern, rather than only a predictor, we have included an analysis of the ocean-atmosphere feedbacks that contribute to the evolution of the SST pattern in summer (Section 4.7). After investigating the surface heat and momentum fluxes and their influences on the ocean and atmosphere with ERA5, remote sensing data, in-situ hydrographic observations from the cold anomaly region, and with the models, we find that wind-driven transports may contribute to the intensification of the SST signal. Thus, we conclude (line 453):

"We conclude that the atmospheric forcing contributes to the development of the SST field but is in turn, forced by it. On the one hand, the consistency of the spatial SST pattern with the wind-driven transports, the intensification of the SST signal in mid-summer, and the vertical extent of the hydrographic anomaly point to the relevance of the atmospheric forcing in driving

the SST signal over the North Atlantic. On the other hand, model simulations, forced with the prescribed, observed SST, reveal the importance of the SST field for the large-scale atmospheric circulation, including for European summer weather. The underlying SST pattern covers the entire North Atlantic with the strongest signal occurring in the subpolar region (Fig. 11b). While further studies are necessary to confirm the dynamical contribution of freshwater anomalies to the large-scale SST pattern, its spatial and temporal characteristics, and the time lag of one year, indicate that enhanced surface freshening from the preceding year may have initiated the chain of air-sea feedbacks."

(4) We are more cautious about the wording. Throughout the analyses, we refer to freshwater as a predictor (e.g. line 350). In the conclusion, we discuss evidence that points to freshwater as a trigger rather than only a predictor, but we are explicit about the involved uncertainties, and we have removed phrases that may previously have caused confusion.

> Indeed, I can't help noticing that the SST patterns being explored in the models (figure 8a and figure 9a and b) look like the cold AMV pattern (e.g., Zhang et al, 2019). Therefore, how can the authors be confident that the atmospheric circulation patterns are being driven by just the subpolar North Atlantic, rather than the tropical North Atlantic?

Thank you for indicating that large-scale ocean-atmosphere feedbacks over the full North Atlantic were insufficiently described.

In the revised manuscript, we now describe the ocean-atmosphere feedbacks more clearly in winter (Section 4.3), and we have added a new analysis on the involved feedbacks in summer (Section 4.7).

The SST in the subtropical and subpolar regions is highly coupled due to large-scale atmospheric feedbacks. For instance, the large-scale atmospheric circulation anomaly, associated with the subpolar cold anomaly in winter, leads to convergent, wind-induced surface currents in the inter-gyre region, and thus a warm anomaly that emphasises the SST gradient. At the same time, the anti-cyclonic, atmospheric circulation anomaly to the south of the SST front leads to upwelling off the coast of Africa, contributing to the characteristic tri-pole pattern of the AMV.

Moreover, we have now removed the previous model analyses and replaced it with a new analysis where the underlying $SST_{FW}$ pattern in summer is more clearly linked to the preceding freshwater anomalies (Section 4.7). Considering that the SST anomalies are largest in the subpolar region and decrease towards the south (Fig. 11b), it can be explained by the large-scale atmospheric feedbacks associated with freshwater anomalies. This includes a range of timescales, including that of the AMV.

However, we are cautious in the wording about the role of freshwater in triggering these feedbacks. We refer to freshwater anomalies as predictor. While we discuss evidence in the conclusions that suggests freshwater may act as a trigger of the large-scale ocean-atmosphere feedbacks, rather than only a predictor, we state this as a hypothesis, not as a result (line 511).

**Minor comments**

Line 1 - The paper discusses here and in the introduction the possibility of long-term forced changes in the arctic having an impact on the mid-latitude weather, but then exclusively focuses on interannual variability? This left me slightly confused about what I was supposed to be taking from the analysis, so maybe it worth rethinking the motivation?

Thank you for pointing this out.

We have now added an analysis of the long-term variability of freshwater and the implications (Section 4.6). Thus, we find that there is a significant freshening trend over the last 70 years, superimposed on substantial interannual variability. This is consistent with seasonal runoff in the subpolar region, which has a high interannual variability but also a trend.

Line 63 - why not just use HadISST rather than merge the two datasets?

We did not manually merge the two datasets ourselves. The merged dataset is freely available and benefits from the optimal interpolation technique used in the NOAA data.

We have now added the link in the data description to avoid misunderstandings (line 59).

Section 2.2 - how do these simulations differ from CMIP style AMIP runs (e.g., Eyring et al, 2016)?

The simulations were designed to facilitate comparison between different models, like those in Eyring et al. (2016). Thus, they have a pre-determined forcing to facilitate the comparison, which is now clarified.

In this case, the simulations were performed with the prescribed, observed SST and sea ice cover and time varying greenhouse gases and ozone (line 90).

Section 2.3 - I am confused by the discussion of a trend, in part due to the framing of the paper on understanding the impact of long-term forced changes in the Arctic on the wider climate, but I think the point is that trends over land may alter the relationship with ocean SST anomalies? Please clarify.

We do not distinguish between trends over land and ocean. The intention of this approach was to remove the effect of Greenhouse gases, not to remove trends in general. This is now clarified (line 97).

Moreover, we have adjusted the subsequent paper to the motivation in the introduction. Thus, we have added an analysis of the trends in the freshening and SST over the last 70 years (Section 4.6).

Line 133 - You state that "we derive indices that exhibit a strong relationship to subpolar temperatures but not to the drivers of density anomalies", but then argue that the summer NAO drives changes in freshwater (which will drive the density anomalies) - can you clarify your point, please?

Thank you for pointing out that this was unclear. We referred to the active drivers of density (or in turn temperature and freshwater) anomalies. We consider freshwater as state variable that in turn has drivers, like precipitation or runoff.

We have now clarified this in the manuscript (for instance line 149). We have also removed the sentence you were referring to, since it raised confusion.

Line 140 - you mention the potential impact for the summer NAO on freshwater, but won't it also affect the heat fluxes too? How do you take account of these in your analysis?

Yes, we do consider the surface fluxes in the mass balance analysis. The surface heat fluxes are included in the buoyancy fluxes, and we also show them separately (Figs. A1d and A2d). Surface heat fluxes can be a driver of SST anomalies or a response. In this case, we find that that the surface heat fluxes in winter cannot explain the SST anomalies. They are too small, and their distribution is inconsistent with the SST anomalies.

We now explicitly state in the revised manuscript that the surface heat fluxes cannot account for the SST anomaly in winter (line 336). In addition, we now show that the surface heat fluxes in summer also cannot account for the SST pattern in summer (Fig. 1C).

Line 392 - "...we select all the years that lead to an increase in the slope of the regression line." - I'm uncomfortable about this - it sounds a bit like cherry picking to get a stronger signal (which in turn would affect your conclusions about how important freshwater changes are). Please could the authors justify this choice physically?

Thank you for suggesting this. Following your suggestion, we have now added a description of the causes of freshwater anomalies associated with high and low NAO summers (Section 4.2). Thus, we find that the subsampled indices are also more strongly linked to the physical causes of freshwater anomalies, supporting the subsampling with a physical explanation.

The objective of the subsampling is to improve the signal-to-noise ratio between the index and freshwater anomalies. The strong signal between the index and the freshwater anomalies is a pre-requisite for the index to represent freshwater anomalies. We do not use it as the conclusion.

We now explain the method more clearly and have integrated the appendices into the main text, making it easier to follow the steps (Section 3.2).

In addition, we have included an analysis of the link between European summer weather and the preceding summer NAO, without any subsampling (Section 4.7). Combined, these analyses show that the results are neither sensitive to the choice of the index (including the sampling) nor the statistical method (regressions or composites, Sections 4.4 and 4.5).

Figure A1 - I can't help but to see that there is a long-term change in the summer NAO with most negative years occurring after 2000. This is consistent with the long-term trend in summer jet. The question this raises is how important is this longer term variability in the sNAO for your results? One thing that is missing is a time series of subpolar SST, which also shows significant decadal timescale variability. Please add this to figure A1 at least. However, following up on this, previous studies have linked these low frequency variability in the jet to the changes in the subpolar temperatures (e.g., Dong et al, 2016) - therefore, how certain are you that the changes in summer NAO are not reflecting decadal time-scale changes in the subpolar gyre?

We now show the timeseries of the summer SST pattern linked to freshwater anomalies (Figs. 9 and 11) since this pattern is relevant for European summer weather. A related timeseries for winter is included in Fig. B1.

While there has been significant decadal variability in the causes of freshwater anomalies in the past, the negative NAO picks out the high-frequency variations, associated with the interannual variability of runoff (see also our response to your second comment). Since the focus of this study is on the influences of freshwater anomalies rather than the causes, we think that showing the time series for the winter SST pattern in the main text would not add substantial information.

Line 405 - It's not clear to me why the geostrophic currents are not important - there is nothing in the 95% confidence lines that are related to constant lines of density. Please can you clarify your reasoning here?

Thank you for pointing out that this was unclear. Geostrophic currents at the surface are along lines of constant pressure and density. Thus, they cannot lead to a net increase or a decrease of density. Importantly, they can, and do, drive freshwater and temperature anomalies. However, the density anomalies associated with freshwater and temperature changes need to compensate each other in geostrophically balanced flows.

We have now clarified that geostrophic surface flows cannot cross density contours (line 554).

Section A3 - it is not clear why you are regressing the JFM heat fluxes onto your summer NAO? In order to rule out the impact of surface heat fluxes on the JFM SSTs it is the integrated heat fluxes between the summer and JFM that are important?

Thank you for pointing out that this was unclear. We are examining the SST signal in winter, which is typically not influenced by the surface fluxes before the winter because the amplitude and variability of the surface fluxes in winter is much larger. In addition, the surface fluxes in autumn respond to re-emerging ocean anomalies in the mixed layer (Timlin et al., 2002), thus balancing the term $M_n$ when integrated over longer periods (line 546). To be sure, we have repeated the surface mass balance for the period from summer to winter and found that the results did not change appreciably. In the case of the $F_E$ subset, the overall amplitude slightly decreases, implying slightly smaller uncertainties. For the reasons stated above, however, we think that the surface fluxes in winter provide the more accurate estimate.

We now motivate the focus on the winter period by the tendency of $B_n$ anomalies to balance anomalies in $M_n$ when integrating over the period of rapid mixed layer deepening. Consistent with earlier studies, we also find that the amplitude and variability of the surface heat and buoyancy fluxes is largest in winter (line 546).

Moreover, we state in the revised manuscript that the results do not change appreciably if we integrate over autumn and winter or only winter (line 226).

417 - the mean mixed layer is the annual mean mixed layer?

Yes, indeed. Since we are investigating the SST anomalies in winter, we also need to use the mean mixed layer in winter.

We now clearly state the averaging and integration periods in the manuscript (line 134, line 580).

Line 440 - Please elaborate on the origin of the uncertainty numbers.

Thank you for pointing out that this was not sufficiently clear. The uncertainty originates from neglecting the terms on the righthand side of the mass balance equation. It is obtained by comparing the size of these terms to the size of the terms on the lefthand side.

We now devote a full section in the main text to the derivation of freshwater anomalies and their uncertainties (Section 4.1).

Figure 3 - what is the data used to plot the SSS anomalies, or are they implied from your mass balance analysis?

Yes, the estimates were obtained from the mass balance analysis. This is now clarified in the revised manuscript (Section 4.1, line 241, and Figure 3 caption). Thank you for letting us know that this was unclear.

Section 4.2 - I think when you mean first summer, you mean the same summer as the summer NAO event? However, it was not clear as you're talking about looking in subsequent summers which I took to be after the winter - please clarify. This is particularly important for figure 6, which is unclear what summers you are looking at.

No, we refer to the European weather in the summer one year after the NAO index. It is not the same summer. This is why the summer NAO can be a such a valuable predictor of European summer weather, one or even two years in advance. We have now clarified this (line 352, line 364, line 420, and captions of Figures 6, 7, 10 and 12). Thank you for pointing out that this was unclear since it is indeed important.

Figure 7 - Are you computing the variance explained using all years, or are you only focusing on the years that you have large changes? Either way, how important is the long-term trend in atmospheric circulation shown in figure 2c for your analysis?

The variance is computed over all the years for which the indices are defined. In addition, we now state the variances explained by the winter SST gradient and the un-subsampled summer NAO over all years (line 474).

Given the low autocorrelations of the summer NAO, we do not expect the long-term trend to substantially affect the explained variances. Also, the subsampling ensured that the high correlations are not due to outliers or clusters (Fig. 2d). Thus, the results are not sensitive to excluding individual years, implying an increased robustness of the results.

Figure 9 - Is the only difference between these figures and that shown in figure 8 (d and f) that they are split across model ensembles? Not clear to me why this is relevant - could you just include the ensemble spread in figure 8 ?

The main intention of the figure was to show the spread across the ensemble members and how they compare with the observations.

We have now removed this figure.

References:

Bamber, J. L., Tedstone, A. J., King, M. D., Howat, I. M., Enderlin, E. M., Van Den Broeke, M. R., & Noel, B. (2018). Land ice freshwater budget of the Arctic and North Atlantic Oceans: 1. Data, methods, and results. *Journal of Geophysical Research: Oceans*, *123*(3), 1827-1837.

Dukhovskoy, D. S., Yashayaev, I., Proshutinsky, A., Bamber, J. L., Bashmachnikov, I. L., Chassignet, E. P., ... & Tedstone, A. J. (2019). Role of Greenland freshwater anomaly in the recent freshening of the subpolar North Atlantic. *Journal of Geophysical Research: Oceans*, *124*(5), 3333-3360.

Hakkinen, S., & Rhines, P. B. (2009). Shifting surface currents in the northern North Atlantic Ocean. *Journal of Geophysical Research: Oceans*, *114*(C4).

Häkkinen, S., Rhines, P. B., & Worthen, D. L. (2011). Warm and saline events embedded in the meridional circulation of the northern North Atlantic. *Journal of Geophysical Research: Oceans*, *116*(C3).

Häkkinen, S., Rhines, P. B., & Worthen, D. L. (2013). Northern North Atlantic sea surface height and ocean heat content variability. *Journal of Geophysical Research: Oceans*, *118*(7), 3670-3678.

Holliday, N. P., Bersch, M., Berx, B., Chafik, L., Cunningham, S., Florindo-López, C., ... & Yashayaev, I. (2020). Ocean circulation causes the largest freshening event for 120 years in eastern subpolar North Atlantic. *Nature communications*, *11*(1), 1-15.

Proshutinsky, A., Dukhovskoy, D., Timmermans, M. L., Krishfield, R., & Bamber, J. L. (2015). Arctic circulation regimes. *Philosophical Transactions of the Royal Society A: Mathematical, Physical and Engineering Sciences*, *373*(2052), 20140160.

Timlin, M. S., Alexander, M. A., & Deser, C. (2002). On the reemergence of North Atlantic SST anomalies. *Journal of climate*, *15*(18), 2707-2712.

Again, we sincerely thank the reviewer for carefully reading and reviewing our manuscript and providing so many detailed and constructive comments and suggestions. Based on your comments and suggestions, we were able to clarify and improve the manuscript. Thank you!

---

## Referee Report (RR1)

The authors use statistical analysis of observations and reanalysis data to examine a potential link between freshwater anomalies in the North Atlantic subpolar gyre region and summer weather in Europe in the following year. They propose that stronger freshwater anomalies resulting from increased glacial runoff in Greenland leads to an stronger meridional SST gradient between the subpolar and subtropical gyre and consequently increased baroclinic instability in the atmosphere above. The resulting deflection of the jet stream alters the advection pathway of maritime air masses over Europe. The foundation of the analysis are freshwater indices derived from a mass balance equation that are used to identify freshwater anomalies in relation to simultaneous sea surface temperature (SST) anomalies linked to the North Atlantic Oscillation (NAO).

The revised version of the manuscript is a substantial improvement to the previous draft, particularly with respect to the derivation of the freshwater indices. However, there is still lack of clarity and detail in some methods, as well as precision in the description of the results. Therefore, the manuscript in its current form is not suitable for publication and I suggest major revisions based on my comments below.

**I. General Comments and Suggestions:**

Throughout the manuscript, multiple indices are used, but their exact definition unclear. Some details regarding the methods are often only mentioned in the figure captions leading to interruptions in the reading flow. Additionally, the description of results often remains rather vague or general, and does not necessarily match the figures. Similarly, I encourage the authors to be more concise and precise in their language, e.g., sometimes it is not immediately clear if they refer to the ocean or atmospheric anomalies or geographic descriptions are very broad. Some of these instances are mentioned in my comments below.

**II. Main Comments:**

1. Section 3: The combination of the appendices from the previous version with the main text helps better understanding and following the construction of the freshwater indices. However, there are a few points that require some clarification.
   1. l. 141-143: Equation (3) already has the mean state subtracted, yet you say "we subtract the mean values from the resulting equation".
   2. l. 145: Please specify the the values of $\alpha$ and $\beta$.
   3. l. 165: There is no Figure 1a that shows the NAO in summer.
   4. Section 3.2: Some important information/motivation is either missing or presented later in the text which makes it difficult to follow the approach and understand the derivation of the freshwater indices:
      1. It is unclear how the resulting freshwater indices are defined. Are they (scaled) summer NAO indices, i.e., $F_E = \{-NAO_{S,i} \mid NAO_S < -0.5, i=F_E \text{ years}\}$ and $F_W = \{NAO_{S,i} \mid NAO_S > -0.5, i=F_W \text{ years}\}$?
      2. l. 173-179: This paragraph is missing some motivation why and how you use the meridional SST gradient to establish a relationship between the summer NAO and freshwater

anomalies in the following winter. I suggest something along the lines of: "Given the theoretical considerations based on the mass balance analysis (Section 3.1), there are certain conditions under which freshwater anomalies associated with the summer NAO are accompanied by temperature anomalies. To identify these conditions, we regress winter SST onto two subsets of the summer NAO…". Please give a rationale for the seasonal lag.

3. l. 174-175 & l. 181: Is the "meridional SST gradient" the same as $\Delta$SST? Please define $\Delta$SST in the text.

4. l. 180: The "conditions c" are not explicitly stated in the text nor referred to in the remainder of the manuscript. I suggest to remove the variable name "c".

5. l. 200-203: The significant SST anomalies associated with $F_W$ extend from the Labrador Sea in the west almost all the way to the British Isles in the east while the SST anomalies associated with $F_E$ occur mostly in the central North Atlantic subpolar gyre region. Thus the naming can be misleading unless you specifically refer to *maximum* cold anomalies.

2. Section 4.7: This section lacks a clear structure, clarity, and important details that make it hard to reconcile the results with the rest of the paper. In Section 3.2, you establish a non-linear relationship between a subset of summer NAO and SST anomalies associated with freshwater anomalies. Here, you use the "full, un-subsampled summer NAO" as a linear predictor to "assess the role of freshwater triggering the SST signal". By regressing different variables onto the summer NAO and resulting SST pattern (which is quite different compared to the SST patterns related to $F_E$ and $F_W$, cf. Figures 6a and 7a with Figure 10a), you describe the associated atmospheric response in both observations and models. As you point out in your concluding sentence of this section, "further studies are necessary to confirm the dynamical contribution of freshwater anomalies to the large-scale SST pattern". Since an explicit link to freshwater anomalies, and their role in triggering your proposed chain reaction, is missing, this section creates confusion for the reader. I suggest to add more details following my comments below or remove this section entirely from the manuscript.

1. Please specify what you are trying to predict using the "un-subsampled summer NAO as predictor".

2. l. 420: Please clarify what you mean by "reduced NAO states".

3. l. 422: I believe this should be Figure 10a-c.

4. l. 423: Please discuss the significance that the warm anomalies occur mostly over land around the Mediterranean Sea, but the dry anomalies over the ocean.

5. l. 429: Please specify that the upwelling occurs in the ocean below the center of the cyclone.

6. l. 442-443: Please define $SST_{FW}$ explicitly. Is this related to the $F_W$ index? If not, I suggest to use a different name to avoid confusion.

7. l. 443-444: This sentence is unclear. Please specify which "observational analysis" you are referring to. Does that mean you define the SST index based on the model simulations?

8. l. 446: Please add more details in the text: Are you referring to temperature and precipitation anomalies from climatology or ensemble mean?

9. l. 449-450: The observed and simulated atmospheric responses agree well over the ocean, while the simulated response over land is much smaller. It is not clear how you determined the statistical significance from the 90 ensemble simulations given that deems very small anomalies

close to zero as statistically significant – please add more details in the text and discuss the implications.

**III. Additional Comments and Suggestions:**

1. l. 4: What are "medium-term climate trajectories"?
2. l. 97: Please state years of "recent period"
3. l. 101-105: The 2 m-temperature trends are different over land than over the ocean (e.g., Simmons, 2022). Your averaging area includes large parts of the North Atlantic. Please discuss if this has any implications for your results.
4. l. 230-231: This is only true in the absence of vertical mixing.
5. l. 238-239: To keep the reader on board it would be helpful to state that "using this relationship, we estimate SSS from SST anomalies for the two $NAO_S$ subsets".
6. Figure 3: Did you actually do the regression? Given the linear relationship between SSS and SST, and assuming a constant coefficients, panels a and b should be equal to $\alpha/\beta$ * (negative areas in Figures 2a and b). However, the SSS anomalies show more structure. Is that due to the choice of contour levels? Please clarify.
7. l. 241-242: Please be more precise: the significant area of $F_W$ SSS anomalies extend further eastward compared to the anomalies associated with $F_E$.
8. l. 244: Please refer to the appendix to ensure that the reader knows how the uncertainties were estimated.
9. l. 252-253: The different phases of the NAO are not associated with "opposite atmospheric circulation patterns". They differ in the strength of the pressure difference between the subpolar low and subtropical high pressure system. In this regard, the wording "lower/higher NAO states" seems slightly odd. I suggest to make explicit links to $F_E$ and $F_W$ throughout the whole text.
10. l. 280-281: It is unclear how you identify "an intensified subpolar gyre circulation in the Labrador Sea" in Figure 4c.
11. Figure 4: What are the vectors in panel b? Please define dSSS in the text: it is unclear what you mean by "newly arriving, seasonal surface freshening from summer (August) to winter (January to March). Do you accumulate or take the mean of the freshwater anomalies during the winter months and subtract it from the August value?
12. l. 299-301: If the "seasonal freshening is mixed down and too small to affect the absolute SST anomaly in winter" when $NAO_S \geq -0.5$, does that not contradict the construction of the $F_W$ indices or does the sentence refer to the years that were excluded in the optimization process?
13. l. 354-355: Please discuss the fact that the SST anomalies in Figure 6a are largely not statistically significant. What are the implications for your conclusions?
14. l. 359-361: This sentence is unclear. I don't understand what you mean by " the anticyclonic circulation anomaly is in part rotated over the continent".
15. l. 362: Interestingly, the warm and dry anomalies during the first summer are not necessarily co-located. Please discuss/speculate why the dry anomalies occur more toward the southeast.
16. l. 364: Is the "cold SST anomaly" in Figure 7a statistically significant other than the very small region around 40°N, 40°W? What are the implications for your conclusions?

17. l. 367: Please discuss why, after $F_W$ freshwater anomalies, the warm anomalies occur over France and Great Britain while the dry anomalies occur more to the northeast over the Baltic region.
18. Figure 6: The coastlines in panels c and d are hard to make out. This makes it difficult to relate it to the other maps, particularly since each row shows a different domain.
19. Figures 6 & 7: Does "SST with the 700-hPa winds (…) on $F_E/F_W$" mean you performed a multiple linear regression? Please provide more detail in the text.
20. l. 385: Please be more precise: Does "cold anomaly over the North Atlantic" refer to the composite of air temperature (Figure 8b) or the SST anomaly in the western subpolar gyre region (Figure 8e)?
21. Figure 8: What are the red lines in panel f?
22. l. 400: Please clarify what you mean by "SST variability each summer". Are you referring to the spatial distribution of summer SST anomalies?
23. l. 403-406: What is the correlation of the SST pattern with the summer NAO? It might be worthwhile adding the NAO time series in Figure 9e to demonstrate the relationship.
24. l. 407-410: To keep the reader onboard, I suggest to add a sentence that links the SST variability back to the freshwater anomalies.
25. Figure 9f: Please add a label for the x-axis.
26. l. 506: Please specify if you refer to an "increased meridional temperature gradient" in the ocean or atmosphere.
27. Figure A2: Why are the regions enclosing the 95% significance level different from Figure 2b?
28. l. 580-581: The cold anomaly regions cover the entire subpolar North Atlantic including areas of deep convection. Please discuss the implications and added uncertainty using a spatially constant value for the mean winter mixed layer depth.

**IV. Typos/Wording:**

I suggest the following changes:

l. 36: "requiring a high grid spacing of ~1/12° " to "requiring ocean models with high grid spacing of at least ~1/12°"
l. 64: "spatial resolution" to "grid spacing"
l. 84: "run at a resolution of" to "run with grid spacing of"
l. 158: "surface temperature" to "ocean surface temperature"
l. 160: "subpolar cold anomalies" to "subpolar SST cold anomalies"
l. 189: Please remove second "warm".
l. 317: "to north of" to either "north of" or "to the north of"
l. 326: "resolution" to "grid spacing"
l. 327: "resolution" to grid spacing"
l. 372: "subsets" to "subset"
l. 416: "autocorrelations" to "autocorrelation"
l. 417: "point" to "point out"
l. 435: "of SST signal" to "of the SST signal"

**References:**

Simmons, Adrian J. "Trends in the Tropospheric General Circulation from 1979 to 2022." *Weather and Climate Dynamics* 3, no. 3 (July 21, 2022): 777–809. https://doi.org/10.5194/wcd-3-777-2022.

---

## Referee Report (RR2)

The authors use statistical analysis of observations and reanalysis data to examine a potential link between freshwater anomalies in the North Atlantic subpolar gyre region and summer weather in Europe in the following year. They propose that stronger freshwater anomalies are associated with a stronger meridional SST gradient between the subpolar and subtropical gyre and consequently increased baroclinic instability in the atmosphere above. The resulting changes in the large-scale circulation lead to significant anomalies in near-surface temperature and precipitation in different parts of Europe. The foundation of the analysis are freshwater indices derived from a mass balance equation that are used to identify freshwater anomalies in relation to simultaneous sea surface temperature (SST) anomalies linked to the North Atlantic Oscillation (NAO).

The current manuscript is another substantial improvement to previous versions and I appreciate the added detail in the derivation of the mass balance and estimation of the freshwater indices, and the more precise description of the results. As outlined in my comments below, there is some remaining uncertainty regarding the construction of the freshwater indices. Additionally, the presentation and description of the central results, i.e., the link between the freshwater anomalies and European summer weather, requires – in my opinion – some larger context and relation to known atmospheric circulation and weather patterns that are described in the literature. Overall, I recommend a series of minor revisions that may appear major due to their extent.

**I. Main Comments:**

1. Section 3.2:
   1. I now understand why this part has been – and, in part, still is – confusing to me: the threshold of $NAO_S < -0.05$ is based on the non-linear relationship shown in Figure 1d. The y-axis of that figure is $\Delta SST$ which "corresponds to the SST difference between the red, subtropical and blue, subpolar 95% confidence regions in panels e (red years) and f (blue years) respectively, relative to the respective means". However, these confidence regions are not known a priori, so I am trying to wrap my head around the specific steps to get from the time series of $NAO_S$ (Figure 1c) and winter SST at each grid point, via the scatter plot in panel d, to the maps in panels a and b. Strictly speaking, you cannot determine the value of $\Delta SST$ for Figure 1d, and therefore the threshold of -0.5, until you have the two maps (panels e and f) after the subsampling. The subsampling itself, however, depends on the relationship in Figure 1d, so there must have been some iteration or trial-and-error. Please add more details in the text (before l. 202) to lay out you analysis steps.
   2. Please define $\Delta SST$ in the text. Is this the difference between the regressed SST or the actual SST values in these regions? Is it relative to the respective spatial or temporal mean (over which period)?
   3. Caption of Figure 1: $-NAO_S/+NAO_S$ is confusing – this could be interpreted as positive and negative phase of the NAO. Maybe " $-1 \times NAO_S$" is more obvious.
   4. l. 235: Have you looked at the differences of regressions/composites between the included an rejected years? This could potentially help identify or constrain a physical mechanism at play for years with a strong relationship.

2. Section 4.4: Given the title of the manuscript, this section describes the central result of the study: the statistical link between summer weather and freshwater anomalies in the subpolar gyre in previous years. However, the presentation of the results leaves me as the reader unsatisfied. While I appreciate the added details compared to the previous version, some some of the conclusions remain slightly hand-wavy and are missing some larger context:

   1. Based on the regressed meridional wind anomalies at 700 hPa, you describe a "northward deflection of the jet stream" following both FE and FW freshwater anomalies that differ in there location between years and subsets. Given that the jet stream occurs at higher altitude, you are rather describing circulation anomalies in the lower troposphere. Regardless of this semantic distinction, I am missing a discussion of the southward anomalies in the 700 hPa winds in Figures 5c and 6b as they can help put these anomalies in the context of known large-scale circulation patterns (e.g., Cassou, 2008; Grams et al., 2017) and their related expressions in surface temperature and precipitation anomalies. Relating your result to previous studies may also help identify physical processes that lead to the anomalies that you describe – is it advection of warmer/dryer air masses or changes in radiation/heat fluxes that can be linked to the large-scale circulation? For example, the anticyclonic circulation anomaly over the North Sea in Figure 6b might be suggestive of a blocking event (reduced winds, increased radiation, less precipitation...) – interestingly, the dry anomalies in Figure 6d  are roughly co-located.
   2. Figure 5: In order to make it easier for the reader to interpret the regressed anomalies, it might be worthwhile adding another column of maps showing the mean conditions.
   3. I do not understand the justification for excluding the 2016 anomalies (caption of Figure 5).

**II. Additional Comments and Suggestions:**

1. l. 36: This wording suggests that it is certain that freshwater initiates the causal chain, but only the physical mechanism is unclear.
2. l. 77-79: You mention that you use monthly – presumably mean – ERA5 output in the analysis. Please clarify: did you estimate the maximum Eady growth rate of the monthly mean circulation or did you compute it from higher-frequency, e.g., daily mean, output that you then averaged over a month? Given the nonlinearities in the equation these two estimates could be quite different, however, I do not expect them to change your results.
3. l. 195, 197, 246: I am still stumbling over the phrase "lower/higher NAO phase". I am not familiar with the detail of the previous studies that you refer to in the first two instances, but I would assume the most of them contrasted the two states of the NAO and therefore, I suggest to use "negative/positive NAO phase". The last instance can be changed to "associated with $NAO_S < -0.5$".
4. l. 323-325: Please clarify: the runoff-$NAO_S$ relationship is calculated over all years, yet you use it as a potential explanation for $F_E$ freshwater anomalies, i.e., only a very specific subset of years. In Figure 3a, this relationship is not that clear if you only consider $NAO_S < -0.5$. If anything, there seems to be a clearer relationship and less spread around the regression line for the $F_W$ years.
5. l. 376-377: Please clarify: do you show that "after strong $F_E$ anomalies, the NAO anomaly switches sign from being strongly negative in summer to being strongly positive in winter" or do you infer

that from Figure 4b? Out of curiosity, what is the correlation between summer NAO and winter NAO with and without conditioning on $F_E$ years?

6. l. 457-459: This sentence is unclear.
7. l. 471: It is easy to get lost here: I think what you are doing is regressing the winter SST on $NAO_S$ for all years, but calculate $\Delta$SST bases on the regions shown in Figure 1e with the resulting time series shown in Figure 7a. Please add more detail to clarify your analysis steps here.
8. l. 484: Notably, the T2m anomalies are offset to the east of the V700 anomaly. Similar to my comments on Section 4.4 above, please discuss this in the context of the existing literature and speculate about the physical mechanism.
9. Figure 7a: What did you normalize the SST difference by? Please add more details in the text.
10. l. 525-526: Please clarify: your predictors are $F_E$ and $F_W$ (i.e., $NAO_S$) and $\Delta$SST, so the common denominator is the atmospheric circulation associated with the summer NAO. How does sea surface salinity constrain weather predictions?
11. l. 535. Please add more details in the discussion of your results with respect to the large-scale circulation (see comments on Section 4.4 above).
12. Please add more details: how exactly do you "trace the cold SST anomaly back to a freshwater anomaly in the preceding winter".
13. l. 557-558: The northward deflection of the jet stream is a bit too hand-wavy for me. Please discuss this in the context of large-scale atmospheric circulation in summer and associated weather patterns.
14. l. 563: Please discuss: the freshwater anomalies themselves are part of the chain reaction that are ultimately linked to the summer NAO. Provocatively asked: if you wanted to create a statistical model to predict summer weather with one or two year lead time, what information is added by knowing the freshwater anomalies? While I do understand the role of the freshwater anomalies as part of the chain that eventually leads to changes in European weather, I am missing a discussion how knowledge of sea surface salinity can constrain the predictions (see also comment II.10).
15. l. 610: What salinity did you use in the evaluation of the buoyancy flux?
16. l. 664: Regarding the "northward deflection of the jet stream", please see my comments above.

**III. Typos/Wording:**

I suggest the following changes:

l. 39: "requiring a fine grid spacing" to "requiring *ocean models* with fine grid spacing"
Caption of Figure 1 d: "SST" to "$\Delta$SST"
l. 227: "0.5" to "-0.5" (minus sign missing)
l. 368: "pariticularly" to "particularly"
l. 371: "circulation" to "atmospheric circulation"
Figure 7: The title fo panel e should be $T_{+1}$
l. 657: "$\Delta$SSS" to "$\Delta$SST"

**References:**

Cassou, C., 2008. Intraseasonal interaction between the Madden-Julian Oscillation and the North Atlantic Oscillation. Nature 455, 523–527. https://doi.org/10.1038/nature07286

Grams, C.M., Beerli, R., Pfenninger, S., Staffell, I., Wernli, H., 2017. Balancing Europe's wind-power output through spatial deployment informed by weather regimes. Nature Clim Change 7, 557–562. https://doi.org/10.1038/nclimate3338

---

## Author Response (AR2)

4. September 2023

Dear Prof. Pfahl,
Dear Reviewers,

We are pleased to submit a revised version of the manuscript "European summer weather linked to North Atlantic freshwater anomalies in preceding years". Based on the detailed and constructive feedback provided by both reviewers, we have carried out a major revision to clarify and improve the manuscript.

The main shortcoming of the last version was a lack of clarity and a poor presentation. This was evident from both reviews. Reviewer 1 criticised a lack of details and precision in the description of the methods and results. Reviewer 2 was additionally concerned about too much certainty in the language, which did not adequately reflect the underlying uncertainties. Both reviewers also recognised a substantial improvement, however.

Considering that the lack of clarity has been an ongoing issue with this manuscript, we have followed the reviewers' suggestions and removed the analyses that evoked most comments and that are not necessary, including the model and trend analyses which had shifted the focus. In the remaining analyses, we have provided more details and clearer explanations of the methods, a more comprehensive uncertainty analysis, and a more precise description of the results. We are also more cautious about the language, clearly separate interpretation from the results, and we have fully removed contentious statements from the manuscript.

Since Reviewer 2 was additionally concerned about the robustness of the results, we have carried out a rigorous statistical analysis, including a new section "Significance and robustness" (Section 4.5) and Appendix B showing that the results are not sensitive to the choices made in the methods. In conclusion, we now demonstrate the link between freshwater anomalies in winter and subsequent European summer weather using three techniques: (1) correlations and regressions, (2) composites, and (3) a multi-taper coherence analysis. Moreover, the correlation and regressions are now based on three freshwater indices – including a new un-subsampled SST index covering all years. We clearly state the advantages and disadvantages of each index and explain the trade-off between the magnitude of the correlations and the number of years included in each index, reflecting the complexity of freshwater variability. Combined, the regressions, composites, and coherence analysis show that the identified links are robust, irrespective of the method that is used, independent of the spatial and temporal characteristics of the freshwater indices, and significant on all resolved timescales from years to decades.

Detailed changes are explained in the response letters, where the responses to the reviewers' comments are shown in blue, and the resulting changes in the manuscript are shown in red. Figure and line numbers refer to the revised manuscript without tracked changes:

- Reviewer 1: page 2
- Reviewer 2: page 17

The manuscript has greatly benefitted from the review. Thus, we thank both reviewers and the editor for reviewing and handling this manuscript, helping us to clarify and improve this study.

Sincerely,
Marilena Oltmanns, on behalf of all authors

**Responses to Reviewer 1**

The authors use statistical analysis of observations and reanalysis data to examine a potential link between freshwater anomalies in the North Atlantic subpolar gyre region and summer weather in Europe in the following year. They propose that stronger freshwater anomalies resulting from increased glacial runoff in Greenland leads to an stronger meridional SST gradient between the subpolar and subtropical gyre and consequently increased baroclinic instability in the atmosphere above. The resulting deflection of the jet stream alters the advection pathway of maritime air masses over Europe. The foundation of the analysis are freshwater indices derived from a mass balance equation that are used to identify freshwater anomalies in relation to simultaneous sea surface temperature (SST) anomalies linked to the North Atlantic Oscillation (NAO).

The revised version of the manuscript is a substantial improvement to the previous draft, particularly with respect to the derivation of the freshwater indices. However, there is still lack of clarity and detail in some methods, as well as precision in the description of the results. Therefore, the manuscript in its current form is not suitable for publication and I suggest major revisions based on my comments below.

We thank the reviewer for reviewing our manuscript once more, and for again providing many detailed and helpful comments and suggestions!

The main comments referred to a lack of clarity and detail in some methods, and a lack of precision in the description of the results. Following your comments and suggestions below, we have carried out a major revision to provide clearer and more detailed explanations of the methods, remove any ambiguities, and describe the results more precisely.

I. General Comments and Suggestions: Throughout the manuscript, multiple indices are used, but their exact definition unclear. Some details regarding the methods are often only mentioned in the figure captions leading to interruptions in the reading flow. Additionally, the description of results often remains rather vague or general, and does not necessarily match the figures. Similarly, I encourage the authors to be more concise and precise in their language, e.g., sometimes it is not immediately clear if they refer to the ocean or atmospheric anomalies or geographic descriptions are very broad. Some of these instances are mentioned in my comments below.

We thank the reviewer for making us aware of the remaining unclarities, vagueness, inaccuracies, or impreciseness in the manuscript.

The revised manuscript includes a clear, unambiguous derivation of the indices. We also motivate the use of multiple indices more clearly and explain their specific advantages and disadvantages. In addition, the text is now more specific and precisely describes the figures. All information in the figure captions is also provided in the main text. Special care is taken to differentiate between ocean and atmospheric anomalies and to clearly describe detailed, geographic features.

II. Main Comments:

1. Section 3: The combination of the appendices from the previous version with the main text helps better understanding and following the construction of the freshwater indices. However, there are a few points that require some clarification.

1. l. 141-143: Equation (3) already has the mean state subtracted, yet you say "we subtract the mean values from the resulting equation".

We agree that the term "resulting equation" was confusing. We have rephrased this paragraph and restructured the associated section. Thus, we now include more steps in the derivation. We have also changed the nomenclature to make it consistent with earlier studies, and we explain the individual terms more clearly (Section 3.1).

2. l. 145: Please specify the values of $\alpha$ and $\beta$.

We apologise that the preceding term "linearised equation of state" was confusing, because $\alpha$ and $\beta$ are not constant in space and time. Thus, we cannot specify their exact values.

We have now removed the term "linearised equation of state" and only specify $\alpha$ and $\beta$ as the thermal and haline expansion coefficients. Moreover, when we calculate the surface mass balance in Section 4.1, we now explain (line 271):

*"We further compute $\alpha$ and $\beta$ using the Gibbs Seawater Routines (McDougall et al., 2009), in accordance with the highest standards of current knowledge. Noting that the effects of salinity and pressure on $\alpha$ and $\beta$ are small and only affect the second decimal place or less, we use nominal values of 35 g kg$^{-1}$ and 10 db for the subpolar region in winter to compute $\alpha$ and $\beta$. The dependence of $\alpha$ and $\beta$ on temperature is larger, however. For instance, $\alpha$ can vary from $5 \cdot 10^{-5}$ to $18 \cdot 10^{-5}$ °C$^{-1}$ across the subpolar ocean surface. Thus, for an enhanced accuracy, we allow $\alpha$ and $\beta$ to vary with temperature."*

In the subsequent paragraph, we also point out that all terms in Eq. (5) are calculated before the regression onto the freshwater indices to ensure that Eq. (5) remains valid after the regression.

3. l. 165: There is no Figure 1a that shows the NAO in summer.

Thank you for pointing out that we referred to the wrong panel. In the revised version, this sentence no longer appears.

4. Section 3.2: Some important information/motivation is either missing or presented later in the text which makes it difficult to follow the approach and understand the derivation of the freshwater indices:

Thank you for letting us know that this section was still unclear. We have rewritten the full section and now provide a clear, unambiguous, mathematical derivation.

  1. It is unclear how the resulting freshwater indices are defined. Are they (scaled) summer NAO indices, i.e., FE = {-NAOS,i | NAOS < -0.5, i=FE years} and FW = {NAOS,i | NAOS > - 0.5, i=FW years}?

You are right that the $F_E$ and $F_W$ indices directly correspond to the $NAO_S$ values. They are not scaled and we now state this in the text (line 217).

Using the formulation you suggest would be misleading since the indices do not contain all $NAO_S$ values above or below -0.5 but subsets of it. Since there exists no simple expression, we now provide a clear mathematical derivation to avoid any ambiguity (Section 3.2, line 209 – 236). The resulting years are uniquely identifiable in Figure 1c, and we now also list them in Appendix B.

At the end of the derivation, we further clarified their naming (line 244): *"Since the associated cold anomalies are strongest over the southeastern subpolar region (Fig. 1e), we refer to the selected 8 years as $F_E$ subset – shown by the clear red coloured bars in Figure 1c. For the other subset, the maximum cold anomalies extend over the full subpolar gyre, including the western part (Fig. 1f). Thus, we refer to the selected 17 years as $F_W$ subset – shown by the clear blue coloured bars in Figure 1c."*

We further added a sensitivity analysis to the number of years included in the subsets (Appendix B). Thus, we show how the results change with alternative definitions, with the corresponding years clearly shown (Figs. B3 to B7).

2. l. 173-179: This paragraph is missing some motivation why and how you use the meridional SST gradient to establish a relationship between the summer NAO and freshwater anomalies in the following winter. I suggest something along the lines of: "Given the theoretical considerations based on the mass balance analysis (Section 3.1), there are certain conditions under which freshwater anomalies associated with the summer NAO are accompanied by temperature anomalies. To identify these conditions, we regress winter SST onto two subsets of the summer NAO…". Please give a rationale for the seasonal lag.

Thank you for this helpful suggestion. We have now integrated your suggestion.

Since we think this information needs to come earlier, we have rewritten the end of Section 3.1 and the beginning of Section 3.2. Thus, we now motivate the use of SST-based indices by the constraint of salinity on the SST (line 174), and then explain the use of SST anomalies associated with the summer NAO specifically (line 194).

We further explain the seasonal lag. First, we motivate focussing the SST analysis on the winter period (line 170): *"Considering the competing influences of salinity and temperature on stratification, the conditions, in which freshwater may impact the temperature, can only occur in autumn and winter, when surface water is cooled by the atmosphere, becomes denser and sinks. In summer, the temperature and salinity do not compete in their influence on stratification and thus, do not constrain each other."*

In the subsequent derivation of the indices, we further explain (line 199): *"[...] Yet, even if the freshening occurs in summer (when melting and runoff is strongest), the effect of the freshwater on the SST would only become visible in autumn and winter (when the freshwater impedes the sinking of surface water). Thus, we focus on the SST in winter to infer the potential existence of freshwater anomalies."*

The seasonal lag of ~6 months is well in line with the observed and theoretical propagation time of seasonal runoff and meltwater from the boundary currents into the interior subpolar region. For instance, a typical length scale from the boundary to the interior is 1,000,000 m and a typical timescale from summer to winter is $3600 \cdot 24 \cdot 30 \cdot 6$ seconds (= 6 months). Thus, the minimum current speeds required to distribute freshwater over the subpolar region is ~0.06 m s$^{-1}$, which is smaller than the observed velocities ranging from up to 1 m s$^{-1}$ near the boundary to 0.1 m s$^{-1}$ in the interior (based on the altimetry).

A more detailed description of the seasonal freshening with additional references is provided in Section 4.2.

3. l. 174-175 & l. 181: Is the "meridional SST gradient" the same as $\Delta$SST? Please define $\Delta$SST in the text.

To avoid confusion, we have removed the word "meridional SST gradient". Instead, we state that we observe an increased SST difference between the subtropical and subpolar gyres. We have also added two panels in Figure 1 (Fig. 1a and b) to confirm this statement and show the associated SST anomalies.

4. l. 180: The "conditions c" are not explicitly stated in the text nor referred to in the remainder of the manuscript. I suggest to remove the variable name "c".

Thank you for suggestion this. We agree with you and have removed the variable name "c", as you suggest.

5. l. 200-203: The significant SST anomalies associated with FW extend from the Labrador Sea in the west almost all the way to the British Isles in the east while the SST anomalies associated with FE occur mostly in the central North Atlantic subpolar gyre region. Thus the naming can be misleading unless you specifically refer to maximum cold anomalies.

Thank you for pointing this out. We have followed your suggestion and now specifically refer to the maximum cold anomalies. In addition, we now describe the different spatial characteristics of the cold anomalies more precisely.

2. Section 4.7: This section lacks a clear structure, clarity, and important details that make it hard to reconcile the results with the rest of the paper. In Section 3.2, you establish a non-linear relationship between a subset of summer NAO and SST anomalies associated with freshwater anomalies. Here, you use the "full, un-subsampled summer NAO" as a linear predictor to "assess the role of freshwater triggering the SST signal". By regressing different variables onto the summer NAO and resulting SST pattern (which is quite different compared to the SST patterns related to FE and FW, cf. Figures 6a and 7a with Figure 10a), you describe the associated atmospheric response in both observations and models. As you point out in your concluding sentence of this section, "further studies are necessary to confirm the dynamical contribution of freshwater anomalies to the largescale SST pattern". Since an explicit link to freshwater anomalies, and their role in triggering your proposed chain reaction, is missing, this section creates confusion for the reader. I suggest to add more details following my comments below or remove this section entirely from the manuscript.

Thank you for pointing out that this section was unclear. We have now removed it entirely from the manuscript. However, we still answer your questions about this section below.

By referring to different subsets of years, different timescales and time periods, the previous Figures 6, 7 and 10 from the last manuscript cannot be compared to each other and do not contradict each other. In Section 3.2, we now clearly explain the temporal and spatial complexity of freshwater variability in the subpolar region, resulting from different drivers of freshwater variations with different timescales and different spatial characteristics. The complex 3-dimensional variability of freshwater in space and time cannot fully and precisely be captured by a single, one-dimensional index. Thus, we use several indices. By regressing the SST (or SSS estimates) on one specific freshwater index, we reduce the influence of other drivers of freshwater variations and obtain a near-linear relationship.

The full, unsampled summer NAO was used as an indicator for one component of the freshening – the component that is attributed to seasonal melting or runoff (Fig. 3a and b). The associated freshening is strongest over the eastern subpolar region, and it is characterised by a significant trend.

However, we agree that the mix of the different NAO indices was confusing. Thus, we have removed this section, and we do not use the full, un-subsampled NAO index anymore. Moreover, in the revised manuscript, we took care to clearly explain the different characteristics, advantages, and disadvantages of each freshwater index.

The following comments (1 to 9) refer to an analysis that we have now fully removed. While the analysis was meant to provide insights into the processes associated with the SST anomalies in summer, we agree that it is not necessary for understanding the link between freshwater and SST anomalies. Thus, to improve clarity and conciseness, we have removed it.

1. Please specify what you are trying to predict using the "un-subsampled summer NAO as predictor".

The un-subsampled summer NAO was used an index for the seasonal freshening and runoff, and it was used as a predictor for the summer weather in the subsequent year.

2. l. 420: Please clarify what you mean by "reduced NAO states".

We referred to years in which the summer NAO index was small (or more negative).

3. l. 422: I believe this should be Figure 10a-c.

Yes, that is correct. Thank you for noticing this.

4. l. 423: Please discuss the significance that the warm anomalies occur mostly over land around the Mediterranean Sea, but the dry anomalies over the ocean.

The regions of the warm and dry anomalies did not exactly coincide which we attribute to their different drivers within extra-tropical weather systems. While we have removed this section, we include this information now in Section 4.4, where we state (line 437):

*"Considering that precipitation anomalies preferentially occur along trailing cold fronts and are shifted southward relative to cyclone centres (Booth et al., 2018; Kodama et al., 2019), the observed displacement of the dry anomalies relative to the warm anomalies is expected from their locations within individual weather systems and consistent with other studies (Yu et al., 2023)."*

5. l. 429: Please specify that the upwelling occurs in the ocean below the center of the cyclone.

Yes, that is correct. Since the Ekman transports on the northern hemisphere are directed to the right relative to the wind direction, all atmospheric cyclones in the extra-tropical region lead to upwelling in their centres.

6. l. 442-443: Please define SSTFW explicitly. Is this related to the FW index? If not, I suggest to use a different name to avoid confusion.

The index referred to the variability of the SST pattern in the panel next to it. We have now removed the index, and we do not use this name anywhere in the revised manuscript.

7. l. 443-444: This sentence is unclear. Please specify which "observational analysis" you are referring to. Does that mean you define the SST index based on the model simulations?

Since the model simulations were forced by the observed SST, the index used in the simulations was the same as that in the observations. We carried out the same regressions of temperature and P-E in the observations (obtained from ERA5) and simulations (obtained from the models) to allow for a direct comparison.

8. l. 446: Please add more details in the text: Are you referring to temperature and precipitation anomalies from climatology or ensemble mean?

We referred to the anomalies obtained by regressing the precipitation and temperature variability in all simulation runs onto the freshwater index. Thus, the anomalies were relative to the ensemble mean climatology.

9. l. 449-450: The observed and simulated atmospheric responses agree well over the ocean, while the simulated response over land is much smaller. It is not clear how you determined the statistical significance from the 90 ensemble simulations given that deems very small anomalies close to zero as statistically significant – please add more details in the text and discuss the implications.

The significance was obtained from Student t-tests. Due to the high number of realisations (90 simulations x 40 years = 3600 realisations), the warm and dry anomalies do not all occur at the same location. They are shifted over different areas and sometimes cancel each other out, for instance when the jet stream is shifted southward or northward. Thus, given the high number of realisations, it is expected that the magnitude of the resulting warm and dry anomalies, obtained from the simulations, is smaller and distributed over a larger area compared to the observations that only include 40 years or realisations. This was consistent with our results.

There is a trade-off between the number of degrees of freedom and the magnitude of the correlations and regressions. It is expected that, with increasing number of simulations and increasing degrees of freedom, the magnitude of the correlations and regressions decreases

because of the increased variance of the temperature and precipitation variability. However, the significance of the correlations and regressions is unaffected because the reduced correlations are compensated for by the higher number of degrees of freedom.

In the revised manuscript, we explain the trade-off between the number of degrees of freedom and the magnitude of the correlation in Sections 3.2 and 4.6 and in Appendix B.

Again, we thank you for providing the above comments. We have fully removed the corresponding analysis (the previous Section 4.6 and 4.7). However, we took care that we have integrated your comments in the other sections to avoid any potential misunderstandings in the revised manuscript.

III. Additional Comments and Suggestions:

1. l. 4: What are "medium-term climate trajectories"?

We referred to seasonal to interannual climate evolution. We have now removed this term and the associated sentence.

2. l. 97: Please state years of "recent period"

We have now removed this term. Instead, we state: *"Over the investigated period [1979 to 2022], the climate system has been characterised by increasing greenhouse gas concentrations."*

3. l. 101-105: The 2 m-temperature trends are different over land than over the ocean (e.g., Simmons, 2022). Your averaging area includes large parts of the North Atlantic. Please discuss if this has any implications for your results.

Thank you for pointing out this study by Simmons (2022). The trends in Simmons (2022) show the 2-m air temperature trends. Over the North Atlantic and Europe, the trends largely show the signal that we explain in our study by changes in the ocean and atmospheric circulations following enhanced freshening in the subpolar North Atlantic. Thus, the study nicely supports our method of subtracting regionally averaged trends.

Since, in this study, we are specifically interested in the warming and cooling effects induced by changes in the ocean and atmospheric circulations, removing this trend signal would be counterproductive. After all, it represents the central motivation behind subtracting regionally averaged trends.

We have now included the reference in the manuscript and explain (line 101):

*"[...] Considering that the freshening trend is an important part of the signal we are investigating, removing trends at each location (or grid point) would remove an important part of a signal that we are interested in. Thus, to reduce the influence of increasing greenhouse gas concentrations on European air temperatures, we subtract regionally averaged trends from the air temperature. The method of subtracting regionally averaged trends is motivated by the*

*observation that greenhouse gases are distributed comparatively uniformly in the atmosphere (Reuter et al., 2020) whereas the observed surface warming exhibits large regional differences (Simmons et al., 2022). These regional differences in surface warming result from changes in the ocean and atmospheric circulations, which are redistributing the excess heat. Since, in this study, we are specifically interested in these dynamic processes associated with changes in the ocean and atmospheric circulations, we are subtracting a spatially uniform warming trend associated with increasing greenhouse gases."*

4. l. 230-231: This is only true in the absence of vertical mixing.

Yes. We now explain in Section 4.1 that vertical mixing cannot explain the SST anomalies. We agree that this sentence raised confusion since we did not clearly explain the different roles of the different terms in the mass balance equation. Vertical mixing can have two causes: (1) passive entrainment in which case it only affects the mixed layer depth but not its density, and (2) an active energy source that mixes denser water upward (against gravity).

In the revised manuscript, we have changed the nomenclature of the mass balance, Eq. (1), to make it consistent with earlier studies (e.g., Griffies and Greatbatch, 2012), and we clearly explain the individual terms. Thus, we clearly distinguish between passive entrainment and active drivers of vertical mixing (line 133):

*"The passive component is defined as the entrainment of mass into the mixed layer that results from mixed layer deepening as the mixed layer density increases. The active component results from externally forced, horizontal and vertical mass fluxes, such as wind-driven Ekman transports and upwelling. The passive component can only change the mixed layer depth, but not its density, while the active component does change the mixed layer density."*

In the subsequent derivation in Section 3.1, the passive entrainment term cancels out since it is not relevant for the mixed layer density. Moreover, we have included additional references providing a detailed evaluation of the energy sources and forcings that drive ocean currents and vertical mixing (Ferrari and Wunsch 2009; Wunsch and Ferrari 2004). These studies agree that, on the timescales and spatial scales we consider, the wind stresses and air-sea fluxes are by far the largest energy source that can drive horizontal and vertical motion and hence, mass fluxes. Additional sources include biological activity, tides, geothermal activity, and others. However, they are negligible on the timescales and spatial scales considered (line 276 – 284).

The influence of the largest and most important, active drivers of vertical mixing over the open ocean, away from topographic boundaries, and on interannual timescales – the wind forcing and air-sea fluxes – is evaluated as part of the mass balance analysis (Section 4.1, Appendix A). Thus, we find that both drivers are not significantly correlated with our freshwater indices, their magnitudes are also too small, and their spatial characteristics also do not agree with those of the identified SST anomalies (line 285 – 292).

In conclusion: Vertical mixing does exist in the mass balance equation, but it cannot drive a substantial mass flux across the base of the mixed layer that is comparable to the density anomaly implied by the identified temperature anomalies.

5. l. 238-239: To keep the reader on board it would be helpful to state that "using this relationship, we estimate SSS from SST anomalies for the two NAOS subsets".

Thank you for suggestion this. We have now included this statement (line 300).

6. Figure 3: Did you actually do the regression? Given the linear relationship between SSS and SST, and assuming a constant coefficients, panels a and b should be equal to α/β * (negative areas in Figures 2a and b). However, the SSS anomalies show more structure. Is that due to the choice of contour levels? Please clarify.

The relationship between SSS and SST anomalies is not linear because $\alpha$ and $\beta$ depend on temperature. Thus, they must show a different structure.

By integrating your second comment in Section 4.1, and explaining how $\alpha$ and $\beta$ are calculated (line 271), we think this has been clarified.

7. l. 241-242: Please be more precise: the significant area of FW SSS anomalies extend further eastward compared to the anomalies associated with FE.

We are now more precise and state (Line 301): *"[] the maximum freshwater anomalies (or minimum SSS anomalies) associated with $F_E$ occur over the central subpolar region (corresponding to the south-eastern subpolar gyre) and are spatially more confined than the maximum freshwater anomalies associated with $F_W$. Moreover, the significant area of $F_W$ freshwater anomalies extends further eastward, westward, and northward compared to $F_E$ freshwater anomalies and the anomalies have a smaller amplitude, consistent with the associated cold SST anomalies (Fig. 1e and f)."*

8. l. 244: Please refer to the appendix to ensure that the reader knows how the uncertainties were estimated.

Following your suggestion, we now refer to the appendix. Thank you for suggesting this.

In addition, we state in the main manuscript (line 307): *"These uncertainties apply to the cold anomaly regions, enclosed by the 95% lines. Uncertainties at each individual grid point can differ."*

9. l. 252-253: The different phases of the NAO are not associated with "opposite atmospheric circulation patterns". They differ in the strength of the pressure difference between the subpolar low and subtropical high pressure system. In this regard, the wording "lower/higher NAO states" seems slightly odd. I suggest to make explicit links to FE and FW throughout the whole text.

We agree that the term "opposite atmospheric circulation patterns" was misleading. Instead, we should have referred to "opposite atmospheric circulation anomalies" since $F_E$ freshwater anomalies are associated with a more anticyclonic atmospheric circulation anomaly over Greenland and the subpolar North Atlantic in the preceding summer, while $F_W$ freshwater anomalies are preceded by a more cyclonic atmospheric circulation anomaly over Greenland and the subpolar North Atlantic.

However, after revising the manuscript, we found that this sentence is not needed and mostly raised confusion. Thus, we have now fully removed it. In addition, we make explicit links to $F_E$ and $F_W$ throughout the text.

10. l. 280-281: It is unclear how you identify "an intensified subpolar gyre circulation in the Labrador Sea" in Figure 4c.

*Indeed, this sentence was misleading. We have now removed it.*

11. Figure 4: What are the vectors in panel b? Please define dSSS in the text: it is unclear what you mean by "newly arriving, seasonal surface freshening from summer (August) to winter (January to March). Do you accumulate or take the mean of the freshwater anomalies during the winter months and subtract it from the August value?

*We now specify: "The arrows represent the mean geostrophic surface flow, obtained from the absolute dynamic topography, averaged from August to March (the period of the freshening)."*

*Regarding your second question: Yes, we subtract the August value from the winter value to calculate the accumulated freshwater anomalies from summer to winter. This is now clarified both in the figure caption and the main text (line 329).*

12. l. 299-301: If the "seasonal freshening is mixed down and too small to affect the absolute SST anomaly in winter" when NAOS ≥ -0.5, does that not contradict the construction of the FW indices or does the sentence refer to the years that were excluded in the optimization process?

*This may have been clarified by answering your preceding question. The summer NAO is linearly correlated with the seasonal SSS change from summer to winter but it has a nonlinear relationship with absolute SSS anomalies in winter.*

*When the $NAO_S$ > -0.5, the seasonal mixed layer is eroded before an absolute cold SST and freshwater anomaly develops. In that case, the seasonal freshening (which is typically confined to a shallow surface layer) is too small to affect the SST because it mixes with deeper and larger volumes of freshwater. In this case, other sources of freshening – particularly the advection of cold and fresh polar water into the subpolar region – are more important. These other sources are filtered out by considering seasonal differences since they have no strong seasonal dependence (when compared to runoff, which has a very strong seasonal dependence).*

*Thus, there is no contradiction. We just need to use a different freshwater index to describe the freshwater variability in this regime.*

*We have now removed the sentence to avoid any confusion. Instead, we no simply state (line 329): "While the full, un-subsampled summer NAO is a suitable indicator of the seasonal freshwater input from summer to winter, it is not necessarily correlated with absolute SSS anomalies in winter. Once a seasonal mixed layer is eroded, the SST and surface salinity are expected to be influenced by other factors, consistent with the nonlinear relationship between the summer NAO and subsequent winter SST anomalies (Fig. 1d)."*

13. l. 354-355: Please discuss the fact that the SST anomalies in Figure 6a are largely not statistically significant. What are the implications for your conclusions?

*The SST difference between the warmer subtropical region and the colder subpolar region is highly significantly correlated with both freshwater indices. The significance of absolute SST anomalies does not show at individual grid points because the exact location of the SST front*

between the subtropical and the subpolar gyre differs across the years and is therefore poorly constrained by the regressions. However, when considering larger areas and spatial differences instead of individual grid points, the increased SST difference between the warm subtropical and the cold subpolar region is very robust.

In the revised manuscript, we explain (line 415):

*"[...] The SST signal in both summers after the freshwater anomalies implies an increased SST difference between the warm subtropical gyre and the cold subpolar gyre. The exact location of the SST front between the subtropical gyre and the subpolar gyre can differ between the years included in the subset and is therefore poorly constrained, resulting in reduced significances at individual grid points. However, the resulting increased SST gradient – which is of greater dynamical relevance than absolute SST anomalies – is highly significant. For instance, the SST difference between the region, in which the SST anomalies exceed 2 °C and the region, in which the SST anomaly falls below –2 °C, includes a substantial area of the extra-tropical North Atlantic (Fig. 5a and b) and is significantly correlated with the $F_E$ index with a correlation coefficient well above 0.7 in both summers (r ≈ 0.76 and 0.84 in the first and second summer respectively), with p-values well below 0.05"*

Accordingly, we have updated the figure panels to include the +/- 2 °C lines (now Figure 5a and b).

14. l. 359-361: This sentence is unclear. I don't understand what you mean by " the anticyclonic circulation anomaly is in part rotated over the continent".

In winter, we show that there is an increased SST gradient between the subpolar and the subtropical gyres (Fig. 1e and f). Associated with this SST signal is a more cyclonic atmospheric circulation anomaly over the subpolar region and a more anticyclonic atmospheric circulation anomaly to the south of the SST front (Fig. 4b).

In summer, there is an additional temperature contrast across the coastline because the land typically warms up faster in summer. Thus, in contrast to winter, there is not only an increased SST front over the North Atlantic, but also between the subpolar North Atlantic region and the European continent. Accordingly, we found that the more anti-cyclonic atmospheric circulation anomaly (that was over the subtropical region in winter) is now shifted over the continent.

We agree that the word "rotated" was misleading. Also, we do not show the anti-cyclonic circulation anomaly since it is not needed. We already show the meridional wind velocities. Thus, we have removed the sentence.

15. l. 362: Interestingly, the warm and dry anomalies during the first summer are not necessarily colocated. Please discuss/speculate why the dry anomalies occur more toward the southeast.

Thank you for pointing this out. We attribute the spatial differences to the different drivers of temperature and precipitation anomalies within individual weather systems such as cyclones and anticyclones. However, we believe a more in-depth discussion of the detailed dynamical features would go beyond the scope of the study. Thus, we now state (line 437):

*"Considering that precipitation anomalies preferentially occur along trailing cold fronts and are shifted southward relative to cyclone centres (Booth et al., 2018; Kodama et al., 2019), the*

*observed displacement of the dry anomalies relative to the warm anomalies is expected from their locations within individual weather systems and consistent with other studies (Yu et al., 2023)."*

16. l. 364: Is the "cold SST anomaly" in Figure 7a statistically significant other than the very small region around 40 °N, 40 °W? What are the implications for your conclusions?

Please refer to our response to your Comment 13 above. We believe this has now been clarified.

17. l. 367: Please discuss why, after FW freshwater anomalies, the warm anomalies occur over France and Great Britain while the dry anomalies occur more to the northeast over the Baltic region.

Please refer to our response to your Comment 15 above. We believe this has now been clarified.

In addition, we now clearly describe the different regions in the text.

18. Figure 6: The coastlines in panels c and d are hard to make out. This makes it difficult to relate it to the other maps, particularly since each row shows a different domain.

Thank you for pointing this out. Since the figure panels c and d were already quite busy, and we already show the arrows in panels a and b, we have removed the arrows from panels c and d. Thus, the coastline is now easier to see (now Fig. 5c and d).

19. Figures 6 & 7: Does "SST with the 700-hPa winds (…) on FE/FW" mean you performed a multiple linear regression? Please provide more detail in the text.

We regressed the SST and the 700 hPa separately onto the freshwater index. This is now clarified in both figure captions. We also better describe the identified wind field more in the text.

20. l. 385: Please be more precise: Does "cold anomaly over the North Atlantic" refer to the composite of air temperature (Figure 8b) or the SST anomaly in the western subpolar gyre region (Figure 8e)?

We are now more precise and specify "cold SST anomaly".

21. Figure 8: What are the red lines in panel f?

Thank you for spotting that there were lines at the edge of the panel. They referred to positive salinity anomalies. However, since we did not calculate SSS in the associated regions, the lines were misplaced.

We have now removed the lines.

22. l. 400: Please clarify what you mean by "SST variability each summer". Are you referring to the spatial distribution of summer SST anomalies?

We referred to the time variability of the spatial SST pattern. However, this analysis has now been removed to improve the overall clarity and conciseness.

23. l. 403-406: What is the correlation of the SST pattern with the summer NAO? It might be worthwhile adding the NAO time series in Figure 9e to demonstrate the relationship.

Thank you for suggesting this. We have now removed this figure and the associated section.

If we would have calculated the correlation, it would have depended on the underlying period. Over the last two decades, the time series would have been highly correlated. Over the earlier period, they would have been only weakly correlated. Along with this switch, we observe a shift of power from decadal variability towards interannual climate variability both in ocean and atmospheric variables. However, to improve clarity and conciseness, we have shifted this analysis into another manuscript.

24. l. 407-410: To keep the reader onboard, I suggest to add a sentence that links the SST variability back to the freshwater anomalies.

Thank you for suggesting this. We have removed this analysis to improve the overall clarity and conciseness of the manuscript.

25. Figure 9f: Please add a label for the x-axis.

Thank you for spotting this. This figure has now been removed.

26. l. 506: Please specify if you refer to an "increased meridional temperature gradient" in the ocean or atmosphere.

We referred to an increased meridional SST gradient, although we would expect that the increased SST gradient affects the meridional temperature gradient in the lower troposphere.

We have now removed this analysis.

27. Figure A2: Why are the regions enclosing the 95% significance level different from Figure 2b?

Thank you for spotting this. In Figure 2b, we had calculated the significance level of the correlation, while in Figure A2 we had used the significance level from the regression, which is the correct one, since we are showing the regressions. The difference is only minor, however.

We have now corrected the significance lines in Figure 1 (previous Figure 2).

28. l. 580-581: The cold anomaly regions cover the entire subpolar North Atlantic including areas of deep convection. Please discuss the implications and added uncertainty using a spatially constant value for the mean winter mixed layer depth.

Since the average was obtained from Argo floats, which are more often found in regions of shallower mixed layers (because these regions cover a larger area), the average may underestimate the true mixed layer depth, particularly in the regions of deep ocean convection. Using a deeper mixed layer depth in the mass balance equation increases the magnitude of the terms on the lefthand side of Eq. (5). Thus, the uncertainties that would result from using deeper

mixed layers would be smaller than the ones we provided. However, using a different mixed layer depth has only small effects on the uncertainty.

To remind the reader that we are using spatial averages, we now state (line 307): *"[...] The uncertainties apply to the cold anomaly regions, enclosed by the 95% lines. Uncertainties at each individual grid point can differ."*

If there are much shallower mixed layer depths compared to the mean mixed layer depth, sampled by the Argo floats, the surface mass balance may underestimate (but not overestimate) the surface freshening. In that case, the stratification must be controlled by salinity and the salinity anomalies even overcompensate the density increase implied by the cold anomalies (to justify that there shallower mixed layers in the regions of cold anomalies).

We now explain this in the text (line 308): *"[...] If the freshwater forcing is very large, the surface mass balance may underestimate the freshening because freshwater anomalies can (in theory) decrease the surface salinity up to a threshold of near zero, while SST anomalies cannot drop below the air temperature. Still, we find that even during the strong observed freshwater anomalies in 2015 and 2016, the surface mass balance provided a good approximation (Appendix A), suggesting that a potential underestimate of the obtained freshwater anomalies is only small."*

IV. Typos/Wording:

I suggest the following changes:

l. 36: "requiring a high grid spacing of ~1/12° " to "requiring ocean models with high grid spacing of at least ~1/12°"
l. 64: "spatial resolution" to "grid spacing"
l. 84: "run at a resolution of" to "run with grid spacing of" l. 158: "surface temperature" to "ocean surface temperature"
l. 160: "subpolar cold anomalies" to "subpolar SST cold anomalies"
l. 189: Please remove second "warm".
l. 317: "to north of" to either "north of" or "to the north of"
l. 326: "resolution" to "grid spacing"
l. 327: "resolution" to grid spacing"
l. 372: "subsets" to "subset"
l. 416: "autocorrelations" to "autocorrelation"
l. 417: "point" to "point out"
l. 435: "of SST signal" to "of the SST signal"

We have now integrated all suggestions in the text. Thank you for providing these suggestions!

Again, we sincerely thank the reviewer for their careful and thorough assessment of our manuscript, providing many helpful and constructive suggestions. Your comments have helped us to remove any remaining ambiguities and make the manuscript much clearer.

**References:**

Ferrari, R., & Wunsch, C. (2009). Ocean circulation kinetic energy: Reservoirs, sources, and sinks. *Annual Review of Fluid Mechanics*, *41*, 253-282.

Gill, A. E. (1982). *Atmosphere-ocean dynamics* (Vol. 30). Academic press.

Griffies, S. M., & Greatbatch, R. J. (2012). Physical processes that impact the evolution of global mean sea level in ocean climate models. *Ocean Modelling*, *51*, 37-72.

Wunsch, C., & Ferrari, R. (2004). Vertical mixing, energy, and the general circulation of the oceans. *Annu. Rev. Fluid Mech.*, *36*, 281-314.

**Responses to Reviewer 2**

I would like to thank the authors for responding to my points. This has led to significant changes to the paper, many of which are positive. However, it also led to many wholly new analyses being presented that didn't quite address the concerns the last time, and the presentation is still confusing in areas, with either insufficient information or statistical rigor to fully follow the arguments. Therefore, although I still find the ideas being presented here to be highly interesting and exciting, I am left feeling that the paper is not publishable in its current state. In particular, I feel as though it just does not do enough to discuss the uncertainties in the analysis, and has too much certainty in its conclusions. I list my major issues below followed by minor points (which may overlap into the more major ones).

We sincerely thank the reviewer for reviewing our manuscript once more and providing many helpful and constructive comments and suggestions!

Following your comments and suggestions below, we have carried out a major revision, paying particular attention to your questions regarding uncertainties, significance, and the phrasing of the conclusions. There has been some degree of repetition in a few comments due to overlapping concerns. We have tried to avoid repetition in our responses by referring to earlier comments but could not avoid some overlap. We apologise for any inconvenience.

Major points

The discussion on subselecting is still confusing - I think this is because there are two steps that are discussed as one and referred to as "subsampling"? For example, there is the choice to split the record into two, i.e., above and below the value of 0.5 for SNAO, and then again where individual years are removed. I understand the first part of this sub-selection, and it seems warranted given the relationship between SST gradients and summer NAO. But I am still unsure of the second, which seems more focused on getting a higher correlation between the two. I still do not understand the physical basis for this, and am worried that it over emphasizes the potential role of this mechanism. Indeed, the authors say they are not worried about the mechanism, but I'm not sure how they can be so certain. I guess in particular, the logic seems not ideal here, as the authors state they are looking for years where SSTs are being driven by freshwater, but then focus on the relationship with the summer NAO. An obvious question is whether the same years come out if you first look for years where you think the SST anomalies are driven by FW changes? This would go some way to better understand the thinking here.

Thank you for pointing out that the method of subsampling was still confusing. To address your concern, we have now made the following changes:

(1) We have removed the sentence that the indices are used for a purely statistical purpose since it was confusing. Please note that the NAO is, itself, only an index. It is by no means more physical than any subset of it. In the manuscript, we now provide a clearer dynamical motivation for the subsampling (line 184 – 194):

*"The challenge in detecting the conditions in which freshwater anomalies may have affected the SST, consists in the complexity of SST and freshwater variability in the subpolar region. In theory, changes in surface freshwater can be influenced by river runoff, sea ice and glacial*

*melting, evaporation and precipitation, mixing, and ocean currents. Considering that multiple factors can contribute to freshwater variations over a range of timescales and spatial scales, it may not be possible to reduce the complexity of freshwater variability in space and time into a single, one-dimensional index.*

*To overcome this challenge, we construct indices over subsets of years that allow us to closely constrain the variability of the SST, over the selected subset. Thus, this approach is different to traditional methods in which the dynamical mechanisms are known a priori, and statistical methods are used to assess the significance of these mechanisms. Here, we first select indices with a strong and significant statistical relationship with the SST, and then look for potential freshening mechanisms that can explain the relationship, assuming that these mechanisms exist but may be masked by other drivers."*

*Subsequently, we find (228): "[The subsampling] represents a powerful method for increasing the statistical relationship between two variables and thus identifying dynamical links, based on the assumption that noise, and other mechanisms, can mask these links. Once a strong statistical connection has been established, the physical basis is assessed by investigating the associated dynamical links with freshwater anomalies."*

(2) In Section 4.2, we show that the subsampled index has a higher correlation with the physical causes of freshwater anomalies compared to the un-subsampled index, providing a physical justification (Fig. 3c and d). We have now clarified and shortened this section, stating (line 346):

*"To assess the role of the wind stress curl and subpolar gyre circulation for the cold and fresh anomalies associated with higher summer NAO states, we inspect the associated absolute dynamic topography in winter. The full, un-subsampled summer NAO only displays a weak and mostly non-significant relationship with the geostrophic surface circulation in the southwest subpolar region (Fig. 3c). When using the sub-sampled summer NAO corresponding to the $F_W$ subset, however, the absolute dynamic topography north of 50 °N in winter is significantly reduced, implying a more cyclonic and hence, stronger subpolar gyre circulation in the northwest subpolar region (Fig. 3d). The strengthened relationship between the subsampled summer NAO and the subpolar gyre circulation thus supports the subsampling by providing a physical explanation for the freshwater anomalies associated with $F_W$ (Fig. 2b)."*

(3) We now show that the results are not sensitive to the subsampling nor the number of years included (Section 4.5 and Appendix B). In the derivation, we explain (line 237):

*"In Section 4.5 and Appendix B, we show that the results are not sensitive to the subsampling or the number of years included. However, having a close relationship between the index and the SST results in reduced uncertainties when estimating the associated freshwater anomalies. In addition, the high correlations help us to identify and assess potential dynamical links more clearly: Freshwater indices that are only poorly correlated with freshwater are only of limited use when assessing links between freshwater and other ocean or atmospheric parameters. Since the indices will be used as a tool for representing freshwater anomalies, high correlations between the indices, the SST and potential freshwater anomalies are a prerequisite, not a conclusion, and we make no assumptions on the suitability of both subsets outside the selected years."*

(4) We have followed your suggestion and introduced a new index, which is solely based on the SST and not subsampled (Section 4.5). As with the other two indices, we find that the un-subsampled SST index is significantly correlated with freshwater anomalies in winter and with European summer weather in the subsequent summer (Fig. 7).

An advantage of the SST index is, that is based on all years and does not involve any subsampling. However, by using an un-subsampled SST index, we need to accept a greater loss in explained variance when linking the identified anomalies in European summer weather back to freshwater anomalies. Specifically, the correlation of the index with the freshwater anomalies is reduced, amounting only to up to 0.8 over the subpolar region (Fig. 7b). Thus, a critical reader might argue: "Are the identified summer weather anomalies really linked to freshwater anomalies if the freshwater index can only explain 64% of the variance of the freshwater variability?"

This loss in explained variance can be understood by considering the spatial and temporal complexity of freshwater variations in the subpolar region. The advantage of using the optimised (subsampled) freshwater indices is that they constrain the freshwater variability more closely over selected subsets. In a statistical sense, if we express the variability of European summer weather ($y_i$) in terms of freshwater ($x_i$), we need to account for two steps in the linear regression for European summer weather ($y$):

$y = r_1 \cdot r_2 \cdot x + residual$, where $r_1$ is the regression of European summer weather on the freshwater index and $r_2$ is the regression of the freshwater index on the actual freshwater variability.

A smaller correlation between freshwater and the freshwater index reduces the capability of the index to represent freshwater anomalies and link the obtained summer weather anomalies back to the freshwater anomalies. Using subsampled indices, the correlations exceed 0.9 over large parts of the subpolar region (Fig. 2c and d). Thus, the loss of explained variance is smaller, supporting the identified links to freshwater more strongly.

(5) In Sections 3.2 and 4.6 and in Appendix B, we explain the trade-off between the number of years included in the index and its correlation. To address this trade-off, we now provide three freshwater indices with different correlations, different spatial and temporal characteristics, and different numbers of years. In each case, we clearly state the advantages, and disadvantages of the respective index.

Overall, we find: The higher the correlation is that the freshwater index has with the freshwater anomalies (and hence the better constrained the freshwater anomalies are), the stronger is also the relationship of the index with European summer weather (line, Fig. 9).

We agree that having an additional index linked to the SST directly, covering all years, nicely complements the analysis. Thus, we have followed your suggestion and added this index.

Thank you for suggesting this!

Although there has been some attempt to explain the physical relationships - I'm still uneasy about the explanation for the two regimes of stronger SST gradients from only the freshwater fluxes related to the summer NAO. In the response the authors say that "The threshold of ~ -

0.5 in the summer NAO corresponds to a critical surface freshening above which the shallower, seasonal freshwater anomalies are mixed down", but supply no evidence. I take it that this is an assumption? I agree that this could be one interpretation, but another is that the SST gradient is not being driven by the same processes (i.e., freshwater) in the other cases?

Thank you for pointing out that the physical cause was still unclear.

We did not attribute the cold anomalies to the freshwater anomalies, and we have clarified this in the revised version. For the $F_W$ regime, specifically, we identify a stronger subpolar gyre circulation advecting cold and fresh polar water into the region, consistent with earlier studies. Thus, we attribute the cold and fresh anomalies to the subpolar gyre circulation rather than attributing the cold anomalies to the freshwater anomalies.

Being cold and fresh however, it is possible that the freshwater affected the SST prior to entering the subpolar region, in the Arctic. The Arctic is defined as a $\beta$-ocean in which stratification is dominated by salinity changes (Stewart and Haine, 2016). Thus, there is a pronounced halocline with a cold fresh layer of polar water overlying a warm, saline layer of Atlantic water (Aagard et al., 1981). If there were no freshwater, the surface water would mix with deeper, warmer water. Thus, one might still argue that the amount of freshwater constrained the vertical mixing and hence, the SST, prior to entering the subpolar region by influencing the T-S characteristics of the polar water. However, we do not make this argument in the text since the connection with the subpolar gyre circulation is the more direct link.

We agree that the statement *"0.5 corresponds to threshold, above seasonal freshwater anomalies are mixed down"* included too much interpretation. Thus, we have removed this sentence.

We now simply state that both NAO regimes are associated with freshwater anomalies (Fig. 2, line). Moreover, for different NAO regimes, different drivers of the freshwater anomalies are important, which is well supported by our findings (Fig. 3, Section 4.2) and earlier studies, both for the $F_E$ subset, which is linked to runoff and seasonal melting (Bamber et al. 2018, Dukhovskoy et al., 2019), and for the $F_W$ subset, which is linked to the subpolar gyre current system (Häkkinen and Rhines 2009; Häkkinen et al., 2011, 2013; Holliday et al., 2020).

I don't fully understand the logic behind the mass balance either, and I worry it is missing key processes (and hence overestimating the role of freshwater). In particular, the authors basically assume that geostrophic currents do not contribute to the SST anomalies (and in the response argue that it is because they are in parallel to density contours). This is obviously the case, but how do you take account of weakened or stronger geostrophic currents and, hence, ocean heat transport convergence anomalies? This may be a bit simple minded, but if the NAC weakens and transports less warm water into the eastern subpolar gyre, but the sea water is being subject to the climatological heat loss, then this will cause a cooling. Another process that isn't taken account for is whether there is a shift in the currents. Confusingly, such a mechanism is discussed in terms of the ocean response to the summer NAO, and is argued to make key changes to temperature, but the mass balance model (which is used to argue that freshwater changes are the key process) doesn't take into account these processes, as far as I can tell.

Thank you for indicating that the mass balance was still unclear.

(1) First, we would like to reassure you that the mass balance considers all physical causes of mass changes. We have added further references using the same equation (e.g., Griffies and Greatbatch, 2012) and associated studies providing a detailed evaluation of the energy sources and forcings that drive ocean currents (Ferrari and Wunsch 2009; Wunsch and Ferrari 2004). These studies agree that, on the timescales and spatial scales we consider, the wind stresses and air-sea fluxes are by far the largest energy source that can drive horizontal and vertical motion and hence, mass fluxes.

(2) Importantly, we analyse the mass budget, not the heat budget. Geostrophic currents cannot contribute to a net mass increase or decrease. Of course, geostrophic currents can, and do, contribute to a net heat convergence. Being geostrophic, however, the mass changes associated with temperature changes must be compensated for by mass changes associated with salinity changes.

Specifically, for a geostrophic flow, the advective term $A_{geo}$ in the mass balance equation can be expressed as:

$$A_{geo} = u_{geo} \cdot \frac{d(\rho \cdot h)}{dx} + v_{geo} \cdot \frac{d(\rho \cdot h)}{dy},$$

where we have integrated the advective term over the surface mixed layer, $u_{geo}$ and $v_{geo}$ are the average zonal and meridional velocities within the surface mixed layer, $h$ is the depth of the surface mixed layer, and $x$ and $y$ are longitude and latitude.

Moreover, the definition for geostrophic flows is: $u_{geo} = -\frac{1}{f \cdot \rho} \frac{d(p)}{dy}$ and $v_{geo} = \frac{1}{f \cdot \rho} \frac{d(p)}{dx}$, where $f$ is the Coriolis parameter and $p$ is pressure. Next, we integrate the hydrostatic balance over the mixed layer: $p = \rho \cdot g \cdot h$. Thus, we obtain for the geostrophic flows: $u_{geo} = -\frac{g}{f \cdot \rho} \frac{d(\rho \cdot h)}{dy}$ and $v_{geo} = \frac{g}{f \cdot \rho} \frac{d(\rho \cdot h)}{dx}$, where $g$ is the gravitational acceleration. Plugging both terms into the expression for $A_{geo}$ above, we find that $A_{geo} = 0$.

To answer your other question:

If the NAC experiences a greater (or lesser) heat loss prior to entering the subpolar region, it will adjust geostrophically. It cannot contribute to a net mass change. The geostrophic adjustment time is 1/f, which is approximately 2 days in the subpolar region, well below the winter averages we are using.

We have clarified the mass budget and changed the nomenclature to make it more consistent with earlier studies. Also, we better explain the individual terms (Section 3.1).

Geostrophic flows constitute by far the largest part of fluid motion over the open ocean, both as eddies and as the subpolar gyre circulation. However, we clearly state now in the main manuscript that geostrophic flows cannot contribute to the mass budget (line 276 – 284).

We have also included references investigating the mass budget (Griffies and Greatbatch, 2012) and potential sources of energy and hence mass fluxes (Ferrari and Wunsch, 2009; Wunsch and Ferrari. 2004), clarifying that the mass equation is well established and includes all relevant

terms. These studies find that the winds and air sea fluxes are by far the strongest forcing for fluid motion. Additional external sources of energy that can drive fluid motion include biological activity, tides, geothermal activity, and others, which are negligible on the timescales and spatial scales considered.

In conclusion, ageostrophic motion (including vertical mixing) requires an energy source (for instance to mix denser water upward against gravity). Over the open ocean, the necessary energy can only be provided by air-sea fluxes and wind stresses, neither of which is correlated with the freshwater indices (Section 4.1, Appendix A). Their magnitudes are much too small to be connected to the SST anomalies and their spatial characteristics do not agree with those of the identified SST anomalies (line 285 – 292).

I still remain confused about the proposed mechanism. Moreover, in order to explain how freshwater affects summer weather, the authors invoke changes in ocean circulation (to explain Fe) and wind-driven changes in heatfluxes and Ekman upwelling. They do this using a very small sample (next point) to produce a range of statistical links, but still appear to conclude that it is freshwater anomalies that drive everything with little discussion of the uncertainties inherent to a coupled system (note that the simulations used still do not test even the specific hypothesis that subpolar SSTs are driving European summer temperatures as opposed to SSTs or external forcing more generally). Therefore, I feel the authors need to do a much better job in reflecting the uncertainties in their conclusions. Ultimately, the way this is written is that the authors seem to think that the freshwater (or really the summer NAO) drives everything, but this seems a bit extreme to me based on the evidence. I would expect a significant amount of discussion time on the limitations of the method and on the exact role of freshwater anomalies (e.g., are the responsible for everything (as presented here), or could they be an important and not fully appreciated feedback?)

Thank you for indicating that the language was still too definite, and for suggesting adding a more comprehensive discussion about the involved uncertainties.

First, we would like to point out that we do not state in the manuscript that the freshwater anomalies or the summer NAO drive everything. We are very cautious about the wording in that we say: *"the NAOs indices are associated with freshwater anomalies"* or *"the cold anomalies are associated with freshwater anomalies"*. We do not use active verbs like *"Freshwater anomalies drive the cold anomalies"* or *"Freshwater anomalies drive European summer weather"*.

We summarise your comment stating that there are three sources of uncertainties: (1) those involved in the freshwater estimates, (2) those involved in the statistical analyses, (3) those involved in linking the statistical connections to dynamical mechanisms. In the revised manuscript, we have clarified these sources of uncertainty and included additional significance analyses. Below, we discuss all three sources.

(1) First, we provide the uncertainty estimates of the freshwater anomalies, which are only small, amounting to 4% and 6% (Section 4.1). This is fully acceptable for the purpose of the analysis. We now do the same for the new, un-subsampled freshwater index (Section 4.5, Appendix A). Also, we point out that these uncertainties are much smaller compared to other, publicly available salinity products.

(2) We demonstrate the links based on the statistical analyses, using three different methods. Specifically, we use composites, correlations/regressions, and a coherence analysis, all of which provide statistically significant results.

(3) Moreover, in the case of the correlation/regression analyses, we use three different indices: $F_E$, $F_W$, $\Delta$SST (Section 4.5), which have different characteristics in terms of autocorrelations, number of years included, and correlations with the freshwater anomalies. Yet, all four indices show statistically significant relationships with freshwater and European summer weather. The significance estimates are based on standard Student t-tests, which do account for the numbers of degrees of freedom (Section 4.5).

(4) We have added an Appendix B showing that the results are not sensitive to any choices that were made in the derivation, like adding or removing individual years, or excluding all consecutive years, or lowpass filtering European summer weather and adjusting the numbers of degrees of freedom for the resulting, increased autocorrelations.

(5) Regarding the link between the statistical correlations and the proposed dynamical mechanisms, we have followed your suggestions below, and removed analyses, paragraphs and sentences that were misleading. In each of the sections, we use words like *"associated with"* or *"linked to"*, when referring to either the freshwater anomalies or the freshwater indices, and we have rephrased the conclusions. We only use active verbs when referring to mechanisms that are shown and well-supported by theory or earlier studies (like *"Ekman transports being driven by the winds" or "surface pressure gradients driving geostrophic currents"*) but we do not use active verbs in connection with freshwater anomalies.

Moreover, when showing the air-sea feedbacks in the revised version, we explicitly state (line 404): *"By being highly correlated with the SST anomalies, the freshwater indices serve as valuable tools for visualising the associated ocean and atmospheric circulations, reinforcing each other (Figs. 1 and 4). However, we do not causally attribute the SST pattern to freshwater anomalies, and we do not infer that the freshwater anomalies act as a trigger for the characteristic tripole SST pattern."*

In addition, we clearly separate interpretation from results. Thus, we only discuss a potential coherent mechanism in the conclusions, where we conclude (line 564): *"This study identified statistically significant links and thus indicates an enhanced predictability of European summer weather arising from freshwater anomalies in the North Atlantic, without attributing the variability of European summer weather to freshwater anomalies as a mechanical trigger."*

We have also gone over all the formulations again and removed anything that could lead to misunderstandings. If there is anything you still find misleading, it would be great if you could let us know the specific sentences.

I'm still very worried about the robustness of the results in a statistical sense. The regressions for Fw are based on, as far as I can tell, 9 sample years(!). Moreover, even if the authors argue that the auto-correlation is small (it doesn't look that small over the 1980-2020 period), most of the years in their samples are not wholly independent (e.g., they come in clusters), and occur on a background of significant decadal change in atmospheric circulation and European temperatures, which both may be driven by external forcing). There is no attempt to address the particular uncertainty of small samples in the whole paper, which begs the question, how

robust are the results regarding European Weather to further sub-sampling? Given the short time-series, I would at least expect some sort of jackknife resampling (i.e., leaving out 1 year at a time), but I would strongly recommend the authors to also compute figures 6 and 7 using non-consecutive years.

We thank the reviewer for encouraging us to add more information on the significance of the results and to add more analyses assessing the robustness of the results. We note there is some overlap with your preceding comments, so we only provide a short summary of the main changes:

(1) First, we are using three different methods (correlations/regressions, composites, coherence estimates), all of which provide statistically significant results.

(2) Using the correlations (which is the only method you criticise), we use three different indices that are characterised by different correlations with the freshwater anomalies and based on different sets of years. The new $\Delta$SST index, specifically, includes 44 years. The $F_W$ and $F_E$ subsets (which are the two indices you are referring to), are based on only 17 and 8 years respectively. Yet, they are both characterised by exceptionally high correlations, compensating for the reduced number of degrees of freedom in the significance estimates.

Importantly, the significance tests we are using (Student t-tests), do account for the low numbers of degrees of freedom in all analyses.

For each correlation that is mentioned in the text, we provide the corresponding p-values. These p-values assess the probability that the high correlations are obtained by chance. In most cases, the p-values are well below 0.05. Thus, the probability that the links are obtained by chance is well below 5%. We also show the corresponding data points in the provided scatter diagrams to rule out that the high correlations are achieved by outliers or clusters.

(3) We consider the autocorrelations in all significance tests: Thus, we now show that autocorrelation of the summer NAO is below the e-folding correlation at one year lag (Fig. 8a). The autocorrelations of the subsets are even smaller. For instance, the autocorrelation of the $F_E$ index at a lag of one "index year" is ~0.07, implying that subsequent freshwater anomalies are different from each other in a rigorous statistical sense, regardless of whether these anomalies are consecutive or not. In addition, we show the autocorrelations of European summer weather (Fig. 8b), which are more relevant and reflect high interannual variability (see also Figure 10a).

Despite the autocorrelations being only small at one year lag, we now show that the results are not sensitive to excluding anomalies in consecutive years (Appendix B). The results remain significant, which is expected since the scatter diagram (Fig. 1d) shows that there are no clusters of point responsible for the high correlations. The data points are evenly spread over the range of each freshwater index.

The autocorrelation of the new $\Delta$SST index is higher, reflecting the low-frequency variability in the causes of the SST and freshwater anomalies. Yet, this study is focused on the subsequent links with European summer weather, for which the autocorrelations are small, suggesting that high-frequency variability contributes to the significance of the relationship.

To be sure that the increased autocorrelation of the $\Delta$SST index does not affect the significance, we lowpass filtered the European summer weather to match the increased autocorrelation of

the $\Delta$SST index. After correcting for the number of degrees of freedom using $N^* = N \cdot \frac{\Delta t}{2T_e} - 2$, where $N$ refers to the number of data points, $\Delta t$ refers to the time interval between them, and $T_e$ is the e-folding timescale of the autocorrelation (Leith, 1973), we still obtain significant relationships (Fig. B8).

(4) Lastly, we have added a multi-taper coherence analysis (Section 4.5), showing that the relationship between freshwater anomalies and the subsequent European summer weather is significant on timescales from years to decades (Fig. 8c and d). Thus, the coherence analysis also indicates that interannual variability is important.

Following your suggestion, we find that randomly selecting 8 summers or winters does not lead to substantially higher or lower correlations. We tested this with 1000 iterations for both summer and winter (Fig. 1 below).

[Figure]

*Fig. 1: Histograms for the correlation between the summer NAO and (a) the SST in the subpolar region in the subsequent winter, and (b,c) the 2-m air temperature and precipitation over Europe in the subsequent summer, obtained by randomly selecting 8 years, using 1000 test cases. We used boxes as regions, extending from 50 °N to 60 °N and 20 °W to 60 °W for the SST and from 30 °N to 60 °N and 10 °W to 30 °W for the European summer weather. However, the results are not sensitive to these choices.*

However, we think that it not necessary to add this analysis of the 1000 random test cases in the manuscript. Importantly, the selection of years with a high correlation between freshwater anomalies and their index has no implications for the statistical significance of the relationships between the freshwater index and any other variable that is independent of freshwater. If there were no statistical links between the freshwater anomaly and European summer weather, the selection of years would be purely random for European summer weather.

In a mathematical sense, the definition of independence is P(A|B) = P(A), where P(A|B) is the probability of A conditioned B (e.g. Evans and Rosenthal, 2004). If there were no statistical links between European summer weather ("A") and the freshwater anomalies ("B"), then the probability to obtain a specific European summer weather anomaly would be unchanged by the subsampling. Thus, the validity of the Student t-tests remains unaffected by the subsampling.

(5) In the main text, we have now clarified that the selection of years with a high correlation between freshwater events and the freshwater index does not affect the significance of a correlation between the index and any other variable that is independent of freshwater events. If there would be no statistical connection between freshwater and any the proposed influences, the probability for randomly obtaining significant statistical connections by chance remains the same (line 457 – 460).

Minor points

Line 166 - should be figure 1c? Relatedly, I would recommend to move figure 1c to 1a (e.g., swap the time-series with the spatial maps). This seems more logical.

Thank you for suggesting this. We have followed your suggestion, removed the sentence with the figure reference, and rearranged the subplots (Fig. 1).

The time series is now shown before the regression maps on $F_E$ and $F_W$. We have also added two more panels at the top to visualise additional explanations provided in the text.

Line 190 - I had not realized previously that you use a different spatial area to define your SST gradient. This seems slightly odd in that you may be picking up different processes (and related to my previous worries about not understanding why there is such a threshold in the SNAO SST regression). How sensitive is the SST gradient, and hence the correlations shown in figure 1d, to using fixed boxes for SST?

Thank you for pointing this out.

Using fixed boxes (for instance between 30 °N and 45 °N and between 45 °N and 60 °N) leads qualitatively to the same relationship when comparing the SST anomalies with the NAO but the correlations decreases since we are also including regions that are not statistically significant. Here, we used the regions in which the relationship is significant to ensure that the significance is not due to outliers or clusters (line 217).

Using slightly different regions is also motivated by the different characteristics of the two types of freshwater anomalies. Still both regions are highly correlated with each other. Thus, the correlation between the two, resulting $\Delta$SST timeseries is r $\approx$ 0.96 (with p $\approx 10^{-23}$), which is now stated in the manuscript (line 471).

We now explain the choice of the regions more clearly. The entire section has been rewritten to provide a clearer motivation of the regions to account for the complexity of freshwater variations in the subpolar region (Section 3.2).

Line 195 - Better to be specific - you mean for values above the -0.5 threshold? The process of "optimizing" the regression slope is very unclear - how do you do this? Did you just resample by randomly picking 17 years, and then pick the highest value? What is the physical basis for this? You say that results are not sensitive to this choice, which may be true, but then why "optimize" the correlation to 0.9 (from 0.68)? I think it would be good to show the sensitivity more clearly in the appendix.

Thank you for pointing out that this was still unclear.

We have now rewritten this section and clearly explain each step in the method (Section 3.2). Regarding the selection, specifically, we have added the following paragraphs (line 209 – 243):

*"Next, we strengthen the identified relationships between the two NAO subsets and subsequent SST anomaly through subsampling. The subsampling is motivated by the objective of achieving a near-linear relationship between the subsampled $NAO_S$ index and the subsequent SST anomaly. Specifically, if $x_i$ correspond to the NAO subset years, and $y_i$ correspond to the SST anomaly, we strive to derive a linear relationship $y = ax + b$, where $a$ and $b$ are constants and in which $|a|$ is high. The higher the magnitude of $a$ is, the higher is the magnitude of $\alpha T$ on the lefthand side of Eq. (5) after regressing Eq. (5) onto the index. Thus, we aim to select NAO years, for which the magnitude of the slope $a = \frac{y_i - y}{x - x_0}$ is large, where $x_0 = x|_{y=\bar{y}_t}$ and $y_i$ corresponds to the mean over $y_i$. At the same time, we strive to obtain a high correlation between the subset and the subsequent SST anomalies. Thus, we aim to select NAO years where $(x_i - x_0)^2$ is large, since this increases the variance of the index and SST anomalies.*

*[...] Following these objectives, we maximise the slope and the variance of the subsampled index by selecting the N years where the term $(y - y_i) \cdot (x - x_0)$ is highest. [...] Graphically, the subsampling is equivalent to increasing the slope of the regression line while keeping a high variance (Fig. 1d). It represents a powerful method for increasing the statistical relationship between two variables and thus identifying dynamical links, based on the assumption that noise, and other mechanisms, can mask these links."*

Further details are provided in Section 3.2 Following your suggestion, we now also show the sensitivity of the results to the subsampling in Appendix B.

Overall, we find: There is a trade-off between the numbers of degrees of freedom and the magnitude of the correlations. A higher number of years included implies an increased variance, reducing the correlations. However, the significance is unaffected because the reduced correlations are compensated for by a higher number of degrees of freedom.

In our responses to your preceding comments, we explain that higher correlations between the index and the freshwater anomalies support the subsequently identified links with European summer weather more strongly. Thus, we motivate the subsampling more clearly. Still, we also realise that a lot of space and time is now spent on the derivation of the indices. Since we show in Appendix B that the subsampling is not strictly needed (it strengthens the relationships but the conclusions remain the same without subsampling), we would be open to removing the subsampling from the manuscript for the sake of clarity if needed.

Line 280 - its not clear to me that figure 4c shows an intensification of the subpolar gyre - if anything, it is a weakening, in the mean (lower ADT in NAC region means weaker NAC, and higher ADT in Eastern subpolar gyre indicates weaker flow?)

Yes, you are right. This sentence was misleading. We have removed it.

Line 283 - what do you mean by subsampling in this case? When you say ADT on Fw, do you really mean it is the ADT when you only use sNAO value > -0.5. Is the field optimized in the same way as the delta SST?

Thank you for pointing out that this was unclear.

The $F_W$ index is the same $F_W$ index as the one used for the freshwater anomalies (Fig. 2b and d) and the SST anomalies (Fig. 1d and f). There would be no physical reason to show the correlation between $NAO_S > -0.5$ and the ADT since we are only using the $F_W$ index. This analysis is meant to show that the correlations with the potential drivers of freshwater anomalies (in this case the subpolar gyre circulation) also increases after the subsampling.

We have rephrased the paragraph as follows (line 346): "*To assess the role of the wind stress curl and subpolar gyre circulation for the cold and fresh anomalies associated with higher summer NAO states, we inspect the associated absolute dynamic topography in winter. The full, un-subsampled summer NAO only displays a weak and mostly non-significant relationship with the geostrophic surface circulation in the southwest subpolar region (Fig. 3c). When using the sub-sampled summer NAO corresponding to the $F_W$ subset, however, the absolute dynamic topography north of 50 °N in winter is significantly reduced, implying a more cyclonic and hence, stronger subpolar gyre circulation in the northwest subpolar region (Fig. 3d). The strengthened relationship between the subsampled summer NAO and the subpolar gyre circulation therefore supports the subsampling by providing a physical explanation for the associated freshwater anomalies (Fig. 2b).*"

Paragraph from line 299 - I found it very confusing to understand what anomalies were being explained by what. It starts by talking about sNAO anomalies >-0.5, but then talks about both Fe and Fw events. This part implies that ocean circulation changes control Fw events, however, this bring up two questions. Firstyly, what sort of circulation anomalies are we talking about? In particular, previous work had talked about shifts in the size and location of the subpolar gyre (e.g., Hatun et al, 2005). Is this the sort of circulation anomaly that the authors are thinking of?

Thank you for pointing out that this was unclear. Indeed, we refer to the subpolar gyre circulation. In line with the study by Hatun et al. (2005), we find that the subpolar gyre circulation is intensified and more confined to the north.

Overall, we find that the study by Hatun et al (2005) nicely supports our results. Thank you also for providing the reference. We are referencing it now in the manuscript.

We have now fully removed the paragraph that you referenced in your comment since we agree that it included too much interpretation.

Instead, we explain more clearly the role of the $F_W$ index in linking the identified freshwater anomalies with the subpolar circulation. The $F_W$ index is highly correlated with the freshwater anomaly (Fig. 2d) and with the subpolar gyre circulation in the north-western subpolar region (Fig. 3d), which advects more cold and fresh polar water into the subpolar region and reduces the import of warm, saline subtropical water (e.g. Häkkinen and Rhines 2009; Häkkinen et al. 2011, 2013; Holliday et al., 2020). This is also in line with the study by Hatun et al. 2005.

In addition, we provide a more precise description of the location by stating: "*When using the sub-sampled summer NAO corresponding to the $F_W$ subset, the absolute dynamic topography north of 50 °N in winter is significantly reduced, implying a more cyclonic and hence, stronger subpolar gyre circulation in the northwest subpolar region (Fig. 3d).* "

Second, the argument that circulation changes are important for Fw seems at odds with the argument that advection doesn't cause the SST anomalies in the derivation of the freshwater signal, where the authors say geostrophic circulation is not important?

Geostrophic currents cannot contribute to mass budget, but they substantially contribute to the heat and freshwater budgets and anomalies. The SST and freshwater anomalies balance each other in their effect on density (now explained in line 276).

We think has now been clarified, following our response to your earlier comments.

Line 358 - I think this is written with far too much certainty ( the SST fronts destabilize the overlying atmosphere, resulting in an enhanced jet stream), and is presented as though it has been proven within this study. I take the point that this is what might be expected from theory, but a regression onto a short time series is not proof. Indeed, the previous section has just been highlighting how wind anomalies can drive SST gradients. Therefore, I would suggest wording here (and elsewhere) that better reflects the uncertainties. E.g., the enhanced jet along the SST front is consistent with…. Ultimately, if I have understood the analysis correctly - this regression is based on 8 years/events - this is an incredibly small sample.

Thank you for indicating that this sentence was written with too much certainty. We agree with your suggestion.

We have now removed the sentence. We have also carefully gone through the manuscript and rephrased all potentially contentious statements.

We still find that the link between the SST and atmospheric circulation is significant in a statistical sense. We note that we identify the same relationships for all indices and the composite (Figs. 5, 6, 7 and 10). Thus, the link is not just based on 8 years.

Still, we do not attribute the atmospheric circulation to the SST, and we do not state the freshwater anomalies triggered the SST pattern. We have also clarified this in the conclusions.

Figure 6 (and other relevant ones) - its not clear what the time period used for this regression - Previously I would have thought it from 1979 onwards using the 8 events - but, now with figure 9, I'm not sure…

Thank you for indicating that this was unclear. All indices are derived from the period 1979 to 2022.

To avoid confusion, we have removed the specific analyses 4.6 and 4.7 that you found unclear regarding the underlying period. Thus, we no longer use the extended period since 1950. We agree that this makes the study clearer and more concise.

On the subject of figure 6, please can you say more about the significance test, please? What is the test, specifically, and how have you ensured that it is robust to a small sample? Also, how do you take account of the fact that many of your years are clustered together? Maybe a jackknife resampling would be appropriate here to test the sensitivity of your results to omitting years from your analysis.

Following your major comments above, we have added a section on significance and robustness (Section 4.5) and an Appendix B, showing that the conclusions to not change after adding or removing years, excluding all consecutive years, and applying no subsampling.

We also explain that we are using standard Student t-tests to assess the significance. As the autocorrelations show, moreover, there are also no high autocorrelations, neither for the $NAO_S$ index, nor for European summer weather. There are also no clusters or outliers responsible for the high correlations. Still, we show that the results remain significant if consecutive anomalies are excluded (Appendix B).

The Figure you refer to is now Figure 5, and the associated Figure that excludes all consecutive events is Figure B2. Both figures look very similar.

Please also see our responses to your earlier comments regarding significance and jackknife sampling.

Section 4.5 - I didn't really understand what this section was trying to say, other than it took a different analysis approach - e.g., composite. However, I found the analysis a bit confusing in the sense that freshwater anomalies are argued to be important, but the SSS anomalies do not look like the SSTs in this case?

Thank you for pointing out that this was unclear. The locations of the SSS and SST anomalies are different because the SSS anomalies are shown in winter whereas the SST anomalies are shown in summer. In winter, the SST and SSS anomalies look very similar (Fig. A4).

We now motivate the section more clearly. Specifically, we state that it assesses the role of freshwater anomalies as a predictor for Europe's warmest summer (line 256) and investigate the extent to which enhanced freshwater anomalies are not only a sufficient but also a necessary condition for warmer European summers (line 533).

Another advantage of this section is that it is based on a temperature index rather than an SST or freshwater index. Showing that the links remain significant regardless of the method or index that is used, increases the robustness of the identified links (line 541).

Since you were previously concerned about autocorrelations, another advantage of this approach is that it starts with a time series that has very low autocorrelations, reflected in a high interannual variability (Fig. 10a).

Figure 8 - The climate pattern seen here looks just like the response to the summer NAO… what is the correlation between your summer NAO index and the Tsummer shown in 8a?

The correlation is r ≈ 0.25 and it is not significant (p ≈ 0.10). There is also no reason for expecting the $T_{Summer}$ timeseries to be correlated with the summer NAO. In this section, we link $T_{Summer}$ to freshwater anomalies in the preceding winter, which have a nonlinear relationship with the summer NAO (Fig. 1d). Moreover, the summer NAO index, on which the freshwater indices were based, already occurs in the preceding year.

Section 4.6 - I found this section very confusing, and it wasn't clear to me what was being argued - in particular, I thought the authors had argued in their response that this analysis was was based on the interannual variability due to the use of the NAO, but then present an analysis

of very long timescale changes. It also raises the question of what is the time periods are being used in all the other analyses in the paper. If this is not a key aspect of the story of the paper, then I suggest removing it - there is already enough in this paper.

Thank you for suggesting this. We had added the analysis since a previous reviewer was asking for it. However, we agree that it might change the shift the focus, and it makes the manuscript less concise.

We have now fully removed this analysis to improve the overall clarity and conciseness of the manuscript.

Section 4.7 - over what time-period do you compute the relationships?

It was computed over the period 1979 to 2022. However, this section has been removed since it was not necessary and mostly led to confusion.

Line 423 - do you mean figure 10?

Indeed, thank you. However, we have removed this figure now.

Line 443 - I don't understand what you mean by "projet the SST each summer onto the observed SST pattern - please elaborate

We define the SST pattern as a mode of variability. This is the same method as in EOF analysis, only that we predefined the pattern based on its link with the freshwater anomalies and then looked at its time variability. Thank you for indicating that this was unclear.

We have now fully removed this analysis since we found that it was not strictly necessary for understanding the results. Instead, it mostly led to confusion.

Line 456 - the model simulations still don't just test the impact of subpolar SST, but include all SST anomalies and external forcings. So I really do not see how you can attribute the signals to the subpolar SST anomalies.

The SST pattern is most pronounced in the subpolar region where we have linked it to a freshwater anomaly. The time variability of this pattern and the lag of one year has also been linked to freshwater anomalies and variations in the supolar region. In the preceding analysis, moreover, we showed that the remainder of the large-scale SST field could fully be explained by atmospheric feedbacks.

Despite the agreement of the temporal and spatial characteristics of this pattern with the freshwater variability – and despite the absence of any other possible physical mechanisms that can initiate this pattern (for instance in the tropics or stratosphere or random fluctuations) – we only stated that this pattern is statistically linked to freshwater anomalies, using the phrase "SST pattern linked to freshwater anomalies". Thus, we explained there is a significant statistical link to freshwater anomalies in the preceding winter and we provided the associated correlations. We did not state that freshwater triggered this pattern, and we did not attribute the SST pattern to freshwater.

Since the analysis was not needed, and to improve clarity and conciseness, we have now fully removed this analysis.

Section 4.8 - Over what time period is the $R^2$ computed? If this is based on just the 9 points shown in figure 2? How robust is this relationship to jackknife resampling? What happens if you remove all consecutive years?

Please also see our responses to your earlier comments.

Again, we sincerely thank the reviewer for providing so many detailed, helpful, and constructive comments and suggestions. We are particularly grateful for the suggestion to add the un-sampled SST index to the analysis. Your comments have helped us to clarify and improve this study!

**References:**

Aagaard, K., Coachman, L. K., & Carmack, E. (1981). On the halocline of the Arctic Ocean. *Deep Sea Research Part A. Oceanographic Research Papers*, *28*(6), 529-545.

Bamber, J. L., Tedstone, A. J., King, M. D., Howat, I. M., Enderlin, E. M., Van Den Broeke, M. R., & Noel, B. (2018). Land ice freshwater budget of the Arctic and North Atlantic Oceans: 1. Data, methods, and results. *Journal of Geophysical Research: Oceans*, *123*(3), 1827-1837.

Dukhovskoy, D. S., Yashayaev, I., Proshutinsky, A., Bamber, J. L., Bashmachnikov, I. L., Chassignet, E. P., ... & Tedstone, A. J. (2019). Role of Greenland freshwater anomaly in the recent freshening of the subpolar North Atlantic. *Journal of Geophysical Research: Oceans*, *124*(5), 3333-3360.

Evans, M. J., & Rosenthal, J. S. (2004). *Probability and statistics: The science of uncertainty*. Macmillan.

Ferrari, R., & Wunsch, C. (2009). Ocean circulation kinetic energy: Reservoirs, sources, and sinks. *Annual Review of Fluid Mechanics*, *41*, 253-282.

Gill, A. E. (1982). *Atmosphere-ocean dynamics* (Vol. 30). Academic press.

Griffies, S. M., & Greatbatch, R. J. (2012). Physical processes that impact the evolution of global mean sea level in ocean climate models. *Ocean Modelling*, *51*, 37-72.

Hakkinen, S., & Rhines, P. B. (2009). Shifting surface currents in the northern North Atlantic Ocean. *Journal of Geophysical Research: Oceans*, *114*(C4).

Häkkinen, S., Rhines, P. B., & Worthen, D. L. (2011). Warm and saline events embedded in the meridional circulation of the northern North Atlantic. *Journal of Geophysical Research: Oceans*, *116*(C3).

Häkkinen, S., Rhines, P. B., & Worthen, D. L. (2013). Northern North Atlantic sea surface height and ocean heat content variability. *Journal of Geophysical Research: Oceans*, *118*(7), 3670-3678.

Holliday, N. P., Bersch, M., Berx, B., Chafik, L., Cunningham, S., Florindo-López, C., ... & Yashayaev, I. (2020). Ocean circulation causes the largest freshening event for 120 years in eastern subpolar North Atlantic. *Nature communications*, *11*(1), 1-15.

Koul, V., Tesdal, J. E., Bersch, M., Hátún, H., Brune, S., Borchert, L., ... & Baehr, J. (2020). Unraveling the choice of the north Atlantic subpolar gyre index. *Scientific reports*, *10*(1), 1-12.

Stewart, K. D., & Haine, T. W. (2016). Thermobaricity in the transition zones between alpha and beta oceans. *Journal of Physical Oceanography*, *46*(6), 1805-1821.

Wunsch, C., & Ferrari, R. (2004). Vertical mixing, energy, and the general circulation of the oceans. *Annu. Rev. Fluid Mech.*, *36*, 281-314.

---

## Author Response (AR3)

8. November 2023

Dear Prof. Pfahl,
Dear Reviewer,

We are pleased to submit a revised version of the manuscript "European summer weather linked to North Atlantic freshwater anomalies in preceding years". Based on the detailed and constructive feedback provided by the reviewer, we have further clarified the manuscript.

The main shortcoming of the last version was the skipping of a step in the derivation of Figure 1, which displays the method, giving rise to confusion. In addition, the description of the main results (the link between freshwater anomalies and European weather anomalies) was not adequately placed into a larger context of existing studies.

Following the reviewer's suggestion, we have carried out a series of minor revisions to add further details and clarifications in the description of the method and the results, and to place the results more adequately into a larger context. Overall, the manuscript has again greatly benefitted from the review. Thus, we thank the editor for the careful handling of our manuscript and the reviewer for, once again, providing such a detailed and constructive assessment.

Detailed changes are explained in the response letter below, where our responses to the reviewer's comments are shown in blue, and the resulting changes are shown in red. Figure and line numbers refer to the revised manuscript without tracked changes.

Sincerely,
Marilena Oltmanns, on behalf of all authors

**Responses to Reviewer 1**

The authors use statistical analysis of observations and reanalysis data to examine a potential link between freshwater anomalies in the North Atlantic subpolar gyre region and summer weather in Europe in the following year. They propose that stronger freshwater anomalies are associated with a stronger meridional SST gradient between the subpolar and subtropical gyre and consequently increased baroclinic instability in the atmosphere above. The resulting changes in the large-scale circulation lead to significant anomalies in near-surface temperature and precipitation in different parts of Europe. The foundation of the analysis are freshwater indices derived from a mass balance equation that are used to identify freshwater anomalies in relation to simultaneous sea surface temperature (SST) anomalies linked to the North Atlantic Oscillation (NAO).

The current manuscript is another substantial improvement to previous versions and I appreciate the added detail in the derivation of the mass balance and estimation of the freshwater indices, and the more precise description of the results. As outlined in my comments below, there is some remaining uncertainty regarding the construction of the freshwater indices. Additionally, the presentation and description of the central results, i.e., the link between the freshwater anomalies and European summer weather, requires – in my opinion – some larger context and relation to known atmospheric circulation and weather patterns that are described in the literature. Overall, I recommend a series of minor revisions that may appear major due to their extent.

We sincerely thank the reviewer for reviewing our manuscript once more, for acknowledging substantial improvements, and for providing additional suggestions to help us clarify the method and place the results into a larger context.

**I. Main Comments:**

1. Section 3.2:

    1. I now understand why this part has been – and, in part, still is – confusing to me: the threshold of $NAO_S < -0.05$ is based on the non-linear relationship shown in Figure 1d. The y-axis of that figure is $\Delta SST$ which "corresponds to the SST difference between the red, subtropical and blue, subpolar 95% confidence regions in panels e (red years) and f (blue years) respectively, relative to the respective means". However, these confidence regions are not known a priori, so I am trying to wrap my head around the specific steps to get from the time series of $NAO_S$ (Figure 1c) and winter SST at each grid point, via the scatter plot in panel d, to the maps in panels a and b. Strictly speaking, you cannot determine the value of $\Delta SST$ for Figure 1d, and therefore the threshold of -0.5, until you have the two maps (panels e and f) after the subsampling. The subsampling itself, however, depends on the relationship in Figure 1d, so there must have been some iteration or trial-and-error. Please add more details in the text (before l. 202) to lay out your analysis steps.

Thank you for letting us know that the derivation of the scatter plot was still confusing. The confusion resulted from a discrepancy between the number of steps described in the text, and the number of steps shown in the figure.

Since the scatter plots are largely insensitive to the exact regions, we only showed the scatter plot for the 95% significance regions obtained after the method has been applied. We chose these regions because they cover a large area, nicely demonstrate the effectiveness of the method, and provide a means to directly inspect the robustness of the correlation.

Changes:

To avoid confusion, we have included additional panels in Figure 1, showing each step described in the text.

In addition, we have updated the text, and we now explain (line 208): "The threshold of ~–0.5 was initially identified using box regions for the subpolar and subtropical regions (for instance with latitudinal boundaries between 45 °N and 60 °N for the subpolar region and between 30 °N and 45 °N for the subtropical region). However, the identified relationships are not sensitive to the exact regions."

With the additional steps shown in the figure, it becomes clearer that the distribution of the points in the scatter plots does not appreciably change as the regions, used for the spatial averaging, are adjusted throughout the optimization (compare Figure 1d, h and l).

2. Please define ΔSST in the text. Is this the difference between the regressed SST or the actual SST values in these regions? Is it relative to the respective spatial or temporal mean (over which period)?

   It is the actual SST value, not the regressed SST. The anomalies are relative to the respective (temporal) mean over each subset. The SST anomalies are a vector quantity with only one dimension. They are already spatially averaged over the 95% significance regions.

   We added the word "observed SST difference" to avoid confusion. We now also clearly state in the caption of Figure 1 and in the text that we subtracted the temporal mean over each subset, and we define ΔSST in the text.

   Thank you for helping us to be more precise.

3. Caption of Figure 1: $-NAO_S$/$+NAO_S$ is confusing – this could be interpreted as positive and negative phase of the NAO. Maybe " $-1 \times NAO_S$" is more obvious.

   Thank you for suggesting this. We have adopted your suggestion.

4. l. 235: Have you looked at the differences of regressions/composites between the included and rejected years? This could potentially help identify or constrain a physical mechanism at play for years with a strong relationship.

Yes, we did. The first outlier was a strong runoff-driven anomaly. However, the anti-cyclonic atmospheric circulation anomaly in the preceding summer (in 2014) extended far to the south. Thus, it was not captured by the NAO index.

The second outlier (with the NAO index in 2019) was also a runoff-driven freshwater anomaly. However, a careful examination of the associated ocean and atmospheric circulation anomalies showed that the subpolar gyre circulation anomaly was still enhanced in this year. Thus, there was not only more freshwater inside the subpolar gyre current system, but in addition, an enhanced redirection of the cold and fresh polar water into the interior subpolar region. As a result, the NAO index underestimated the fresh and cold SST anomaly.

Likewise, we found that for the years, rejected in the second step of the subsampling, the relationship between the subpolar gyre circulation and the SSS anomalies still holds. However, in these years the subpolar gyre circulation is not well captured by the NAO index. Thus, the years, rejected in the second step of the subsampling still correspond to circulation-driven anomalies.

In the end, we decided not to include a detailed description of the outliers and rejected years for the sake of brevity. After all, the underlying mechanisms are still the same as those described in the manuscript. The disagreement with the other years is mostly a matter of the increased variance that inevitably results from including more years. This increased variance is not adequately captured by the NAO index. However, it is not a third, distinct mechanism that is responsible for the associated fresh and cold anomalies.

Through the previous inclusion of Section 4.5, we also clarified that the link between fresh and cold anomalies (and their link to European summer weather) still holds for the rejected years. They are just not well captured by the NAO index.

One of the advantages of the subsampling is to ensure spatial consistency within each subset. This is important because small shifts in the location of the SST front are associated with corresponding shifts in the atmospheric circulation and warm and dry anomalies over Europe in the subsequent summer. For instance, if two years with very different spatial patterns are included in one subset, they can partially cancel each other out.

We clarified that the link between freshwater and cold SST anomalies still holds during the outliers and rejected years (line 227 and Section 4.5).

In addition, we now state that, for the rejected years, the subpolar gyre circulation is still the main driver of cold and fresh anomalies, but the circulation is not well captured by the NAO index in these years (line 363).

Lastly, we now also mention the advantage of the subsampling to reduce the spatial SST variance and thus ensure spatial consistency within the subset (line 480 and 504).

2. Section 4.4: Given the title of the manuscript, this section describes the central result of the study: the statistical link between summer weather and freshwater anomalies in the subpolar gyre in previous years. However, the presentation of the results leaves me as the reader unsatisfied. While I appreciate the added details compared to the previous version, some of the conclusions remain slightly hand-wavy and are missing some larger context:

1.  Based on the regressed meridional wind anomalies at 700 hPa, you describe a "northward deflection of the jet stream" following both FE and FW freshwater anomalies that differ in there location between years and subsets. Given that the jet stream occurs at higher altitude, you are rather describing circulation anomalies in the lower troposphere.

    Thank you for pointing this out. We did investigate the winds at higher altitudes and found that the anomalies extend at least up to 500 hPa (the highest altitude we checked). However, we only show the lower component since we expected them to affect the surface weather more directly.

    We have replaced the word "jet stream" by "lower tropospheric winds" at all instances in the results and conclusions sections.

    Regardless of this semantic distinction, I am missing a discussion of the southward anomalies in the 700 hPa winds in Figures 5c and 6b as they can help put these anomalies in the context of known large-scale circulation patterns (e.g., Cassou, 2008; Grams et al., 2017) and their related expressions in surface temperature and precipitation anomalies. Relating your result to previous studies may also help identify physical processes that lead to the anomalies that you describe – is it advection of warmer/dryer air masses or changes in radiation/heat fluxes that can be linked to the large- scale circulation? For example, the anticyclonic circulation anomaly over the North Sea in Figure 6b might be suggestive of a blocking event (reduced winds, increased radiation, less precipitation...) – interestingly, the dry anomalies in Figure 6d are roughly co-located.

    Thank you for suggesting relating our findings to previous studies. We have now added a longer discussion of the observed flow anomalies to place them into a larger context of existing literature. First, we have included a longer discussion of the observed large-scale atmospheric circulation anomaly (for instance line 450). In a subsequent step, we then relate it to earlier studies. For instance, in line 470, we write:

    "Placing the identified atmospheric anomalies into a larger context described in the literature, we find that is representative of blocking anticyclones (Brunner et al., 2018; Kautz et al., 2022). In summer, blocking anticyclones over Europe are typically associated with increased surface pressure and higher surface air temperatures (Brunner et al., 2018; Kautz et al., 2022). Considering that the maximum temperature anomalies in summers after enhanced $F_E$ freshwater anomalies occur east of the northward wind deflection, in the centre of the anticyclonic circulation anomaly, the location of the increased air temperature anomalies is consistent with earlier studies which have attributed the warm

anomalies to enhanced shortwave radiation (Kautz et al., 2022; Pfahl, 2014; Sousa et al., 2017). Moreover, the occurrence of the dry anomalies to the east of the warm surface air temperature anomalies likely results from a reduced passage of cyclonic weather systems, which are blocked by the large-scale anticyclonic circulation anomalies (Sousa et al., 2016)."

Accordingly, we have also updated other instances in the text where we describe the large-scale atmospheric circulation anomaly.

2. Figure 5: In order to make it easier for the reader to interpret the regressed anomalies, it might be worthwhile adding another column of maps showing the mean conditions.

We have now added another column showing the mean conditions (Fig. 5).

3. I do not understand the justification for excluding the 2016 anomalies (caption of Figure 5).

The freshwater anomaly in Summer 2016 extended over a larger area further south of the North Atlantic Current compared to other years, resulting in enhanced mixing and a patchy meridional SST gradient east of the European coastline, with alternating warm and cold SST anomalies of reduced amplitudes. Consistent with the underlying SST field, we identified a split zonal wind around 0 °E to 10 °E, with one branch extending northward along the European coastline, and another one along the southern Mediterranean Sea. In conclusion, the relationship between the SST, winds, and European temperature anomaly still holds in this year but the spatial patterns just do not map onto the other years, included in that subset.

We are now acknowledging the limitations of our method to ensure spatial consistency across the events. Specifically, we explain (line 479):

"A downside of the statistical approach arises from the sensitivity of European summer weather to the exact location of the SST front between the subtropical and subpolar gyres. Small deviations in the spatial characteristics of the SST pattern and lower tropospheric circulation between two years can lead to shifts in the location of the maximum warm and dry anomalies over Europe, partially cancelling each other. Thus, we found that the spatial patterns in Summer 2016 did not match those of the other years included in the $F_E$ subset (Fig. C1). The cold SST anomaly extended further south of the North Atlantic Current, resulting in enhanced mixing and a patchy meridional SST gradient just west of the European coast with two cold anomalies of reduced amplitudes. Consistent with the underlying SST field, we identified a split zonal wind between ~0 °E to ~10 °E, with one branch extending northward along the European coastline, and another one crossing the southern Mediterranean Sea. Accordingly, one warm surface air temperature anomaly covered northern Africa and another warm anomaly occurred along the northwest European coastline (Fig. C1). So, even though the spatial SST pattern in Summer 2016 did not match those in the other summers, we still identify the same close relationship between the SST, tropospheric winds, and European weather anomaly."

Later, we mention (line 504):

"Considering that – from all the years included in each subset (17 and 8 respectively) – only the Summer 2016 exhibited a diverging spatial pattern in SST and atmospheric anomalies, the results suggest that (1) the statistical method is overall successful in selecting years with similar spatial structures, and (2) the spatial consistency for which we selected in winter is, in most cases, maintained through to the subsequent summer."

**II. Additional Comments and Suggestions:**

1. l. 36: This wording suggests that it is certain that freshwater initiates the causal chain, but only the physical mechanism is unclear.

   Thank you for pointing this out. We have replaced "will" by "could":

2. l. 77-79: You mention that you use monthly – presumably mean – ERA5 output in the analysis. Please clarify: did you estimate the maximum Eady growth rate of the monthly mean circulation or did you compute it from higher-frequency, e.g., daily mean, output that you then averaged over a month? Given the nonlinearities in the equation these two estimates could be quite different, however, I do not expect them to change your results.

   Yes. Thank you for spotting that this has been unclear from the previous phrasing of this section.

   The maximum Eady growth rate is the only nonlinear variable that we computed based on monthly means. Thus, we only use it for a qualitative assessment of the lower tropospheric instability. We agree that using high-frequency output in the calculation is unlikely to affect the overall conclusion that the maximum Eady growth rate is enhanced over the SST front.

   We now clarified that we are using monthly mean output to calculate the maximum Eady growth rate by explaining (line 78): "The maximum Eady growth rate was calculated using monthly mean output from ERA5 to qualitatively assess the baroclinic instability in the lower troposphere over increased meridional SST gradients."

3. l. 195, 197, 246: I am still stumbling over the phrase "lower/higher NAO phase". I am not familiar with the detail of the previous studies that you refer to in the first two instances, but I would assume the most of them contrasted the two states of the NAO and therefore, I suggest to use "negative/positive NAO phase". The last instance can be changed to "associated with $NAO_S < - 0.5$".

   Thank you for suggesting this. We have adopted your suggestions.

4. l. 323-325: Please clarify: the runoff-$NAO_S$ relationship is calculated over all years, yet you use it as a potential explanation for $F_E$ freshwater anomalies, i.e.,

only a very specific subset of years. In Figure 3a, this relationship is not that clear if you only consider $NAO_s$ < -0.5. If anything, there seems to be a clearer relationship and less spread around the regression line for the $F_W$ years.

Yes, you are right that the relationship between the summer NAO and runoff is less clear for NAO values smaller -0.5. However, we do not use this relationship to explain differences in the freshwater anomalies between individual $F_E$ years. Instead, we use this relationship to explain the difference between $F_E$ and $F_W$ anomalies.

Specifically, we state (line 330): "[...], we find a significant anticorrelation between the summer NAO and runoff (Fig. 3a)". Thus, $F_E$ anomalies are associated with more runoff than $F_W$ anomalies. We are comparing the drivers of freshwater anomalies associated with $F_E$ and $F_W$ with each other, rather than comparing different $F_E$ years with themselves.

Further studies are required to inspect the differences between individual $F_E$ years. However, we think that the wind forcing associated with the NAO index may play a role, by favouring an enhanced surface flow convergence in the subpolar region.

We now explain that the relationship between the NAO index and runoff cannot explain differences within the $F_E$ subset. Instead, it explains the existence of the $F_E$ subset in the first place (line 341).

5. l. 376-377: Please clarify: do you show that "after strong $F_E$ anomalies, the NAO anomaly switches sign from being strongly negative in summer to being strongly positive in winter" or do you infer that from Figure 4b? Out of curiosity, what is the correlation between summer NAO and winter NAO with and without conditioning on $F_E$ years?

We have followed your suggestion and replaced the sentence with a more quantitative statement (line 388):

"Over the period 1979 to 2022, and without conditioning on $F_E$ years, the correlation between the NAO in summer (July and August) and the NAO in the subsequent winter (January to March) is r = ~0.12, which is not significant (p = ~0.46). With conditioning on $F_E$ years, the correlation is r= ~–0.74, which is significant (p = ~0.03). Moreover, we find that after all but the two weakest $F_E$ years, the NAO changed from its strongly negative state in summer into a positive state in the subsequent winter."

6. l. 457-459: This sentence is unclear.

We have split the sentence into smaller parts and simplified it.

7. l. 471: It is easy to get lost here: I think what you are doing is regressing the winter SST on $NAO_s$ for all years, but calculate ΔSST bases on the regions shown in Figure 1e with the resulting time series shown in Figure 7a. Please add more detail to clarify your analysis steps here.

Thank you for pointing out that this was unclear.

The ΔSST index does not involve any new regression on the NAO index. It is defined as the observed subpolar-subtropical SST difference in any given year. We defined the underlying regions based on the significance lines shown in the previous Figure 1e (now Figure 1j). We selected these regions since they cover such a large area of the subpolar and subtropical gyre and clearly define both regions. However, the results are not sensitive to this choice.

We have clarified the text and clearly explain that the SST index covers all years. It is defined as the SST difference between the regions shown in Figure 1j in each individual year, with the resulting timeseries shown in Figure 7a.

8.  l. 484: Notably, the T2m anomalies are offset to the east of the V700 anomaly. Similar to my comments on Section 4.4 above, please discuss this in the context of the existing literature and speculate about the physical mechanism.

The northward V700 anomaly form the western component of a large-scale anticyclonic circulation anomaly over western Europe (see for instance the arrows in Figure 6b). The T2m anomalies are strongest in the centre of the anticyclonic anomalies, which is to the east of the V700 anomalies. This suggests that the temperature anomalies are driven by the radiative forcing in the centre of the anticyclone rather than the direct effect of the horizontal wind anomalies.

We have now added a longer discussion on the identified, large-scale atmospheric circulation anomalies and placing them into the context of existing literature (see also our response to your Main Comment 3).

9.  Figure 7a: What did you normalize the SST difference by? Please add more details in the text.

We now specify that we normalised the time series by the standard deviation. This also explains why it has no units.

10. l. 525-526: Please clarify: your predictors are $F_E$ and $F_W$ (i.e., NAO$_S$) and ΔSST, so the common denominator is the atmospheric circulation associated with the summer NAO. How does sea surface salinity constrain weather predictions?

The ΔSST time series is not related to the summer NAO. The common denominator is the correlation with the SST and freshwater anomalies in winter. Thus, we think this comment has partially been addressed by clarifying your earlier comment (Comment 7).

We now explain that all indices are representing freshwater (and SST) anomalies. In addition, we have rephrased the sentence to clarify that we are not making any actual weather predictions. Thus, we have removed the word "predictions" from the sentence you referred to.

We further explain in this section (Section 4.6, line 581) and in the discussion (line 630), that estimation of the extent, amplitude, and type of freshwater anomalies in winter can help constrain the subsequent ocean-atmosphere evolution into the summer based on the evolution of past freshwater anomalies with similar spatial patterns. The extent to which past freshwater anomalies can help constrain the future evolution is quantified by the explained variances.

Lastly, we have added a paragraph in the discussion section on the advantages of connecting European summer weather to North Atlantic freshwater anomalies specifically, rather than just using the SST (line 640).

11. l. 535. Please add more details in the discussion of your results with respect to the large-scale circulation (see comments on Section 4.4 above).

We now explain that the northward deflection of the lower tropospheric winds forms part of a large-scale atmospheric circulation anomaly over the North Atlantic and Europe. The circulation patterns are indicated by the arrows in Figures 5a, 6a and b, and 7c and their characteristics are similar to the patterns described by earlier studies (see our response to your Main Comment 3).

Accordingly, we have now also updated the discussions in the subsequent parts of the manuscript.

12. Please add more details: how exactly do you "trace the cold SST anomaly back to a freshwater anomaly in the preceding winter".

We have removed the word "trace back". Instead, we state:

"Based on composites [...], we again identify a significant freshwater anomaly in the preceding winter, with the freshwater anomaly covering a large part of the subpolar North Atlantic (Fig. 10f). [...] The similarity of the ocean and atmospheric conditions with those described in the preceding sections supports the relevance of freshwater anomalies in winter for the subsequent ocean-atmosphere evolution into the summer. In addition, the composites suggest that enhanced freshwater anomalies in the subpolar North Atlantic in winter could serve as early warning signs of Europe's warmest and coldest summers approximately half a year in advance."

Further implications are now discussed in the discussion section.

13. l. 557-558: The northward deflection of the jet stream is a bit too hand-wavy for me. Please discuss this in the context of large-scale atmospheric circulation in summer and associated weather patterns.

We now explain that the northward deflection of the jet stream forms part of a large-scale atmospheric circulation anomaly and cite earlier studies investigating this study (please also see our responses to your Comment 3).

14. l. 563: Please discuss: the freshwater anomalies themselves are part of the chain reaction that are ultimately linked to the summer NAO. Provocatively

asked: if you wanted to create a statistical model to predict summer weather with one or two year lead time, what information is added by knowing the freshwater anomalies? While I do understand the role of the freshwater anomalies as part of the chain that eventually leads to changes in European weather, I am missing a discussion how knowledge of sea surface salinity can constrain the predictions (see also comment II.10).

If we start in any given winter and would like to constrain the subsequent weather, we would first identify a set of years with similar spatial patterns (but possibly different amplitudes) in the SST. We would then include these years in a regression model to assess the future evolution of the current year. The associated explained variances would provide a measure of the uncertainty.

However, given the limited set of years in the observational period, it would be advantageous if models would better capture the subpolar hydrography and represent freshwater anomalies more accurately. Models could quantify the predictability more exactly by providing a higher number of simulations. In that case, we would not rely on a regression model based on observations only.

We now explain in the discussion section that estimations of the strength, amplitude, and type of freshwater anomalies in winter help constrain the subsequent ocean-atmosphere evolution into the summer, based on the evolution of past freshwater anomalies with similar spatial patterns. The extent to which European summer weather is constrained is quantified by means of the explained variances.

We further explain that the connection to freshwater specifically adds an enhanced predictability on longer timescales since the drivers of freshwater anomalies are traditionally more narrowly defined than for the SST. Runoff and melting, specifically, will likely increase in future, which is expected to increase the meridional SST gradient and associated air-sea feedbacks.

Please also see our responses to your Comment 10.

15. l. 610: What salinity did you use in the evaluation of the buoyancy flux?

Thank you for pointing out that we did not specify the value. We now explain that we used a mean observed salinity of 34.5 g kg$^{-1}$ in the subpolar region.

We point that the results are not sensitive to this value. Typical salinity variations in the subpolar region in winter range from 34.5 to 35. Since the salinity appears in the nominator, and the surface buoyancy flux is over one order of magnitude smaller than the density anomaly associated with the cold anomaly, the exact value has no appreciable effect on the results.

16. l. 664: Regarding the "northward deflection of the jet stream", please see my comments above.

Following your comments above, we have now added more details on the large-scale atmospheric circulation anomaly.

**III. Typos/Wording:**

I suggest the following changes:

l. 39: "requiring a fine grid spacing" to "requiring *ocean models* with fine grid spacing"
Caption of Figure 1 d: "SST" to "ΔSST"
l. 227: "0.5" to "-0.5" (minus sign missing)
l. 368: "pariticularly" to "particularly"
l. 371: "circulation" to "atmospheric circulation"
Figure 7: The title of panel e should be $T_{+1}$
l. 657: "ΔSSS" to "ΔSST"

Thank you for all these suggestions and for spotting the typos. We have adopted all your suggestions.

**References:**

Cassou, C., 2008. Intraseasonal interaction between the Madden-Julian Oscillation and the North Atlantic Oscillation. Nature 455, 523–527. https://doi.org/10.1038/nature07286

Grams, C.M., Beerli, R., Pfenninger, S., Staffell, I., Wernli, H., 2017. Balancing Europe's wind-power output through spatial deployment informed by weather regimes. Nature Clim Change 7, 557–562. https://doi.org/10.1038/nclimate3338

Thank you for your detailed and constructive feedback helping us to clarify and improve this study. It is much appreciated!

---

## Author Response (AR4)

30. November 2023

Dear Prof. Pfahl,

Thank you for your suggestions. I have now updated the author contributions and turned the Appendix into a Supplement. Thus, the manuscript is now shorter.

I sincerely thank you for your letter and for handling this manuscript.

Best wishes,
Marilena